# Stroke genetics informs drug discovery and risk prediction across ancestries

Previous genome-wide association studies (GWASs) of stroke — the second leading cause of death worldwide — were conducted predominantly in populations of European ancestry[1,2]. Here, in cross-ancestry GWAS meta-analyses of 110,182 patients who have had a stroke (five ancestries, 33% non-European) and 1,503,898 control individuals, we identify association signals for stroke and its subtypes at 89 (61 new) independent loci: 60 in primary inverse-variance-weighted analyses and 29 in secondary meta-regression and multitrait analyses. On the basis of internal cross-ancestry validation and an independent follow-up in 89,084 additional cases of stroke (30% non-European) and 1,013,843 control individuals, 87% of the primary stroke risk loci and 60% of the secondary stroke risk loci were replicated (P < 0.05). Effect sizes were highly correlated across ancestries. Cross-ancestry fine-mapping, in silico mutagenesis analysis[3], and transcriptome-wide and proteome-wide association analyses revealed putative causal genes (such as *SH3PXD2A* and *FURIN*) and variants (such as at *GRK5* and *NOS3*). Using a three-pronged approach[4], we provide genetic evidence for putative drug effects, highlighting F11, KLKB1, PROC, GP1BA, LAMC2 and VCAM1 as possible targets, with drugs already under investigation for stroke for F11 and PROC. A polygenic score integrating cross-ancestry and ancestry-specific stroke GWASs with vascular-risk factor GWASs (integrative polygenic scores) strongly predicted ischaemic stroke in populations of European, East Asian and African ancestry[5]. Stroke genetic risk scores were predictive of ischaemic stroke independent of clinical risk factors in 52,600 clinical-trial participants with cardiometabolic disease. Our results provide insights to inform biology, reveal potential drug targets and derive genetic risk prediction tools across ancestries.

Stroke is the second leading cause of death worldwide, responsible for approximately 12% of total deaths, with an increasing burden particularly in low-income countries[6]. Characterized by a neurological deficit of sudden onset, stroke is predominantly caused by cerebral ischaemia (of which the main aetiological subtypes are large-artery atherosclerotic stroke (LAS), cardioembolic stroke (CES), and small-vessel stroke (SVS)) and, less often, by intracerebral haemorrhage (ICH). The frequency of stroke subtypes differs between ancestry groups as exemplified by a higher prevalence of SVS and ICH in Asian and African populations compared with European populations. Most genetic loci associated with stroke have been identified in populations of European ancestry. The largest published GWAS meta-analysis to date (67,162 cases and 454,450 control individuals, MEGASTROKE) reported 32 stroke risk loci[1]. To identify new genetic associations and provide insights into stroke pathogenesis and putative drug targets, we first performed a cross-ancestry GWAS of 1,614,080 participants, including 110,182 patients who had a stroke, and followed up genome-wide significant signals in an independent dataset of 89,084 patients who had a stroke and 1,013,843 control individuals. We then characterized the identified stroke risk loci by leveraging expression and protein quantitative trait loci, cross-ancestry fine-mapping and shared genetic variation with other traits. Finally, we used a series of approaches for genomics-driven drug discovery for stroke prevention and treatment, and examined the

prediction of stroke with polygenic scores (PGSs) across ancestries in the setting of both population-based studies and clinical trials.

## Genetic discovery from GWASs

We performed a fixed-effect inverse-variance weighted (IVW) GWAS meta-analysis on 29 population-based cohorts or biobanks with incident stroke ascertainment and 25 clinic-based case–control studies, comprising up to 110,182 patients who had a stroke and 1,503,898 control individuals (of whom 45.5% were in longitudinal cohorts or biobanks), nearly doubling the number of cases in previous stroke GWASs (the GIGASTROKE initiative; Supplementary Table 1 and Extended Data Fig. 1). Genome-wide genotyping and imputation characteristics are described in Supplementary Table 2. The cohorts included individuals of European (66.7% of the patients who had a stroke), East Asian (24.8%), African American (3.7%), South Asian (3.3%) and Hispanic (1.4%) ancestry. Analyses were performed for any stroke (AS; comprising ischaemic stroke, ICH, and stroke of unknown or undetermined type), any ischaemic stroke regardless of subtype (AIS; n = 86,668) and ischaemic stroke subtypes (LAS, n = 9,219; CES, n = 12,790; SVS, n = 13,620). We also conducted separate GWAS analyses of incident AS and AIS (n = 32,903 and n = 16,863, respectively) in longitudinal population-based cohort studies.

A list of authors and their affiliations appears online. ✉e-mail: Martin.Dichgans@med.uni-muenchen.de; stephanie.debette@u-bordeaux.fr

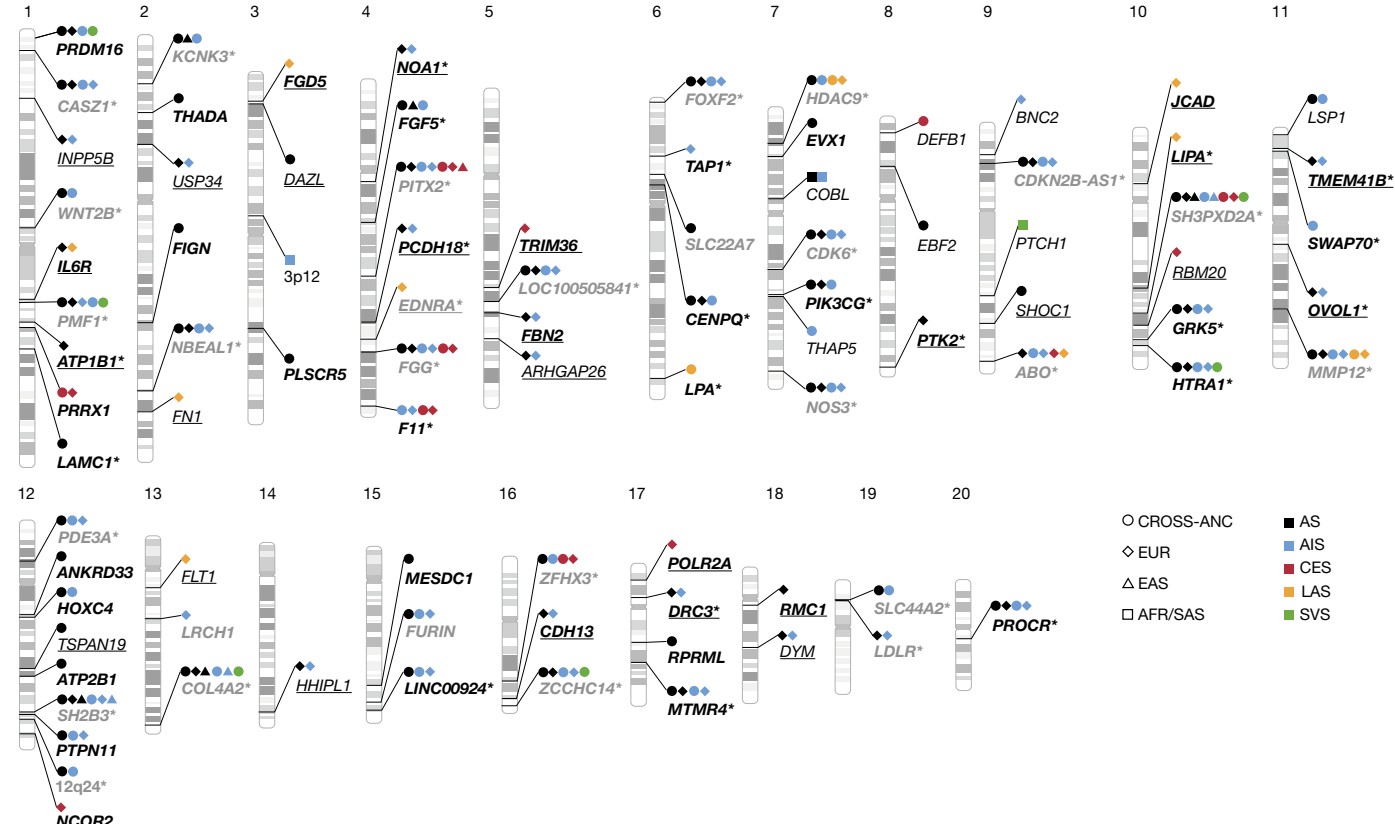

**Fig. 1 | Identifying genetic variants that influence stroke risk.** Ideogram showing 89 genome-wide significant stroke-risk loci. The shapes correspond to ancestry: circles, cross-ancestry (CROSS-ANC); diamonds, Europeans (EUR); triangles, East Asians (EAS); squares, African Americans (AFR) or South Asians (SAS). Colours correspond to stroke types: green, AS; red, AIS; light blue, SVS; dark blue, CES; purple, LAS. The nearest genes to lead variants are displayed.

Loci are characterized as follows, on the basis of replication results (Methods): bold with asterisk, high confidence; bold without asterisk, intermediate confidence; not bold, low confidence; underlined, loci identified in secondary MR-MEGA and MTAG analyses. Black and grey font indicate new and known loci, respectively. The numbers at the top indicate the chromosome.

We tested up to around 7,588,359 single-nucleotide polymorphisms (SNPs) with a minor allele frequency (MAF) of ≥0.01 for association with stroke. The linkage-disequilibrium score intercepts for our ancestry-specific GWAS meta-analyses ranged from 0.91 to 1.12, suggesting that there was no systematic inflation of association statistics (Supplementary Table 3). By performing IVW GWAS meta-analyses, we identified variants associated with stroke at genome-wide significance ($P < 5 \times 10^{-8}$) at 60 loci, of which 33 were new (Fig. 1 and Supplementary Table 4). Lead variants at all of the new loci were common (MAF ≥ 0.05), except for low-frequency intronic variants in *THAP5* (MAF = 0.02, in complete association ($r^2 = 1$) with variants in the 5′ UTR of *NRCAM*) associated with cross-ancestry incident AS/AIS, and in *COBL* (MAF = 0.04) associated with AS/AIS in South Asian individuals. Most of the associations for these 60 loci were with AS (48 loci, 23 new) and AIS (45 loci, 18 new), and one of the AIS loci was associated only with incident AIS (Supplementary Table 4c). Although AIS subtypes were not available in some population-based cohorts (Supplementary Table 1), genome-wide significance was reached for 4 loci for LAS, 8 for CES and 7 for SVS (of which 1, 3 and 3 were new, respectively; Supplementary Table 4). Our results include a large and comprehensive description of stroke genetic risk variants in each of the five represented ancestries. In cross-ancestry meta-analyses, 53 loci (51 loci after controlling for ancestry-specific linkage-disequilibrium score intercepts) reached genome-wide significance (Supplementary Table 4), whereas 42 loci were genome-wide significant in individual ancestries (35 in Europeans, 6 in East Asians, 1 in South Asians and 2 in African Americans; Supplementary Table 4). Using conditional and joint analysis (GCTA-COJO)[7], we confirmed three independent signals at *PITX2* and two at *SH3PXD2A*[1] (CES in Europeans;

Supplementary Table 5). We also performed cross-ancestry gene-based association tests using VEGAS2[8] and MAGMA[9], which revealed 267 gene-wide significant associations ($P < 2.63 \times 10^{-6}$) at 39 loci, of which 14 were in 8 new loci that did not reach genome-wide significance in the single-variant analyses (*AGAP5/SYNPO2L/SEC24C/CHCHD1*, *CD96*, *HNRNPA0*, *MAMSTR*, *PPM1H*, *RALGAPA1*, *USP34* and *USP38*; Supplementary Tables 6 and 7).

Next, we conducted a secondary cross-ancestry GWAS meta-analysis using MR-MEGA[10], which accounts for the allelic heterogeneity between ancestries. We identified three additional genome-wide significant loci for AS (all new), near *TSPAN19*, and in introns of *DAZL* and *SHOC1*, all showing high heterogeneity in allelic effects across ancestries (heterogeneity $P < 0.01$; Supplementary Table 8). To further enhance the statistical power for AIS subtypes, we conducted secondary multitrait analyses of GWASs (MTAG)[11] in Europeans and East Asians, including traits correlated with specific stroke subtypes, namely (1) coronary artery disease (CAD) for LAS, both caused by atheroma; (2) atrial fibrillation for CES, as its main underlying cause; and (3) white matter hyperintensity volume (WMH, an MRI-marker of cerebral small vessel disease) for SVS (available in Europeans only). In Europeans, 11 additional loci were associated with LAS (10 new), 3 with SVS (all reported in a recent SVS GWAS[2]) and 5 with CES (all new; Supplementary Tables 9–11). Moreover, 18 and 15 additional genome-wide significant associations were identified (all new) for AS and AIS, respectively, using MTAG with WMH, CAD and atrial fibrillation (Supplementary Tables 12 and 13). In East Asian individuals, one locus was associated with AS (*FGF5*) and one with LAS (*HDAC9*, new in East Asians) using MTAG. This brings the number of identified stroke-risk loci from primary (IVW)

and secondary (MR-MEGA and MTAG) analyses to 89 in total (61 new), of which 69 were associated with AS, 45 with AIS, 15 with LAS, 13 with CES and 10 with SVS (of these 44, 33, 11, 8 and 3 were new, respectively; Fig. 1 and Supplementary Tables 4, 8 and 9–14).

## Independent follow-up of GWAS signals

We followed up genome-wide significant stroke-risk loci both internally and externally. First, we sought to replicate the 42 stroke-risk loci that reached genome-wide significance in individual ancestries in at least one other ancestry group among the discovery samples. We successfully replicated, with consistent directionality, 10 of these loci at $P < 1.19 \times 10^{-3}$ (accounting for the number of loci tested), of which 7 were genome-wide significant in Europeans, 1 in East Asians, and 2 in both Europeans and East Asians. An additional 15 loci showed nominal association ($P < 0.05$) in at least one other ancestry (Supplementary Table 15).

Second, we gathered an independent dataset of 89,084 individuals who had a stroke (AS; of which 85,546 AIS; 70.0% European, 15.6% African American, 10.1% East Asian, 4.1% Hispanic and 0.1% South Asian) and 1,013,843 control individuals, mostly from large biobanks, for external replication (the biobank setting did not allow suitable ischaemic stroke subtype analyses). Out of the 60 loci that reached genome-wide significance in the IVW meta-analyses, 48 loci (80%) replicated at $P < 0.05$ with consistent directionality (Extended Data Fig. 2), of which 31 (52%) replicated at $P < 8.2 \times 10^{-4}$ (accounting for the number of loci tested) (Supplementary Table 16). When considering both the internal and external follow-up, 52 (87%) of the 60 IVW loci replicated, of which 37 replicated with high confidence, and 15 with intermediate confidence (Methods, Fig. 1 and Supplementary Table 14). The 8 loci that did not replicate were labelled as low confidence (Methods and Supplementary Table 14). Four of these were ethnic specific and three were low-frequency variants that were monomorphic in some ancestries and were therefore probably underpowered for replication.

Within the secondary analyses, none of the three MR-MEGA loci replicated, although one was borderline significant (Supplementary Table 16). Of the 26 MTAG loci, 18 (69%) replicated with AS or AIS at $P < 0.05$, of which 9 (35%) replicated with high confidence ($P < 1.7 \times 10^{-3}$, accounting for 29 secondary loci tested; Supplementary Table 16). Of the eight MTAG loci that did not replicate, seven showed a consistent directionality and four were subtype specific and were therefore underpowered to detect associations with AS or AIS.

## Cross-ancestry effects and fine-mapping

For the 60 loci associated with stroke risk derived from the IVW meta-analyses, we first demonstrated the added value in terms of locus discovery of including non-European samples, showing a clear gain in power beyond sample size increase, compared with the incremental addition of European ancestry samples (Extended Data Fig. 3). We next compared the per-allele effect size across the three ancestries with the largest sample size (European, East Asian, African American). Correlations of per-allele effect sizes of index variants varied from $r = 0.55$ (European with African American) to $r = 0.66$ (European with East Asian) and $r = 0.74$ (East Asian with African American; Fig. 2a).

To identify putative causal variants at stroke-risk loci identified through IVW meta-analyses, we performed multiple-causal-variant fine-mapping using SuSiE[12], separately in European and East Asian participants (Methods). Across stroke types, we identified 110 and 16 95% credible set–trait pairs in European and East Asian participants, respectively, each of which having a 95% posterior probability of containing a causal variant, with multiple credible sets identified at 6 (in Europeans) and 1 (in East Asians) stroke-risk loci (Supplementary Tables 17–19). Within the credible sets identified in European participants, 17 variants were found to have a posterior inclusion probability

(PIP) of >0.9. We found overlapping credible sets between European and East Asian participants at *SH3PXD2A* (19 overlapping variants), suggesting that there is cross-ancestry-shared genetic architecture at this locus (Fig. 2b). Two loci had credible sets with a single variant (rs10886430 at *GRK5* (PIP = 0.999), associated with *GRK5* platelet gene expression and thrombin-induced platelet aggregation[13], and rs1549758 at *NOS3*, PIP = 0.995), probably representing strong targets for functional validation.

Although there were six non-synonymous variants among credible sets (rs671 (*ALDH2*), rs8071623 (*SEPT4*), rs35212307 (*WDR12*), rs72932557 (*CARF*), rs11906160 (*MYH7B*) and rs2501968 (*CENPQ*)), exonic variants for coding RNA within credible sets were few (1.2%). To detect putative causal regulatory variants, we conducted an in silico mutagenesis analysis using MENTR, a machine-learning method to precisely predict transcriptional changes caused by causal variants[3]. From credible sets, we obtained 78 robust predictions of variant–transcript-model sets comprising 13 variants and 19 transcripts (Supplementary Table 20), involving multiple cell types, consistent with the diversity of mechanisms that underlie stroke aetiology. For example, the G allele of rs12476527 (5′ UTR of *KCNK3*) is a risk allele for stroke and was predicted to increase *KCNK3* expression in kidney cortex tubule cells, despite no expression quantitative trait loci (eQTL) of this variant being reported in Genotype-Tissue Expression (GTEx, v.8) or eQTLgen (2019-12-23). The same G allele has been associated with higher systolic blood pressure[14]. Furthermore, three variants (rs12705390 at *PIK3CG*, rs2282978 at *CDK6* and rs2483262 at *PRDM16*) were predicted to affect the expression of a long non-coding RNA and enhancer RNAs, predominantly in endothelial cells, as well as other vascular cells and visceral preadipocytes, whereas a promoter variant of *SH3PXD2A* was predicted to modulate its expression in macrophages.

## Characterizing stroke-associated loci

VEGAS2Pathway[15] analysis revealed significant enrichment ($P < 5.01 \times 10^{-6}$) of stroke-risk loci in pathways involved in (1) carboxylation of amino-terminal glutamate residues required for the activation of proteins involved in blood clot formation and regulation; (2) negative regulation of coagulation; and (3) angiopoietin receptor Tie2-mediated signalling, involved in angiogenesis (Supplementary Table 21).

We examined shared genetic variation with 12 (in Europeans) and 10 (in East Asians) vascular risk factors and disease traits (Methods and Supplementary Methods). In Europeans, the lead variants for stroke at 57 of the 89 primary and secondary risk loci (64.0%) were associated ($P < 5 \times 10^{-8}$) with at least one vascular trait, most frequently blood pressure (33 loci, 37.1%; Extended Data Fig. 4 and Supplementary Table 22). After correction for multiple testing (Methods; $P < 4.17 \times 10^{-3}$), all of the vascular-risk traits except for low-density lipoprotein (LDL)-cholesterol showed significant genetic correlation ($r_g$) with at least one stroke type, the strongest correlations being for CAD and LAS ($r_g = 0.73$), atrial fibrillation and CES ($r_g = 0.63$), and systolic blood pressure (SBP) with all stroke types ($r_g$ ranging from 0.21 for CES to 0.49 for LAS and SVS; Extended Data Fig. 5 and Supplementary Table 23). Using two-sample Mendelian randomization (MR), we found evidence for a possible causal association for every vascular-risk trait except for triglycerides with at least one stroke type ($P < 4.17 \times 10^{-3}$), with some subtype-specific association patterns. Genetic liability to WMH was associated with increased risk of SVS but not other stroke subtypes, whereas genetic liability to venous thromboembolism was associated with AS, AIS, CES and LAS, but not SVS (Extended Data Fig. 5 and Supplementary Table 24). Owing to a limited overlap between the European GIGASTROKE sample and cohorts included in GWASs for the exposure traits, we ran sensitivity analyses weighting our genetic instruments on the basis of a sub-sample of the UK Biobank, excluding cases included in GIGASTROKE[16]. The notable consistency of these with the main analyses confirmed their robustness against weak instrument bias (Supplementary Table 25).

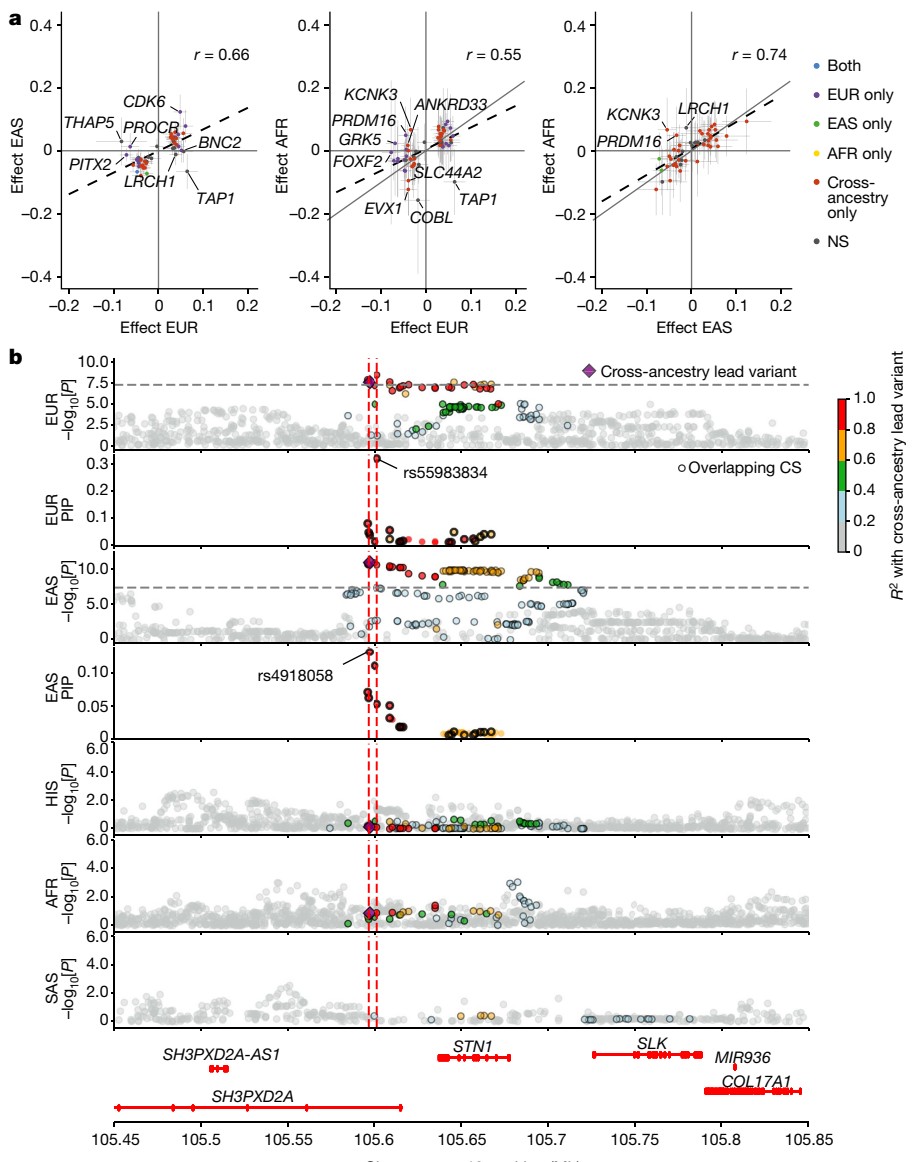

**Fig. 2 | Effect-size comparison across ancestry groups of lead variants identified in stroke GWASs and cross-ancestry fine-mapping. a**, Plots showing the Pearson's correlation coefficient ($r$) between the effect sizes ($\beta$) of the 60 stroke-risk alleles on AS significant after multiple-testing correction ($P < 0.017$) in Europeans and East Asians (left; $r$ (95% CI) = 0.66 (0.47–0.79), $P = 1 \times 10^{-7}$); Europeans and African Americans (middle; $r$ (95% CI) = 0.55 (0.33–0.71), $P = 2 \times 10^{-5}$); and East Asians and African Americans (right; $r$ (95% CI) = 0.74 (0.58–0.85), $P = 8 \times 10^{-10}$). $n = 60$ independent stroke-risk variants from the IVW meta-analyses were used to compute Pearson's correlation coefficients ($r$) of the effect sizes between ancestries. The nearest gene is reported for SNPs showing a difference in effect size ($\beta$, absolute value) of >0.05 between a pair of ancestries. The dots represent the effect-size ($\beta$) estimates and the bars represent the 95% CI of the estimates. Two-sided $P$ values of the deviation of Pearson's correlation coefficient from zero are reported. Colour corresponds to genome-wide significant association ($P < 5 \times 10^{-8}$) in individual ancestries:

purple, European only (±cross-ancestry); green, East Asian only (±cross-ancestry); yellow, African American only (±cross-ancestry); blue, both ancestries (±cross-ancestry); red, cross-ancestry only; grey, not genome-wide significant in two plotted ancestries and in cross-ancestry. **b**, Locus plots of variants at *SH3PXD2A* in five ancestries. Fine-mapped variants are shown only in European and East Asian individuals (insufficient power for other ancestries). Variants are coloured on the basis of their linkage disequilibrium with the cross-ancestry lead variant (rs4918058), shown by the purple diamonds. In the fine-mapping plots, variants in the SuSiE 95% credible sets (CS) are shown. Shared variants between credible sets of European and East Asian participants are indicated by black circles. The red vertical lines represent the position of the lead variants in European (rs55983834) and East Asian (rs4918058) participants. The grey dashed horizontal lines represent $P = 5 \times 10^{-8}$. The linkage disequilibrium of each ancestry was derived from the 1000 Genomes Project.

We confirmed directionality using the Steiger test (Supplementary Table 24) and ruled out reverse causation with reverse MR (Supplementary Table 26). In East Asian individuals, SBP, diastolic blood pressure (DBP), body mass index (BMI) and atrial fibrillation showed significant genetic correlation with AS ($r_g$ = 0.45, 0.39, 0.24 and 0.32 versus $r_g$ = 0.36, 0.21, 0.22 and 0.44 in Europeans) and AIS (except for BMI), with evidence for a causal association of SBP and DBP with AS,

AIS and SVS; CAD with AS, AIS and LAS; and atrial fibrillation with CES (Extended Data Fig. 6 and Supplementary Tables 23 and 24). Notably, MR analyses performed with binary exposures should be interpreted with caution owing to the potential violations of the exclusion restriction assumption[16].

Next, to generate hypotheses of target genes and directions of effect, we conducted transcriptome-wide association studies (TWAS) using

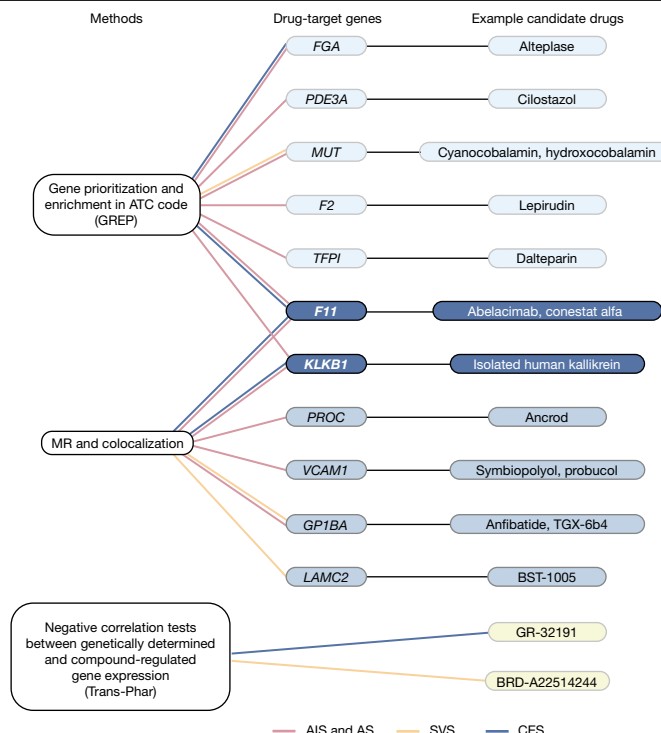

| Methods | Drug-target genes | Example candidate drugs |
|---|---|---|

Gene prioritization and enrichment in ATC code (GREP)
- FGA — Alteplase
- PDE3A — Cilostazol
- MUT — Cyanobobalamin, hydroxocobalamin
- F2 — Lepirudin
- TFPI — Dalteparin

MR and colocalization
- F11 — Abelacimab, conestat alfa
- KLKB1 — Isolated human kallikrein
- PROC — Ancrod
- VCAM1 — Symbiopolyol, probucol
- GP1BA — Anfibatide, TGX-6b4
- LAMC2 — BST-1005

Negative correlation tests between genetically determined and compound-regulated gene expression (Trans-Phar)
- GR-32191
- BRD-A22514244

— AIS and AS  — SVS  — CES

**Fig. 3 | Genomics-driven drug discovery.** Overlap enrichment analysis using GREP[22] (top). Middle, integrating MR results using *cis*- and *trans*-pQTLs as instrumental variables with data from drug databases. Bottom, negative correlation tests between compound-regulated gene expression profiles and genetically determined case–control gene expression profiles using Trans-Phar.

TWAS-Fusion and eQTL based on RNA-sequencing (RNA-seq) analyses in different tissues[17–20]. We identified 27 genes of which the genetically regulated expression is associated with stroke and its subtypes at the transcriptome-wide level and colocalized in at least one tissue (10 genes in arteries and heart; 6 genes in brain tissue; 17 genes across tissues). Of these genes, 18 overlapped with 11 genome-wide significant stroke-risk loci (Extended Data Fig. 7 and Supplementary Table 27). For several genes of which bulk tissue expression levels showed evidence for association with stroke, human single-nucleus sequencing data of brain cells in the dorsolateral prefrontal cortex (DLPFC) showed distinct cell-specific gene expression patterns suggesting that multiple genes could be involved through different cell types[21] (Extended Data Fig. 8). Overall, we observed a significant enrichment mostly in brain vascular endothelial cells and astrocytes, possibly reflecting the importance of both vascular pathology and brain response to the vascular insult in modulating stroke susceptibility (Extended Data Fig. 8 and Supplementary Tables 28 and 29). Furthermore, using proteome-wide association studies (PWAS) in DLPFC brain tissue, we found evidence for the association of ICA1L with AS and AIS through its *cis*-regulated protein abundance, with colocalization evidence (Extended Data Fig. 8 and Supplementary Table 30). In both TWAS and PWAS, lower *ICA1L* transcript or protein abundance in the DLPFC was associated with a higher risk of stroke.

## Genomics-driven drug discovery

We used a three-pronged approach for genomics-driven discovery of drugs for the prevention or treatment of stroke[4] (Methods and Fig. 3). First, using GREP[22], we observed significant enrichment of stroke-associated genes (MAGMA[9] or VEGAS2[8] false-discovery rates (FDR) < 0.05) in drug-target genes for blood and blood-forming organs

(Anatomical Therapeutic Chemical Classification System B drugs, for AS, AIS and CES). This encompasses the previously described *PDE3A* and *FGA* genes[1], which encode targets for cilostazol (antiplatelet agent) and alteplase (thrombolytic drug acting through plasminogen[23]), respectively, as well as *F11*, *KLKB1*, *F2*, *TFPI* and *MUT*, which encode targets for conestat alfa, ecallantide (both used for hereditary angioedema), lepirudin, dalteparin (both used to treat recurrent thromboembolism) and vitamin B12, respectively (Supplementary Table 31). Notably, the results for AS are probably driven by AIS (the vast majority of AS in the current study) and cannot be extrapolated to ICH. Second, we used Trans-Phar[24] to test the negative correlations between genetically determined case–control gene expression associated with stroke (TWAS using all GTEx v.7 tissues[17]) and compound-regulated gene expression profiles. At FDR < 0.10, we observed significant negative correlations for BRD.A22514244 (for SVS; drug target unknown) and GR.32191 (for CES; Supplementary Table 32). GR-32191 is a thromboxane A2 receptor antagonist that has been proposed as an alternative antiplatelet therapy for stroke prevention[25], and further drugs of this class are under development[26]. Note that one of those drugs, terutroban, was evaluated in a phase III study but did not show non-inferiority against aspirin[27]. Third, we used protein quantitative trait loci (pQTL) for 218 drug-target proteins as instruments for MR and found evidence for causal associations of 9 plasma proteins with stroke risk (4 *cis*-pQTL and 6 *trans*-pQTL), of which 7 were supported by colocalization analyses, with no evidence for reverse causation using the Steiger test (PROC, VCAM1, F11, KLKB1, MMP12, GP1BA and LAMC2; Supplementary Table 33). All of these replicated (at FDR < 0.05) with consistent directionality using at least one independent plasma pQTL resource and cerebrospinal fluid pQTL for PROC and KLKB1, with evidence for colocalization for PROC, F11, KLKB1 and MMP12, but not for GP1BA (for which both concordant and discordant directionality was observed) and LAMC2 (pQTL available in one replication dataset only; FDR = 0.08). Using public drug databases, we curated drugs targeting those proteins in a direction compatible with a beneficial therapeutic effect against stroke based on MR estimates and identified such drugs for VCAM1, F11, KLKB1, GP1BA, LAMC2 (inhibitors) and PROC (activators; Supplementary Table 34). Drugs targeting F11 (NCT04755283, NCT04304508, NCT03766581) and PROC (NCT02222714) are currently under investigation for stroke, and our results provide genetic support for this. Notably, *F11* and *KLKB1* are adjacent genes with a long-range linkage-disequilibrium pattern and complex co-regulation[28], as illustrated here by the presence of a shared *trans*-pQTL in *KNG1* (Supplementary Table 33). Additional studies are needed to disentangle causal associations and the most appropriate drug target in this region[29,30]. Next, for the five genes targeted by inhibitors, *VCAM1*, *F11*, *KLKB1*, *GP1BA* and *LAMC2*, we examined the associations of rare deleterious variants (MAF < 0.01) with stroke and stroke-related traits, applying gene-based burden tests to whole-exome sequencing data from >450,000 UK Biobank participants to support potential therapeutic targets for inhibitors[31]. We observed one significant protective association of rare deleterious variants in *F11* with venous thromboembolism (odds ratio (OR) = 0.471, $P = 2.46 \times 10^{-4}$), in a direction concordant with that of MR estimates (Supplementary Table 35). To further validate the candidate drugs and estimate their potential side effects, we investigated whether the drug-target genes were associated with stroke-related phenotypes using a phenome-wide association study (PheWAS) approach. We conducted PheWAS in the Estonian Biobank (EstBB) for pQTL variants for the *PROC*, *VCAM1*, *F11*, *KLKB1*, *GP1BA* and *LAMC2* genes. A *cis*-pQTL for *F11*, rs2289252, was associated with higher risk of venous thromboembolic disorders ($P < 3.45 \times 10^{-6}$), as previously described[32], and showed suggestive association ($P = 3.44 \times 10^{-3}$) with cerebral artery occlusion with cerebral infarction (Phecode 433.21; Extended Data Fig. 9 and Supplementary Table 36). By contrast, we observed no significant association with non-stroke-related phenotypes, suggesting the safety of targeting F11. Similar profiles were observed in the UK Biobank (https://pheweb.org/

UKB-SAIGE/variant/4-187207381-C-T) and FinnGen (https://r7.finngen.fi/variant/4-186286227-C-T), with no significant associations with other disorders and no overlap of subthreshold signals with side-effects reported in clinical trials[33]. We further confirmed the association of rs2289252 with venous thromboembolic disorders and that it has no association with other non-stroke-related phenotypes using the Phenoscanner database (Supplementary Table 37).

Overall, combining evidence from genomics-driven drug discovery approaches, characterization of stroke-risk loci (missense variants, TWAS, PWAS, colocalization, pathway enrichment, MR with pQTL, MENTR and PoPS[34]), and previous knowledge from monogenic disease models and experimental data, we found evidence for the potential functional implication of 56 genes that should be prioritized for further functional follow-up, with evidence from multiple approaches for 20 genes (Supplementary Table 38).

## Integrative polygenic risk prediction

We investigated the risk prediction potential of stroke GWASs, alone and in combination with vascular-risk-trait GWASs, first in Europeans and East Asians, using ancestry-specific PGSs. PGSs were based on ancestry-specific and cross-ancestry GWAS summary statistics. We first derived single PGS (sPGS) models from single stroke GWAS summary data (Supplementary Table 39). We then constructed integrative PGS (iPGS) models, which combined multiple GWAS summary data of different traits into a PGS using elastic-net logistic regression[5] (Extended Data Fig. 10). The iPGS analysis used two datasets for each ancestry for model training and evaluation, respectively. The participants in the training and evaluation datasets did not overlap and were not included in the input GWAS summary data.

For Europeans, we constructed the iPGS model using 1,003 prevalent AIS cases and 8,997 controls, followed by evaluation of the model using 1,128 incident AIS cases among 102,099 participants, all from the EstBB. The improvement in predictive ability ($\Delta C$-index) was assessed over a base model including age, sex and the top 5 principal components (PCs) for population stratification. The iPGS model for Europeans incorporated 10 GIGASTROKE GWAS analyses (all stroke types, using the European and cross-ancestry analysis) and 12 vascular-risk-trait GWAS analyses (Extended Data Fig. 10 and Supplementary Table 40). The iPGS model achieved a $\Delta C$-index of 0.027 (Supplementary Table 41), 93% higher than that for a previously constructed iPGS model for Europeans, derived from 5 MEGASTROKE GWAS analyses and similar vascular-risk-trait GWASs ($\Delta C$-index = 0.014)[5]. The age-, sex- and top 5 PC-adjusted hazard ratio (HR) per s.d. of the iPGS was 1.26 (95% confidence interval (CI) = 1.19–1.34, $P = 2.0 \times 10^{-15}$) for the GIGASTROKE-based iPGS model compared to 1.19 (95% CI = 1.12–1.26, $P = 4.2 \times 10^{-9}$) for the MEGASTROKE-based iPGS model. Compared with participants in the middle 10% (45–55%) of the GIGASTROKE-based iPGS model, those in the top 1% showed a >2.5-fold higher hazard of AIS (HR = 2.56, 95% CI = 1.59–4.10, $P = 9.6 \times 10^{-5}$; Fig. 4a and Supplementary Table 42). We further confirmed the GIGASTROKE-based European iPGS model trained on the EstBB in 403,489 European-ancestry participants of the Million Veteran Program (MVP) study, of whom 8,392 developed an AIS: HR per s.d. = 1.19 (95% CI = 1.16–1.21, $P = 6.94 \times 10^{-52}$), with a $\Delta C$-index of 0.010 (Supplementary Table 43).

For East Asians, we derived the iPGS model using 577 cases of prevalent AIS and 9,232 control individuals, and evaluated the model using 1,470 cases of prevalent AIS and 40,459 control individuals from Biobank Japan (BBJ). A base model including age, sex and the top 5 PCs showed an area under the curve (AUC) of 0.634. The iPGS model was constructed by integrating 10 GIGASTROKE GWAS analyses and 12 vascular-risk-trait GWAS analyses (Extended Data Fig. 10 and Supplementary Table 44). The iPGS model for East Asians showed an improvement in AUC ($\Delta$AUC) of 0.019 (Supplementary Table 45). The age-, sex- and top 5 PC-adjusted odds ratio (OR) per s.d. of PGS was 1.33 (95% CI = 1.26–1.40, $P = 9.9 \times 10^{-26}$) for the iPGS model. The MEGASTROKE- and GIGASTROKE-based iPGS models for Europeans achieved a lower AUC improvement ($\Delta$AUC = 0.007 and 0.009, respectively) than the GIGASTROKE-based iPGS model for East Asians. While this suggests that the transferability of iPGS models from Europeans to East Asians might be limited (Supplementary Table 45), it does indicate that an ancestry-specific stroke iPGS approach yields similar improvement in predictive ability relative to their base models.

Participants in the top 1% of the iPGS showed 1.9-fold higher odds of AIS (OR = 1.90, 95% CI = 1.20–2.91, $P = 0.004$) compared with the middle 10% (Fig. 4b and Supplementary Table 46). We further confirmed the GIGASTROKE-based East Asian iPGS model trained on the BBJ in 1,399 cases of prevalent AIS and 86,283 controls from the Taiwan Biobank (TWB): OR per s.d. = 1.18 (95% CI = 1.12–1.25, $P = 1.1 \times 10^{-9}$), with a $\Delta$AUC of 0.003 (Supplementary Table 47).

Notably, iPGS models derived from cross-ancestry stroke GWASs had a higher predictive ability compared with iPGS models derived from ancestry-specific stroke GWASs both in Europeans and East Asians (Supplementary Table 48).

Next, we evaluated the predictive ability of the European-derived GIGASTROKE-based iPGS model in African American and indigenous African (Nigerian and Ghanaian) datasets. In 107,343 African American MVP participants, of whom 2,227 developed an AIS, the GIGASTROKE-based iPGS model showed a significant association with AIS incidence (HR per 1 s.d. = 1.11, 95% CI = 1.06–1.17, $P = 1.8 \times 10^{-5}$, $\Delta C$-index = 0.003; Supplementary Table 49), although weaker than in European MVP participants (Supplementary Table 43). The participants in the top 1% of the iPGS showed 1.5-fold higher odds of AIS (HR = 1.53, 95% CI, 1.04–2.25, $P = 0.03$) compared with participants in the middle 10% (Fig. 4c and Supplementary Table 50). In 1,691 cases and 1,743 control participants from the indigenous African (Nigerian and Ghanaian) SIREN case–control study, the GIGASTROKE-based iPGS also showed a significant association with the odds of AIS (OR per 1 s.d. = 1.09, 95% CI = 1.02–1.17, $P = 0.010$, $\Delta$AUC = 0.007; Supplementary Table 51). The GIGASTROKE-based iPGS model showed a stronger association with AIS and a larger improvement in predictive ability compared with the MEGASTROKE-based iPGS model in both MVP and SIREN (Supplementary Tables 49 and 51).

## Risk prediction in clinical trials

Following up on previous work[1,35], we further examined whether a genetic risk score (GRS) based on genome-wide significant risk loci from the cross-ancestry IVW AS meta-analyses could identify individuals who are at higher risk of AIS after accounting for established risk factors in five clinical trials across the spectrum of cardiometabolic disease[35]. The primary analysis was conducted in 51,288 European participants of whom 960 developed an incident ischaemic stroke (AIS) over a 3 year follow-up. In a Cox model adjusted for age, sex and vascular risk factors (Methods), a higher GIGASTROKE GRS was significantly associated with increased risk of AIS in Europeans (adjusted HR = 1.17, 95% CI = 1.09–1.24 per s.d. increase, $P = 2 \times 10^{-6}$; Supplementary Table 52). This association was substantially stronger than the association with the earlier MEGASTROKE GRS based on 32 genome-wide significant stroke-risk loci (HR = 1.07, 95% CI = 1.00–1.14, $P = 0.036$)[1,35]. Compared with patients in the lowest GIGASTROKE GRS tertile, patients in the top GRS tertile had an adjusted HR of 1.35 (95% CI = 1.16–1.58) for developing AIS, whereas those in the middle tertile had an adjusted HR of 1.13 (95% CI = 0.96–1.33, $P_{trend} = 1.4 \times 10^{-4}$; Fig. 4e). The performance of the GRS was stronger in individuals who had not previously had a stroke ($n = 44,095$; adjusted HR of the top versus lowest tertile = 1.37, 95% CI = 1.14–1.65) compared with in those who previously had a stroke ($n = 7,193$; adjusted HR = 1.15, 95% CI = 0.87–1.54). Similar associations were observed when using effect estimates from stroke GWAS meta-analyses in Europeans or for AIS (Supplementary Table 52). In secondary analyses, we examined

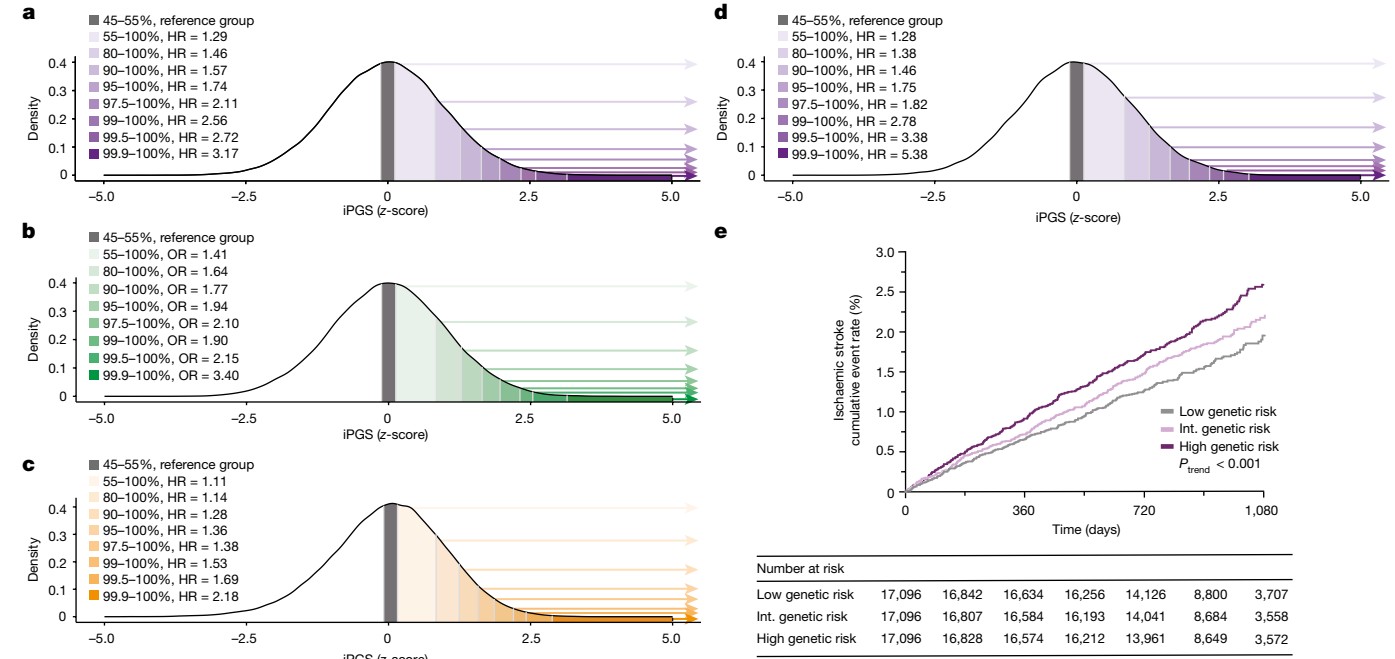

**Fig. 4 | Risk prediction in a population and trial setting. a–d**, The association of iPGS with ischaemic stroke (AIS) in European (Estonian Biobank) (**a**), East Asian (BioBank Japan) (**b**), African American (Million Veteran Program) (**c**) and European participants in clinical trials (**d**). Compared with the middle decile (45–55%) of the population as a reference group, the risk of high-iPGS groups with varying percentile thresholds was estimated using a Cox proportional hazards model for European and African American individuals and logistic regression models for East Asian individuals with adjustments for age, sex and the top five genetic principal components. **e**, Kaplan–Meier event rates for ischaemic stroke in European participants in five clinical trials (Methods) by tertile of GRS at 3 years (the GRS uses effect estimates of the cross-ancestry AS GWAS as weights) showing higher GRS increases risk of ischaemic stroke ($P_{trend} = 1.4 \times 10^{-4}$). The two-sided $P_{trend}$ value was computed using Cox regression. Int., intermediate.

the association of the GIGASTROKE cross-ancestry AS GRS with incident AIS in the much smaller East Asian sample (1,312 participants of whom 27 developed an incident AIS over a 3 year follow-up), and found consistent associations (adjusted HR = 1.49, 95% CI = 1.00–2.21 per s.d. increase, $P = 0.048$; Supplementary Table 52), whereas the MEGASTROKE GRS was not associated with incident AIS in East Asians (adjusted HR = 0.82, 95% CI = 0.55–1.23, $P = 0.34$). Finally, in European trial participants (there were too few East Asian individuals for this analysis), the GIGASTROKE-based iPGS was also significantly associated with increased AIS incidence (HR per 1 s.d. increase = 1.19, 95% CI = 1.11–1.27, $P = 3.2 \times 10^{-7}$, $\Delta C$-index = 0.008), performing better than the MEGASTROKE-based iPGS (Supplementary Table 53). Compared with the middle 10% of the participants, those in the top 1% had a 2.8-fold higher hazard of AIS (HR = 2.78, 95% CI = 1.67–4.61, $P = 7.9 \times 10^{-5}$) (Fig. 4d and Supplementary Table 54).

## Discussion

Our GWAS meta-analyses, including 110,182 patients who had a stroke and 1,503,898 control participants from five different ancestries (33% of patients who had a stroke were non-European), identified 89 (61 new) risk loci for stroke and stroke subtypes (60 through primary IVW and 29 through secondary MR-MEGA and MTAG analyses). We observed substantial shared susceptibility to stroke across ancestries, with a strong correlation of effect sizes. On the basis of internal cross-ancestry validation and independent follow-up in 89,084 cases of stroke (30% non-European) and 1,013,843 control individuals, mostly from large biobanks with information on AS and AIS only, the level of confidence of these loci was intermediate or high for 87% of primary stroke-risk loci and 60% of secondary loci. Effect estimates for variants that were common across ancestries were typically similar, whereas, expectedly, variants that were rare or low frequency in one or more populations

showed differences in effect size, for example, at *PROCR*, *TAP1* or *BNCZ-CNTLN* (MAF ≤ 0.05 in East Asians), or at *GRK5*, *FOXF2* or *COBL* (MAF ≤ 0.05 in African Americans). Ancestry-specific meta-analyses in smaller non-European populations detected fewer loci than in Europeans that were nevertheless biologically plausible, for example, 3p12 and *PTCH1* for SVS in African Americans. Rare variants at 3p12 were recently shown to be associated with WMH volume[36], whereas common variants at *PTCH1* were associated with functional outcome after ischaemic stroke (in European individuals)[37]. New association signals from cross-ancestry GWASs included, for example, variants at *PROCR*, *GRK5* and *F11* (thrombosis), *LPA* and *ATP2B1* (lipid metabolism, hypertension and atherosclerosis), *SWAP70* (membrane ruffling) and *LAMC1* (cerebrovascular matrisome).

Extensive bioinformatics analyses highlight genes for prioritization in functional follow-up studies (Supplementary Table 38). For example, a promoter variant of *SH3PXD2A*, which encodes an adaptor protein that is involved in extracellular matrix degradation through invadopodia and podosome formation, was predicted to modulate its expression in macrophages[38]. *FURIN* expression levels across tissues were associated with an increased stroke risk. *FURIN* has previously been implicated in CAD[39] as well as in atherosclerotic lesion progression in mice[40]. It also has a key role in SARS-CoV-2 infectivity[41], and patients with COVID-19 are at increased risk of AIS, especially LAS[42]; the *FURIN* locus was predominantly associated with LAS in our data (Supplementary Table 55).

Our results provide genetic evidence for putative drug effects using three independent approaches, with converging results from two methods (gene enrichment analysis and pQTL-based MR) for drugs targeting F11 and KLKB1. F11 and F11a inhibitors (such as abelacimab, BAY 2433334 and BMS-986177) are currently being examined in phase 2 trials for primary or secondary stroke prevention (NCT04755283, NCT04304508, NCT03766581). pQTL-based MR suggested PROC as a potential drug target for stroke. A recombinant variant of human activated protein C

(encoded by *PROC*) was found to be safe[43] for the treatment of acute ischaemic stroke after thrombolysis, mechanical thrombectomy or both in phase 1 and 2 trials (3K3A-APC, NCT02222714)[43,44], and is poised for an upcoming phase 3 trial. 3K3A-APC is proposed as a neuroprotectant, with evidence for the protection of white matter tracts and oligodendrocytes against ischaemic injury in mice[45]. Weaker evidence was found for GP1BA, VCAM1 and LAMC2 as potential drug targets for stroke, with evidence for colocalization in only one pQTL dataset. Anfibatide, a GPIbα antagonist, reduced blood–brain barrier disruption after ischaemic stroke in mice[46] and is being tested as an antiplatelet drug in myocardial infarction (NCT01585259). Although specific VCAM1 inhibitors are not available, probucol—a lipid lowering drug with pleiotropic effects including VCAM1 inhibition—was tested for secondary prevention against atherosclerotic events in patients with CAD (PROSPECTIVE, UMIN000003307)[47].

We investigated stroke PGSs across ancestries. PGSs integrating cross-ancestry and ancestry-specific stroke GWASs with vascular-risk-factor GWASs (iPGS) analyses showed strong prediction of ischaemic stroke risk in Europeans and, importantly, in East Asians, in whom stroke incidence is highest[6]. These results were confirmed in several independent datasets. The iPGS performed better than stroke PGS alone and better than the previous best iPGS models in Europeans[5]. The transferability of European-specific iPGS models to East Asians was limited. While there were not enough African participants to generate an African-specific stroke PGS, the European iPGS showed a significant association with AIS in both African American and indigenous African participants, although expectedly weaker than in European participants. Individuals in the top 1% of the PGS distribution had a 2- to 2.5-fold risk of ischaemic stroke in East Asian and European participants compared with those in the middle 10%, whereas this risk was 1.5-fold in African American participants. Although caution is warranted when interpreting risk estimates owing to the wide CIs, these results suggest that GIGASTROKE-based iPGS models may be useful to stratify individuals exposed to genetically high risk of ischaemic stroke, especially in Europeans and East Asians. Our results highlight the importance of ancestry-specific and cross-ancestry genomic studies for the transferability of genomic risk prediction across populations, and the urgent need to substantially increase participant diversity in genomic studies, especially from the most under-represented regions such as Africa, to avoid exacerbation of health disparities in the era of precision medicine and precision public health[48].

Finally, leveraging data from 5 clinical trials in 52,600 patients with cardiometabolic disease, we showed that a cross-ancestry GRS predicted ischaemic stroke, independently of clinical risk factors, and outperforming previous genetic risk evaluation[35]. Notably, although the trials included predominantly European participants, consistent results were observed in East Asian participants. We further confirmed the GIGASTROKE iPGS in these clinical trials.

Our study includes a considerable contribution of non-European stroke genetics resources (*n* = 61,528/616,014 cases/controls for the GWASs and follow-up and an additional *n* = 1,718/3,055 for the PGS/GRS studies). Despite substantial efforts to enhance non-European contributions to GIGASTROKE, we still had limited power for identifying shared causal variants through cross-ancestry fine-mapping. We provided independent validation of the vast majority of identified genome-wide significant associations and graded loci by level of confidence based on these findings. Despite the notable size of the follow-up study sample, with nearly 90,000 additional patients who had a stroke, this analysis remains underpowered, especially for low-frequency variants and ancestry- and subtype-specific associations, as most follow-up studies were derived from large biobanks with event ascertainment based on electronic health records and no suitable stroke subtype information. The muted risk prediction in clinical-trial participants with previous stroke history possibly points to the impact of selection or index event biases and secondary prevention therapy[49].

In conclusion, our genomic findings derived from >200,000 patients who had a stroke worldwide provide critical insights to inform future biological research on stroke pathogenesis, highlight potential drug targets for intervention and provide tools for genetic risk prediction across ancestries.

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

Aniket Mishra[1,418], Rainer Malik[2,418], Tsuyoshi Hachiya[3,418], Tuuli Jürgenson[4,5,418], Shinichi Namba[6,418], Daniel C. Posner[7,418], Frederick K. Kamanu[8,9], Masaru Koido[10,11], Quentin Le Grand[1], Mingyang Shi[11], Yunye He[11], Marios K. Georgakis[2,12,13], Ilana Caro[1], Kristi Krebs[4], Yi-Ching Liaw[14,15], Felix C. Vaura[16,17], Kuang Lin[18], Bendik Slagsvold Winsvold[19,20,21], Vinodh Srinivasasainagendra[22], Livia Parodi[12,13], Hee-Joon Bae[23], Ganesh Chauhan[24], Michael R. Chong[25,26], Liisa Tomppo[27], Rufus Akinyemi[28,29], Gennady V. Roshchupkin[30,31], Naomi Habib[32], Yon Ho Jee[33], Jesper Qvist Thomassen[34], Vida Abedi[35,36], Jara Cárcel-Márquez[37,38], Marianne Nygaard[39,40], Hampton L. Leonard[41,42,43], Chaojie Yang[44,45], Ekaterina Yonova-Doing[46,47], Maria J. Knol[30], Adam J. Lewis[48], Renae L. Judy[49], Tetsuro Ago[50], Philippe Amouyel[51,52,53], Nicole D. Armstrong[54], Mark K. Bakker[55], Traci M. Bartz[56,57], David A. Bennett[58], Joshua C. Bis[56], Constance Bordes[1], Sigrid Børte[20,59,60], Anael Cain[32], Paul M. Ridker[61,62], Kelly Cho[7], Zhengming Chen[18,63], Carlos Cruchaga[64,65], John W. Cole[66,67], Phil L. de Jager[13,68], Rafael de Cid[69], Matthias Endres[70,71,72,73], Leslie E. Ferreira[74], Mirjam I. Geerlings[75], Natalie C. Gasca[57], Vilmundur Gudnason[76,77], Jun Hata[78], Jing He[48], Alicia K. Heath[79], Yuk-Lam Ho[7], Aki S. Havulinna[80,81], Jemma C. Hopewell[82], Hyacinth I. Hyacinth[83], Michael Inouye[46,84,85,86,87], Mina A. Jacob[88], Christina E. Jeon[89], Christina Jern[90,91], Masahiro Kamouchi[92], Keith L. Keene[93], Takanari Kitazono[50], Steven J. Kittner[67,94], Takahiro Konuma[6], Amit Kumar[24], Paul Lacaze[95], Lenore J. Launer[96], Keon-Joo Lee[97], Kaido Lepik[4,98,99,100], Jiang Li[35], Liming Li[101], Ani Manichaikul[44], Hugh S. Markus[102], Nicholas A. Marston[8,9], Thomas Meitinger[103,104], Braxton D. Mitchell[105,106], Felipe A. Montellano[107,108], Takayuki Morisaki[10], Thomas H. Mosley[109], Mike A. Nalls[41,42,43], Børge G. Nordestgaard[110,111], Martin J. O'Donnell[112], Yukinori Okada[6,113,114,115,116,117], N. Charlotte Onland-Moret[75], Bruce Ovbiagele[118], Annette Peters[119,120,121], Bruce M. Psaty[56,122,123], Stephen S. Rich[44], Jonathan Rosand[12,13,124], Marc S. Sabatine[8,9], Ralph L. Sacco[125,126], Danish Saleheen[127], Else Charlotte Sandset[128,129], Veikko Salomaa[80], Muralidharan Sargurupremraj[130], Makoto Sasaki[3], Claudia L. Satizabal[130,131], Carsten O. Schmidt[132], Atsushi Shimizu[3], Nicholas L. Smith[122,133,134], Kelly L. Sloane[135], Yoichi Sutoh[3], Yan V. Sun[136,137], Kozo Tanno[3], Steffen Tiedt[2], Turgut Tatlisumak[138], Nuria P. Torres-Aguila[37], Hemant K. Tiwari[22], David-Alexandre Trégouët[1], Stella Trompet[139,140], Anil Man Tuladhar[88], Anne Tybjærg-Hansen[34,111], Marion van Vugt[141], Riina Vibo[142], Shefali S. Verma[143], Kerri L. Wiggins[56], Patrik Wennberg[144], Daniel Woo[83], Peter W. F. Wilson[136,145], Huichun Xu[105], Qiong Yang[131,146], Kyungheon Yoon[147], The COMPASS Consortium*, The INVENT Consortium*, The Dutch Parelsnoer Initiative (PSI) Cerebrovascular Disease Study Group*, The Estonian Biobank*, The PRECISE4Q Consortium*, The FinnGen Consortium*, The NINDS Stroke Genetics Network (SiGN)*, The MEGASTROKE Consortium*, The SIREN Consortium*, The China Kadoorie Biobank Collaborative Group*, The VA Million Veteran Program*, The International Stroke Genetics Consortium (ISGC)*, The Biobank Japan*, The CHARGE Consortium*, The GIGASTROKE Consortium*, Iona Y. Millwood[18,63], Christian Gieger[148], Toshiharu Ninomiya[78], Hans J. Grabe[149,150], J. Wouter Jukema[140,151,152], Ina L. Rissanen[75], Daniel Strbian[27], Young Jin Kim[147], Pei-Hsin Chen[15], Ernst Mayerhofer[12,13], Joanna M. M. Howson[46,47], Marguerite R. Irvin[54], Hieab Adams[153,154], Sylvia Wassertheil-Smoller[155], Kaare Christensen[39,40,156], Mohammad A. Ikram[30], Tatjana Rundek[125,126], Bradford B. Worrall[157,158], G. Mark Lathrop[159], Moeen Riaz[95], Eleanor M. Simonsick[160], Janika Kõrv[142], Paulo H. C. França[74], Ramin Zand[161,162], Kameshwar Prasad[24], Ruth Frikke-Schmidt[34,111], Frank-Erik de Leeuw[88], Thomas Liman[71,75,163], Karl Georg Haeusler[108], Ynte M. Ruigrok[55], Peter Ulrich Heuschmann[107,164,165], W. T. Longstreth[122,166], Keum Ji Jung[167], Lisa Bastarache[48], Guillaume Paré[25,26,168,169], Scott M. Damrauer[170,171], Daniel I. Chasman[61,62], Jerome I. Rotter[172], Christopher D. Anderson[12,13,124,173], John-Anker Zwart[19,20,59], Teemu J. Niiranen[16,17,174], Myriam Fornage[175,176], Yung-Po Liaw[15,177], Sudha Seshadri[130,131,178], Israel Fernández-Cadenas[37], Robin G. Walters[18,63], Christian T. Ruff[8,9,419], Mayowa O. Owolabi[28,179,419], Jennifer E. Huffman[7,419], Lili Milani[4,419], Yoichiro Kamatani[11,419], Martin Dichgans[2,180,181,419✉] & Stephanie Debette[1,182,419✉]

[1]Bordeaux Population Health Research Center, University of Bordeaux, Inserm, UMR 1219, Bordeaux, France. [2]Institute for Stroke and Dementia Research (ISD), University Hospital, LMU Munich, Munich, Germany. [3]Iwate Tohoku Medical Megabank Organization, Iwate Medical University, Iwate, Japan. [4]Estonian Genome Centre, Institute of Genomics, University of Tartu, Tartu, Estonia. [5]Institute of Mathematics and Statistics, University of Tartu, Tartu, Estonia. [6]Department of Statistical Genetics, Osaka University Graduate School of Medicine, Suita, Japan. [7]Massachusetts Veterans Epidemiology Research and Information Center (MAVERIC), VA Boston Healthcare System, Boston, MA, USA. [8]TIMI Study Group, Boston, MA, USA. [9]Division of Cardiovascular Medicine, Brigham and Women's Hospital, Harvard Medical School, Boston, MA, USA. [10]Division of Molecular Pathology, Institute of Medical Sciences, The University of Tokyo, Tokyo, Japan. [11]Laboratory of Complex Trait Genomics, Graduate School of Frontier Sciences, The University of Tokyo, Tokyo, Japan. [12]Center for Genomic Medicine, Massachusetts General Hospital, Boston, MA, USA. [13]Program in Medical and Population Genetics, Broad Institute of Harvard and the Massachusetts Institute of Technology, Cambridge, MA, USA. [14]Laboratory of Clinical Genome Sequencing, Department of Computational Biology and Medical Sciences, Graduate School of Frontier Sciences, The University of Tokyo, Tokyo, Japan. [15]Department of Public Health and Institute of Public Health, Chung Shan Medical University, Taichung, Taiwan. [16]Department of Internal Medicine, University of Turku, Turku, Finland. [17]Department of Public Health and Welfare, Finnish Institute for Health and Welfare, Turku, Finland. [18]Nuffield Department of Population Health, University of Oxford, Oxford, UK. [19]Department of Research and Innovation, Division of Clinical Neuroscience, Oslo University Hospital, Oslo, Norway. [20]K. G. Jebsen Center for Genetic Epidemiology, Department of Public Health and Nursing, Faculty of Medicine and Health Sciences, Norwegian University of Science and Technology (NTNU), Trondheim, Norway. [21]Department of Neurology, Oslo University Hospital, Oslo, Norway. [22]Department of Biostatistics, School of Public Health, University of Alabama at Birmingham, Birmingham, AL, USA. [23]Department of Neurology and Cerebrovascular Disease Center, Seoul National University Bundang Hospital, Seoul National University College of Medicine, Seongnam, Republic of Korea. [24]Rajendra Institute of Medical Sciences, Ranchi, India. [25]Thrombosis and Atherosclerosis Research Institute, David Braley Cardiac, Vascular and Stroke Research Institute, Hamilton, Ontario, Canada. [26]Department of Pathology and Molecular Medicine, Michael G. DeGroote School of Medicine, McMaster University, Hamilton, Ontario, Canada. [27]Department of Neurology, Helsinki University Hospital and University of Helsinki, Helsinki, Finland. [28]Center for Genomic and Precision Medicine, College of Medicine, University of Ibadan, Ibadan, Nigeria. [29]Neuroscience and Ageing Research Unit Institute for Advanced Medical Research and Training, College of Medicine, University of Ibadan, Ibadan, Nigeria. [30]Department of Epidemiology, Erasmus MC University Medical Center Rotterdam, Rotterdam, The Netherlands. [31]Department of Radiology and Nuclear Medicine, Erasmus MC University Medical Center Rotterdam, Rotterdam, The Netherlands. [32]The Edmond and Lily Safra Center for Brain Sciences, The Hebrew University of Jerusalem, Jerusalem, Israel. [33]Department of Epidemiology, Harvard T. H. Chan School of Public Health, Boston, MA, USA. [34]Department of Clinical Biochemistry, Copenhagen University Hospital—Rigshospitalet, Copenhagen, Denmark. [35]Department of Molecular and Functional Genomics, Weis Center for Research, Geisinger Health System, Danville, VA, USA. [36]Department of Public Health Sciences, College of Medicine, The Pennsylvania State University, State College, PA, USA. [37]Stroke Pharmacogenomics and Genetics Laboratory, Biomedical Research Institute Sant Pau (IIB Sant Pau), Barcelona, Spain. [38]Departament de Medicina, Universitat Autònoma de Barcelona, Barcelona, Spain. [39]The Danish Twin Registry, Department of Public Health, University of Southern Denmark, Odense, Denmark. [40]Department of Clinical Genetics, Odense University Hospital, Odense, Denmark. [41]Center for Alzheimer's and Related Dementias, National Institutes of Health, Bethesda, MD, USA. [42]Laboratory of Neurogenetics, National Institute on Aging, National Institutes of Health, Bethesda, MD, USA. [43]Data Tecnica International, Glen Echo, MD, USA. [44]Center for Public Health Genomics, University of Virginia, Charlottesville, VA, USA. [45]Department of Biochemistry and Molecular Genetics, University of Virginia, Charlottesville, VA, USA. [46]British Heart Foundation Cardiovascular Epidemiology Unit, Department of Public Health and Primary Care, University of Cambridge, Cambridge, UK. [47]Department of Genetics, Novo Nordisk Research Centre Oxford, Oxford, UK. [48]Department of Biomedical Informatics, Vanderbilt University Medical Center, Nashville, TN, USA. [49]Department of Surgery, University of Pennsylvania, Philadelphia, PA, USA. [50]Department of Medicine and Clinical Science, Graduate School of Medical Sciences, Kyushu University, Fukuoka, Japan. [51]University of Lille, INSERM U1167, RID-AGE, LabEx DISTALZ, Risk Factors and Molecular Determinants of Aging-Related Diseases, Lille, France. [52]CHU Lille, Public Health Department, Lille, France. [53]Institut Pasteur de Lille, Lille, France. [54]Department of Epidemiology, University of Alabama at Birmingham, Birmingham, AL, USA. [55]UMC Utrecht Brain Center, Department of Neurology and Neurosurgery, University Medical Center Utrecht, University Utrecht, Utrecht, The Netherlands. [56]Cardiovascular Health Research Unit, Department of Medicine, University of Washington, Seattle, WA, USA. [57]Department of Biostatistics, University of Washington, Seattle, WA, USA. [58]Rush Alzheimer's Disease Center, Rush University Medical Center, Chicago, IL, USA. [59]Institute of Clinical Medicine, Faculty of Medicine, University of Oslo, Oslo, Norway. [60]Research and Communication Unit for Musculoskeletal Health (FORMI), Department of Research and Innovation, Division of Clinical Neuroscience, Oslo University Hospital, Oslo, Norway. [61]Division of Preventive Medicine, Brigham and Women's Hospital, Boston, MA, USA. [62]Harvard Medical School, Boston, MA, USA. [63]MRC Population Health Research Unit, University of Oxford, Oxford, UK. [64]Department of Psychiatry, Washington University School of Medicine, Saint Louis, MO, USA. [65]NeuroGenomics and Informatics Center, Washington University School of Medicine, Saint Louis, MO, USA. [66]VA Maryland Health Care System, Baltimore, MD, USA. [67]Department of Neurology, University of Maryland School of Medicine, Baltimore, MD, USA. [68]Center for Translational and Computational Neuroimmunology, Department of Neurology, Columbia University Medical Center, New York, NY, USA. [69]GenomesForLife—GCAT Lab Group, Germans Trias i Pujol Research Institute (IGTP), Badalona, Spain. [70]Klinik und Hochschulambulanz für Neurologie, Charité—Universitätsmedizin Berlin, Berlin, Germany. [71]Center for Stroke Research Berlin, Berlin, Germany. [72]German Center for Neurodegenerative Diseases (DZNE), partner site Berlin, Berlin, Germany. [73] German Centre for Cardiovascular Research (DZHK), partner site Berlin, Berlin, Germany. [74]Post-Graduation Program on Health and Environment, Department of Medicine and Joinville Stroke Biobank, University of the Region of Joinville, Santa Catarina, Brazil. [75]Department of Epidemiology, Julius Center for Health Sciences and Primary Care, University Medical Center Utrecht, Utrecht University,

Utrecht, The Netherlands. [76]Icelandic Heart Association, Kopavogur, Iceland. [77]Faculty of Medicine, University of Iceland, Reykjavik, Iceland. [78]Department of Epidemiology and Public Health, Graduate School of Medical Sciences, Kyushu University, Fukuoka, Japan. [79]Department of Epidemiology and Biostatistics, School of Public Health, Imperial College London, London, UK. [80]Department of Public Health and Welfare, Finnish Institute for Health and Welfare, Helsinki, Finland. [81]Institute for Molecular Medicine Finland, FIMM-HiLIFE, Helsinki, Finland. [82]Clinical Trial Service and Epidemiological Studies Unit (CTSU), Nuffield Department of Population Health, University of Oxford, Oxford, UK. [83]Department of Neurology and Rehabilitation Medicine, University of Cincinnati College of Medicine, Cincinnati, OH, USA. [84]Cambridge Baker Systems Genomics Initiative, Department of Public Health and Primary Care, University of Cambridge, Cambridge, UK. [85]Cambridge Baker Systems Genomics Initiative, Baker Heart and Diabetes Institute, Melbourne, Victoria, Australia. [86]Health Data Research UK Cambridge, Wellcome Genome Campus and University of Cambridge, Cambridge, UK. [87]British Heart Foundation Centre of Research Excellence, University of Cambridge, Cambridge, UK. [88]Department of Neurology, Donders Center for Medical Neuroscience, Radboud University Medical Center, Nijmegen, The Netherlands. [89]Los Angeles County Department of Public Health, Los Angeles, CA, USA. [90]Institute of Biomedicine, Department of Laboratory Medicine, the Sahlgrenska Academy, University of Gothenburg, Gothenburg, Sweden. [91]Department of Clinical Genetics and Genomics, Sahlgrenska University Hospital, Gothenburg, Sweden. [92]Department of Health Care Administration and Management, Graduate School of Medical Sciences, Kyushu University, Fukuoka, Japan. [93]Department of Biology, Brody School of Medicine Center for Health Disparities, East Carolina University, Greenville, NC, USA. [94]Department of Neurology and Geriatric Research and Education Clinical Center, VA Maryland Health Care System, Baltimore, MD, USA. [95]Department of Epidemiology and Preventive Medicine, School of Public Health and Preventive Medicine, Monash University, Melbourne, Victoria, Australia. [96]Intramural Research Program, National Institute on Aging, NIH, Baltimore, MD, USA. [97]Department of Neurology, Korea University Guro Hospital, Seoul, Republic of Korea. [98]Department of Computational Biology, University of Lausanne, Lausanne, Switzerland. [99]Swiss Institute of Bioinformatics, Lausanne, Switzerland. [100]University Center for Primary Care and Public Health, Lausanne, Switzerland. [101]Department of Epidemiology and Biostatistics, School of Public Health, Peking University Health Science Center, Beijing, China. [102]Stroke Research Group, Department of Clinical Neurosciences, University of Cambridge, Cambridge, UK. [103]Institute of Human Genetics, Technical University of Munich, Munich, Germany. [104]Institute of Human Genetics, Helmholtz Zentrum München, German Research Center for Environmental Health, Neuherberg, Germany. [105]Department of Medicine, University of Maryland School of Medicine, Baltimore, MD, USA. [106]Geriatrics Research and Education Clinical Center, Baltimore Veterans Administration Medical Center, Baltimore, MD, USA. [107]Institute of Clinical Epidemiology and Biometry, University of Würzburg, Würzburg, Germany. [108]Department of Neurology, University Hospital Würzburg, Würzburg, Germany. [109]The MIND Center, University of Mississippi Medical Center, Jackson, MS, USA. [110]Department of Clinical Biochemistry, Copenhagen University Hospital—Herlev and Gentofte, Copenhagen, Denmark. [111]Department of Clinical Medicine, University of Copenhagen, Copenhagen, Denmark. [112]College of Medicine Nursing and Health Science, NUI Galway, Galway, Ireland. [113]Department of Genome Informatics, Graduate School of Medicine, The University of Tokyo, Tokyo, Japan. [114]Laboratory for Systems Genetics, RIKEN Center for Integrative Medical Sciences, Yokohama, Japan. [115]Laboratory of Statistical Immunology, Immunology Frontier Research Center (WPI-IFReC), Osaka University, Suita, Japan. [116]Integrated Frontier Research for Medical Science Division, Institute for Open and Transdisciplinary Research Initiatives, Osaka University, Suita, Japan. [117]Center for Infectious Disease Education and Research (CiDER), Osaka University, Suita, Japan. [118]Weill Institute for Neurosciences, University of California, San Francisco, San Francisco, CA, USA. [119]Institute of Epidemiology, Helmholtz Zentrum München,, German Research Center for Environmental Health, Neuherberg, Germany. [120]Institute for Medical Information Processing, Biometry and Epidemiology, Ludwig Maximilian University Munich, Munich, Germany. [121]German Centre for Cardiovascular Research (DZHK), partner site Munich, Munich, Germany. [122]Department of Epidemiology, University of Washington, Seattle, WA, USA. [123]Department of Health Systems and Population Health, University of Washington, Seattle, WA, USA. [124]McCance Center for Brain Health, Massachusetts General Hospital, Boston, MA, USA. [125]Department of Neurology, University of Miami Miller School of Medicine, Miami, FL, USA. [126]Evelyn F. McKnight Brain Institute, Gainesville, FL, USA. [127]Division of Cardiology, Department of Medicine, Columbia University, New York, NY, USA. [128]Stroke Unit, Department of Neurology, Oslo University Hospital, Oslo, Norway. [129]Research and Development, The Norwegian Air Ambulance Foundation, Oslo, Norway. [130]Glenn Biggs Institute for Alzheimer's and Neurodegenerative Diseases, University of Texas Health Sciences Center, San Antonio, TX, USA. [131]Framingham Heart Study, Framingham, MA, USA. [132]University Medicine Greifswald, Institute for Community Medicine, SHIP/KEF, Greifswald, Germany. [133]Kaiser Permanente Washington Health Research Institute, Kaiser Permanente Washington, Seattle, WA, USA. [134]Department of Veterans Affairs Office of Research and Development, Seattle Epidemiologic Research and Information Center, Seattle, WA, USA. [135]Department of Neurology, University of Pennsylvania, Philadelphia, PA, USA. [136]Atlanta VA Health Care System, Decatur, GA, USA. [137]Department of Epidemiology, Emory University Rollins School of Public Health, Atlanta, GA, USA. [138]Department of Clinical Neuroscience, Institute of Neuroscience and Physiology, Sahlgrenska Unviersity Hospital, Gothenburg, Sweden. [139]Department of Internal Medicine, Section of Gerontology and Geriatrics, Leiden University Medical Center, Leiden, The Netherlands. [140]Department of Cardiology, Leiden University Medical Center, Leiden, The Netherlands. [141]Division Heart & Lungs, Department of Cardiology, University Medical Center Utrecht, Utrecht University, Utrecht, The Netherlands. [142]Department of Neurology and Neurosurgery, University of Tartu, Tartu, Estonia. [143]Department of Pathology and Laboratory Medicine, University of Pennsylvania, Philadelphia, PA, USA. [144]Department of Public Health and Clinical Medicine, Umeå University, Umeå, Sweden. [145]Department of Medicine, Division of Cardiovascular Disease, Emory University School of Medicine, Atlanta, GA, USA. [146]Department of Biostatistics, Boston University School of Public Health, Boston, MA, USA. [147]Division of Genome Science, Department of Precision Medicine, National Institute of Health, Cheongju, Republic of Korea. [148]Research Unit Molecular Epidemiology, Institute of Epidemiology, Helmholtz Zentrum München, German Research Center for Environmental Health, Neuherberg, Germany. [149]Department of Psychiatry and Psychotherapy, University Medicine Greifswald, Greifswald, Germany. [150]German Center for Neurodegenerative Diseases (DZNE), site Rostock/Greifswald, Rostock, Germany. [151]Netherlands Heart Institute, Utrecht, The Netherlands. [152]Einthoven Laboratory for Experimental Vascular Medicine, LUMC, Leiden, The Netherlands. [153]Department of Clinical Genetics, Department of Radiology and Nuclear Medicine, Erasmus MC, Rotterdam, The Netherlands. [154]Latin American Brain Health (BrainLat), Universidad Adolfo Ibáñez, Santiago, Chile. [155]Department of Epidemiology and Population Health, Albert Einstein College of Medicine, New York, NY, USA. [156]Department of Clinical Biochemistry and Pharmacology, Odense University Hospital, Odense, Denmark. [157]Department of Neurology, University of Virginia, Charlottesville, VA, USA. [158]Department of Public Health Science, University of Virginia, Charlottesville, VA, USA. [159]McGill Genome Centre, Montreal, Quebec, Canada. [160]Longitudinal Studies Section, Translational Gerontology Branch, National Institute on Aging, Baltimore, MD, USA. [161]Geisinger Neuroscience Institute, Geisinger Health System, Danville, PA, USA. [162]Department of Neurology, College of Medicine, The Pennsylvania State University, State College, PA, USA. [163]Klinik für Neurologie, Carl von Ossietzky University of Oldenburg, Oldenburg, Germany. [164]Comprehensive Heart Failure Center, University Hospital Würzburg, Würzburg, Germany. [165]Clinical Trial Center, University Hospital Würzburg, Würzburg, Germany. [166]Department of Neurology, University of Washington, Seattle, WA, USA. [167]Institute for Health Promotion, Graduate School of Public Health, Yonsei University, Seoul, Republic of Korea. [168]Department of Health Research Methods, Evidence and Impact, McMaster University, Hamilton, Ontario, Canada. [169]Population Health Research Institute, David Braley Cardiac, Vascular and Stroke Research Institute, Hamilton, Ontario, Canada. [170]Department of Surgery and Department of Genetics, University of Pennsylvania, Philadelphia, PA, USA. [171]Corporal Michael Crescenz VA Medical Center, Philadelphia, PA, USA. [172]The Institute for Translational Genomics and Population Sciences, Department of Pediatrics, The Lundquist Institute for Biomedical Innovation at Harbor-UCLA Medical Center, Torrance, CA, USA. [173]Department of Neurology, Brigham and Women's Hospital, Boston, MA, USA. [174]Division of Medicine, Turku University Hospital, Turku, Finland. [175]Brown Foundation Institute of Molecular Medicine, McGovern Medical School, University of Texas Health Science Center at Houston, Houston, TX, USA. [176]Human Genetics Center, School of Public Health, University of Texas Health Science Center at Houston, Houston, TX, USA. [177]Department of Medical Imaging, Chung Shan Medical University Hospital, Taichung, Taiwan. [178]Department of Neurology, Boston University School of Medicine, Boston, MA, USA. [179]Department of Medicine, University of Ibadan, Ibadan, Nigeria. [180]Munich Cluster for Systems Neurology, Munich, Germany. [181]German Center for Neurodegenerative Diseases (DZNE), Munich, Germany. [182]Department of Neurology, Institute for Neurodegenerative Diseases, CHU de Bordeaux, Bordeaux, France. [418]These authors contributed equally: Aniket Mishra, Rainer Malik, Tsuyoshi Hachiya, Tuuli Jürgenson, Shinichi Namba, Daniel C. Posner. [419]These authors jointly supervised this work: Christian T. Ruff, Mayowa O. Owolabi, Jennifer E. Huffman, Lili Milani, Yoichiro Kamatani, Martin Dichgans, Stephanie Debette. *A list of authors and their affiliations appear online.

**The COMPASS Consortium**

Keith L. Keene[93], Hyacinth I. Hyacinth[83], Joshua C. Bis[56], Steven J. Kittner[67,94], Braxton D. Mitchell[105,106], Guillaume Paré[25,26,168,169], Michael R. Chong[25,26], Stephen S. Rich[44], Mike A. Nalls[41,42,43], Hugh S. Markus[102], Thomas H. Mosley[109], Daniel Woo[83], W. T. Longstreth[122,166], Bruce M. Psaty[56,122,123], Ralph L. Sacco[125,126], Tatjana Rundek[125,126], Jin-Moo Lee[183], Carlos Cruchaga[64,65], Martin Dichgans[2,180,181,419], Rainer Malik[2,418], Bradford B. Worrall[157,158], Myriam Fornage[175,176], Yu-Ching Cheng[184], Martin J. O'Donnell[112], James F. Meschia[185], Wei Min Chen[44], Michèle M. Sale[44], Alan B. Zonderman[186], Michele K. Evans[186], James G. Wilson[187], Adolfo Correa[187], Matthew Traylor[188], Cathryn M. Lewis[189], Cara L. Carty[190], Alexander Reiner[122,191], Jeffrey Haessler[191], Carl D. Langefeld[192], Rebecca F. Gottesman[193,194,195], Kristine Yaffe[196], Yong Mei Liu[192], Charles Kooperberg[191], Leslie A. Lange[197], Karen L. Furie[198], Donna K. Arnett[199], Oscar R. Benavente[200], Raji P. Grewal[201] & Leema Reddy Peddareddygari[201]

[183]Department of Neurology, Washington University School of Medicine, Saint Louis, MO, USA. [184]Baltimore Veterans Administration Medical Center and University of Maryland School of Medicine, Baltimore, MD, USA. [185]Mayo Clinic Florida, Jacksonville, FL, USA. [186]Laboratory of Epidemiology and Population Science, National Institute on Aging, National Institutes of Health, Baltimore, MD, USA. [187]University of Mississippi Medical Center, Jackson, MS, USA. [188]William Harvey Research Institute, Barts and The London School of Medicine and Dentistry, Queen Mary University of London, London, UK. [189]Social, Genetic and Developmental Psychiatry Centre, King's College London, London, UK. [190]Initiative for Research and Education to Advance Community Health, Washington State University, Seattle, WA, USA. [191]Division of Public Health Sciences, Fred Hutchinson Cancer Research Center, Seattle, WA, USA. [192]Division of Public Health Sciences, Wake Forest School of Medicine, Winston-Salem, NC, USA. [193]Johns Hopkins University School of Medicine, Baltimore, MD, USA. [194]Department of Neurology, Johns Hopkins University School of Medicine, Baltimore, MD, USA. [195]Stroke Branch, National Institute of Neurological Disorders and Stroke, Bethesda, MD, USA. [196]University of California, San Francisco, San Francisco, CA, USA. [197]University of Colorado Anschutz Medical Campus, Denver, CO, USA. [198]Brown University Warren Alpert Medical School, Providence, RI, USA. [199]College of Public Health, University of Kentucky, Lexington, KY, USA. [200]University of British Columbia, Vancouver, British Columbia, Canada. [201]Neuroscience Institute, Saint Francis Medical Center, Trenton, NJ, USA.

**The INVENT Consortium**

Daniel I. Chasman[61,62], Kerri L. Wiggins[56], Jerome I. Rotter[172], Bruce M. Psaty[56,122,123], Charles Kooperberg[191], Kristian Hveem[20,202,203], Paul M. Ridker[61,62], David-Alexandre Trégouët[1], Nicholas L. Smith[122,133,134], Sara Lindstrom[122,191], Lu Wang[204], Erin N. Smith[205,206], William Gordon[122], Astrid van Hylckama Vlieg[207], Mariza de Andrade[208], Jennifer A. Brody[56], Jack W. Pattee[209], Jeffrey Haessler[191], Ben M. Brumpton[20,210,211], Pierre Suchon[212,213], Ming-Huei Chen[131], Kelly A. Frazer[205,206,214], Constance Turman[215], Marine Germain[1], James MacDonald[204], Sigrid K. Braekkan[206,216], Sebastian M. Armasu[208], Nathan Pankratz[217], Rebecca D. Jackson[218,219], Jonas B. Nielsen[20,220,221], Franco Giulianini[61], Marja K. Puurunen[131], Manal Ibrahim[212], Susan R. Heckbert[122,133], Theo K. Bammler[204], Bryan M. McCauley[208], Kent D. Taylor[172,222,223], James S. Pankow[224], Alexander P. Reiner[122,191], Maiken E. Gabrielsen[20], Jean-François Deleuze[225,226], Chris J. O'Donnell[131,227], Jihye Kim[215], Barbara McKnight[57,191], Peter Kraft[215], John-Bjarne Hansen[206,216], Frits R. Rosendaal[207], John A. Heit[208], Weihong Tang[224], Pierre-Emmanuel Morange[212,213,228], Andrew D. Johnson[131] & Christopher Kabrhel[229,230]

[202]HUNT Research Center, Department of Public Health and Nursing, Faculty of Medicine and Health Sciences, Norwegian University of Science and Technology (NTNU), Trondheim, Norway. [203]Department of Research, Innovation and Education, St Olavs Hospital, Trondheim University Hospital, Trondheim, Norway. [204]Department of Environmental and Occupational Health Sciences, University of Washington, Seattle, WA, USA. [205]Department of Pediatrics and Rady Children's Hospital, University of California San Diego, La Jolla, CA, USA. [206]K. G. Jebsen Thrombosis Research and Expertise Center, Department of Clinical Medicine, UiT—The Arctic University of Norway, Tromsø, Norway. [207]Department of Clinical Epidemiology, Leiden University Medical Center, Leiden, The Netherlands. [208]Department of Health Sciences Research, Mayo Clinic, Rochester, MN, USA. [209]Division of Biostatistics, School of Public Health, University of Minnesota, Minneapolis, MN, USA. [210]MRC Integrative Epidemiology Unit, University of Bristol, Bristol, UK. [211]Clinic of Thoracic and Occupational Medicine, St Olavs Hospital, Trondheim University Hospital, Trondheim, Norway. [212]Laboratory of Haematology, La Timone Hospital, Marseille, France. [213]C2VN, University of Aix Marseille, INSERM, INRAE, C2VN, Marseille, France. [214]Institute of Genomic Medicine, University of California, San Diego, La Jolla, CA, USA. [215]Program in Genetic Epidemiology and Statistical Genetics, Harvard T. H. Chan School of Public Health, Boston, MA, USA. [216]Division of Internal Medicine, University Hospital of North Norway, Tromsø, Norway. [217]Department of Laboratory Medicine and Pathology, School of Medicine, University of Minnesota, Minneapolis, MN, USA. [218]Division of Endocrinology, Diabetes and Metabolism, The Ohio State University, Columbus, OH, USA. [219]Department of Internal Medicine and the Center for Clinical and Translational Science, Ohio State University, Columbus, OH, USA. [220]Department of Internal Medicine, Division of Cardiology, University of Michigan, Ann Arbor, MI, USA. [221]Department of Epidemiology Research, Statens Serum Institut, Copenhagen, Denmark. [222]Institute for Translational Genomics and Population Sciences, Los Angeles Biomedical Research Institute at Harbor–UCLA Medical Center, Torrance, CA, USA. [223]Division of Genomic Outcomes, Department of Pediatrics, Harbor–UCLA Medical Center, Torrance, CA, USA. [224]Division of Epidemiology and Community Health, School of Public Health, University of Minnesota, Minneapolis, MN, USA. [225]Centre National de Recherche en Génomique Humaine, Direction de la Recherche Fondamentale, CEA, Evry, France. [226]CEPH-Fondation Jean Dausset, Paris, France. [227]Million Veteran's Program, Veteran's Administration, Boston, MA, USA. [228]CRB Assistance Publique—Hôpitaux de Marseille, HemoVasc (CRB AP-HM HemoVasc), Marseille, France. [229]Center for Vascular Emergencies, Department of Emergency Medicine, Massachusetts General Hospital, Boston, MA, USA. [230]Department of Emergency Medicine, Harvard Medical School, Boston, MA, USA.

**The Dutch Parelsnoer Initiative (PSI) Cerebrovascular Disease Study Group**

Mark K. Bakker[55], Ynte M. Ruigrok[55], Ewoud J. van Dijk[88], Peter J. Koudstaal[231], Gert-Jan Luijckx[232], Paul J. Nederkoorn[233], Robert J. van Oostenbrugge[234], Marieke C. Visser[233], Marieke J. H. Wermer[235] & L. Jaap Kappelle[55]

[231]Department of Neurology, Erasmus University Medical Center, Rotterdam, The Netherlands. [232]Department of Neurology, University Medical Center Groningen, Groningen, The Netherlands. [233]Department of Neurology, Amsterdam UMC, University of Amsterdam, Amsterdam, The Netherlands. [234]Department of Neurology, Cardiovascular Research Institute Maastricht, Maastricht University Medical Center, Maastricht, The Netherlands. [235]Department of Neurology, Leiden University Medical Center, Leiden, The Netherlands.

**The Estonian Biobank**

Tuuli Jürgenson[4,5,418], Kristi Krebs[4], Kaido Lepik[4,98,99,100], Lili Milani[4,419], Tõnu Esko[4], Andres Metspalu[4], Reedik Mägi[4] & Mari Nelis[4]

**The PRECISE4Q Consortium**

Lili Milani[4,419]

**The FinnGen Consortium**

Felix C. Vaura[16,17], Teemu J Niiranen[16,17,174], Aki S. Havulinna[80,81]

**The NINDS Stroke Genetics Network (SiGN)**

Rainer Malik[2,418], Michael R. Chong[25,26], Liisa Tomppo[27], Vida Abedi[35,36], Carlos Cruchaga[64,65], John W. Cole[66,67], Christina Jern[90,91], Steven J. Kittner[67,94], Hugh S. Markus[102], Braxton D. Mitchell[105,106], Jonathan Rosand[12,13,124], Ralph L. Sacco[125,126], Danish Saleheen[127], Turgut Tatlisumak[138], Daniel Woo[83], Huichun Xu[105], Christian Gieger[148], Daniel Strbian[27], Marguerite R. Irvin[54], Sylvia Wassertheil-Smoller[155], Tatjana Rundek[125,126], Bradford B. Worrall[157,158], Ramin Zand[161,162], Frank-Erik de Leeuw[88], Guillaume Paré[25,26,168,169], Israel Fernández-Cadenas[37], Yoichiro Kamatani[11,419], Martin Dichgans[2,180,181,419], Stephanie Debette[1,182,419], Christopher R. Levi[236], Jane Maguire[237], Jordi Jiménez-Conde[238], Pankaj Sharma[239,240], Cathie L. M. Sudlow[241,242], Kristiina Rannikmäe[241,242], Reinhold Schmidt[243], James F. Meschia[185], Agnieszka Slowik[244], Joanna Pera[244], Vincent N. S. Thijs[245,246], Arne G. Lindgren[247,248], Andreea Ilinca[247,248], Olle Melander[249], Gunnar Engström[249], Raji P. Grewal[201], Leema Reddy Peddareddygari[201], Kathryn M. Rexrode[250], Peter M. Rothwell[251], Leslie A. Lange[197], Tara M. Stanne[90], Julie A. Johnson[252,253], John Danesh[254,255,256,257], Adam S. Butterworth[254,258], Jin-Moo Lee[183], Laura Heitsch[183,259], Giorgio B. Boncoraglio[260], Michiaki Kubo[261], Alessandro Pezzini[262], Arndt Rolfs[263], Anne-Katrin Giese[264], David Weir[265], Rebecca D. Jackson[218,219], Owen A. Ross[185], Robin Lemmons[266,267], Martin Soderholm[248,249], Mary Cushman[268], Katarina Jood[269,270], Caitrin W. McDonough[252], Steven Bell[102], Birgit Linkohr[119], Tsong-Hai Lee[271] & Jukka Putaala[27]

[236]John Hunter Hospital, Hunter Medical Research Institute and University of Newcastle, Newcastle, New South Wales, Australia. [237]Faculty of Health, University of Technology Sydney, Ultimo, New South Wales, Australia. [238]Department of Neurology, IMIM-Hospital del Mar, Neurovascular Research Group, IMIM (Institut Hospital del Mar d'Investigacions Mèdiques), Universitat Autònoma de Barcelona/DCEXS-Universitat Pompeu Fabra, Barcelona, Spain. [239]Institute of Cardiovascular Research, Royal Holloway University of London, London, UK. [240]Ashford and St Peters Hospital, Chertsey, UK. [241]Usher Institute of Population Health Sciences and Informatics, University of Edinburgh, Edinburgh, UK. [242]Centre for Clinical Brain Sciences, University of Edinburgh, Edinburgh, UK. [243]Department of Neurology, Medical University of Graz, Graz, Austria. [244]Department of Neurology, Jagiellonian University, Krakow, Poland. [245]Stroke Division, Florey Institute of Neuroscience and Mental Health, University of Melbourne, Heidelberg, Melbourne, Victoria, Australia. [246]Department of Neurology, Austin Health, Heidelberg, Victoria, Australia. [247]Department of Clinical Sciences Lund, Neurology, Lund University, Lund, Sweden. [248]Department of Neurology, Rehabilitation Medicine, Memory Disorders, Geriatrics, Skåne University Hospital, Lund, Sweden. [249]Department of Clinical Sciences, Lund University, Malmö, Sweden. [250]Department of Medicine, Brigham and Women's Hospital, Boston, MA, USA. [251]Wolfson Centre for Prevention of Stroke and Dementia, Nuffield Department of Clinical Neurosciences, University of Oxford, Oxford, UK. [252]Department of Pharmacotherapy and Translational Research and Center for Pharmacogenomics, College of Pharmacy, University of Florida, Gainesville, FL, USA. [253]Division of Cardiovascular Medicine, College of Medicine, University of Florida, Gainesville, FL, USA. [254]BHF Cardiovascular Epidemiology Unit, Department of Public Health and Primary Care, University of Cambridge, Cambridge, UK. [255]NIHR Blood and Transplant Research Unit in Donor Health and Genomics, Department of Public Health and Primary Care, University of Cambridge, Cambridge, UK. [256]Wellcome Trust Sanger Institute, Cambridge, UK. [257]British Heart Foundation Cambridge Centre of Excellence, Department of Medicine, University of Cambridge, Cambridge, UK.

[258]National Institute for Health Research Blood and Transplant Research Unit in Donor Health and Genomics, University of Cambridge, Cambridge, UK. [259]Department of Emergency Medicine, Washington University School of Medicine, Saint Louis, MO, USA. [260]Department of Cerebrovascular Diseases, Fondazione IRCCS Istituto Neurologico 'Carlo Besta', Milan, Italy. [261]Laboratory for Statistical Analysis, RIKEN Center for Integrative Medical Sciences, Yokohama, Japan. [262]Department of Clinical and Experimental Sciences, Neurology Clinic, University of Brescia, Brescia, Italy. [263]Albrecht Kossel Institute, University Clinic of Rostock, Rostock, Germany. [264]Department of Neurology, Massachusetts General Hospital (MGH), Harvard Medical School, Boston, MA, USA. [265]Survey Research Center, University of Michigan, Ann Arbor, MI, USA. [266]Department of Neurosciences, Experimental Neurology, KU Leuven–University of Leuven, Leuven, Belgium. [267]Department of Neurology, VIB Center for Brain & Disease Research, University Hospitals Leuven, Leuven, Belgium. [268]Department of Medicine, Larner College of Medicine at the University of Vermont, Burlington, VT, USA. [269]Department of Clinical Neuroscience, Institute of Neuroscience and Physiology, Sahlgrenska Academy at University of Gothenburg, Gothenburg, Sweden. [270]Region Västra Götaland, Department of Neurology, Sahlgrenska University Hospital, Gothenburg, Sweden. [271]Chang Gung Memorial Hospital, Linkou Medical Center in Taiwan, Taoyuan City, Taiwan.

**The MEGASTROKE Consortium**

Aniket Mishra[1,418], Rainer Malik[2,418], Ganesh Chauhan[24], Michael R. Chong[25,26], Liisa Tomppo[27], Tetsuro Ago[50], Philippe Amouyel[51,52,53], Traci M. Bartz[56,57], Paul M. Ridker[61,62], Carlos Cruchaga[64,65], John W. Cole[66,67], Vilmundur Gudnason[76,77], Jun Hata[78], Aki S. Havulinna[80,81], Jemma C. Hopewell[82], Hyacint I. Hyacint[83], Christina Jern[90,91], Masahiro Kamouchi[92], Keith L. Keene[93], Takanari Kitazono[50], Steven J. Kittner[67,94], Ani Manichaikul[44], Hugh S. Markus[102], Braxton D. Mitchell[105,106], Thomas H. Mosley[109], Mike A. Nalls[41,42,43], Martin J. O'Donnell[112], Yukinori Okada[6,113,114,115,116,117], Bruce M. Psaty[56,122,123], Stephen S. Rich[44], Jonathan Rosand[12,13,124], Ralph L. Sacco[125,126], Danish Saleheen[127], Veikko Salomaa[80], Muralidharan Sarguruprempraj[130], Makoto Sasaki[3], Claudia L. Satizabal[130,131], Carsten O. Schmidt[132], Kozo Tanno[3], Steffen Tiedt[2], Turgut Tatlisumak[138], Nuria P. Torres-Aguila[64], Stella Trompet[139,140], Kerri L. Wiggins[56], Daniel Woo[83], Huichun Xu[105], Qiong Yang[131,146], Toshiharu Ninomiya[78], J. Wouter Jukema[140,151,152], Daniel Strbian[27], Joanna M. M. Howson[46,47], Marguerite R. Irvin[54], Hieab Adams[153,154], Sylvia Wassertheil-Smoller[155], Mohammad A. Ikram[30], Tatjana Rundek[125,126], Bradford B. Worrall[157,158], W. T. Longstreth[122,166], Guillaume Paré[25,26,168,169], Daniel I. Chasman[61,62], Jerome I. Rotter[172], Christopher D. Anderson[12,13,124,173], Myriam Fornage[175,176], Sudha Seshadri[130,131,178], Israel Fernández-Cadenas[37], Yoichiro Kamatani[11,419], Martin Dichgans[2,180,181,419], Stephanie Debette[1,182,419], Matthew Traylor[188], Oscar L. Lopez[272], Cara L. Carty[190], Adolfo Correa[187], Raji P. Grewal[201], Jeffrey Haessler[191], Susan R. Heckbert[122,133], Xueqiu Jian[130,131], Jordi Jiménez-Conde[38], Charles Kooperberg[191], Leslie A. Lange[197], Carl D. Langefeld[192], Jin-Moo Lee[183], Cathryn M. Lewis[189], James F. Meschia[185], Alexander Reiner[122,191], Ulf Schminke[273], James G. Wilson[187], Donna K. Arnett[199], Yu-Ching Cheng[184], Natalia Cullell[37,274], Pilar Delgado[275], Laura Ibañez[64], Jerzy Krupinski[276], Vasileios Lioutas[131,277], Koichi Matsuda[14], Joan Montaner[278], Elena Muiño[37], Leema Reddy Peddareddygari[201], Jaume Roquer[238], Chloe Sarnowski[279], Naveed Sattar[280], Gerli Sibolt[27], Alexander Teumer[132,281], Loes Rutten-Jacobs[102], Masahiro Kanai[261,282,283], Anne-Katrin Giese[264], Solveig Gretarsdottir[284], Natalia S. Rost[264,285], Salim Yusuf[286], Peter Almgren[249], Hakan Ay[285,287], Steve Bevan[188], Giorgio B. Boncoraglio[260], Robert D. Brown Jr[289], Adam S. Butterworth[254,258], Caty Carrera[275,290], Julie E. Buring[61,62], Wei-Min Chen[44], Ioana Cotlarciuc[239,240], John Danesh[254,255,256,257], Paul I. W. de Bakker[75,291], Anita L. DeStefano[131,146,292], Marcel den Hoed[293], Qing Duan[294], Stefan T. Engelter[295,296], Guido J. Falcone[13,297], Rebecca F. Gottesman[195], Stefan Gustafsson[298], Ahamad Hassan[299], Elizabeth G. Holliday[300,301], George Howard[22], Fang-Chi Hsu[302], Erik Ingelsson[298,303], Tamara B. Harris[186], Julie A. Johnson[252,253], Brett M. Kissela[83], Dawn O. Kleindorfer[83], Michiaki Kubo[261], Claudia Langenberg[304,305,306], Robin Lemmens[266,267], Didier Leys[307], Wei-Yu Lin[254,308], Arne G. Lindgren[247,248], Erik Lorentzen[309], Patrik K. Magnusson[310], Jane Maguire[284], Patrick F. McArdle[105], Sara L. Pulit[55,291], Kristiina Rannikmäe[241,242], Kathryn M. Rexrode[250], Kenneth Rice[57], Peter M. Rothwell[251], Saori Sakaue[283,311], Bishwa R. Sapkota[312], Reinhold Schmidt[243], Pankaj Sharma[239,240], Agnieszka Slowik[244], Cathie L. M. Sudlow[241,242], Christian Tanislav[313], Vincent N. S. Thijs[245,246], Gudmar Thorleifsson[284], Unnur Thorsteinsdottir[284], Christophe Tzourio[1,314], Cornelia M. van Duijn[18,30], Matthew Walters[315], Nicholas J. Wareham[266], Najaf Amin[30], Hugo J. Aparicio[131,178], John Attia[316], Alexa S. Beiser[131,146], Claudine Berr[317], Mariana Bustamante[318], Valeria Caso[319], Seung Hoan Choi[48,320], Ayesha Chowhan[131,178], Jean-François Dartigues[1,321], Hossein Delavaran[247,248], Marcus Dörr[281,322], Gunnar Engström[249], Ian Ford[323], Wander S. Gurpreet[324], Anders Hamsten[325], Laura Heitsch[183,259], Atsushi Hozawa[326], Andreea Ilinca[247,248], Martin Ingelsson[327,328,329], Motoki Iwasaki[330], Rebecca D. Jackson[218,219], Katarina Jood[269,270], Sara Kaffashian[1], Lalit Kalra[331], Olafur Kjartansson[332], Manja Kloss[333], Peter J. Koudstaal[231], Daniel L. Labovitz[334], Cathy C. Laurie[57], Christopher R. Levi[236], Linxin Li[335], Lars Lind[335], Cecilia M. Lindgren[336,337], Yong Mei Liu[192], Oscar L. Lopez[272], Hirata Makoto[338], Naoko Minegishi[326], Andrew P. Morris[337,339], Martina Müller-Nurasy id[121,340,341], Bo Norrving[247], Soichi Ogishima[219], Eugenio A. Parati[260], Nancy L. Pedersen[310], Joanna Pera[244], Markus Perola[80], Pekka Jousilahti[80], Alessandro Pezzini[262], Silvana Pileggi[342], Raquel Rabionet[343], Iolanda Riba-Llena[275], Marta Ribasés[344], Jose R. Romero[131,178], Anthony G. Rudd[345,346], Antti-Pekka Sarin[81,130], Ralhan Sarju[324], Mamoru Satoh[284], Norie Sawada[330], Ásgeir Sigurdsson[284], Albert Smith[347], Tara M. Stanne[90], O. Colin Stine[348], David J. Stott[349], Konstantin Strauch[340,350], Takako Takai[326], Hideo Tanaka[351,352], Emmanuel Touze[353,354], Shoichiro Tsugane[330], Andre G. Uitterlinden[259], Einar M. Valdimarsson[356], Sven J. van der Lee[30], Kenji Wakai[351], David Weir[265], Stephen R. Williams[156], Charles D. A. Wolfe[345,346], Quenna Wong[57], Taiki Yamaji[330], Dharambir K. Sanghera[312,357,358], Olle Melander[249], Arndt Rolfs[263], Kari Stefansson[77,284], Kent D. Taylor[172,222,223], Nicolas Martinez-Majander[27], Kenji Sobue[3], Carolina Soriano-Tárraga[359] & Henry Völzke[132]

[272]Department of Neurology, School of Medicine, University of Pittsburgh, Pittsburgh, PA, USA. [273]Department of Neurology, University Medicine Greifswald, Greifswald, Germany.

[274]Stroke Pharmacogenomics and Genetics Laboratory, Fundación Docència I Recerca Mútua Terrassa, Hospital Mútua Terrassa, Terrassa, Spain. [275]Neurovascular Research Laboratory, Vall d'Hebron Institute of Research, Universitat Autònoma de Barcelona, Barcelona, Spain. [276]Neurology Service, Hospital Universitari Mútua Terrassa, Terrassa, Spain. [277]Department of Neurology, Beth Israel Deaconess Medical Center, Boston, MA, USA. [278]Institute de Biomedicine of Seville, IBiS/Hospital Universitario Virgen del Rocío/CSIC/University of Seville and Department of Neurology, Hospital Universitario Virgen Macarena, Seville, Spain. [279]Department of Epidemiology, Human Genetics, and Environmental Sciences (EHGES), UTHealth Science Center, School of Public Health, Houston, TX, USA. [280]BHF Glasgow Cardiovascular Research Centre, Faculty of Medicine, University of Glasgow, Glasgow, UK. [281]German Centre for Cardiovascular Research (DZHK), partner site Greifswald, Greifswald, Germany. [282]Program in Bioinformatics and Integrative Genomics, Harvard Medical School, Boston, MA, USA. [283]Department of Statistical Genetics, Osaka University Graduate School of Medicine, Osaka, Japan. [284]deCODE genetics/AMGEN, Reykjavik, Iceland. [285]J. Philip Kistler Stroke Research Center, Department of Neurology, MGH, Boston, MA, USA. [286]Population Health Research Institute, McMaster University, Hamilton, Ontario, Canada. [287]AA Martinos Center for Biomedical Imaging, Department of Radiology, MGH, Harvard Medical School, Boston, MA, USA. [288]School of Life Science, University of Lincoln, Lincoln, UK. [289]Department of Neurology, Mayo Clinic, Rochester, MN, USA. [290]Stroke Pharmacogenomics and Genetics, Fundacio Docència i Recerca MutuaTerrassa, Terrassa, Spain. [291]Department of Medical Genetics, University Medical Center Utrecht, Utrecht, The Netherlands. [292]Boston University School of Public Health, Boston, MA, USA. [293]The Beijer Laboratory and Department of Immunology, Genetics and Pathology, Uppsala University and Science for Life Laboratory, Uppsala, Sweden. [294]Department of Genetics, University of North Carolina, Chapel Hill, NC, USA. [295]Department of Neurology and Stroke Center, Basel University Hospital, Basel, Switzerland. [296]Neurorehabilitation Unit, University of Basel and University Center for Medicine of Aging and Rehabilitation Basel, Felix Platter Hospital, Basel, Switzerland. [297]Department of Neurology, Yale University School of Medicine, New Haven, CT, USA. [298]Department of Medical Sciences, Molecular Epidemiology and Science for Life Laboratory, Uppsala University, Uppsala, Sweden. [299]Department of Neurology, Leeds General Infirmary, Leeds Teaching Hospitals NHS Trust, Leeds, UK. [300]Public Health Stream, Hunter Medical Research Institute, New Lambton, New South Wales, Australia. [301]College of Health, Medicine and Wellbeing, The University of Newcastle, Newcastle, New South Wales, Australia. [302]Department of Biostatistics and Data Science, Division of Public Health Sciences, Wake Forest University School of Medicine, Winston-Salem, NC, USA. [303]Department of Medicine, Division of Cardiovascular Medicine, Stanford University School of Medicine, Stanford, CA, USA. [304]MRC Epidemiology Unit, School of Clinical Medicine, Institute of Metabolic Science, University of Cambridge, Cambridge, UK. [305]Computational Medicine, Berlin Institute of Health at Charité – Universitätsmedizin Berlin, Berlin, Germany. [306]Precision Healthcare Institute, Queen Mary University of London, London, UK. [307]INSERM U 1172, CHU Lille, Université Lille, Lille, France. [308]MRC Biostatistics Unit, University of Cambridge, Cambridge, UK. [309]Bioinformatics Core Facility, University of Gothenburg, Gothenburg, Sweden. [310]Department of Medical Epidemiology and Biostatistics, Karolinska Institutet, Stockholm, Sweden. [311]Department of Allergy and Rheumatology, Graduate School of Medicine, University of Tokyo, Tokyo, Japan. [312]Department of Pediatrics, College of Medicine, University of Oklahoma Health Sciences Center, Oklahoma City, OK, USA. [313]Department of Neurology, Justus Liebig University, Giessen, Germany. [314]Department of Public Health, Bordeaux University Hospital, Bordeaux, France. [315]School of Medicine, Dentistry and Nursing at the University of Glasgow, Glasgow, UK. [316]University of Newcastle and Hunter Medical Research Institute, New Lambton, New South Wales, Australia. [317]INM, University of Montpellier Inserm U1298, Montpellier, France. [318]Centre for Research in Environmental Epidemiology, Barcelona, Spain. [319]Department of Neurology, Università degli Studi di Perugia, Umbria, Italy. [320]Broad Institute, Cambridge, MA, USA. [321]Department of Neurology, Memory Clinic, Bordeaux University Hospital, Bordeaux, France. [322]Department of Internal Medicine B, University Medicine Greifswald, Greifswald, Germany. [323]Robertson Center for Biostatistics, University of Glasgow, Glasgow, UK. [324]Hero DMC Heart Institute, Dayanand Medical College & Hospital, Ludhiana, India. [325]Atherosclerosis Research Unit, Department of Medicine Solna, Karolinska Institutet, Stockholm, Sweden. [326]Tohoku Medical Megabank Organization, Sendai, Japan. [327]Department of Public Health and Caring Sciences/Geriatrics, Uppsala University, Uppsala, Sweden. [328]Krembil Brain Institute, University Health Network, Toronto, Ontario, Canada. [329]Department of Medicine and Tanz Centre for Research in Neurodegenerative Diseases, University of Toronto, Toronto, Ontario, Canada. [330]Epidemiology and Prevention Group, Center for Public Health Sciences, National Cancer Center, Tokyo, Japan. [331]Department of Basic and Clinical Neurosciences, King's College London, London, UK. [332]Departments of Neurology & Radiology, Landspitali National University Hospital, Reykjavik, Iceland. [333]Department of Neurology, Heidelberg University Hospital, Heidelberg, Germany. [334]Montefiore Medical Center, Albert Einstein College of Medicine, New York, NY, USA. [335]Department of Medical Sciences, Uppsala University, Uppsala, Sweden. [336]Genetic and Genomic Epidemiology Unit, Wellcome Trust Centre for Human Genetics, University of Oxford, Oxford, UK. [337]Wellcome Trust Centre for Human Genetics, Oxford, UK. [338]BioBankJapan, Laboratory of Clinical Sequencing, Department of Computational Biology and Medical Sciences, Graduate School of Frontier Sciences, University of Tokyo, Tokyo, Japan. [339]Department of Biostatistics, University of Liverpool, Liverpool, UK. [340]Institute of Genetic Epidemiology, Helmholtz Zentrum München—German Research Center for Environmental Health, Neuherberg, Germany. [341]Department of Medicine I, Ludwig-Maximilians-Universität, Munich, Germany. [342]Translational Genomics Unit, Department of Oncology, IRCCS Istituto di Ricerche Farmacologiche Mario Negri, Milan, Italy. [343]Department of Genetics, Microbiology and Statistics, University of Barcelona, Barcelona, Spain. [344]Psychiatric Genetics Unit, Group of Psychiatry, Mental Health and Addictions, Vall d'Hebron Research Institute (VHIR), Universitat Autònoma de Barcelona,Biomedical Network Research Centre on Mental Health (CIBERSAM), Barcelona, Spain. [345]National Institute for Health Research Comprehensive Biomedical Research Centre, Guy's & St Thomas' NHS Foundation Trust and King's College London, London, UK. [346]School of Lifecourse and Population Sciences, King's College London, London, UK. [347]Icelandic Heart Association, Reykjavik, Iceland. [348]Department of Epidemiology, University of Maryland School of Medicine, Baltimore, MD, USA. [349]Institute of Cardiovascular and

Medical Sciences, College of Medical, Veterinary and Life Sciences, University of Glasgow, Glasgow, UK. [350]IBE, Faculty of Medicine, LMU Munich, Munich, Germany. [351]Division of Epidemiology and Prevention, Aichi Cancer Center Research Institute, Nagoya, Japan. [352]Department of Epidemiology, Nagoya University Graduate School of Medicine, Nagoya, Japan. [353]Department of Neurology, Caen University Hospital, Caen, France. [354]University of Caen Normandy, Caen, France. [355]Department of Internal Medicine, Erasmus University Medical Center, Rotterdam, The Netherlands. [356]Landspitali University Hospital, Reykjavik, Iceland. [357]Department of Pharmaceutical Sciences, College of Pharmacy, University of Oklahoma Health Sciences Center, Oklahoma City, OK, USA. [358]Oklahoma Center for Neuroscience, Oklahoma City, OK, USA. [359]IMIM (Hospital del Mar Medical Research Institute), Barcelona, Spain.

**The SIREN Consortium**

Vinodh Srinivasasainagendra[22], Rufus Akinyemi[28,29], Bruce Ovbiagele[118], Hemant K. Tiwari[22], Mayowa O. Owolabi[28,179,419], Onoja Akpa[360,361], Fred S. Sarfo[362], Albert Akpalu[363], Reginald Obiako[364], Kolawole Wahab[365], Godwin Osaigbovo[366], Lukman Owolabi[367], Morenikeji Komolafe[368], Carolyn Jenkins[369], Oyedunni Arulogun[370], Godwin Ogbole[364], Abiodun M. Adeoye[28,179], Joshua Akinyemi[360], Atinuke Agunloye[371], Adekunle G. Fakunle[179], Ezinne Uvere[179], Abimbola Olalere[179] & Olayinka J. Adebajo[179]

[360]Department of Epidemiology and Medical Statistics, University of Ibadan, Ibadan, Nigeria. [361]Institute of Cardiovascular Diseases, University of Ibadan, Ibadan, Nigeria. [362]Department of Medicine, Kwame Nkrumah University of Science and Technology, Kumasi, Ghana. [363]Department of Medicine, University of Ghana Medical School, Accra, Ghana. [364]Department of Medicine, Ahmadu Bello University, Zaria, Nigeria. [365]Department of Medicine, University of Ilorin Teaching Hospital, Ilorin, Nigeria. [366]Jos University Teaching Hospital, Jos, Nigeria. [367]Department of Medicine, Aminu Kano Teaching Hospital, Kano, Nigeria. [368]Department of Medicine, Obafemi Awolowo University Teaching Hospital, Ile-Ife, Nigeria. [369]Medical University of South Carolina, Charleston, SC, USA. [370]Department of Health Promotion and Education, University of Ibadan, Ibadan, Nigeria. [371]Department of Radiology, College of Medicine, University of Ibadan, Ibadan, Nigeria.

**The China Kadoorie Biobank Collaborative Group**

Kuang Lin[18], Zhengming Chen[18,63], Liming Li[101], Iona Y. Millwood[18,63], Robin G. Walters[18,63], Junshi Chen[372], Robert Clarke[18], Rory Collins[18], Yu Guo[373], Chen Wang[374,375], Jun Lv[101], Richard Peto[18], Yiping Chen[18,63], Zammy Fairhurst-Hunter[18], Michael Hill[18,63], Alfred Pozaricki[18], Dan Schmidt[18], Becky Stevens[18], Iain Turnbull[18] & Canqing Yu[101]

[372]China National Center For Food Safety Risk Assessment, Beijing, China. [373]Fuwai Hospital, Chinese Academy of Medical Sciences, National Center for Cardiovascular Diseases, Beijing, China. [374]Chinese Academy of Medical Sciences, Beijing, China. [375]Peking Union Medical College, Beijing, China.

**The VA Million Veteran Program**

Daniel C. Posner[7,418], Kelly Cho[7], Yuk-Lam Ho[7], Yan V. Sun[136,137], Peter W. F. Wilson[136,145] & Jennifer E. Huffman[7,419]

**The International Stroke Genetics Consortium (ISGC)**

Aniket Mishra[1,418], Rainer Malik[2,418], Quentin Le Grand[1], Yunye He[11], Marios K. Georgakis[2,12,13], Ilana Caro[1], Livia Parodi[12,13], Hee-Joon Bae[23], Ganesh Chauhan[24], Michael R. Chong[25,26], Liisa Tomppo[27], Rufus Akinyemi[28,29], Vida Abedi[35,36], Jara Cárcel-Márquez[37,38], Maria J. Knol[30], Mark K. Bakker[55], Joshua C. Bis[56], Zhengming Chen[18,63], Carlos Cruchaga[64,65], John W. Cole[66,67], Leslie E. Ferreira[74], Jemma C. Hopewell[82], Hyacinth I. Hyacinth[83], Christina Jern[90,91], Keith L. Keene[93], Steven J. Kittner[67,94], Paul Lacaze[95], Keon-Joo Lee[97], Hugh S. Markus[102], Braxton D. Mitchell[105,106], Yukinori Okada[6,113,114,115,116,117], Stephen S. Rich[44], Jonathan Rosand[12,13,124], Ralph L. Sacco[125,126], Muralidharan Sargurupremraj[130], Claudia L. Satizabal[130,131], Turgut Tatlisumak[138], David-Alexandre Trégouët[1], Daniel Woo[83], Iona Y. Millwood[18,63], Daniel Strbian[27], Ernst Mayerhofer[12,13], Christopher D. Anderson[12,13,124,173], Sylvia Wassertheil-Smoller[155], Tatjana Rundek[125,126], Bradford B. Worrall[157,158], Janika Kõrv[142], Paulo H. C. França[74], Ramin Zand[161,162], Kameshwar Prasad[24], Frank-Erik de Leeuw[88], Ynte M. Ruigrok[55], W. T. Longstreth[122,166], Guillaume Paré[25,26,168,169], Daniel I. Chasman[61,62], Jerome I. Rotter[172], Myriam Fornage[175,176], Sudha Seshadri[130,131,178], Israel Fernández-Cadenas[37], Robin G. Walters[18,63], Mayowa O. Owolabi[28,179,419], Yoichiro Kamatani[11,419], Martin Dichgans[2,180,181,419] & Stephanie Debette[1,182,419]

**The Biobank Japan**

Masaru Koido[10,11], Takayuki Morisaki[10], Yoichiro Kamatani[11,419], Akiko Nagai[376], Yoishinori Murakami[10] & Koichi Matsuda[14]

[376]Department of Public Policy, Institute of Medical Sciences, The University of Tokyo, Tokyo, Japan.

**The CHARGE Consortium**

Aniket Mishra[1,418], Quentin Le Grand[1], Ilana Caro[1], Gennady V. Roshchupkin[30,31], Hampton L. Leonard[41,42,43], Chaojie Yang[44,45], Maria J. Knol[30], Traci M. Bartz[56,57], Joshua C. Bis[56], Constance Bordes[1], Paul M. Ridker[61,62], Mirjam I. Geerlings[75], Natalie C. Gasca[57], Vilmundur Gudnason[76,77], Paul Lacaze[95], Lenore J. Launer[96], Ani Manichaikul[44], Thomas H. Mosley[109], Mike A. Nalls[41,42,43], Stephen S. Rich[44], Muralidharan Sargurupremraj[130], Claudia L. Satizabal[130,131], Carsten O. Schmidt[132], Stella Trompet[139,140], Marion van Vugt[141], Qiong Yang[149,150], Hans J. Grabe[149,150], J. Wouter Jukema[140,151,152], Ina L. Rissanen[75], Hieab Adams[153,154], Sylvia Wassertheil-Smoller[155], Mohammad A. Ikram[30], Moeen Riaz[95], Eleanor M. Simonsick[160], W. T. Longstreth[122,166], Daniel I. Chasman[61,62], Jerome I. Rotter[172], Myriam Fornage[175,176], Sudha Seshadri[130,131,178], Stephanie Debette[1,182,419], Rebecca F. Gottesman[195], David J. Stott[349], Naveed Sattar[280], David J. Stott[349], Eric J. Shiroma[186], Oscar L. Lopez[272], Sigurdur Sigurdsson[76], Mohsen Ghanbari[30], Ulf Schminke[273], Eric Boerwinkle[176,377], Hugo J. Aparicio[131,178], Alexa S. Beiser[131,146], Jose R. Romero[131,178], Vasileios Lioutas[131,277], Xueqiu Jian[130,131], Bernard Fongang[30,131], Ruiqi Wang[131,146], Chloe Sarnowski[279], Mohammad K. Ikram[30,231], Alexander Teumer[132,281] & Uwe Völker[281,378]

[377]Human Genome Sequencing Center, Baylor College of Medicine, Houston, TX, USA. [378]Interfaculty Institute for Genetics and Functional Genomics, University Medicine Greifswald, Greifswald, Germany.

**The GIGASTROKE Consortium**

Aniket Mishra[1,418], Rainer Malik[2,418], Tsuyoshi Hachiya[3,418], Tuuli Jürgenson[4,5,418], Shinichi Namba[6,418], Daniel C. Posner[7,418], Frederick K. Kamanu[8,9], Masaru Koido[10,11], Quentin Le Grand[1], Mingyang Shi[11], Yunye He[11], Marios K. Georgakis[2,12,13], Ilana Caro[1], Kristi Krebs[4], Yi-Ching Liaw[14,15], Felix C. Vaura[16,17], Kuang Lin[18], Bendik Slagsvold Winsvold[19,20,21], Vinodh Srinivasasainagendra[22], Livia Parodi[12,13], Hee-Joon Bae[23], Ganesh Chauhan[24], Michael R. Chong[25,26], Liisa Tomppo[27], Rufus Akinyemi[28,29], Gennady V. Roshchupkin[30,31], Naomi Habib[32], Yon Ho Jee[33], Jesper Qvist Thomassen[34], Vida Abedi[35,36], Jara Cárcel-Márquez[37,38], Marianne Nygaard[39,40], Hampton L. Leonard[41,42,43], Chaojie Yang[44,45], Ekaterina Yonova-Doing[46,47], Maria J. Knol[30], Adam J. Lewis[48], Renae L. Judy[49], Tetsuro Ago[50], Philippe Amouyel[51,52,53], Nicole D. Armstrong[54], Mark K. Bakker[55], Traci M. Bartz[56,57], David A. Bennett[58], Joshua C. Bis[56], Constance Bordes[1], Sigrid Børte[20,59,60], Anael Cain[32], Paul M. Ridker[61,62], Kelly Cho[7], Zhengming Chen[18,63], Carlos Cruchaga[64,65], John W. Cole[66,67], Phil L. de Jager[13,68], Rafael de Cid[69], Matthias Endres[70,71,72,73], Leslie E. Ferreira[74], Mirjam I. Geerlings[75], Natalie C. Gasca[57], Vilmundur Gudnason[76,77], Jing He[48], Alicia K. Heath[79], Yuk-Lam Ho[7], Aki S. Havulinna[80,81], Jemma C. Hopewell[82], Hyacinth I. Hyacinth[83], Michael Inouye[46,84,85,86,87], Mina A. Jacob[88], Christina E. Jeon[89], Christina Jern[90,91], Masahiro Kamouchi[92], Keith L. Keene[93], Takanari Kitazono[50], Steven J. Kittner[67,94], Takahiro Konuma[6], Amit Kumar[24], Lenore J. Launer[96], Keon-Joo Lee[97], Kaido Lepik[4,98,99,100], Jiang Li[35], Liming Li[101], Ani Manichaikul[44], Hugh S. Markus[102], Nicholas A. Marston[8,9], Thomas Meitinger[103,104], Braxton D. Mitchell[105,106], Felipe A. Montellano[107,108], Takayuki Morisaki[10], Thomas H. Mosley[109], Mike A. Nalls[41,42,43], Børge G. Nordestgaard[110,111], Martin J. O'Donnell[112], Yukinori Okada[6,113,114,115,116,117], N. Charlotte Onland-Moret[75], Bruce Ovbiagele[118], Annette Peters[119,120,121], Bruce M. Psaty[56,122,123], Stephen S. Rich[44], Jonathan Rosand[12,13,124], Marc S. Sabatine[8,9], Ralph L. Sacco[125,126], Danish Saleheen[127], Else Charlotte Sandset[128,129], Veikko Salomaa[80], Muralidharan Sargurupremraj[130], Makoto Sasaki[3], Claudia L. Satizabal[130,131], Carsten O. Schmidt[132], Atsushi Shimizu[3], Nicholas L. Smith[122,133,134], Kelly L. Sloane[135], Yoichi Sutoh[3], Yan V. Sun[136,137], Kozo Tanno[3], Steffen Tiedt[2], Turgut Tatlisumak[138], Nuria P. Torres-Aguila[37], Hemant K. Tiwari[22], David-Alexandre Trégouët[1], Stella Trompet[139,140], Anil Man Tuladhar[88], Anne Tybjærg-Hansen[34,111], Marion van Vugt[141], Riina Vibo[142], Shefali S. Verma[143], Kerri L. Wiggins[56], Patrik Wennberg[144], Daniel Woo[83], Peter W. F. Wilson[136,145], Huichun Xu[105], Qiong Yang[131,146], Kyungheon Yoon[147], Iona Y. Millwood[18,63], Christian Gieger[148], Toshiharu Ninomiya[78], Hans J. Grabe[149,150], J. Wouter Jukema[140,151,152], Ina L. Rissanen[75], Daniel Strbian[27], Young Jin Kim[147], Pei-Hsin Chen[15], Ernst Mayerhofer[12,13], Christopher D. Anderson[12,13,124,173], Marguerite R. Irvin[54], Hieab Adams[153,154], Sylvia Wassertheil-Smoller[155], Kaare Christensen[39,40,156], Mohammad A. Ikram[30], Tatjana Rundek[125,126], Bradford B. Worrall[157,158], G. Mark Lathrop[159], Moeen Riaz[95], Eleanor M. Simonsick[160], Janika Kõrv[142], Paulo H. C. França[74], Ramin Zand[161,162], Kameshwar Prasad[24], Ruth Frikke-Schmidt[34,111], Frank-Erik de Leeuw[88], Thomas Liman[71,75,163], Karl Georg Haeusler[108], Ynte M. Ruigrok[55], Peter Ulrich Heuschmann[107,164,165], W. T. Longstreth[122,166], Keum Ji Jung[18,167], Lisa Bastarache[48], Guillaume Paré[25,26,168,169], Scott D. Damrauer[170,171], Daniel I. Chasman[61,62], Jerome I. Rotter[172], John-Anker Zwart[19,20,59], Teemu J. Niiranen[16,17,174], Myriam Fornage[175,176], Yung-Po Liaw[15,177], Sudha Seshadri[130,131,178], Israel Fernández-Cadenas[37], Robin G. Walters[18,63], Christian T. Ruff[8,9,419], Mayowa O. Owolabi[28,179,419], Jennifer E. Huffman[7,419], Lili Milani[4,419], Yoichiro Kamatani[11,419], Martin Dichgans[2,180,181,419] & Stephanie Debette[1,182,419]

**Regeneron Genetics Center**

**The ODYSSEY Study**

Mina A. Jacob[88], Anil Man Tuladhar[88], Frank-Erik de Leeuw[88], Karlijn F. de Laat[379], Anouk G. W. van Norden[380], Paul L. de Kort[381], Sarah E. Vermeer[382], Paul J. A. M. Brouwers[383], Rob A. R. Gons[384], Paul J. Nederkoorn[233], Tom den Heijer[385], Robert J. van Oostenbrugge[234], Gert W. van Dijk[386] & Frank G. W. van Rooij[387]

**HUNT All-In Stroke**

Bendik Slagsvold Winsvold[19,20,21], Sigrid Børte[20,59,60], Else Charlotte Sandset[128,129], John-Anker Zwart[19,20,59], Anne H. Aamodt[21,388], Anne H. Skogholt[20], Ben M. Brumpton[20], Cristen J. Willer[220,389], Ingrid Heuch[19], Knut Hagen[388], Lars G. Fritsche[390], Linda M. Pedersen[19], Maiken E. Gabrielsen[20], Hanne Ellekjær[388,391], Wei Zhou[392,393], Amy E. Martinsen[19,20,59], Espen S. Kristoffersen[19,394,395], Jonas B. Nielsen[20,220,221], Kristian Hveem[20,202,203] & Laurent F. Thomas[20,396,397,398]

**The SICFAIL Study**

Felipe A. Montellano[107,108], Karl Georg Haeusler[108], Peter Ulrich Heuschmann[107,164,165], Christoph Kleinschnitz[399], Stefan Frantz[400] & Kathrin Ungethüm[107]

**The Generacion Study**

Jara Cárcel-Márquez[37,38], Carlos Cruchaga[64,65], Rafael de Cid[69], Nuria P. Torres-Aguila[37], Elena Muiño[37], Cristina Gallego-Fabrega[37,401], Natalia Cullell[37,274], Miquel Lledós[37,402], Laia Llucià-Carol[37], Tomas Sobrino[403], Francisco Campos[403], José Castillo[403], Marimar Freijó[404], Juan Francisco Arenillas[405], Victor Obach[406], José Álvarez-Sabín[407], Carlos A. Molina[407], Marc Ribó[407], Jordi Jiménez-Conde[238], Jaume Roquer[238], Lucia Muñoz-Narbona[408], Elena Lopez-Cancio[409], Mònica Millán[408], Rosa Diaz-Navarro[410], Cristòfol Vives-Bauza[410], Gemma Serrano-Heras[411], Tomás Segura[411], Laura Ibañez[64], Laura Heitsch[183,259], Pilar Delgado[275], Rajat Dhar[183], Jerzy Krupinski[276], Raquel Delgado-Mederos[401], Luis Prats-Sánchez[401], Pol Camps-Renom[401], Natalia Blay[69], Lauro Sumoy[412], Joan Montaner[278], Jin-Moo Lee[183], Joan Martí-Fàbregas[401] & Israel Fernández-Cadenas[37]

**The Copenhagen City Heart Study**

Jesper Qvist Thomassen[34], Børge G. Nordestgaard[110,111], Anne Tybjærg-Hansen[34,111], Ruth Frikke-Schmidt[34,111], Peter Schnohr[413], Gorm B. Jensen[413], Marianne Benn[34,111], Shoaib Afzal[110,111] & Pia R. Kamstrup[110]

**The SMART Study**

Marion van Vugt[141], Ina L. Rissanen[75], Mirjam I. Geerlings[75], Jessica van Setten[141], Sander W. van der Laan[414] & Jet M. J. Vonk[75,415]

**Clinical Research Collaboration for Stroke in Korea (CRCS-K) and Korea Biobank Array (KBA) Project**

Hee-Joon Bae[23], Keon-Joo Lee[97], Kyungheon Yoon[147], Young Jin Kim[147] & Bong-Jo Kim[148]

**Helsinki Stroke Project**

Liisa Tomppo[27], Veikko Salomaa[80], Turgut Tatlisumak[138], Gerli Sibolt[27], Sami Curtze[27], Marjaana Tiainen[27] & Janne Kinnunen[27]

**Follow-up Studies**

Vilas Menon[416], Yun Ju Sung[64,65], Chengran Yang[64,65], Florence Saillour-Glenisson[1,314] & Simon Gravel[417]

**EPIC-CVD**

Ekaterina Yonova-Doing[46,47], N. Charlotte Onland-Moret[75], Alicia K. Heath[79], Christopher D. Anderson[12,13,124,173] & Patrik Wennberg[144]

[379]Department of Neurology, Haga Hospital, The Hague, The Netherlands. [380]Department of Neurology, Amphia Hospital, Breda, The Netherlands. [381]Department of Neurology, Elisabeth-Tweesteden Hospital, Tilburg, The Netherlands. [382]Department of Neurology, Rijnstate Hospital, Arnhem, The Netherlands. [383]Department of Neurology, Medisch spectrum Twente, Enschede, The Netherlands. [384]Department of Neurology, Catharina Hospital, Eindhoven, The Netherlands. [385]Department of Neurology, Franciscus Gasthuis & Vlietland, Rotterdam, The Netherlands. [386]Department of Neurology, Catharina Wilhelmina Hospital, Nijmegen, The Netherlands. [387]Department of Neurology, Medisch Center Leeuwarden, Leeuwarden, The Netherlands. [388]Department of Neuromedicine and Movement Science, Faculty of Medicine and Health Sciences, Norwegian University of Science and Technology (NTNU), Trondheim, Norway. [389]Department of Human Genetics, University of Michigan, Ann Arbor, MI, USA. [390]Center for Statistical Genetics, Department of Biostatistics, University of Michigan, Ann Arbor, MI, USA. [391]Stroke Unit, Department of Internal Medicine, St Olavs Hospital, Trondheim University Hospital, Trondheim, Norway. [392]Department of Computational Medicine and Bioinformatics, University of Michigan, Ann Arbor, MI, USA. [393]Analytic and Translational Genetics Unit, Massachusetts General Hospital, Boston, MA, USA. [394]Department of General Practice, University of Oslo, Oslo, Norway. [395]Department of Neurology, Akershus University Hospital, Lørenskog, Norway. [396]BioCore—Bioinformatics Core Facility, Norwegian University of Science and Technology (NTNU), Trondheim, Norway. [397]Clinic of Laboratory Medicine, St Olavs Hospital, Trondheim University Hospital, Trondheim, Norway. [398]Department of Clinical and Molecular Medicine, Norwegian University of Science and Technology (NTNU), Trondheim, Norway. [399]Department of Neurology and Center for Translational Neuro- and Behavioral Sciences (C-TNBS), University Hospital Essen, University Duisburg-Essen, Essen, Germany. [400]Department of Medicine I, University Hospital Würzburg, Würzburg, Germany. [401]Stroke Unit, Department of Neurology, Biomedical Research Institute Sant Pau, Hospital de la Santa Creu i Sant Pau, Barcelona, Spain. [402]Institute for Biomedical Research of Barcelona (IIBB), National Spanish Research Council (CSIC), Barcelona, Spain. [403]Clinical Neurosciences Research Laboratory (LINC), Health Research Institute of Santiago de Compostela (IDIS), Santiago de Compostela, Spain. [404]Department of Neurology, Biocruces-Bizkaia Health Research Institute, Bilbao, Spain. [405]Stroke Unit, Department of Neurology, University Hospital of Valladolid, Valladolid, Spain. [406]Department of Neurology, Hospital Clínic de Barcelona, IDIBAPS, Barcelona, Spain. [407]Stroke Unit, Department of Neurology, Hospital Universitari Vall d'Hebron, Barcelona, Spain. [408]Department of Neurosciences, Hospital Germans Trias I Pujol, Universitat Autònoma de Barcelona, Barcelona, Spain. [409]Department of Neurology, University Hospital Central de Asturias (HUCA), Oviedo, Spain. [410]Department of Neurology, Son Espases University Hospital, Illes Balears Health Research Institute (IdISBa), Palma, Spain. [411]Department of Neurology, University Hospital of Albacete, Albacete, Spain. [412]High Content Genomics and Bioinformatics Unit, Germans Trias i Pujol Research Institute (IGTP), Badalona, Spain. [413]The Copenhagen City Heart Study, Copenhagen University Hospital—Bispebjerg and Frederiksberg, Copenhagen, Denmark. [414]Central Diagnostics Laboratory, Division Laboratories, Pharmacy, and Biomedical genetics, University Medical Center Utrecht, Utrecht University, Utrecht, The Netherlands. [415]Taub Institute for Research on Alzheimer's Disease and the Aging Brain, College of Physicians and Surgeons, Columbia University, New York, NY, USA. [416]The Centre for Translational and Computational Neuroimmunology, Columbia University Medical Center, New York, NY, USA. [417]Department of Human Genetics, McGill University, Montreal, Quebec, Canada.

## Methods

All human research was approved by relevant boards and/or institutions for each study (Supplementary Table 56) and was conducted according to the Declaration of Helsinki. All of the participants provided written informed consent.

### Study design and phenotypes
Information on participating studies (discovery and follow-up), study design, and definitions of stroke and stroke subtypes is provided in the Supplementary Information. Population characteristics of individual studies are provided in Supplementary Table 1.

### Genotyping, imputation and GWASs
Genotyping methods, pre-imputation quality control of genotypes and imputation methods of individual cohorts (discovery and follow-up) are presented in Supplementary Table 2. High-quality samples and SNPs underwent imputation using mostly Haplotype Reference Consortium (HRC) or 1000 Genomes phase 1 or phase 3 reference panels and, less often, TOPMed, HapMap or biobank-specific reference panels. Individual studies performed a GWAS using logistic regression (or Cox regression in some longitudinal population-based cohorts) testing association of genotypes with five stroke phenotypes (AS, AIS, CES, LAS and SVS) under an additive effect model, adjusting for age, sex, principal components of population stratification and study-specific covariates when needed (Supplementary Table 2).

The R package EasyQC along with custom harmonization scripts were used to perform the quality control of individual GWAS summary results. Marker names and alleles were harmonized across studies. Meta-analyses were restricted to autosomal biallelic SNPs from the HRC panel. Duplicate markers were removed. Before the meta-analysis, we removed variants with extreme effect size values (log[OR] > 5 or log[OR] < −5), minor allele frequency (MAF) < 0.01, imputation quality scores of less than 0.50 and effective allele counts (EAC = 2 × number of cases × MAF × imputation quality score) of less than 6.

The overall analytical strategy is shown in Extended Data Fig. 1. We conducted ancestry-specific fixed-effect IVW meta-analyses in European, East Asian, African American, Hispanic and South Asian populations, followed by cross-ancestry meta-analyses using METAL[50]. In each meta-analysis we removed variants with heterogeneity $P < 1 \times 10^{-6}$ and variants available in less than one third of the total number of cases and less than one third of the total number of contributing studies. We applied the covariate adjusted linkage disequilibrium score regression (cov-LDSC) method to ancestry-specific GWAS meta-analyses without GC correction to test for genomic inflation and to compute robust SNP-heritability estimates in admixed populations[51]. We conducted cross-ancestry GWAS meta-analyses without genomic correction and with correction of the linkage-disequilibrium score intercept for genomic inflation observed in individual ancestry-specific GWASs. We conducted separate GWAS analyses of incident AS and AIS ($n = 32,903$ and $n = 16,863$) in longitudinal population-based cohort studies. For the meta-analysis combining both incident and prevalent stroke studies, a few incident stroke studies were removed because they were already part of a meta-analysis of stroke GWASs used as an input of the overall meta-analysis (WHI, Hisayama, REGARDS, JHS). We considered loci to be genome-wide significant for $P < 5 \times 10^{-8}$.

We applied the conditional and joint analysis approach[7] implemented in the Genome-wide Complex Trait Analysis software[52] (GCTA-COJO) to identify potentially independent signals within the same genomic region. We performed GCTA-COJO analyses on (1) European GWAS meta-analysis summary statistics using HRC imputed data of 6,489 French participants from the 3C study as in ref. [53] and (2) East Asian-ancestry-specific GWAS meta-analysis summary statistics using BBJ data as reference (Supplementary Information).

We also performed a cross-ancestry meta-regression using MR-MEGA[10]. Before the meta-analysis using MR-MEGA, we applied the 'genomic inflation' correction option to all of the input files, and removed variants with extreme effect size values (log[OR] > 5 or log[OR] < −5), MAF < 0.01, imputation quality scores of less than 0.50 and effective allele counts (EAC = 2 × number of cases × MAF × imputation quality score) of less than 6. After the meta-analysis, we considered loci to be genome-wide significant for MR-MEGA $P < 5 \times 10^{-8}$ and showing nominal association ($P < 0.05$) in at least one third of studies in any individual ancestry group (European, East Asian, African American, Hispanic and South Asian).

### Multitrait association study
To identify additional stroke-risk loci we used MTAG[11] in Europeans and East Asians, including traits correlated with specific stroke subtypes, namely CAD for LAS, atrial fibrillation[54] for CES, and WMH[55] (an MRI marker of cerebral small vessel disease, available in Europeans only) for SVS. We also ran an MTAG analysis of AS and AIS, including all three correlated traits (CAD, atrial fibrillation, WMH (European)). In European individuals, we used summary statistics of published GWAS analyses for CAD[56], AF[54] and WMH[55]. In East Asians, we used summary statistics of published GWAS analyses for CAD[57] and atrial fibrillation[58] (Supplementary Information). Associations were retained when the following three conditions were verified: (1) MTAG $P$ value for stroke $< 5 \times 10^{-8}$; (2) $P$ value for stroke < 0.05 in the univariate GWAS; and (3) MTAG $P$ value for stroke less than the $P$ value for any of the included traits in univariate GWASs.

### Independent follow-up of GWAS signals
First, we sought to replicate internally the 42 stroke-risk loci reaching genome-wide significance in IVW meta-analyses within individual ancestries, in at least one other ancestry group among the discovery samples, considering both nominal replication levels ($P < 0.05$) and multiple-testing corrected significance at $P < 1.19 \times 10^{-3}$ (0.05/42). Second, we gathered independent datasets totalling 89,084 AS (including 85,546 AIS; and 70.0% European, 15.6% African American, 10.1% East Asian, 4.1% Hispanic and 0.1% South Asian) and 1,013,843 controls for external replication of associations with AS and AIS (Supplementary Tables 1 and 2). These comprised eight biobanks (82,263 cases, 930,988 controls) and four hospital-based cohorts (6,821 cases, 82,855 controls). We considered both nominal replication levels ($P < 0.05$) and multiple-testing corrected significance at $P < 8.2 \times 10^{-4}$ (0.05/60) and $P < 1.3 \times 10^{-3}$ (0.05/29) for follow-up of genome-wide significant loci from the IVW and the MR-MEGA/MTAG meta-analyses, respectively (two-sided $P$ values were used for both discovery and replication analyses). We considered stroke-risk loci as high confidence in the case of significant internal inter-ancestry and/or external replication after accounting for the number of loci tested, nominally significant replication in both internal and external replication analyses, or evidence of involvement in monogenic stroke; intermediate confidence in the case of nominal significance in either internal inter-ancestry or external replication analyses but not both; and low confidence in the absence of formal replication.

### Gene-based analyses
We performed gene-based tests of common variant associations using VEGAS2[8] and MAGMA[9]. Both VEGAS2 and MAGMA considered variants in the gene or within 10 kb on either side of a gene's transcription site to compute a gene-based $P$ value. We performed MAGMA tests using the default parameters, whereas the VEGAS2 analyses were performed using the '-top 10' parameter that tests enrichment of the top 10% variants assigned to a gene accounting for the linkage disequilibrium between variants and the total number of variants within a gene. We used 1000 Genomes phase 3 continental reference samples of European, East Asian, African, South Asian and South American (for our Hispanic

samples) ancestry and to compute the linkage disequilibrium between variants for respective ancestry-specific gene-based analyses. We then meta-analysed ancestry-specific gene-based results, using Stouffer's method for sample-size-weighted combination of $P$ values. Gene-wide significance was defined as $P < 2.72 \times 10^{-6}$, correcting for 18,371 autosomal protein-coding genes tested.

## Pathway-based analyses

We used the ancestry-specific gene-based association $P$ values generated using VEGAS2 to perform pathway analyses for individual ancestry groups, testing enrichment of gene-based $P$ values in Biosystems pathways with VEGAS2Pathway[8,15]. For each stroke phenotype, we meta-analysed the ancestry-specific pathway association $P$ values using Stouffer's method considering the number of cases in each ancestry-specific GWAS; for example, for AS, we considered 73,652, 27,413, 3,961, 1,516 and 3,640 cases in European-, East Asian-, African American-, Hispanic- and South Asian-specific GWAS analyses to combine the respective ancestry-specific pathway association $P$ values. Pathway-wide significance was defined at $P < 5.01 \times 10^{-6}$ correcting for 9,977 Biosystems pathways tested.

## Shared genetic variation

We examined shared genetic variation with 12 vascular risk factors and related disease traits in Europeans using summary statistics of GWASs on SBP[59], DBP[59], BMI and waist-to-hip ratio[60], high density lipoprotein (HDL) cholesterol[61], LDL cholesterol[61], triglycerides[61], type 2 diabetes[62], WMH volume[55], atrial fibrillation[54], CAD[56] and venous thromboembolism[32]. We extracted sentinel stroke-risk variants (or a proxy ($r^2 > 0.9$)) that showed genome-wide significant association ($P < 5 \times 10^{-8}$) with the aforementioned vascular-risk traits.

We then systematically examined genetic correlations and potentially causal associations between vascular-risk traits and risk of stroke using linkage-disequilibrium score regression (LDSC) and MR analyses, with 12 (in Europeans) and 6 (in East Asians) vascular-risk traits. In individuals of European ancestry, we used summary statistics of the aforementioned GWASs[32,54–56,59–62]. For the analysis in East Asians, we used unpublished GWAS analyses for SBP, DBP, LDL and HDL cholesterol, triglycerides and BMI in up to 53,323 participants of the independent Tohoku Medical Megabank Project (Supplementary Information).

We used cov-LDSC to compute genetic correlations between stroke and vascular-risk traits, using European and East Asian GWAS summary files and 1000Gp3v5 reference data of respective continental ancestries (considering the recommended subset of high-quality HapMap3 SNPs only).

For MR analyses, we constructed genetic instruments for each vascular-risk trait based on genome-wide significant associations ($P < 5 \times 10^{-8}$) after clumping for linkage disequilibrium at $r^2 < 0.01$ (based on European and East Asian 1000 Genomes reference panels). We applied two-sample MR analyses in the GIGASTROKE summary statistics separately for individuals of European and East Asian ancestry based on variant associations derived from the aforementioned sources. After extraction of the association estimates and harmonization of their direction-of-effect alleles, we computed MR estimates with fixed-effect IVW analyses[63]. As a measure of pleiotropy, we assessed heterogeneity across the MR estimates for each instrument in the IVW MR analyses with Cochran's $Q$ statistic ($P < 0.05$ was considered to be significant)[64]. We further applied alternative MR methods that are more robust to the use of pleiotropic instruments: the weighted median estimator enables the use of invalid instruments under the assumption that at least half of the instruments used in the MR analysis are valid[65]; MR-Egger regression allows for the estimation of an intercept term, provides less precise estimates and relies on the assumption that the strengths of potential pleiotropic instruments are independent of their direct associations with the outcome[66]. The intercept obtained from MR-Egger regression was used as a measure of directional pleiotropy

($P < 0.05$ indicated significance)[66]. MR analyses were performed in R v.4.1.1 using the Mendelian Randomization package.

For all genetic correlation and MR analyses, we set statistical significance at Bonferroni-corrected $P < 4.17 \times 10^{-3}$ in Europeans (correcting for 12 vascular-risk traits) and $P < 8.33 \times 10^{-3}$ in East Asians (correcting for 6 vascular-risk traits).

## Cross-ancestry fine mapping

Fine-mapping was performed separately for Europeans and East Asians using susieR v.0.9.1[12] on all variants within 3 Mb of the lead variant of each genomic risk locus (60 loci reached genome-wide significance in the IVW meta-analysis). Unrelated individuals from the UK Biobank ($n = 420,000$) and BBJ ($n = 170,000$) were used as ancestry-matched linkage-disequilibrium reference panels that fulfil the sample size requirement[67]. After extracting variants present in the linkage disequilibrium reference panel, the default settings of susieR were used while allowing for a maximum of 10 putative causal variants in each locus. The fine-mapping results were checked for potential false-positive findings using a diagnostic procedure implemented in SuSiE. In brief, we compared observed and expected $z$-scores for each variant at a given locus and removed the variant if the difference between the observed and expected $z$-score was too high after manual inspection. We compared the variants in credible sets of the same loci between Europeans and East Asians.

To detect putative causal regulatory variants, we conducted an in silico mutagenesis analysis using MENTR (mutation effect prediction on non-coding RNA transcription; https://github.com/koido/MENTR), a machine-learning method to precisely predict transcriptional changes induced by causal variants[3,68]. The in silico mutations predicted to have strong effects are highly concordant with the observed effects of known variants in a cell-type-dependent manner. Furthermore, MENTR does not use population datasets and is therefore less susceptible to linkage-disequilibrium-dependent association signals, enabling precise prediction of the effects of causal variants on transcriptional changes. From 1,274 variants in the credible sets from the European and East Asian fine-mapping, we searched FANTOM5 promoters and enhancers, obtained by cap analysis of gene expression, within ±100 kb from each variant. As a result, we found 37,878 variant–transcript pairs comprising 1,270 variants and 2,350 transcripts. We used MENTR with the pretrained FANTOM5 347 cell/tissue models + LCL models[69–72] and extracted reliable predictions using the predetermined robust threshold (absolute in silico mutation effects $\geq 0.1$, achieving >90% concordance for predicting effects on expression).

## TWAS and PWAS

We performed TWAS using TWAS-Fusion[19] to identify genes of which the expression is significantly associated with stroke risk. We restricted the analysis to tissues considered to be relevant for cerebrovascular disease, and used precomputed functional weights from 21 publicly available eQTL reference panels from blood (Netherlands Twin Registry; Young Finns Study)[19,20], arterial and heart (GTEx v.7)[17] and brain tissues (GTEx v.7, CommonMind Consortium)[17,18]. Moreover, we used the newly developed cross-tissue weights generated in GTEx v.8 using sparse canonical correlation analysis (sCCA) across 49 tissues available on the TWAS-Fusion website, including gene expression models for the first three canonical vectors (sCCA1–3), which were shown to capture most of the gene expression signal[73]. TWAS-Fusion was then used to estimate the TWAS association statistics between predicted gene expression and stroke by integrating information from expression reference panels (SNP-expression weights), GWAS summary statistics (SNP-stroke effect estimates) and linkage disequilibrium reference panels (SNP correlation matrix)[19]. Transcriptome-wide significant genes (eGenes) and the corresponding eQTLs were determined using Bonferroni correction, based on the average number of features (5005.8 genes) tested across all reference panels and correcting for the 5 stroke

phenotypes ($P < 2.0 \times 10^{-6}$). eGenes were then tested in conditional analysis as implemented using the Fusion software[19]. To ensure that the observed associations did not reflect random correlation between gene expression and non-causal variants associated with stroke, we performed a colocalization analysis on the conditionally significant genes ($P < 0.05$) to estimate the posterior probability of a shared causal variant between the gene expression and trait association (PP4)[74]. We used a prior probability of $P < 2.0 \times 10^{-6}$ for the stroke association. Genes presenting a PP4 ≥ 0.75, for which eQTLs did not reach genome-wide significance in association with stroke, and were not in linkage disequilibrium ($r^2 < 0.01$) with any of the lead SNPs of genome-wide significant risk loci for stroke, were considered to be new, i.e. not within a genome-wide significant stroke risk locus.

Using similar parameters in TWAS-Fusion[19], we also performed a proteome-wide association study. For this analysis, we used the precomputed weights for protein expression in DLPFC[75] from the ROS/MAP study ($n = 376$ individuals, $n = 1,475$ proteins)[76] and the Banner Sun Health Institute study ($n = 152$ individuals, $n = 1,145$ proteins)[77]. Proteome-wide significant genes and the corresponding pQTLs were determined using Bonferroni correction, on the number of proteins tested across the reference panel and correcting for the 5 stroke phenotypes ($P < 1.7 \times 10^{-4}$ for ROS/MAP and $P < 2.2 \times 10^{-8}$ for the Banner Sun Health Institute study). We then followed the same method as described for the TWAS.

### Brain single-cell expression analyses

Single-nucleus RNA-sequencing data of the DLPFC region of 24 ageing individuals chosen to represent the range of pathologic and clinical diagnoses of AD dementia, from the ROS/MAP cohorts, was obtained[21]. RNA profiles of cells annotated as endothelial, pericytes or smooth muscle cells and vascular leptomeningeal cells (VLMC) were used, and a pseudobulk RNA profile was generated for each cell type by averaging the expression of all genes across the cells. Average expression levels and the percentage of expressed genes were calculated for genes of interest using the DotPlot function from the Seurat package v.4.0.4 in R v.4.1.1.

We also conducted a cell-type enrichment analysis using the STEAP pipeline (https://github.com/ComPopBio/STEAP). This is an extension of CELLECT and uses S-LDSC[78], MAGMA[9] and H-MAGMA[79] for enrichment analysis. Stroke GWAS summary statistics were first munged. Expression specificity profiles were then calculated using human and mouse single-cell RNA-seq databases (Supplementary Table 28). Cell-type enrichment was calculated using three models: MAGMA, H-MAGMA (incorporating chromatin interaction profiles from human brain tissues in MAGMA) and stratified linkage-disequilibrium score regression. $P$ values were corrected for the number of independent cell types in each database (Bonferroni correction).

### Genomics-driven drug discovery

We used three methodologies for in-depth genomics-driven drug discovery as described previously[4]: (1) an overlap enrichment analysis of disease-risk genes in drug-target genes in medication categories; (2) negative correlation tests between genetically determined case–control gene expression profiles and compound-regulated gene expression profiles; and (3) endophenotype MR. Details of the methods are described in the following sections. For the overlap enrichment analysis and the endophenotype MR-nominated drug targets, we curated drug candidates from four major drug databases: DrugBank[23], Therapeutic Target Database (TTD)[80], PharmGKB[81] and Open Target Platform[82]. As for the endophenotype MR, we curated drugs with opposite effects against the signs of the MR effect estimates. By contrast, the negative correlation tests directly prioritized candidate compounds. We manually curated supporting evidence for candidate drugs and compounds.

### Overlap enrichment analysis of disease-risk genes in drug-target genes in medication categories. We ran MAGMA[9] and VEGAS2[8] to

summarize variant-level $P$ values into gene level and used the genes with FDR < 0.05 in either MAGMA or VEGAS2 as the disease-risk genes. We then used GREP[22] to perform a series of Fisher's exact tests for the enrichment of the disease-risk genes in the drug-target genes involved in the drug indication categories, Anatomical Therapeutic Chemical Classification System codes.

### Negative correlation tests between genetically determined and compound-regulated gene expression profiles. We nominated the compounds with inverse effects on gene expression against genetically determined gene expression by using Trans-Phar[24]. In brief, genetically determined case–control gene expression was inferred for 44 tissues in the Genotype-Tissue Expression project (v.7)[17] with FOCUS[83], and the genes in the top decile for the absolute value of the $z$-score were used for the following correlation analysis. The Library of Integrated Network-based Cellular Signatures project (LINCS) CMAP L1000 library data[84] were used for the compound library. After matching the tissues in GTEx with the cell lines in the LINCS L1000 library, we performed a series of Spearman's rank correlation tests for 308,872 pairs of genetically determined and compound-perturbed tissue- or cell-type specific gene expression profiles. We prioritized compounds with FDR < 0.1, as we previously showed that the compounds with FDR < 0.1 contained plausible therapeutic targets with literature supports[4].

### Endophenotype MR. To pin-point the disease-causing proteins that were targeted by existing drugs, we performed MR analyses (specifically, a Wald ratio test) by using lead variants in pQTL as instrumental variables and five stroke phenotypes as outcomes: AS, AIS, CES, LAS and SVS. We used the tier 1 lead variants defined in ref. [85] to avoid confounding by horizontal pleiotropy. The tier 1 variants, summarized from five pQTL studies ($n = 997$ to 6,861)[86–90], did not include variants with heterogeneous effect sizes among the studies or with a number of associated proteins of larger than five. We restricted the lead variants to the variants associated with drug-target proteins. For the lead variants of pQTLs that were missing in the stroke GWAS summary statistics, the proxy variants with the largest $r^2$ were used if the $r^2$ was greater than 0.8 (1000 Genomes, European). In total, we used 277 lead variants for 218 drug-target proteins for MR and considered FDR < 0.05 as the threshold to identify significant associations. We used the TwoSampleMR R package[91] for MR analysis. As post-MR quality controls, we performed (1) a directionality check of causal relationships by Steiger filtering[92] and (2) colocalization analysis for the proteins with FDR < 0.05. To examine colocalization assuming multiple causal variants per locus, coloc[74] was applied to the decomposed signals by SuSiE[12] for the variants within 500 kb upstream and downstream of the lead variants (coloc + SuSiE)[93]. If SuSiE did not converge after 10,000 iterations, coloc was used instead. coloc + SuSiE and coloc were run with their respective default parameters. For the two pQTL studies without public summary statistics[86,90], we compared the $r^2$ between the lead variants of the pQTL study and the stroke GWAS. We considered that colocalization occurred when the maximum posterior probability (that is, PP.H4) was greater than 0.75 or $r^2$ was greater than 0.8.

To provide further support for our findings, we conducted MR analyses with two additional recent independent pQTL datasets, using the same methodology and significance thresholds (FDR < 0.05 for MR and PP.H4 > 0.75 for colocalization) as above: one study comprised both plasma ($n = 529$) and cerebrospinal fluid ($n = 835$) pQTL datasets[94], the second is one of the largest plasma pQTL studies conducted in 35,559 Icelandic individuals[95].

### Protective rare variants

For the five genes targeted by inhibitors—VCAM1, F11, KLKB1, LAMC2 and GP1BA—we extracted the associations of rare deleterious variants (MAF < 0.01) with stroke and stroke-related traits from the gene-based burden tests in the whole-exome sequencing data of >450,000 UK

Biobank participants[31]. As stroke and stroke-related traits, we extracted 30 traits belonging to 9 vascular risk factor and disease categories (Supplementary Table 35). We applied Bonferroni correction and the corrected $P$-value threshold was $0.05/5/30 = 3.33 \times 10^{-4}$ (5 and 30 represent the number of tested genes and traits, respectively).

## PheWAS

PheWAS analysis was performed using R (v.4.0.3). We used the PheWAS R package[96] (https://github.com/PheWAS/PheWAS) function create-Phenotypes to translate ICD10 diagnosis codes into phecodes for the PheWAS analysis. We tested the associations between phecodes and genetic variants using logistic regression and adjusting for sex, birth year and ten genotype PCs. We applied Bonferroni correction to select statistically significant associations (number of tested phecodes: 1,809; number of tested SNPs: 8; corrected $P$-value threshold: $0.05/(1,809 \times 8) = 3.45 \times 10^{-6}$). The results were visualized using the PheWAS library. To further characterize the associations of the genetic variants with other phenotypes, we searched for all eight SNPs in the PhenoScanner database[97,98].

## Polygenic risk prediction

We constructed iPGS models for stroke in European and East Asian individuals (Extended Data Fig. 10). For each ancestry, independent datasets were used for model training and evaluation. We used as input summary statistics data of multiple GWAS analyses for stroke outcomes and vascular-risk traits to derive iPGS models. We denote the number of input GWASs as $N$. For each of the $N$ GWAS summary data, 37 candidate single-trait polygenic score (sPGS) models were generated using the P+T[99,100], LDpred[101] and PRScs[102] algorithms with an ancestry-specific linkage-disequilibrium reference panel from the 1000 Genomes Project[103] (Supplementary Methods). The plink (v.1.90b6.8)[104], LDpred (v.1.0.11)[101] and PRScs.py (5 June 2021)[102] programs were used to compute the P+T, LDpred and PRScs models, respectively. Subsequently, among the 37 candidate models, the best sPGS model, which was defined as the model that showed a maximal improvement in AUC over a base model (age, sex and top five PCs were included in the base model), was selected using the model training dataset[5,100]. Then, $N$ best sPGS models were selected from the $N$ input GWASs. Among the $N$ best sPGS models, we retained models that were significantly associated with AIS in the model-training dataset (Bonferroni-corrected $P < 0.05$).

Then, each retained best sPGS was $z$-transformed (zero mean and unit s.d.) over the model-training dataset, followed by elastic-net logistic regression[105] to model the associations between the $N$ sPGS variables and AIS with the adjustments for age, sex and top five genetic PCs. Two regularization parameters ($\alpha$ and $\lambda$) were optimized using tenfold cross-validation. Coefficients (weights) for the retained sPGS models were then determined by elastic-net logistic regression with the optimal regularization parameters, followed by integration of the sPGS models into a single iPGS model according to a formula presented previously[5]. Elastic-net regression was performed using the glmnet R package[106].

The predictive ability of the iPGS model was estimated using the model-evaluation dataset, whereby we evaluated the improvement in $C$-index for a prospective cohort dataset or AUC for a case-control dataset over a base model that includes age, sex and top five genetic PCs.

We used EstBB data for the model training and evaluation of iPGS model in Europeans. The model-training dataset was composed of 1,003 cases of prevalent AIS at the baseline and 8,997 control individuals. The control individuals were randomly selected among EstBB participants who had no history of AS at the baseline and who did not develop AS during the follow-up. The remaining 102,099 EstBB participants were used for the model evaluation (mean ± s.d. age at the baseline, 44.0 ± 15.7 years; 37.8% men). Among the participants in the model-evaluation dataset, 1,128 cases of incident AIS were observed during 4.6 ± 4.8 years. To derive the European iPGS model, we incorporated 5 ancestry-specific and 5 cross-ancestry stroke GWAS analyses (AS, AIS, LAS, SVS and CES)

from the GIGASTROKE project, and 12 GWAS analyses of vascular-risk traits from other groups (Extended Data Fig. 10). To avoid the overlap of participants across datasets, the GWAS summary statistics for stroke outcomes were recalculated for the iPGS analysis by excluding the EstBB from the meta-analysis of GIGASTROKE studies. To enable comparison with a previous European iPGS model based on the MEGASTROKE GWAS[5], we incorporated 12 GWAS analyses of vascular-risk traits (atrial fibrillation, CAD, T2D, SBP, DBP, TC, LDL-C, HDL-C, TG, BMI, height and smoking)[54,56,59–61,107,108] into the GIGASTROKE-based iPGS model. The iPGS model for Europeans was further evaluated in two external cohorts of European ancestry (MVP and pooled data of clinical trials) as well as in two studies of participants with African ancestry (MVP and SIREN).

For the East Asian iPGS model, we used BBJ data for the model training and evaluation. The model-training dataset was composed of 577 cases of AIS and 9,232 control individuals, whereas there were 1,470 cases of AIS and 40,459 control individuals in the model-evaluation dataset. The mean ± s.d. of age at recruitment was 69.2 ± 10.8 years for cases and 66.5 ± 12.5 years for controls in the model evaluation dataset. The percentage of male participants was 70.0% for cases and 53.1% for controls. The two case–control datasets were not included in the meta-analysis of GIGASTROKE studies and, therefore, the overlap of participants across datasets was avoided. To derive the East Asian iPGS model, we incorporated 5 ancestry-specific and 5 cross-ancestry stroke GWAS analyses (AS, AIS, LAS, SVS and CES) from the GIGASTROKE project, and 12 GWAS analyses of vascular-risk traits (Extended Data Fig. 10). The iPGS model for East Asian individuals was further evaluated in an external study of East Asian ancestry (TWB).

## GRS in clinical trials

Participants who had consented for genetic testing and who were of European ancestry from the ENGAGE AF-TIMI 48 (effective anticoagulation with factor Xa next generation in atrial fibrillation)[109], SOLID-TIMI 52 (stabilization of plaques using darapladib)[110], SAVOR-TIMI 53 (saxagliptin assessment of vascular outcomes recorded in patients with diabetes mellitus)[111], PEGASUS-TIMI 54 (prevention of cardiovascular events in patients with prior heart attack using ticagrelor compared to placebo on a background of aspirin)[112] and FOURIER (further cardiovascular outcomes research with PCSK9 inhibition in patients with elevated risk)[113] trials were included in this analysis. Methods for genotyping and imputation have previously been published[35,114] and are summarized in Supplementary Table 2. A set of 58 sentinel variants at stroke-risk loci identified in the IVW meta-analysis was used to calculate a GRS for each trial participant and identify tertiles of genetic risk (Supplementary Table 57). A Cox model was used to estimate HRs for ischaemic stroke associated with the quantitative GRS and across genetic risk groups, adjusted for clinical risk factors (age, sex, hypertension, hyperlipidaemia, diabetes, smoking, CAD, atrial fibrillation and congestive heart failure) and the first five principal components of population stratification. Analyses were conducted primarily in participants of European ancestry ($n = 51,288$, with 960 incident AIS)—with secondary analyses in the much smaller East Asian ($n = 1,312$, with 27 incident AIS) ancestry subset—using the AS cross-ancestry IVW meta-analysis effect estimates as weights for the primary analysis and ancestry-specific, as well as AIS effect estimates for secondary analyses. We also looked separately at associations with incident stroke in participants with and without previous stroke.

## Reporting summary

Further information on research design is available in the Nature Research Reporting Summary linked to this article.

## Data availability

Summary statistics generated by the GIGASTROKE consortium across ancestries and stroke subtypes are available in the GWAS Catalog

(GCST90104534–GCST90104563). The integrated polygenic risk score models of stroke in Europeans and East Asians are available in the PGS Catalog (PGS002724 and PGS002725). Individual level data can be requested directly from the authors of the contributing studies, listed in Supplementary Table 1. Single-nucleus RNA-seq data have been deposited in the SYNAPSE database as part of the Religious Orders Study and Memory and Aging Project (ROSMAP) (https://www.synapse.org) and through the RADC Resource Sharing Hub (https://www.radc.rush.edu). We used publicly available data from GTEx (https://gtexportal.org/home/), the Gusev laboratory (http://gusevlab.org/projects/fusion/), the FinnGen Freeze 7 cohort (https://www.finngen.fi/en/access_results), PhenoScanner v.2 database (http://www.phenoscanner.medschl.cam.ac.uk), pQTL summary statistics (https://doi.org/10.1038/s41588-020-0682-6, http://www.phpc.cam.ac.uk/ceu/proteins/, http://metabolomics.helmholtz-muenchen.de/pgwas/index.php, https://zenodo.org/record/264128), deCODE genetics (https://www.decode.com/summarydata/) and summary statistics using the UK Biobank whole-exome sequencing (https://doi.org/10.1038/s41586-021-04103-z).

## Code availability

The code for computation of the integrated polygenic risk score of stroke are available at GitHub (https://github.com/hacchy1983/iPGS-construction). The drug discovery analysis was conducted using the following publicly available tools: GREP (https://github.com/saori-sakaue/GREP), Trans-Phar (https://github.com/konumat/Trans-Phar), and the TwoSampleMR (https://mrcieu.github.io/TwoSampleMR/), coloc (https://chr1swallace.github.io/coloc/) and susieR (https://stephenslab.github.io/susieR/index.html) R packages.

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

**Acknowledgements** Detailed acknowledgements are provided in the Supplementary Information. We thank the participants and staff of contributing studies. We also thank Michèle M. Sale, who passed away in 2018, for her important contributions to some of the studies included in this manuscript, as principal investigator of one of the grants that funded "The Sea Islands Genetics Network (SIGNET)" (R01 DK084350). She contributed specifically to SIGNET-REGARDS from the REasons for Geographic And Racial Differences in Stroke (REGARDS, U01 NS041588) and served as a key contributor to SiGN, COMPASS and METASTROKE.

**Author contributions** S. Debette, M.D., Y.K., L.M., J.E.H., M.O.O. and C.T.F. jointly supervised the research. S. Debette and M.D. designed and conceived the study. D.C., M.F., M.N., S.N., T. Konuma, Y.O., J.Q.T., R.F.-S., S. Trompet, J.B., T.B., K.W., M.R., Y.-H.J., B.W., S.B., H.L., M.A.N., C.Y., A. Mishra, S.R., J.I.R., M.C., F.K., T.H., Y.S., A.S., G.C., A.K., D. Strbian, Q.Y., F.V., J.L., A.C., N.H., T.J., K.K., L. Lepik, J.C.-M., N.P.T.-A., R.M., M.G., J.E.H., E.Y.-D., M. Shi, Y.H., M. Koido, A. Mishra, Q.L.-G., I.C., M.V.V., R.W., K. Lin, M.J.K., A.L., D.P., G.-V.R., H.-J.B., H.T., J.E.H., J. He, K.-J.L., L.T., L.B., V. Srinivasasainagendra, Y.J.K. and Y.-C.L. contributed to bioinformatics analyses. Y.R., M.B., C.A., D.W., P.R., T. Meitinger, K. Cho, K. Christensen, Yi-Ching Liaw, Yung-Po Liaw, B.N., A.T-H., R.F.-S., J.W.J., M.E., T.B., K.W., M.J., F.-E.d.L., P.L., M.R., L.F., P.F., C.J., K.L.K., H.H., Y.-H.J., C.E.J., J.K., R.V., B.W., S.B., E.C.S., J.-A.Z., H.S.M., N.G., M.C., G.P., M.O'D., N.M., F.K., M. Sasaki, C.R., K.T., M.S.S., K.P., D. Strbian, J.C.H., S.S., A.H., L.L., V.G., N.S., D.-A.T., R.S., T.R., H.A., M.A.I., P.H., K.-G.H., F.M., V.A., R.Z., S.W.-S., M.A.I., S.K., B.M., H.X., J.C., C.-O.S., L.M., J.R., D. Saleheen, R.D.C., J. Hata, J.M.M.H., T.N., T.A., M. Koido, T. Kitazono, S. Tiedt, M.D., C.G., A.P., T. Morisaki, T. Meitinger, M. Kamouchi, Y.K., S. Debette, I.R., M.I., N.A., I.M., A.L., A.K.H., C.C., D.P., D.B., J. He, K.S., L.T., L.B., M.O., M.I.G., N.C.O.-M., P.W., P.D.-J., R.J., R.A., S. Damrauer, S.V., T.T., V. Salomaa, Y.-L.H., P.-H.C., K. Lee and Y.-P.L. contributed samples and phenotyping. S. Debette, M.D., Y.K., L.M., J.E.H., M.O.O., C.T.F., A. Mishra, R.M., T.H., T.J., Y.H., M. Koido, M. Shi, K.K. and S.N. wrote and edited the manuscript. All of the authors provided critical revision.

**Competing interests** C.D.A. has received sponsored research support from Bayer, and has consulted for ApoPharma; T. Konuma is an employee of JAPAN TOBACCO; M. E. reports grants from Bayer and fees paid to the Charité from Abbot, AstraZeneca, Bayer, Boehringer Ingelheim, BMS, Daiichi Sankyo, Amgen, GSK, Sanofi, Covidien, Novartis and Pfizer, all outside the submitted work; B.M.P. serves on the steering committee of the Yale Open Data Access Project funded by Johnson & Johnson; P.A. works with Fondation Alzheimer (nonprofit foundation) and Genoscreen (biotech company); H.L.L.'s participation in this project was part of a competitive contract awarded to Data Tecnica International by the National Institutes of Health to support open science research; M.A.N.'s participation in this project was part of a competitive contract awarded to Data Tecnica International by the National Institutes of Health to support open science research, he also currently serves on the scientific advisory board for Clover Therapeutics and is an advisor to Neuron23; N.A.M. declares institutional research grants to the TIMI Study Group at Brigham and Women's Hospital from Amgen, Pfizer, Ionis, Novartis, AstraZeneca and NIH. The TIMI Study Group has received institutional research grant support through Brigham and Women's Hospital from Abbott, Amgen, Anthos Therapeutics, ARCA Biopharma, AstraZeneca, Daiichi-Sankyo, Eisai, Intarcia, Ionis Pharmaceuticals, MedImmune, Merck, Novartis, Pfizer, Regeneron Pharmaceuticals, Roche, The Medicines Company, Zora Biosciences, Janssen Research and Development, Siemens Healthcare Diagnostics and Softcell Medical; F.K.K. declares that the TIMI Study Group has received institutional research grant support through Brigham and Women's Hospital from Abbott, Amgen, Anthos Therapeutics, ARCA Biopharma, AstraZeneca, Daiichi-Sankyo, Eisai, Intarcia, Ionis Pharmaceuticals, MedImmune, Merck, Novartis, Pfizer, Regeneron Pharmaceuticals, Roche, The Medicines Company and Zora Biosciences; M.S.S. has consultancies with Althera, Amgen, Anthos Therapeutics, AstraZeneca, Beren Therapeutics, Bristol-Myers Squibb, DalCor, Dr. Reddy's Laboratories, Fibrogen, IFM Therapeutics, Intarcia, Merck, Moderna, Novo Nordisk and Silence Therapeutics, and research grant support through Brigham and Women's Hospital from Abbott, Amgen, Anthos Therapeutics, AstraZeneca, Bayer, Daiichi-Sankyo, Eisai, Intarcia, Ionis, Medicines Company, MedImmune, Merck, Novartis, Pfizer, Quark Pharmaceuticals; C.T.R.

has consultancies with Anthos, Bayer, Bristol Myers Squibb, Boehringer Ingelheim, Daiichi Sankyo, Janssen and Pfizer, institutional research grant to the TIMI Study Group at Brigham and Women's Hospital from Anthos, AstraZeneca, Boehringer Ingelheim, Daiichi Sankyo, Janssen, National Institutes of Health and Novartis, and consultancies with Anthos, Bayer, Bristol Myers Squibb, Boehringer Ingelheim, Daiichi Sankyo, Janssen and Pfizer. T.H. receives personal fees from Genome Analytics Japan; J.C.H. is supported by a personal fellowship from the British Heart Foundation (FS/14/55/30806), and acknowledges additional support from the Nuffield Department of Population Health (NDPH), University of Oxford, the British Heart Foundation Centre for Research Excellence, Oxford, and the Oxford Biomedical Research Centre. J.C.H. holds steering committee and Data and Safety Monitoring Board (DSMB) positions for various cardiovascular randomized controlled trials, and is a principal investigator/co-principal investigator of research grants from industry related to cardiovascular clinical trials and observational studies that are governed by University of Oxford contracts that protect personal independence. NDP.H also has a staff policy of not taking personal payments from industry (further details can be found online; https://www.ndph.ox.ac.uk/files/about/ndph-independence-of-research-policy-jun-20.pdf/@@download); S.S. has consultancies with Biogen; P.U.H. reports grants from German Ministry of Research and Education, during the conduct of the study, research grants from the German Ministry of Research and Education, European Union, Charité–Universitätsmedizin Berlin, Berlin Chamber of Physicians, German Parkinson Society, University Hospital Würzburg, Robert Koch Institute, German Heart Foundation, Federal Joint Committee (G-BA) within the Innovationfond, German Research Foundation, Bavarian State (ministry for science and the arts), German Cancer Aid, Charité–Universitätsmedizin Berlin (within Mondafis; supported by an unrestricted research grant to the Charité from Bayer), University Göttingen (within FIND-AF randomized; supported by an unrestricted research grant to the University Göttingen from Boehringer- Ingelheim), University Hospital Heidelberg (within RASUNOA-prime; supported by an unrestricted research grant to the University Hospital Heidelberg from Bayer, BMS, Boehringer-Ingelheim and Daiichi Sankyo), outside the submitted work; K.G.H. reports a study grant by Bayer, lecture fees/advisory board fees from Abbott, Alexion, AMARIN, AstraZeneca, Bayer, Biotronik, Boehringer Ingelheim, Bristol-Myers-Squibb, Daiichi Sankyo, Edwards Lifesciences, Medtronic, Pfizer, Premier Research, SUN Pharma and W. L. Gore & Associates; H.J.G. has received travel grants and speakers honoraria from Fresenius Medical Care, Neuraxpharm, Servier and Janssen Cilag as well as research funding from Fresenius Medical Care; J.M.M.H. is full time employee of Novo Nordisk; E.Y.-D. is full-time employee of Novo Nordisk. S. Damrauer receives research support from RenalytixAI and personal consulting fees from Calico Labs, outside the scope of the current research. H.B. reports grants from AstraZeneca, AstraZeneca Korea, Bayer Korea, Boehringer Ingelheim Korea, Boryung Pharmaceutical, Bristol Myers Squibb, Bristol Myers Squibb Korea, Chong Gun Dang Pharmaceutical, Daiichi Sankyo, Daiichi Sankyo Korea, Dong-A ST, Esai, Jeil Pharmaceutical, JLK, Korean Drug, SAMJIN Pharm., Servier Korea, Shinpoong Pharm., Shire International and Yuhan Corporation, and personal fees from Amgen Korea, Esai Korea, Otsuka Korea, Takeda Korea and Viatris Korea outside the submitted work. C.C. has received research support from GSK. The funders of the study had no role in the collection, analysis or interpretation of data, in the writing of the report or in the decision to submit the paper for publication. C.C. is a member of the advisory board of Vivid genetics; F.A.M. is supported by the German Research Foundation (Deutsche Forschungsgemeinschaft (DFG) within the UNION-CVD Clinician-Scientist Programme (project number 413657723) and has been previously supported by a MD/PhD Fellowship of the Interdisciplinary Center for Clinical Research, University Hospital Würzburg. A. Mishra, R.M., T.J., S.N., D.C.P., M. Koido, Q.L.G., M. Shi, Y.H., M.K.G., I.C., K.K., Yi-Ching Liaw, F.C.V., K. Lin., B.S.W., V. Srinivasasainagendra, L.P., G.C., M.R.C., L.T., R.A., G.V.R., N.H., Y.H.J., J.Q.T., V.A., J.C., M.N., C.Y., E.Y., M.J.K., A.J.L., R.L., T.A., N.D.A., M.K.B., T.M.B., D.A.B., J.C.B., C.B., S.B., A.C., P.M.R., K. Cho, Z.C., J.W.C., P.L.d., R.d.C., M.E., L.E.F., M.I.G., N.C.G., V.G., J. Hata, J. He., A.K.H., Y. Ho., A.S.H., H.I.H., M.I., M.A.J., C.E.J., C.J., M. Kamouchi, K.L.K., T. Kitazono, S.J.K., A.K., P.L., L.J.L., K. Lee, K. Lepik, J.L., L.L., A. Manichaikul, H.S.M., T. Meitinger, B.D.M., T. Morisaki, T.H.M., B.G.N., M.J.O., Y.O., N.C.O., B.O., A.P., S.S.R., J.R., M.S.S., R.L.S., D. Saleheen, E.C.S., V. Salomaa, M. Sasaki, C.L.S., C.O.S., A.S., N.L.S., K.S., Y.S., Y.V.S., K.T., S. Tiedt, T.T., N.P.T., H.K.T., D.T., S. Trompet, A.M.T., A.T., M.v.V., R.V., S.S.V., K.L.W., P.W., D.W., P.W.W., H.X., Q.Y., K.Y., I.Y.M., C.G., T.N., J.W.J., I.L.R., D. Strbian, Y.J.K., P.C., E.M., M.R.I., H.A., S.W., K. Christensen, M.A.I., T.R., B.B.W., G.M.L., M.R., E.M.S., J.K., P.H.F., R.Z., K.P., R.F., F.d.L., T.L., Y.M.R., W.T.L., K.J.J., L.B., G.P., D.I.C., J.I.R., J.Z., T.J.N., M.F., Yung-Po Liaw, I.F., R.G.W., M.O.O., J.E.H., L.M., Y.K., M.D. and S. Debette declare no competing interests.

**Additional information**

**Correspondence and requests for materials** should be addressed to Martin Dichgans or Stephanie Debette.

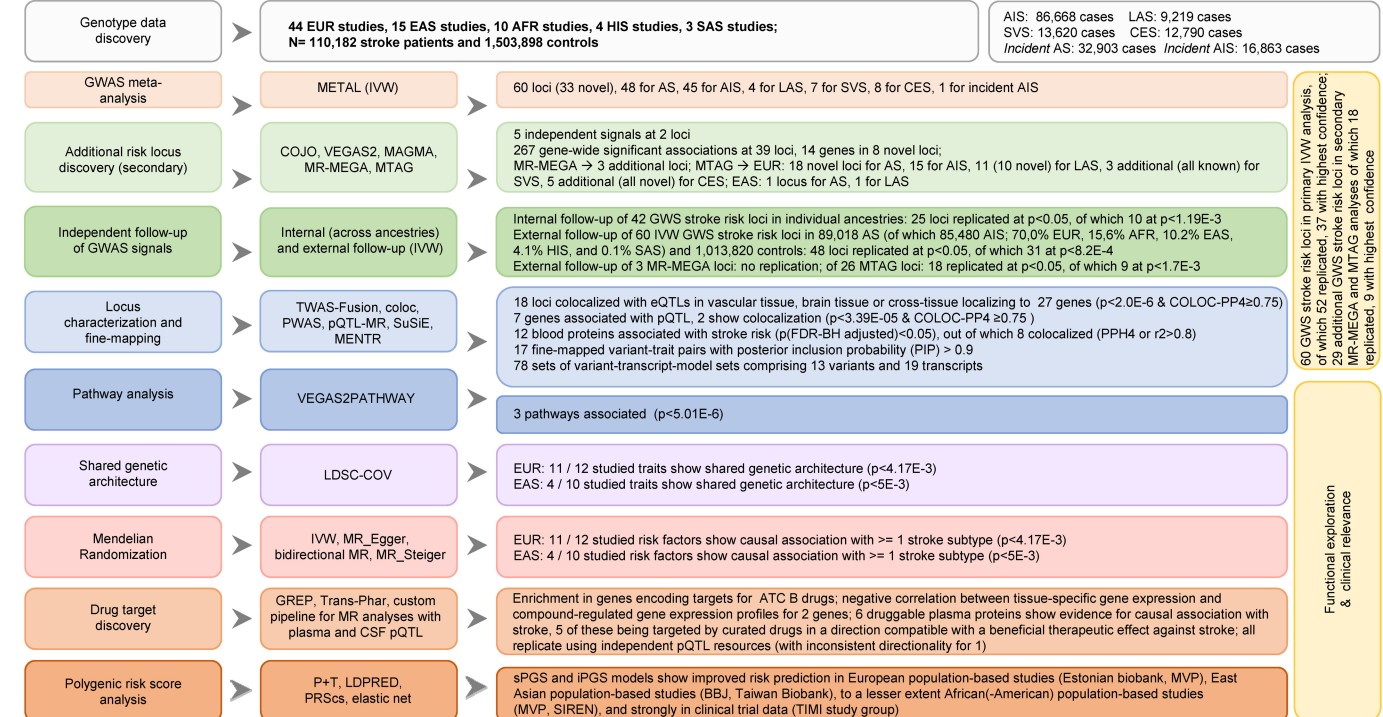

**Extended Data Fig. 1 | GIGASTROKE study workflow.** Study workflow and rationale. EUR: European; EAS: East-Asian; AFR: African; HIS: Hispanic; SAS: South Asian; AS: any stroke; AIS: any ischaemic stroke; LAS: large artery stroke; CES: cardioembolic stroke; SVS: small vessel stroke; GWAS: genome-wide association study; IVW: inverse-variance weighted; MR-MEGA: meta-regression of multi-ethnic genetic association; COJO: conditional and joint analysis; VEGAS2: versatile gene-based association study 2; MTAG: multi-trait analysis of GWAS; TWAS: Transcriptome-wide association study; coloc: Colocalisation Test; PWAS: Proteome-wide association studies; pQTL-MR: protein quantitative trait loci Mendelian Randomization; SuSiE: sum of single effects model; MENTR: Mutation Effect prediction on Non-coding RNA TRanscription; PIP: posterior probability; FDR: false discovery rate; LDSC-COV: covariate-adjusted LD score regression; MR-Egger: Mendelian randomization-Egger; GREP: genome for REPositioning drugs; ATC: Anatomical Therapeutic Chemical; P+T: pruning and thresholding; PRScs: polygenic risk score under continuous shrinkage; BBJ: Biobank Japan; TIMI: thrombolysis in myocardial infarction; MVP: Million Veteran Program; SIREN: Stroke Investigative Research and Educational Network.

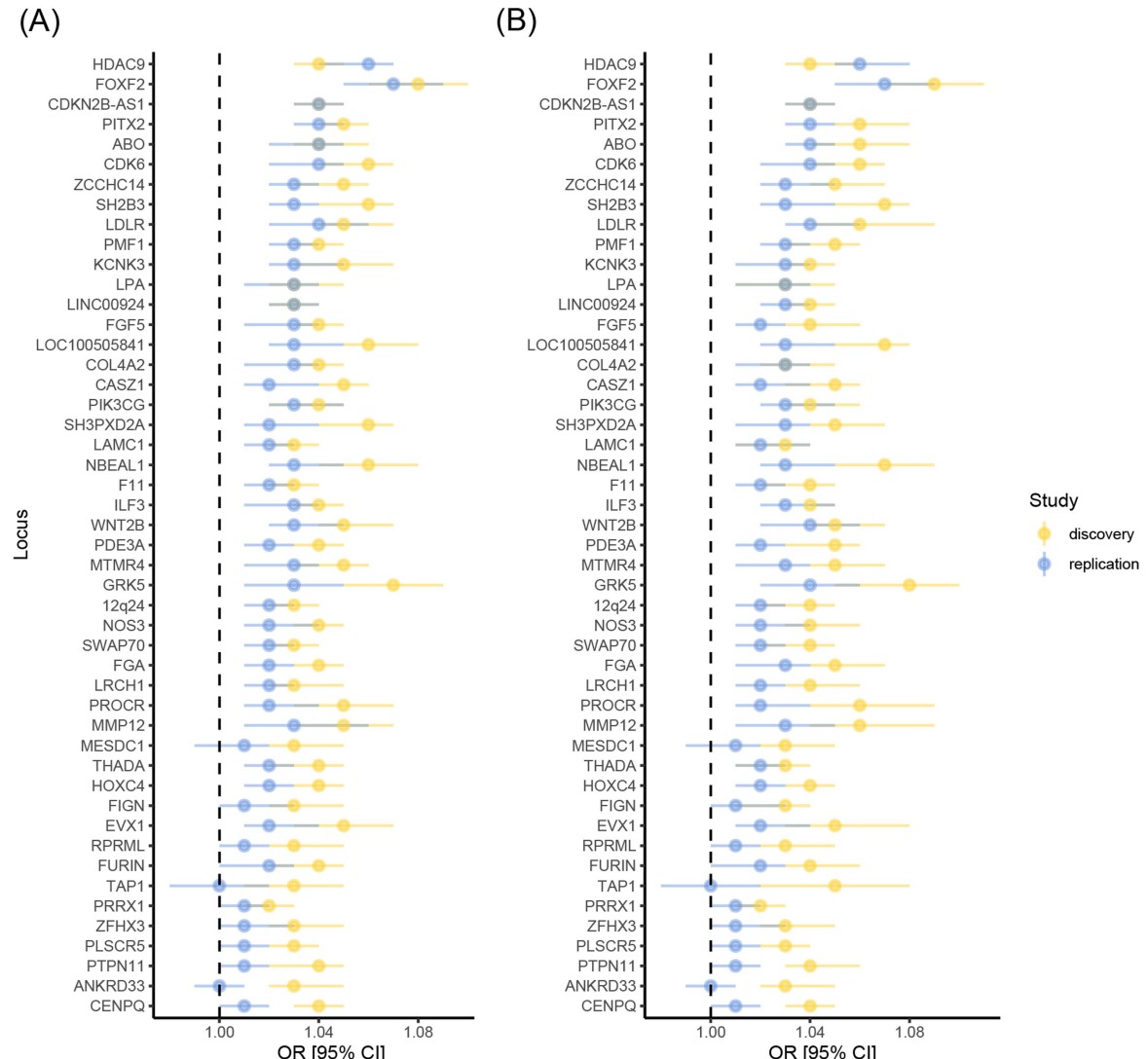

**Extended Data Fig. 2 | Graphical representation of replication results.** Shown are effect sizes and 95% confidence intervals for the 48 replicated IVW loci for (A) cross-ancestry discovery and replication association results for any stroke and (B) cross-ancestry discovery and replication association results for any ischaemic stroke. OR = odds ratio; 95% CI = 95% confidence interval.

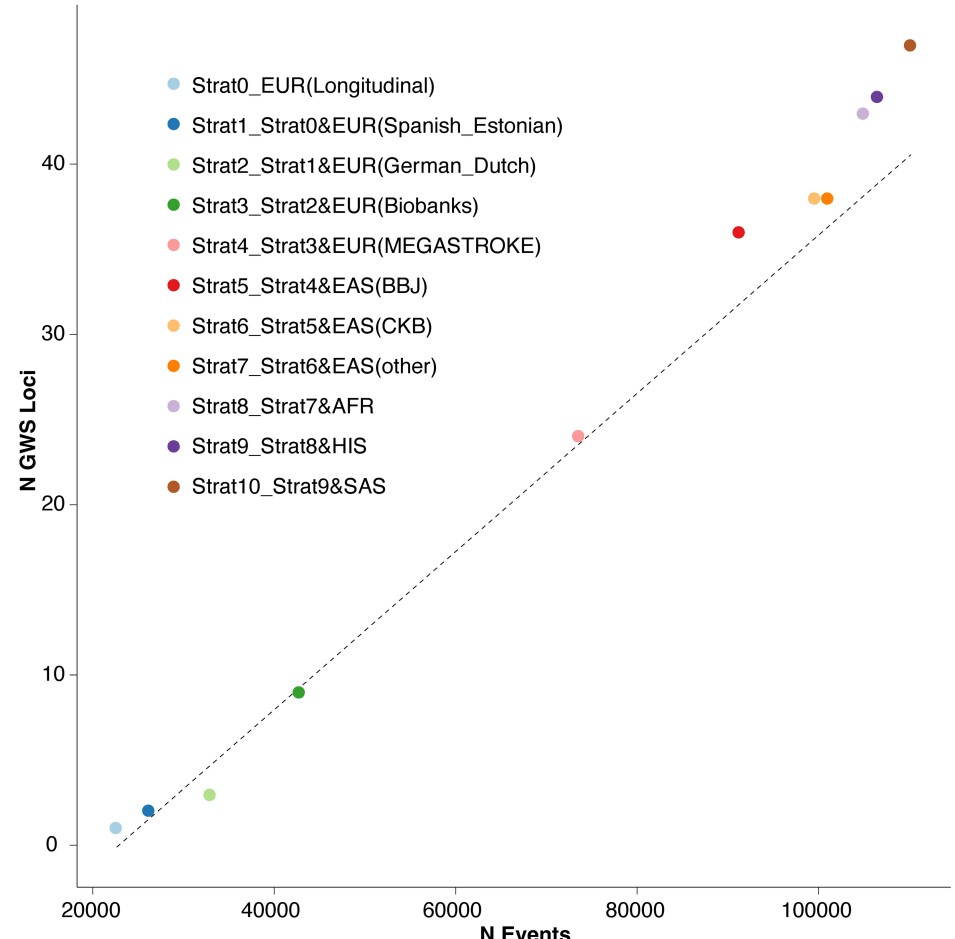

**Extended Data Fig. 3 | Increase in power with increase in population diversity.** The scatter plot shows the number of loci identified with incremental increase in sample size and population diversity. The diagonal line reflects the increase in number of genome-wide significant loci with increasing sample size of European ancestry only. The increase in number of loci departs from this line when adding non-European ancestry samples; EUR: European; EAS: East Asian; AFR: African; HIS: Hispanic; SAS: South Asian; BBJ: Biobank Japan; CKB: China Kadoorie Biobank. Strat0_EUR: N = 22,634 stroke cases (European population-based longitudinal cohorts); Strat1_Strat0&EUR: N = 26,253 stroke cases, adding Spanish and Estonian samples; Strat2_Strat1&EUR: N = 32,980 stroke cases, adding German and Dutch samples; Strat3_Strat2&EUR: N = 42,783, adding large European biobanks; Strat4_Strat3&EUR: N = 73,652 stroke cases, adding MEGASTROKE European; Strat5_Strat4&EAS: N = 91,303 stroke cases, adding Japanese BBJ; Strat6_Strat5&EAS: N = 99,661 stroke cases, adding Chinese CKB; Strat7_Strat6&EAS: N = 101,065 stroke cases, adding other East Asian samples; Strat8_Strat7&AFR: N = 105,026 stroke cases, adding African ancestry samples; Strat9_Strat8&HIS: N = 106,542 stroke cases, adding South American ancestry samples; Strat10_Strat9&SAS, N = 110,182 stroke cases, adding South Asian ancestry samples.

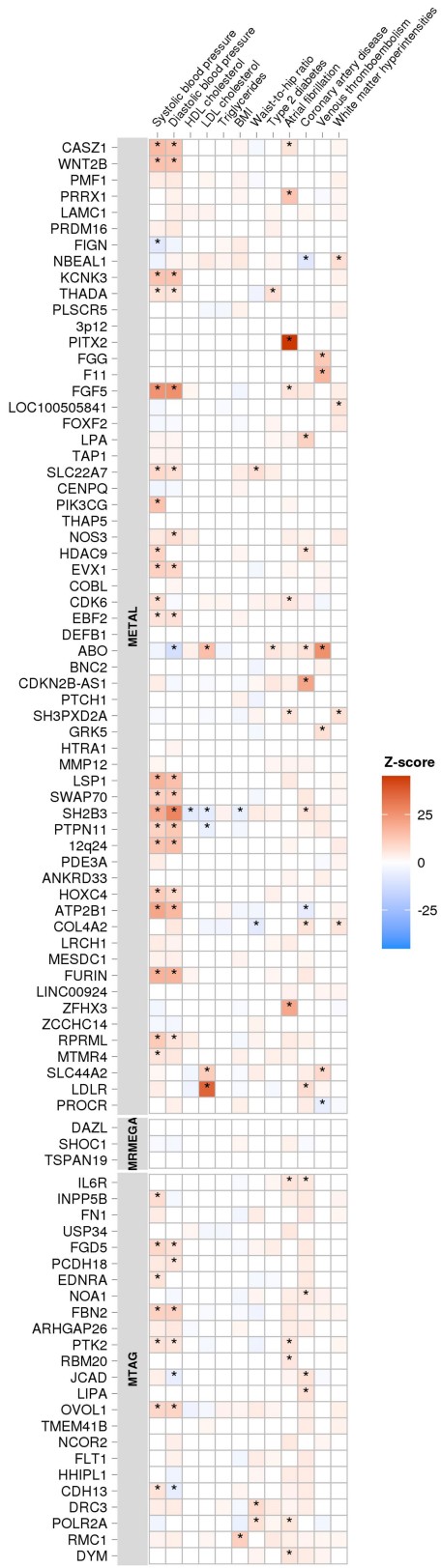

**Extended Data Fig. 4 | Association of stroke risk variants with vascular risk traits.** We report only associations for which the stroke lead variant of a proxy in very high LD ($r^2 > 0.9$) showed genome-wide significant association with the vascular risk trait in a prior GWAS. Colours represent the Z-scores of association of stroke risk increasing alleles with the trait.

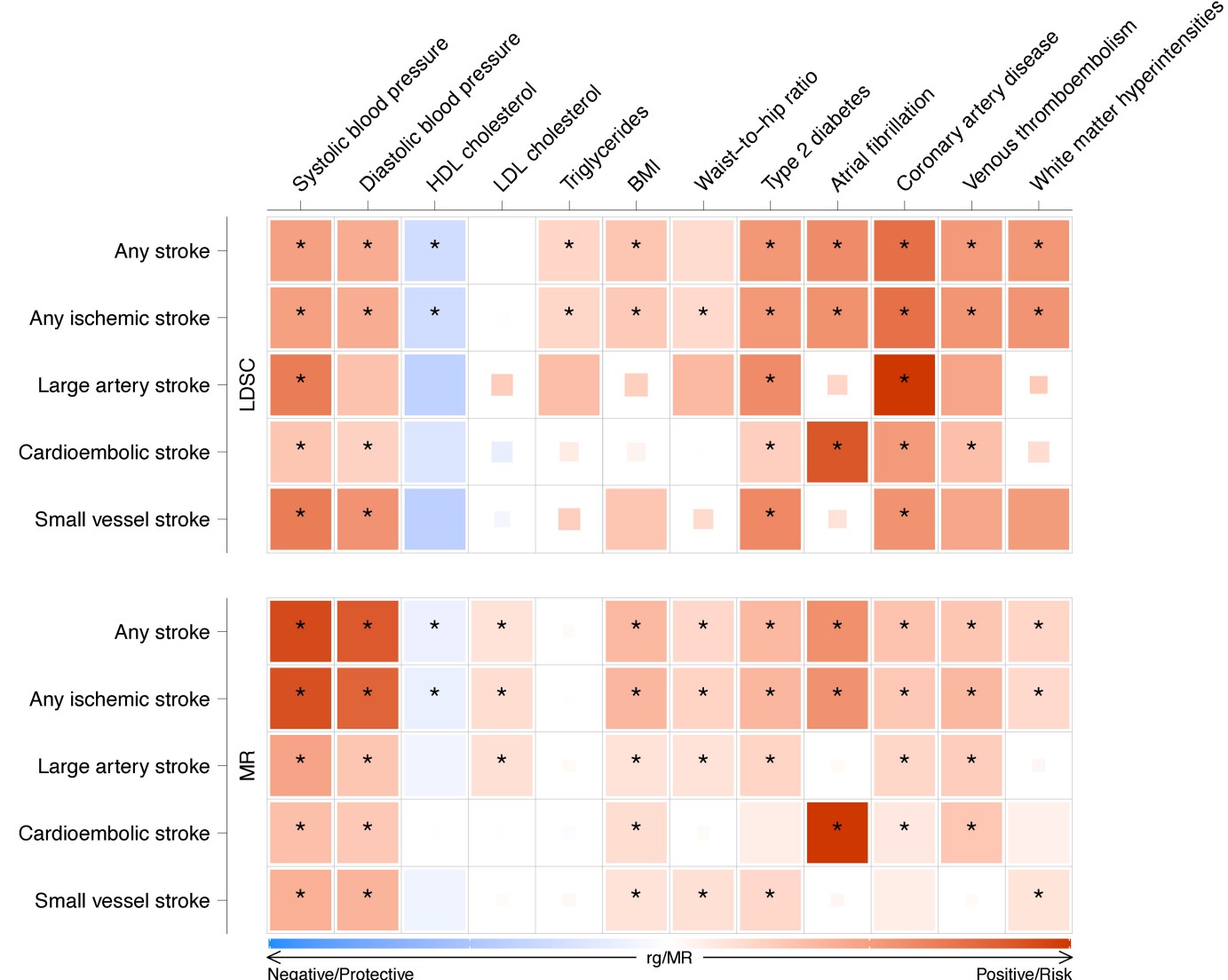

**Extended Data Fig. 5 | Genetic correlations and Mendelian randomization causal estimates of 12 vascular risk factors and disease traits with stroke (any and stroke subtypes), in European ancestry participants.** Larger squares correspond to more significant P-values, with genetic correlations or Mendelian randomization (MR) causal estimates (expressed in Z-scores) significantly different from zero at a P < 0.05 shown as a full-sized square. Genetic correlations or causal estimates that are significant after multiple testing correction (P < 4.17 × 10$^{-3}$) are marked with an asterisk. Two-sided P-values were calculated using LD score regression (LDSC) for genetic correlations and inverse variance weighted analysis for MR.

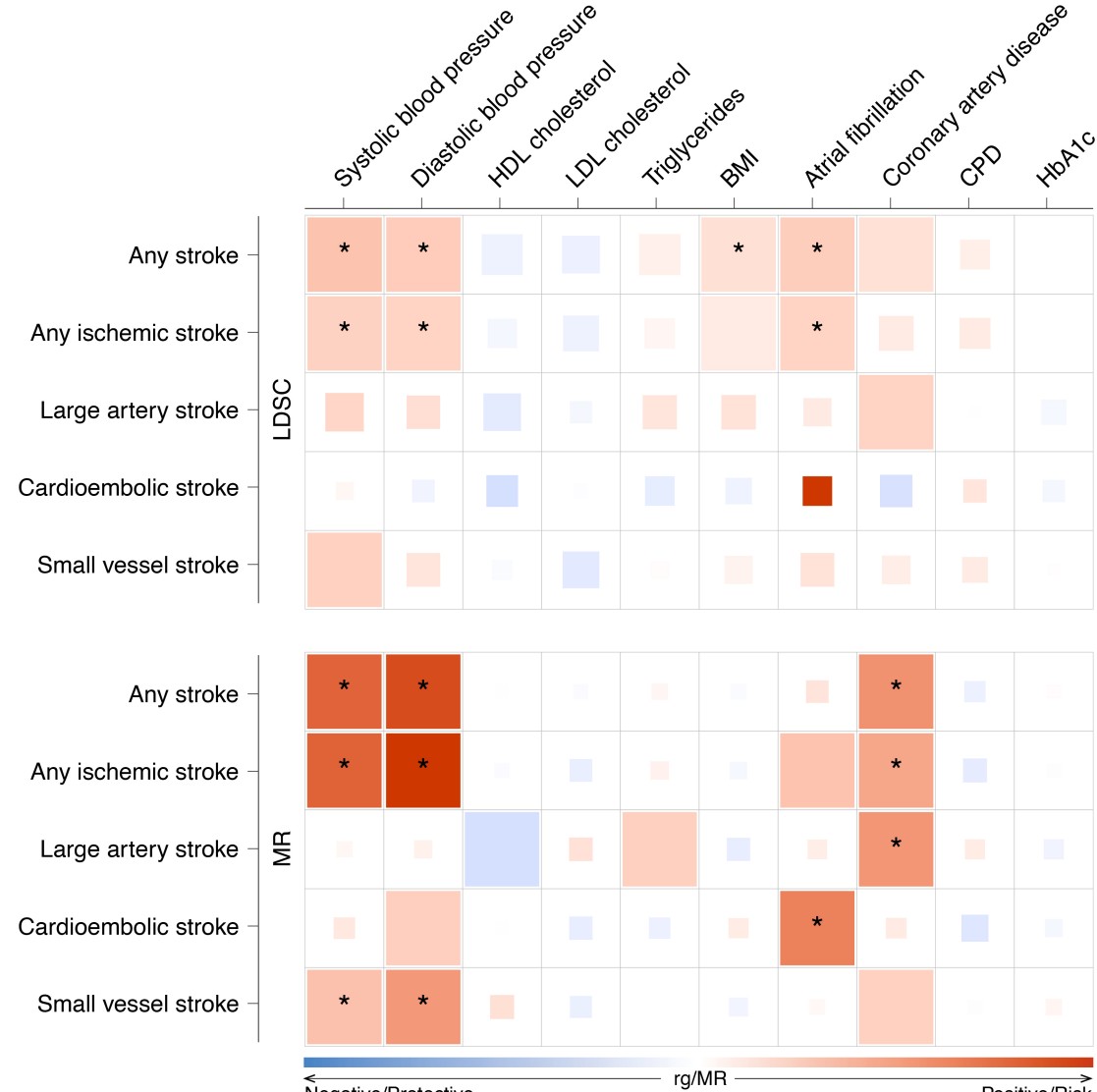

**Extended Data Fig. 6 | Genetic correlations and Mendelian randomization causal estimates of 10 vascular risk factors and disease traits with stroke (any and stroke subtypes), in East Asian ancestry participants.** Larger squares correspond to more significant P-values, with genetic correlations or Mendelian randomization (MR) causal estimates significantly different from zero at a P < 0.05 shown as a full-sized square. Genetic correlations or causal estimates (expressed in Z-scores) that are significant after multiple testing correction (P < 5 × 10⁻³) are marked with an asterisk. Two-sided P-values were calculated using LD score regression (LDSC) for genetic correlations and inverse variance weighted analysis for MR. CPD: cigarettes per day.

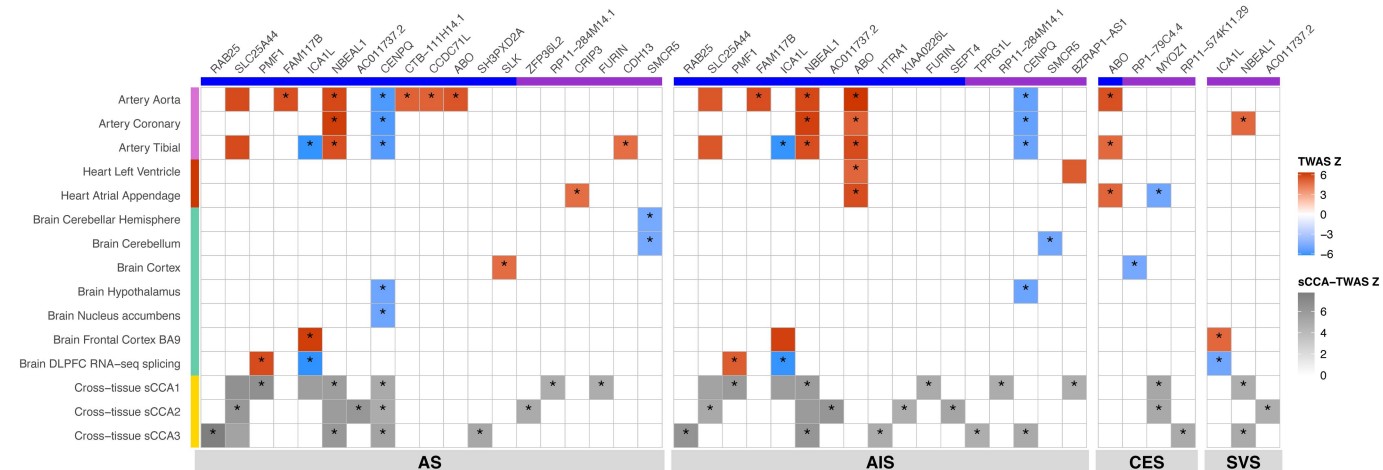

**Extended Data Fig. 7 | Transcriptome-wide association study of stroke in multiple tissues.** Heatmap of the transcriptome-wide association studies (TWAS) of stroke (any stroke and stroke subtypes) showing transcriptome-wide significant associations with supporting evidence from colocalization; Coloured squares are TWAS significant associations based on two-sided p-values after multiple testing correction (p < 2.0 × 10⁻⁶); * Conditionally significant (p < 0.05) and COLOC PP4 ≥ 0.75; Genes are presented on the x-axis, those underlined in blue are in a stroke GWAS locus, those underlined in purple are not within a genome-wide significant stroke risk locus (Methods); Tissue types are on the y-axis (blue: cross-tissue weights; pink: arterial; orange: heart; green: brain).

(A) Single-cell gene expression data of TWAS-COLOC genes in dorsolateral prefrontal cortex (ROS-MAP study)

(C) Cell-type enrichment in human and mouse single cell RNA-seq databases using STEAP

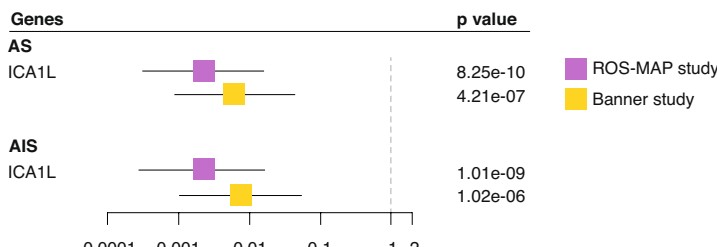

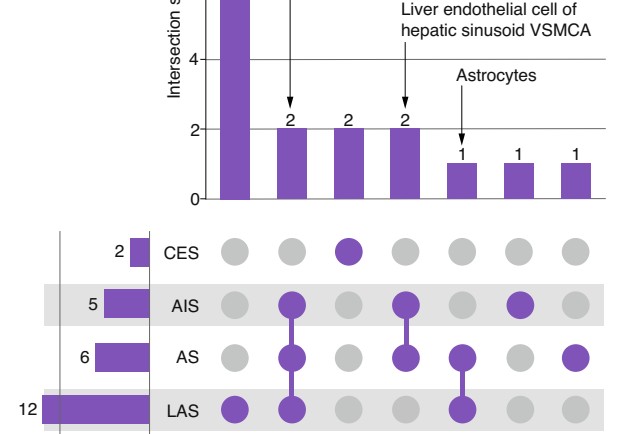

(B) Proteome-wide association study (PWAS) of stroke in brain tissue

**Extended Data Fig. 8 | Single-nucleus gene expression/enrichment analysis and proteome-wide association study (PWAS) of stroke in brain tissue.**
(A) Single-nucleus gene expression data of TWAS-COLOC genes in dorsolateral prefrontal cortex (ROS-MAP study)[21]; Dot plot of the mean expression level in expressing cells (colour) and percent of expressing cells (circle size) of selected genes across different cell types; (B) Proteome-wide association study (PWAS) of stroke in brain tissue; Box plot showing effect estimates (odds ratio) for associations of pQTL of ICA1L in the ROS-MAP (N = 376 independent samples) and Banner (N = 152 independent samples) studies with any stroke (AS) and any ischaemic stroke (AIS), identified in PWAS after multiple testing correction. Odds ratios ± 95% CIs are shown. Dashed line indicates an odds ratio of 1. Two-sided p-values were computed using the TWAS-COLOC approach.

(C) Cell-type enrichment in human and mouse single cell RNA-seq databases using STEAP; the UpSet plot displays the number of significant enrichment results, by stroke subtype (horizontally; 2 for CES, 5 for AIS, 6 for AS, and 12 for LAS) and by cell subtype (vertically; 2 cell-types show significant enrichment in LAS, AIS, and AS, 2 cell-types in AIS and AS, and 1 cell-type in LAS and AS, while 9 cell-types show significant enrichment in LAS only, 2 in CES only and 1 in AS and AIS respectively); details are displayed in Supplementary Table 29. AS: any stroke; AIS: any ischaemic stroke; LAS: large artery stroke; CES: cardioembolic stroke; VLMC: vascular and leptomeningeal cells, OPC: oligodendrocyte progenitor cells; SMC: smooth muscle cells; VSMCA: vascular smooth muscle cells, arterial.

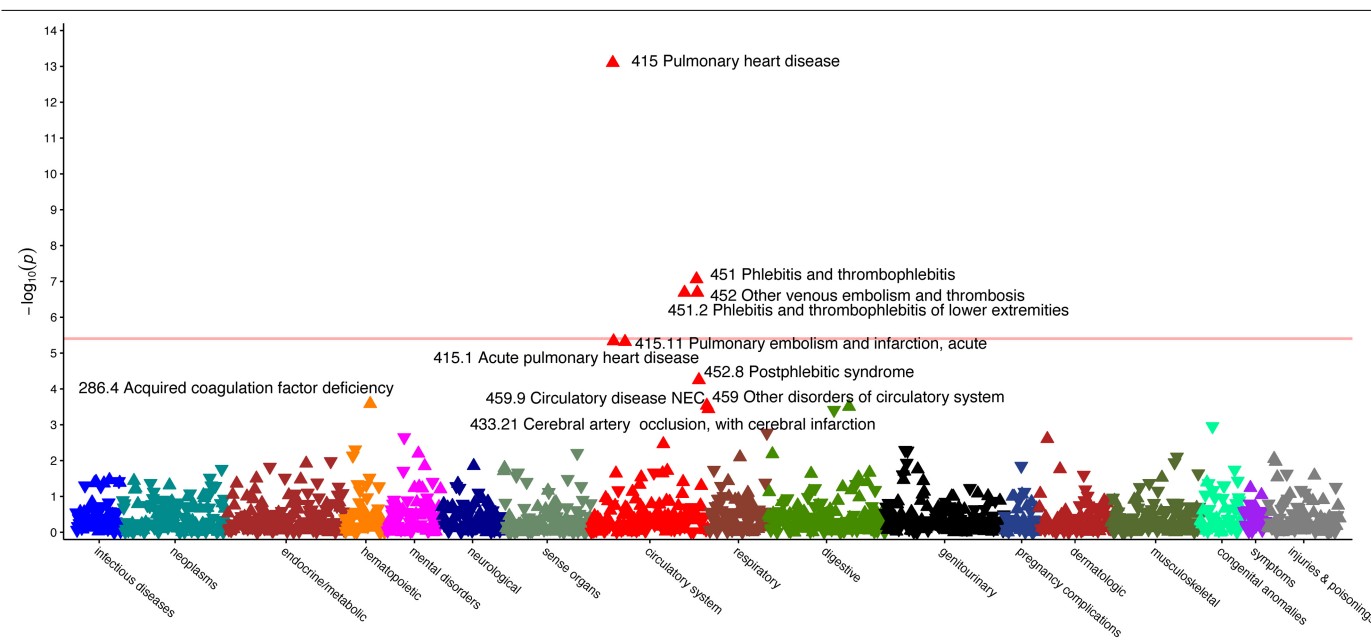

**Extended Data Fig. 9 | Drug target pQTL PheWAS.** PheWAS in Estonian biobank for rs2289252, a cis-pQTL of F11. Each triangle in the plot represents one Phecode and the direction of the triangle represents direction of effect. Two-sided P-values were calculated using logistic regression to test association between the pQTL and Phecodes (p = 3.45 × 10⁻⁶ for phenome-wide significance).

**(A) Derivation of standard PGS models**

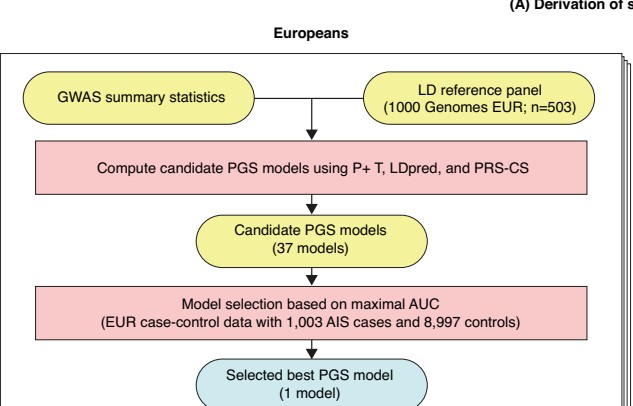
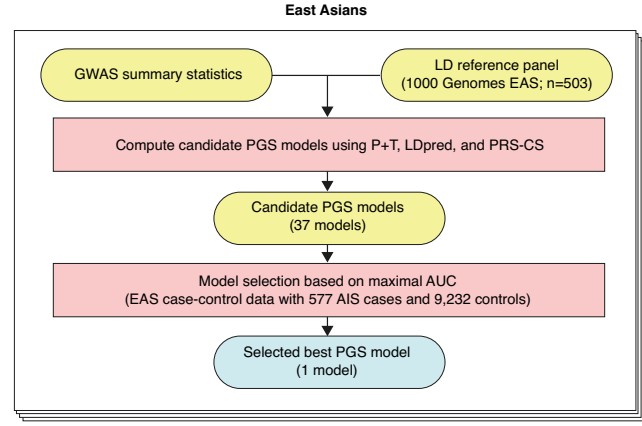

**(B) Derivation and evaluation of integrative PGS models**

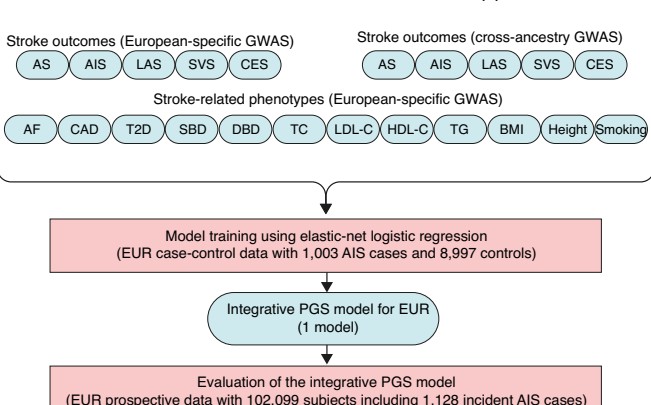
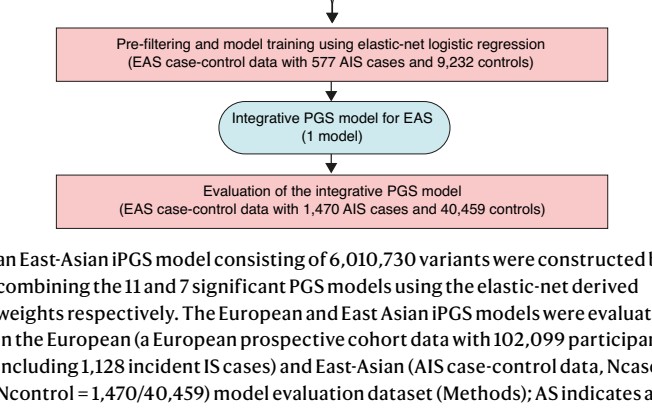

**Extended Data Fig. 10 | Derivation and evaluation of integrative polygenic score models for Europeans and East Asians.** (A) With summary statistics of 22 GWAS (10 GIGASTROKE and 12 on vascular risk factors) and linkage disequilibrium reference data of 1000 Genomes Europeans (n = 503) and East Asians (n = 504), we computed 37 candidate PGS models using P+T, LDpred, and PRScs algorithms. For each GWAS, the best PGS model was selected based on the maximal area under the curve (AUC) values in the training dataset of Europeans (any ischaemic stroke [AIS] case-control data, Ncases/Ncontrols = 1,003/8,997) and East Asians (AIS case-control data, Ncases/Ncontrols = 577/9,232). Out of 22 selected PGS models derived from the 22 GWAS, 11 and 7 were significantly associated with AIS in the European and East Asian training dataset respectively (Bonferroni-corrected P < 0.05). (B) The significant PGS models were used as the variables for elastic-net logistic regression and the weights for the variables were trained using the model training dataset. The European iPGS model consisting of 1,213,574 variants and an East-Asian iPGS model consisting of 6,010,730 variants were constructed by combining the 11 and 7 significant PGS models using the elastic-net derived weights respectively. The European and East Asian iPGS models were evaluated in the European (a European prospective cohort data with 102,099 participants including 1,128 incident IS cases) and East-Asian (AIS case-control data, Ncases/Ncontrol = 1,470/40,459) model evaluation dataset (Methods); AS indicates any stroke; AIS, any ischaemic stroke; LAS, large artery stroke; SVS, small vessel stroke; CES, cardioembolic stroke; AF, atrial fibrillation; CAD, coronary artery disease; T2D, type 2 diabetes; SBP, systolic blood pressure; DBP, diastolic blood pressure; TC, total cholesterol; LDL-C, low-density lipoprotein cholesterol; HDL-C, high-density lipoprotein cholesterol; TG, triglyceride; BMI, body mass index; AUC indicates area under the curve; EUR, European; EAS, East Asian; GWAS, genome-wide association study; LD, linkage disequilibrium; PGS, polygenic score.

# Reporting Summary

## Statistics

For all statistical analyses, confirm that the following items are present in the figure legend, table legend, main text, or Methods section.

| n/a | Confirmed | |
|---|---|---|
| ☐ | ☒ | The exact sample size ($n$) for each experimental group/condition, given as a discrete number and unit of measurement |
| ☐ | ☒ | A statement on whether measurements were taken from distinct samples or whether the same sample was measured repeatedly |
| ☐ | ☒ | The statistical test(s) used AND whether they are one- or two-sided<br>*Only common tests should be described solely by name; describe more complex techniques in the Methods section.* |
| ☐ | ☒ | A description of all covariates tested |
| ☐ | ☒ | A description of any assumptions or corrections, such as tests of normality and adjustment for multiple comparisons |
| ☐ | ☒ | A full description of the statistical parameters including central tendency (e.g. means) or other basic estimates (e.g. regression coefficient) AND variation (e.g. standard deviation) or associated estimates of uncertainty (e.g. confidence intervals) |
| ☐ | ☒ | For null hypothesis testing, the test statistic (e.g. $F$, $t$, $r$) with confidence intervals, effect sizes, degrees of freedom and $P$ value noted<br>*Give P values as exact values whenever suitable.* |
| ☒ | ☐ | For Bayesian analysis, information on the choice of priors and Markov chain Monte Carlo settings |
| ☒ | ☐ | For hierarchical and complex designs, identification of the appropriate level for tests and full reporting of outcomes |
| ☐ | ☒ | Estimates of effect sizes (e.g. Cohen's $d$, Pearson's $r$), indicating how they were calculated |

*Our web collection on statistics for biologists contains articles on many of the points above.*

## Software and code

Policy information about availability of computer code

| Data collection | No software was used for data collection |
|---|---|
| Data analysis | Each individual study that contributed genetic-phenotype association summary statistics to the consortium carried out their association analyses independently of the consortium (study-specific information outlined in Supplementary Table 2). Please refer Supplementary Table 2 for the software used for quality control, imputation and GWAS analyses by individual studies. Moreover we used publicaly available METAL v2020-05-05, MR-MEGA v.0.1.6 and MTAG v1.0.8 software for GWAS meta-analysis. For post GWAS analysis we used VEGASv2, MAGMA v1.08, cov-LDSC v1.0.0, GCTA-COJO v1.26.0, susieR, R v4.1.1 (MendelianRandomization package), MENTR v1, R v4.1.1 (TWAS-Fusion package), STEAP pipeline v1, GREP v1.0.0, Trans-Phar v1, R v4.0.3 (PheWAS package), plink v1.90b6.8, LDpred v.1.0.11, PRScs.py v2021-01-04. These are publicaly available software, original manuscript of these software was cited whenever these software were menioned in the manuscript |

For manuscripts utilizing custom algorithms or software that are central to the research but not yet described in published literature, software must be made available to editors and reviewers. We strongly encourage code deposition in a community repository (e.g. GitHub). See the Nature Portfolio guidelines for submitting code & software for further information.

## Data

Policy information about availability of data

All manuscripts must include a data availability statement. This statement should provide the following information, where applicable:
- Accession codes, unique identifiers, or web links for publicly available datasets
- A description of any restrictions on data availability
- For clinical datasets or third party data, please ensure that the statement adheres to our policy

Summary statistics generated by the GIGASTROKE consortium across ancestries and stroke subtypes are available in the GWAS Catalog (study code GCST90104534-

GCST90104563)
The integrated polygenic risk score models of stroke in Europeans and East Asians are available in the PGS Catalog (PGS002724 and PGS002725). Individual level data can be requested directly from contributing studies, listed in Supplementary Table 1. Single nucleus RNA-seq (snRNA-seq) data is deposited in the SYNAPSE database as part of the Religious Orders Study and Memory and Aging Project (ROSMAP) (https://www.synapse.org) and through the RADC Resource Sharing Hub (https://www.radc.rush.edu). We used publicly available data from GTEx (https://gtexportal.org/home/), the Gusev lab (http://gusevlab.org/projects/fusion/), the FinnGen Freeze 7 cohort (https://www.finngen.fi/en/access_results), PhenoScanner v2 database (http://www.phenoscanner.medschl.cam.ac.uk), the pQTL summary statistics (https://doi.org/10.1038/s41588-020-0682-6, http://www.phpc.cam.ac.uk/ceu/proteins/, http://metabolomics.helmholtz-muenchen.de/pgwas/index.php, https://zenodo.org/record/264128), the deCODE genetics (https://www.decode.com/summarydata/), the summary statistics using the UK Biobank whole-exome sequencing (https://doi.org/10.1038/s41586-021-04103-z).

# Field-specific reporting

Please select the one below that is the best fit for your research. If you are not sure, read the appropriate sections before making your selection.

☒ Life sciences ☐ Behavioural & social sciences ☐ Ecological, evolutionary & environmental sciences

For a reference copy of the document with all sections, see nature.com/documents/nr-reporting-summary-flat.pdf

# Life sciences study design

All studies must disclose on these points even when the disclosure is negative.

| | |
|---|---|
| Sample size | We performed meta-analysis of GWAS on 29 population-based cohorts or biobanks with incident stroke ascertainment and 25 clinic-based case-control studies, comprising up to 110,182 stroke patients and 1,503,898 controls. We also gathered an independent dataset of 89,084 any stroke cases and 1,013,843 controls, mostly from large biobanks, for external replication. We included clinic-based studies with minimum of n=100 cases and n=100 controls, while we were more inclusive for population-based cohorts with longitudinal information on incident stroke and considered all population-based studies with more than 20 incident stroke cases. No statistical calculation for adequate sample size was performed, but the results identifying multiple genomic regions at genome-wide significance threshold indicates adequate power for genetic discovery. |
| Data exclusions | Individual level phenotype and genotype data exclusions were performed by each individual study, described in supplementary appendix. However in population-based longitudinal cohorts we considered well ascertained incident stroke cases only, the self-reported stroke cases at baseline were excluded from this analysis. |
| Replication | To verify the reproducibility of our findings, first, we replicated 42 loci discovered in one ancestry into the internal data of other ancestries (internal cross-ancestry validation). We successfully replicated 10 out of 42 loci after accounting for the number of loci tested, of which 7 were genome-wide significant in European (EUR), 1 in East Asian (EAS), and 2 in both (EUR) and (EAS). Additional 15 loci showed nominal association (p<0.05) in at least one other ancestry (Supplementary Table 15). Second, we also gathered an independent dataset of 89,084 any stroke cases and 1,013,843 controls, mostly from large biobanks, for external replication. Out of the 60 loci reaching genome-wide significance in primary inverse-variance weighted (IVW) meta-analyses, in this independent external dataset 48 loci (80%) replicated at p<0.05 with consistent directionality, of which 31 (52%), at $p<8.2 \times 10^{-4}$ (accounting for the number of loci tested).
Based on these follow-up results we characterized the level of confidence of identified loci as follows: high confidence in case of significant internal 'cross-ancestry' and/or external replication after accounting for the number of loci tested, or nominally significant replication in both internal and external replication, or evidence of involvement in monogenic stroke; intermediate confidence in case of nominal significance in either internal 'inter-ancestry' or external replication but not both; and low confidence in the absence of formal replication.
Overall, out of the 60 loci reaching genome-wide significance in the main IVW GWAS meta-analysis, 52 (87%) replicated at p<0.05 with consistent direction, of which 37 (61.7%) with high confidence, and 15 with intermediate confidence (25%). The 8 loci that did not replicate were labeled as "low confidence". Four of these were ethnic specific and 3 were low frequency variants that were monomorphic in some ancestries, limiting our ability for replication.
Within the secondary analyses (MR-MEGA and MTAG), none of the 3 MR-MEGA loci replicated, although one was borderline significant (Supplementary Table 16). Of the 26 MTAG loci, 18 (69%) replicated with AS or AIS at p<0.05, of which 9 (35%) with high confidence. Of the 8 MTAG loci that did not replicate, 7 showed a consistent directionality (borderline significant for one), and 4 were subtype-specific, limiting our ability for replication with AS or AIS.
While we have clearly labeled "low confidence" variants, we have not removed them from bioinformatics functional follow-up analyses. Indeed, we feel that despite the important worldwide effort that enabled to gather nearly 90,000 additional stroke cases, several issues still affect our ability to replicate some of the identified stroke risk loci:
• limits of statistical power, considering a smaller sample size than in the discovery and the winner's curse phenomenon;
• we cannot rule out some degree of misclassification in the follow-up samples that were, with two smaller exceptions, nearly exclusively derived from large biobanks with stroke ascertainment based on ICD codes only (Turnbull, Lancet Reg Health West Pac. 2022; Rannikmae, Neurology 2020), while a large proportion of stroke cases in the discovery were recruited and deeply phenotyped in a hospital-based setting;
• a substantial proportion of genetic risk for stroke is subtype specific, which is not fully captured in the replication because of the limited availability of stroke subtype data |
| Randomization | No randomization was performed because there was no allocation of samples to experimental groups. |
| Blinding | Blinding was not relevant to this study. The investigators of each study evaluated the case status of individual samples. Individual studies performed a genome-wide association study (GWAS) using logistic regression (or cox regression in some longitudinal population-based cohorts) testing association of genotypes with five stroke phenotypes (AS, AIS, CES, LAS, and SVS) under an additive effect model, adjusting for age, sex, principal components of population stratification, and study specific covariates when needed, details are provided in Supplementary table 2. The consortium meta-analysed summary statistics from these case/control studies, not individual level data. |

# Reporting for specific materials, systems and methods

We require information from authors about some types of materials, experimental systems and methods used in many studies. Here, indicate whether each material, system or method listed is relevant to your study. If you are not sure if a list item applies to your research, read the appropriate section before selecting a response.

## Materials & experimental systems

| n/a | Involved in the study |
|-----|----------------------|
| ☒ | ☐ Antibodies |
| ☒ | ☐ Eukaryotic cell lines |
| ☒ | ☐ Palaeontology and archaeology |
| ☒ | ☐ Animals and other organisms |
| ☐ | ☒ Human research participants |
| ☒ | ☐ Clinical data |
| ☒ | ☐ Dual use research of concern |

## Methods

| n/a | Involved in the study |
|-----|----------------------|
| ☒ | ☐ ChIP-seq |
| ☒ | ☐ Flow cytometry |
| ☒ | ☐ MRI-based neuroimaging |

## Human research participants

Policy information about studies involving human research participants

**Population characteristics**

We performed meta-analysis of GWAS on 29 population-based cohorts or biobanks with incident stroke ascertainment and 25 clinic-based case-control studies, comprising up to 110,182 stroke patients and 1,503,898 controls. The cohorts included individuals of European (EUR, 66.7% of stroke patients), East-Asian (EAS, 24.8%), African-American (AFR, 3.7%), South-Asian (SAS, 3.3%), and Hispanic (HIS, 1.4%) ancestry. Analyses were performed for any stroke (AS: comprising ischemic stroke, ICH, and stroke of unknown or undetermined type), any ischemic stroke regardless of subtype (AIS, N=86,668), and ischemic stroke subtypes (LAS, N=9,219; CES, N=12,790; SVS, N=13,620).
We also gathered an independent dataset of 89,084 AS (of which 85,546 AIS; 70.0% EUR, 15.6% AFR, 10.1% EAS, 4.1% HIS, and 0.1% SAS) and 1,013,843 controls, mostly from large biobanks, for external replication.
Population characteristics of all individual studies are provided in Supplementary Table 1.

**Recruitment**

As summarized in Supplementary Table 1 and described in greater detail in the Supplementary Appendix, participants were recruited in three different settings: (1) population-based studies (about one third of the discovery dataset); (2) clinic-based case-control studies; (3) biobanks (mostly hospital-based, about 90% of follow-up studies). In clinic/hospital-based studies patients with very severe strokes making informed consent more challenging or very minor strokes not leading to any hospitalization are less likely to be included. Stroke ascertainment in population-based studies is more comprehensive as it is conducted prospectively. Stroke subtyping is more detailed in clinic-based case-control studies than in population-based studies and biobanks.

**Ethics oversight**

Each contributing studies received ethical approval by the respective Institutional Review Boards (IRB). Detailed descriptions for each contributing study are given in supplementary appendix.

Note that full information on the approval of the study protocol must also be provided in the manuscript.

