## [Peer Review File · Nature]

Manuscript Title: Stroke genetics informs drug discovery and risk prediction across ancestries

Reviewer Comments & Author Rebuttals

Reviewer Reports on the Initial Version:

Referee expertise:

Referee #1: neurogenomics

Referee #2: complex trait genetics

Referee #3: vascular genetics

Referees' comments:

Referee #1 (Remarks to the Author):

Stroke genetics informs drug discovery and risk prediction across ancestries
Mishra et al.

In this manuscript, Mishra and colleagues conducted the largest GWAS of stroke and stroke subtypes to date in five ancestries, which discovered over 60 novel stroke risk loci. The authors performed a range of follow-up analyses to identify putative causal genes and possible drug targets, and improved polygenic prediction of stroke in European and East Asian populations. Overall this work represents a comprehensive set of analysis that advances our understanding of the genetics of stroke. I have a number of concerns/comments that maybe useful.

- Replication: A major limitation of this study in my opinion is the lack of replication of discovered risk loci. This raises concerns about the reliability of the risk loci especially those that were identified by borrowing power from other GWAS via MTAG. While it may be difficult to find additional independent and sufficiently large cohorts for a replication analysis, the authors could more comprehensively investigate the replication rate (at different levels of stringency) across populations, or in left-out cohorts. I think this direction is under-explored.

- Contribution of population diversity: Although the authors put together a diverse dataset from 5 ancestries, the majority of the analyses were conducted in European and East Asian samples only. I think there is a missed opportunity to investigate the contributions of genomic diversity to the loci discovery and risk prediction. For example, it's unclear whether increased discovery power can be simply attributed to sample size increase, and it's unclear how the polygenic score performs in non-European non-Asian populations and whether increasing the diversity in the training dataset improves prediction across populations (including in European and Asian samples).

- Presentation of significant loci: As there are population-specific GWAS and cross-ancestry meta-analysis for multiple stroke subtypes, it's sometimes difficult to figure out the number of overlapping loci between subtypes and how the total number of (novel) loci was calculated. For example, the authors reported 56 genome-wide significant loci in cross-ancestry meta-analysis (IVW and MR-MEGA). In the next paragraph, per-allele effect sizes were compared across 60 IVW meta-analysis loci. It's unclear where this number (60) came from. When exploring the overlap between stroke risk loci and vascular risk factors, the authors reported 57 of 88 loci had pleiotropic associations, but why not 89 as reported in the abstract? I'd suggest (i) double check all the reported numbers are consistent and explain how they were calculated; and (ii) find a better way to present the discovered loci and their relationships between populations and subtypes to improve the flow of the manuscript (additional figures/tables may be needed in supplementary).

- Fine-mapping: it is a little surprising that there was very limited overlap of credible sets between European and East Asian populations, given that effect sizes were highly correlated across populations. Does this indicate that non-European GWAS are still underpowered for reliable fine-mapping analysis? It's also unclear how to functionally interpret the findings from the MENTR analysis which seemed to pinpoint a range of tissues and cell types with no clear convergence of biology.

- Data release: it seems that only summary statistics for meta-analysis will be publicly available. I would encourage the authors to release summary statistics for all population-specific GWAS and GWAS for all stroke subtypes. Population-specific data will be hugely useful as the field continues to develop methods (e.g. cross-ancestry fine-mapping and polygenic prediction) that can integrate data from multiple populations for improved genomic discovery and prediction.

- Two-sample Mendelian randomization: Is there sample overlap between the GIGASTROKE study and some vascular risk factor GWAS, thus violating the assumption of two-sample MR? For example, both GIGASTROKE and the blood pressure GWAS included the UK Biobank sample?

- SuSiE: Please elaborate on how LD patterns were checked to identify false positives or use a more objective way to filter out false positive signals.

- PheWAS: Consider using PheCodes to reduce noise in ICD codes and combine information from diseases that are individually rare to increase statistical power.

- iPGS analysis: The overall predictive performance of PGS looked fairly weak, requiring an extreme cutoff of 0.1% to reach an OR of 3. Is it possible to assess the prediction performance of iPGS using completely independent training and evaluation datasets? Currently training and evaluation sets were created from the same biobank, which increases the risk of overfitting to specific sample characteristics of the biobank and does not fully assess the generalizability of the iPGS. Why the European iPGS was constructed from 14 secondary traits but the East Asian iPGS was built from many more (37) traits? Would that be better to make a fair comparison between the two iPGS (i.e., using the same set of GWAS for training and evaluation)? It would be helpful to separate the contribution from stroke GWAS and secondary GWAS to the predictions.

- PGS in clinical setting: Is there any reason not to use the genome-wide PGS or iPGS constructed by P+T, LDpred or PRSCs in this analysis?

- Methods Page 15 Line 335: "1,470 IS cases and 40,459 controls in the model training dataset" -> "... model evaluation dataset"

Referee #2 (Remarks to the Author):

Mishra et al. present a genome-wide association study (GWAS) of 110,000 cases of prevalent or incident stroke with extensive computational follow-up analyses. This sample is 40% larger than the previous largest study (MEGASTROKE), and importantly, includes more individuals from diverse ancestries. While the current study reveals incrementally more genomic loci, credible sets, genes, and pathways—which I would argue is mainly of interest to an expert statistical genetics audience—I consider the following findings of particular importance and expect them to appeal to readers across disciplines:

- Dozens of potential drug candidates for stroke are highlighted in the study by combining stroke genetics with public transcriptomic and proteomic data. The large stroke GWAS sample size and innovative use of multiple drug discovery methods lead to compelling evidence for F11 and KLKB1 as drug targets, with the former likely having minimal off-target effects.

- Genetic risk scores from the cross-ancestry GWAS provide meaningful improvement in stroke incidence prediction in addition to standard clinical risk factors, for individuals with EUR ancestry and individuals with EAS ancestry. (Granted I am not qualified to assess whether the selected clinical risk factors used were appropriate)

I believe the study is methodologically sound and the conclusions are largely supported by the evidence presented. There are a small number of statistical checks I expect the authors to perform to verify their results, as outlined in the “minor comments” section below.

My only issue with the current work is its presentation of newly discovered genomic loci. The authors acknowledge their study lacks a formal replication dataset—which is understandable for a GWAS of this size—but despite this limitation, they are not conservative with their definition of stroke risk loci and genes.

Main comments:

The authors claim to identify 89 risk loci for stroke and stroke subtypes (61 novel) based on a genome-wide association threshold of 5×10^{-8} .

However, the authors did not adjust any of the input GWAS for their LD-score regression intercept. While they are correct there is no systematic inflation across ancestries, the European GWAS has an inflated intercept of 1.10 and contributes the largest sample to the cross-ancestry meta-analysis. I would expect the standard errors of each ancestry-specific set of GWAS summary statistics to be

adjusted for any inflation before meta-analysis to minimize false positive discoveries.

Secondly, throughout the manuscript, the authors investigate three stroke subtypes (LAS, CES, SVS) and two stroke summary traits (AS, AIS), but do not adjust for multiple testing of these phenotypes. It is clear these stroke traits are correlated, but the authors do not quantify this correlation, and (from a statistical point of view) treat their five phenotypes as one. The authors should either 1) quantify the (in)dependence of their traits and adjust any significance thresholds for the number independent components (e.g. using genetic correlation and PCA) or 2) qualify statements regarding the number of discovered loci and genes.

Minor comments:

Were summary statistics checked for potential sample overlap before meta-analysis? For example, the METASTROKE analysis appears to contain the WHI cohort, which is also listed as a new dataset in Supplementary Table 1. It would be useful to see the phenotypic correlations between individual cohort GWAS to rule out any sample overlap, using e.g. the cross-trait LD-score regression intercept or alternative methods (see <https://doi.org/10.3389/fgene.2021.665252>),

Mendelian randomisation can be quite susceptible to horizontal pleiotropy and reverse causality. However, the authors do not test for reverse causality in their MR analysis of vascular risk factors on stroke traits. As they have GWAS summary statistics for all phenotypes, they should perform a bidirectional MR. This step is particularly important when assessing the causal role of BMI and WHR, where alternative pathways could explain the observed association with stroke phenotypes.

Why do the authors use VEGAS to aggregate genome-wide P values into gene-wide association statistics, but then use MAGMA to calculate these gene-wide statistics when selecting genes as GREP input? I also note the MAGMA results are not provided in any Supplementary Tables, and can therefore not be compared with VEGAS2 results.

Editorial comments:

Line 115 - It is not mentioned what weights were used in Stouffer's p value meta-analysis of VEGAS2 pathway results.

Line 170 - the phrase "checking the LD pattern" does not make it entirely clear which criteria were used to justify removing specific credible sets.

Line 269 - Which population/source was used for R² calculations of proxy instruments in the protein MR & colocalization analysis?

Figure 2 - error bars are not defined (95% confidence interval or standard error?)

The x-axis of Figure 4D is not defined.

Supplementary Table 19 has two sub-tables which are mistakenly labelled 26A and 26B.

Supplementary Tables 21b, 30, and 38 have coauthor comments which have not been resolved.

The LDSC label and some of the vascular trait labels in Extended Data Fig 3. are not properly visible in the PDF file.

Cathie LM Sudlow appears twice in the second tier author list.

The total participants in the Dutch PSI cohort is reported as 1375 in the Supplementary Appendix, but the number listed in Supplementary Table 1 is 1,377.

Referee #3 (Remarks to the Author):

In this manuscript, Mishra et al perform a GWAS of stroke and stroke subtypes in over 110,000 cases and 1.5 million controls of diverse ancestries nearly doubling the sample size of the largest prior stroke GWAS (MEGASTROKE). The authors identify many novel stroke loci, and perform a series of downstream analyses including conditional analysis (GCTA COJO), multi trait analysis (MTAG), fine mapping, enrichment analysis, Mendelian randomization, and a PRS analysis. Most compelling to this reviewer was the drug discovery analysis, identifying a series of putative stroke drug targets combining multiple lines of evidence. The manuscript represents a landmark paper in the understanding of stroke genetics with substantial novelty in many of its features, and is well written. Most of the conclusions are supported by the data, with appropriate methodology, and the authors cite relevant prior work. However, this reviewer has several concerns (major/minor) outlined below.

Major concerns:

- 1) While this manuscript is a multi ancestry meta-analysis, no independent replication for novel genomic loci are provided. While the authors' efforts in aggregating as many cases as possible should be commended, at a minimum some form of internal replication would improve the manuscript. For example, as utilized in a BP GWAS from 2018 (PMID: 30224653), genome-wide significance plus a p value of 0.01 in 2 (roughly) evenly divided strata of participants would reduce concerns for false positive findings. These concerns are additionally important in light of the fact that multiple GWAS in distinct stroke subtypes (AIS, LAS, SVS) are performed without further correction for multiple testing.
- 2) From this reviewer's understanding, the authors identify a series of novel loci in their primary IVW analysis, and then perform MTAG to further identify loci with correlated and/or causal stroke traits. This MTAG analysis identifies a series of additional loci, which the authors then claim as novel stroke loci in the abstract/paper. In the opinion of this reviewer, this representation of the data is misleading. MTAG results should be interpreted as largely exploratory, and not be claimed as distinctly novel stroke loci. This is particularly true in light of the fact that there is likely substantial overlap among samples in these analyses. While MTAG accounts for sample overlap, these results should not be considered "novel stroke loci" with the same level of supportive evidence.
- 3) In some cases Mendelian randomization analyses are performed with binary exposures -> binary outcomes. There is literature that suggests that this can lead to violation of the exclusion restriction assumption (PMID: 30039250), and that these results should be interpreted in terms of liability

(PMID: 34570226). The nuance here is important and should be appropriately addressed by the authors in their results/figures.

4) A substantial amount of the paper focuses on the multi-ancestry applicability of the PRS generated by the authors. While this is interesting, much of this data is more limited in novelty in light of the recent GLGC paper recently published in Nature (PMID: 34887591).

5) While the authors use SUSIE/finemapping techniques to identify putative causal variants, minimal effort is spent in an attempt to identify candidate causal genes at each locus for all identified loci. More recent GWAS manuscripts often integrate multiple lines of evidence (software like PoPs, colocalization with eQTL data in appropriate tissue(s), nearest gene, TWAS, protein altering variants in high LD with sentinel variants, etc) to interpret and identify what they believe is the likely causal gene acting at a given locus. In many cases this may be unclear, and can be stated.

6) This reviewer believes the interpretation of the FGA locus as a direct link to alteplase (a tissue plasminogen activator) is a bit of a reach. While FGA and a large component of thrombus (fibrin mesh) are certainly critical in this pharmacologic mechanism, the nuance of alteplase and its effect in plasminogen in thrombus degradation is completely overlooked. This should be mentioned in greater detail to the reader.

Minor concerns:

1) There appears to be a theme of highlighting novel loci with [] following a numerical statement of overall loci. This reviewer found this presentation distracting.

2) Single cell data from the pFC is briefly mentioned, but it is not clear to this reviewer what this information/data adds to improve the paper. This is particularly evident given how strong other sections of the manuscript are.

3) The PheWAS analysis is somewhat limited and could easily be expanded in light of the plethora of open source/access tools that can be used for this purpose (Phenoscaner, ieu open GWAS project, etc).

4) Figure 2b was somewhat blurry on the PDF version of the article this reviewer read. The resolution on this image could be enhanced

5) In reference to point #1, the discussion mentions that independent validation is available through the use of population studies/clinical trials. This reviewer believes this sentence is misleading and should be amended.

Author Rebuttals to Initial Comments:

Dear Dr. Trenkmann,

On behalf of our co-authors we thank you very much for giving us the opportunity to revise our manuscript, 2021-12-19966: "Stroke genetics informs drug discovery and risk prediction across ancestries". We also thank you and the Referees for the helpful and constructive comments. We have addressed these point-by-point below and updated the manuscript accordingly (changes are highlighted in the text).

Major changes are summarized below:

- 1. The genomics-driven drug discovery section, which was mentioned to be one of the most attractive features of our study, has been further strengthened by the following additions.** First, we re-ran the drug target enrichment analysis with GREP using the gene-based association results not only from MAGMA, but also from VEGAS2: all previously described genes were confirmed and two additional genes were prioritized. Second, we explored two additional independent pQTL resources (in plasma and cerebrospinal fluid, CSF) to strengthen the evidence for a causal association of drug target protein levels with stroke risk. All previously described pQTL associations replicated at $FDR < 0.05$, with consistent directionality for all except one gene (*GPIBA*), prompting caution for the latter. Third, for genes targeted by inhibitors, we examined associations of rare deleterious variants with stroke and related traits using whole-exome sequencing in >450,000 UK Biobank participants, and observed one significant protective association supporting the corresponding gene (*FII*) as a potential therapeutic target for inhibitors.
- 2. In order to address concerns about the confidence in reported loci, we gathered an independent dataset of 89,084 stroke patients (of which 85,546 ischemic strokes; 70.0% EUR, 15.6% AFR, 10.1% EAS, 4.1% HIS, and 0.1% SAS) and 1,013,843 controls,** mostly from large biobanks, for follow-up of identified loci. We additionally sought to validate the 42 stroke risk loci reaching genome-wide significance in individual ancestries in at least one other ancestry group among the discovery samples. Moreover, we have now more clearly separated our primary analysis (IVW meta-analysis) from secondary analyses (MR-MEGA and MTAG). Overall, we provide independent validation of the vast majority of identified genome-wide significant associations and graded our loci by level of confidence based on these "internal" and "external" follow-up findings. Despite the notable size of the follow-up study sample, with nearly 90,000 additional stroke patients and the compelling follow-up results, the power to replicate low frequency variants, ancestry-specific associations, and subtype-specific associations is limited. In regards to the latter, we note that most of the follow-up studies were derived from large biobanks with event ascertainment based on electronic health records, without suitable stroke subtype information. Loci that did not show evidence for replication (13% of our primary loci) were clearly labeled as "low confidence" pending additional follow-up in the future when additional, larger subtype- and ancestry-specific samples become available.
- 3. We further expanded the polygenic score (PGS) section of our manuscript by adding three types of complementary analyses.** First, we harmonized the number of risk factors included in European and East-Asian integrative PGS (iPGS) and confirmed associations of these with ischemic stroke in additional, completely independent evaluation datasets (from the Million Veteran Program, MVP, for Europeans and Taiwan Biobank for East-Asians). Second, we also validated the European iPGS in a clinical trial setting. Third, we evaluated the predictive ability of

the iPGS model in additional, unique African-American (MVP) and indigenous African (Nigerian and Ghanaian, SIREN) datasets. While African ancestry samples were too small to generate an African-specific stroke PGS, the European iPGS showed a significant association with ischemic stroke in both African-American and indigenous African (Nigerian and Ghanaian) participants, although expectedly weaker than in European participants. This is the first study assessing the performance of iPGS for ischemic stroke in East-Asian and African-ancestry populations. Moreover, the GIGASTROKE iPGS clearly outperforms an earlier iPGS derived from the smaller MEGASTROKE studies, in all ancestries.

4. **We provide different lines of evidence highlighting the added value of the cross-ancestry approach taken in our manuscript (with one third of non-European samples in both discovery and follow-up studies).** First, for loci derived from the IVW meta-analyses we show a clear gain in power beyond sample size increase, compared to the incremental addition of European ancestry samples. Second, we show that iPGS models derived from cross-ancestry stroke GWAS had a higher predictive ability than iPGS models derived from ancestry-specific stroke GWAS only, both in Europeans and East Asians.

A number of additional sensitivity analyses have been applied, such as controlling for ancestry specific LD score intercepts, removing overlapping samples from Mendelian randomization (MR) analyses, performing bidirectional MR and Steiger tests to rule out reverse causation, confirming PheWAS results obtained in Estonian Biobank using the Phenoscanner, etc. We have also expanded our description of the bioinformatics follow-up of identified stroke risk loci.

We hope that you will find this revised version suitable for publication in *Nature*.

Thank you very much for your consideration.

Best regards,

Stéphanie Debette and Martin Dichgans

.....

Point-by-point response to the editor and referees

We would specifically like to point out that referees have raised concerns about the confidence in the novel reported loci and we ask that you provide further supporting evidence (and replication as far as that is possible).

→ We thank the editor and the referees for raising this point. In order to address this point we gathered an independent dataset of 89,084 stroke patients (of which 85,546 ischemic strokes; 70.0% EUR, 15.6% AFR, 10.1% EAS, 4.1% HIS, and 0.1% SAS) and 1,013,843 controls, mostly from large biobanks, for external replication (Table A).

Following the referees' advice, we have now followed up genome-wide significant stroke risk loci both internally and externally. First, we sought to replicate the 42 stroke risk loci reaching genome-wide significance in individual ancestries in at least one other ancestry group among the discovery samples. Second, we aimed for replication in the additional independent dataset (Table A). The biobank setting did not allow suitable ischemic stroke subtype analyses.

Table A: GIGASTROKE external follow-up samples

Total ALL discovery		AS	AIS	Controls
EUR	MVP	39 165	39 165	425 512
	FinnGen	9027	8140	115 190
	FinnHosp	1600	1600	800
	Interstroke	1312	1223	1577
	MGB	3745	3745	28 281
	BioVU	4686	3582	60 936
	EstBB	1571	1372	4062
	PMBB	1223	1018	28795
Total EUR		62 329	59 845	665 153
AFR	MVP	11 126	11 126	111 914
	Interstroke	356	245	380
	PMBB	730	647	10309
	SI REN	1691	1691	1743
Total AFR		13 903	13 709	124 346
HIS	MVP	3 234	3 234	48 901
	Interstroke	400	288	422
Total HIS		3 634	3 522	49 323
EAS	Interstroke	162	97	184
	CKBB	5692	5692	10805
	TBB	2064	1399	86283
	CRCS-K and KBA GWAS	1120	1116	77583
Total EAS		9 038	8 304	174 855
SAS	Interstroke	114	100	143
Total SAS		114	100	143
Total ALL follow-up		89 018	85 480	1 013 820

Based on these follow-up results we characterized the level of confidence of identified loci as follows: high confidence in case of significant internal ‘cross-ancestry’ and/or external replication after accounting for the number of loci tested, or nominally significant replication in both internal and external replication, or evidence of involvement in monogenic stroke; intermediate confidence in case of nominal significance in either internal ‘inter-ancestry’ or external replication but not both; and low confidence in the absence of formal replication.

Overall, out of the 60 loci reaching genome-wide significance in the main IVW GWAS meta-analysis, 52 (87%) replicated at $p < 0.05$ with consistent direction, of which 37 (61.7%) with high confidence, and 15 with intermediate confidence (25%). The 8 loci that did not replicate were labeled as “low

confidence". Four of these were ethnic specific and 3 were low frequency variants that were monomorphic in some ancestries, limiting our ability for replication.

Within the secondary analyses (MR-MEGA and MTAG), none of the 3 MR-MEGA loci replicated, although one was borderline significant (Supplementary Table 16). Of the 26 MTAG loci, 18 (69%) replicated with AS or AIS at $p < 0.05$, of which 9 (35%) with high confidence. Of the 8 MTAG loci that did not replicate, 7 showed a consistent directionality (borderline significant for one), and 4 were subtype-specific, limiting our ability for replication with AS or AIS.

While we have clearly labeled "low confidence" variants, we have not removed them from bioinformatics functional follow-up analyses. Indeed, we feel that despite the important worldwide effort that enabled to gather nearly 90,000 additional stroke cases, several issues still affect our ability to replicate some of the identified stroke risk loci:

- limits of statistical power, considering a smaller sample size than in the discovery and the winner's curse phenomenon;
- we cannot rule out some degree of misclassification in the follow-up samples that were, with two smaller exceptions, nearly exclusively derived from large biobanks with stroke ascertainment based on ICD codes only (Turnbull, Lancet Reg Health West Pac. 2022; Rannikmae, Neurology 2020), while a large proportion of stroke cases in the discovery were recruited and deeply phenotyped in a hospital-based setting;
- a substantial proportion of genetic risk for stroke is subtype specific, which is not fully captured in the replication because of the limited availability of stroke subtype data

In this revised manuscript we have:

- included the replication results and grading in Supplementary Tables 14-16

- revised Figure 1 to provide information on confidence levels for each locus:

- amended the text as follows:

Results (p. 16, line 436-437): "Next, we conducted a secondary cross-ancestry meta-analysis with MR-MEGA,¹² which accounts for the allelic heterogeneity between ancestries."

Results (p. 16, line 440-442): "To further enhance statistical power for AIS subtypes, we conducted secondary multi-trait analyses of GWAS (MTAG)¹³ in Europeans and East-Asians, including traits correlated with specific stroke subtypes, (...)"

Results (p. 16, line 450-454): "This brings the number of identified stroke risk loci from primary (IVW) and secondary (MR-MEGA and MTAG) analyses to 89 in total (61 novel), of which 69 were associated with AS, 45 with AIS, 15 with LAS, 13 with CES, and 10 with SVS (of these 44, 33, 11, 8, and 3 were novel respectively, Fig. 1, Supplementary Table 4, 8, and 9-14)."

Results (p. 16 last paragraph line 455-459 and p. 17 first paragraph, line 460-479): “Independent follow-up of GWAS signals

We followed up genome-side significant stroke risk loci both internally and externally. First, we sought to replicate the 42 stroke risk loci reaching genome-wide significance in individual ancestries in at least one other ancestry group among the discovery samples. We successfully replicated, in a consistent direction, 10 of these loci at $p < 1.19 \times 10^{-3}$ (accounting for the number of loci tested), of which 7 were genome-wide significant in EUR, 1 in EAS, and 2 in both EUR and EAS. Additional 15 loci showed nominal association ($p < 0.05$) in at least one other ancestry (Supplementary Table 15).

Second, we gathered an independent dataset of 89,084 AS (of which 85,546 AIS; 70.0% EUR, 15.6% AFR, 10.1% EAS, 4.1% HIS, and 0.1% SAS) and 1,013,843 controls, mostly from large biobanks, for external replication (the biobank setting did not allow suitable ischemic stroke subtype analyses). Out of the 60 loci reaching genome-wide significance in IVW meta-analyses, 48 loci (80%) replicated at $p < 0.05$ with consistent directionality, of which 31 (52%), at $p < 8.2 \times 10^{-4}$ (accounting for the number of loci tested) (Supplementary Table 16). When considering both the internal and external follow-up, 52 (87%) of the 60 IVW loci replicated, of which 37 with “high confidence”, and 15 with “intermediate confidence” (Methods, Supplementary Table 14). The 8 loci that did not replicate were labeled as “low confidence” (Methods, Supplementary Table 14). Four of these were ethnic specific and 3 were low frequency variants that were monomorphic in some ancestries, restricting our ability to obtain replication.

Within the secondary analyses, none of the 3 MR-MEGA loci replicated, although one was borderline significant (Supplementary Table 16). Of the 26 MTAG loci, 18 (69%) replicated with AS or AIS at $p < 0.05$, of which 9 (35%) with high confidence ($p < 1.7 \times 10^{-3}$, accounting for 29 secondary loci tested, Supplementary Table 16). Of the 8 MTAG loci that did not replicate, 7 showed a consistent directionality (borderline significant for one), and 4 were subtype-specific and hence with less power to detect associations with AS or AIS.”

Discussion (p. 26, line 723-731): “Our GWAS meta-analyses gathering 110,182 stroke patients and 1,503,898 controls from five different ancestries (33% non European stroke patients) identified 89 risk loci for stroke and stroke subtypes (60 through primary IVW and 29 through secondary MR-MEGA and MTAG analyses), of which 61 were novel. We observed substantial shared susceptibility to stroke across ancestries, with strong correlation of effect sizes. Based on internal cross-ancestry validation and independent external follow-up in 89,084 stroke cases (30% non European) and 1,013,843 controls, mostly from large biobanks with information on AS and AIS only, the level of confidence of these loci was intermediate or high for 87% of primary stroke risk loci and 60% of secondary loci.”

Discussion (p. 28, last paragraph, line 803-809): “We provided independent validation of the vast majority of identified genome-wide significant associations and graded our loci by level of confidence based on these findings. Despite the notable size of the follow-up study sample, with nearly 90,000 additional stroke patients, we remain limited in our ability to validate low frequency variants, ancestry- and subtype-specific associations in particular. In regards to the latter, most of the follow-

up studies were derived from large biobanks with event ascertainment based on electronic health records and without suitable stroke subtype information.”

Methods (p. 1, line 3-5): *“Information on participating studies (discovery and follow-up), study design, and definition of stroke and stroke subtype is provided in the Supplementary Appendix. Population characteristics of individual studies are provided in Supplementary Table 1.”*

Methods (p. 1, line 9-10): *“Genotyping methods, pre-imputation quality control (QC) of genotypes and imputation methods of individual cohorts (discovery and follow-up) are presented in Supplementary Table 2.”*

Methods (p. 3 second paragraph, line 65-80): *“First, we sought to replicate “internally” the 42 stroke risk loci reaching genome-wide significance in IVW meta-analyses within individual ancestries, in at least one other ancestry group among the discovery samples, considering both nominal replication levels ($p < 0.05$) and multiple testing corrected significance at $p < 1.19 \times 10^{-3}$ ($0.05/42$). Second, we gathered independent datasets totaling 89,084 AS (of which 85,546 AIS; 70.0% EUR, 15.6% AFR, 10.1% EAS, 4.1% HIS, and 0.1% SAS) and 1,013,843 controls for “external” replication of associations with AS and AIS (Supplementary Table 1-2). These comprised 8 biobanks (82,263/930,988 cases/controls) and 4 hospital-based cohorts (6,821/82,855 cases/controls). We considered both nominal replication levels ($p < 0.05$) and multiple testing corrected significance at $p < 8.2 \times 10^{-4}$ ($0.05/60$) and $p < 1.3 \times 10^{-3}$ ($0.05/29$) for follow-up of genome-wide significant loci from the IVW and the MR-MEGA/MTAG meta-analyses respectively. We considered stroke risk loci as high confidence in case of significant internal ‘cross-ancestry’ and/or external replication after accounting for the number of loci tested, or nominally significant replication in both internal and external replication, or evidence of involvement in monogenic stroke; intermediate confidence in case of nominal significance in either internal ‘inter-ancestry’ or external replication but not both; and low confidence in the absence of formal replication.”*

We also note that the referees have mentioned the drug target section as one of the most attractive features of the study and we encourage you to add further supporting evidence in this direction.

→ We thank the editor for this comment. To further strengthen the drug target section, we performed the following complementary analyses:

1. **We re-ran the drug target enrichment analysis with GREP using the gene-based association results not only from MAGMA, but also from VEGAS2** (in response to a comment from Referee 1). All previously significant enrichments are maintained, and two additional genes were prioritized: *F2* and *TFPI*, targets of Lepirudin and Dalteparin respectively, both involved in the coagulation process and used for treatment of recurrent thromboembolism.

→ The manuscript has been amended as follows:

Results (p. 20, last paragraph, line 569-577): *“First, using GREP²² we observed significant enrichment of stroke-associated genes (MAGMA¹¹ or VEGAS2¹⁰ false discovery rates [FDR] < 0.05)*

in drug-target genes for blood and blood-forming organs (Anatomical Therapeutic Chemical Classification System [ATC] B drugs, for AS, AIS, and CES). This encompasses the previously described *PDE3A* and *FGA* genes,²⁴ encoding targets for cilostazol (antiplatelet agent) and alteplase (thrombolytic drug acting via plasminogen²⁵), respectively, as well as *F11*, *KLKB1*, *F2*, *TFPI*, and *MUT* encoding targets for conestat alfa, ecallantide (both used for hereditary angioedema), *lepirudin*, *dalteparin* (both used to treat recurrent thromboembolism), and vitamin B12, respectively (Supplementary Table 31)."

Methods (p. 9, line 233-235): "We ran *MAGMA*²³ and *VEGAS2*¹⁰ to summarize variant-level *p*-values into gene-level and used the genes with false discovery rates (FDR) less than 0.05 in either *MAGMA* or *VEGAS2* as the disease-risk genes."

Supplementary Table 25 has been updated.

- 2. We explored two additional independent pQTL resources to strengthen the evidence for a causal association of drug target protein levels with stroke risk.** In the initial manuscript, we nominated six proteins with Mendelian randomization (MR) and colocalization evidence, *VCAM1*, *F11*, *KLKB1*, *PROC*, *GP1BA*, and *MMP12*, by leveraging a plasma protein quantitative trait loci (pQTL) dataset. For five of these proteins, we curated drugs targeting those proteins in a direction compatible with a beneficial therapeutic effect against stroke based on MR estimates: inhibitors for *VCAM1*, *F11*, *KLKB1*, and *GP1BA* and activators for *PROC*.

To add further support to these results, we conducted two types of analyses, (i) MR with additional two pQTL datasets and (ii) an investigation of protective associations between rare deleterious variants and stroke risk traits.

First, we conducted MR for the six proteins using an independent cerebrospinal fluid (CSF) and plasma pQTL dataset (Yang *et al.*, *Nature Neuroscience*, 2021). We confirmed significant causal associations (false discovery rate [FDR] < 0.05) for *KLKB1*, *PROC*, *F11*, and *MMP12*; *F11* and *MMP12* were further supported by colocalization analyses. Of the four proteins, the causal associations of *KLKB1* and *PROC* were confirmed using the CSF pQTL dataset. Notably, the directions of MR estimates were the same as the initial results for all four proteins, supporting the beneficial therapeutic effects of the drugs curated in this study. As we noted in the manuscript, *F11* and *KLKB1* are adjacent genes with a long-range linkage disequilibrium pattern and complex co-regulation. Whereas the causal associations of *KLKB1* on stroke subtypes were initially inferred using a trans-pQTL, this time, we confirmed these associations using a cis-pQTL and observed borderline colocalization for ischemic stroke (AIS). No genome-wide significant pQTL was available for *VCAM1* and *GPA1B* in this smaller dataset.

We also conducted MR using one of the largest plasma pQTL studies (Ferkingstad *et al.*, *Nature Genetics*, 2021). We observed significant associations (FDR < 0.05) for all six proteins with consistent directionality except *GP1BA* (for which both concordant and discordant directionality was observed). Colocalization was confirmed for *F11*, *KLKB1*, *MMP12*, and *PROC*, with

directions of effect consistent with the initial results. Together, we confirmed causal associations for six proteins and colocalization for four proteins using independent pQTL datasets.

Next, we investigated protective associations between rare deleterious variants and stroke risk traits. As described in Backman *et al.* (*Nature*, 2021), genes for which deleterious variants are associated with lower disease risk are potential therapeutic targets for inhibitors. Since this approach does not use pQTL datasets, we can leverage it to independently validate the beneficial therapeutic effects of the inhibitors nominated in this study. For the four genes targeted by inhibitors, *VCAM1*, *F11*, *KLKB1*, and *GP1BA*, we examined the associations of rare deleterious variants (minor allele frequency < 0.01) with stroke and stroke-related traits using gene-based burden tests in the whole-exome sequencing data of >450,000 UK Biobank participants (Backman *et al.*, *Nature*, 2021). We observed one significant association, where rare deleterious variants in *F11* had protective associations with venous thromboembolism (OR=0.471 and $p=2.46 \times 10^{-4} < 0.05 / 4 / 30$; 4 and 30 represent the number of tested genes and traits, respectively). The direction of this association was concordant with that of MR estimates, supporting the therapeutic effect of F11 inhibitors on stroke.

The approach used here is different from the phenome-wide association study (PheWAS) in Estonian Biobank because we used rare deleterious variants not captured by genotyping arrays and focused on their associations with stroke-risk traits to validate the potential efficacy of inhibitors. The PheWAS observed no significant association of a cis-pQTL for F11 with non-stroke-related phenotypes. Together, the additional approach used in this revised version and the PheWAS indicate the potential efficacy and safety of inhibiting F11.

→ The manuscript has been amended as follows:

Results (p. 21, line 584-607): *“Third, we used protein quantitative trait loci (pQTL) for 218 drug-target proteins as instruments for MR and found evidence for causal associations of 9 plasma proteins with stroke risk (4 cis-pQTL, 6 trans-pQTL), of which 6 were supported by colocalization analyses, with no evidence for reverse causation using the Steiger test (PROC, VCAM1, F11, KLKB1, MMP12, and GP1BA, Supplementary Table 33). All of these replicated (at FDR < 0.05), with consistent directionality except GP1BA (for which both concordant and discordant directionality was observed), using at least one independent plasma pQTL resource and CSF pQTL for PROC and KLKB1, and with evidence for colocalization for PROC, F11, KLKB1, and MMP12. Using public drug databases we curated drugs targeting those proteins in a direction compatible with a beneficial therapeutic effect against stroke based on MR estimates and identified such drugs for VCAM1, F11, KLKB1, GP1BA (inhibitors) and PROC (activators, Supplementary Table 34). Drugs targeting F11 (NCT04755283, NCT04304508, NCT03766581) and PROC (NCT02222714) are currently under investigation for stroke, and our results provided genetic support for this. **Of note, F11 and KLKB1 are adjacent genes with a long range linkage disequilibrium pattern and complex co-regulation,³⁰ as illustrated here by the presence of a shared trans-pQTL in KNG1 (Supplementary Table 33). Additional studies are needed to disentangle causal associations and the most appropriate drug target in this region.^{31,32} Next, for the four genes targeted by inhibitors,***

VCAM1, F11, KLKB1, and GP1BA, we examined the associations of rare deleterious variants (MAF<0.01) with stroke and stroke-related traits using gene-based burden tests using whole-exome sequencing in >450,000 UK Biobank participants to support potential therapeutic targets for inhibitors.³³ We observed one significant protective association of rare deleterious variants in F11 with venous thromboembolism (OR=0.471, p=2.46×10⁻⁴), in a direction concordant with that of MR estimates (Supplementary Table 35)."

Figure 3: has been updated accordingly and now also includes **F2** and **TFPI** .

Discussion (p. 27, second paragraph, line 753-769): *"Our results provide genetic evidence for putative drug effects using three independent approaches, with converging results from two methods (gene enrichment analysis and pQTL-based MR) for drugs targeting F11 and KLKB1. F11 and F11a inhibitors (e.g. abelacimab, BAY 2433334, BMS-986177) are currently explored in phase-2 trials for primary or secondary stroke prevention (NCT04755283, NCT04304508, NCT03766581). Additional evidence from pQTL-based MR suggested PROC as a potential drug target for stroke. A recombinant variant of human activated protein C (encoded by PROC) was found to be safe for the treatment of acute ischemic stroke following thrombolysis, mechanical thrombectomy or both in phase 1 and 2 trials (3K3A-APC, NCT02222714),^{54,55} and is poised for an upcoming phase 3 trial. 3K3A-APC is proposed as a neuroprotectant, with evidence for protection of white matter tracts and oligodendrocytes from ischemic injury in mice.⁵⁶ Weaker evidence was found for GP1BA and VCAM1 as potential drug targets for stroke, with evidence for colocalization with one but not all pQTL datasets. Anfibatide, a GPIIb/IIIa antagonist, reduced blood-brain barrier disruption following ischemic stroke in mice⁵⁷ and is being tested as an antiplatelet drug in myocardial infarction (NCT01585259). While specific VCAM1 inhibitors are not available, probucol, a lipid lowering drug with pleiotropic effects including VCAM1 inhibition was tested for secondary prevention of atherosclerotic events in CAD patients (PROSPECTIVE, UMIN000003307).⁵⁸"*

Methods (p. 10, paragraph 2, line 272-275): *"To provide further support for our findings we conducted MR analyses with two additional recent independent pQTL datasets, using the same methodology as above: one study comprised both plasma (N=529) and cerebrospinal fluid (CSF, N=835) pQTL datasets,¹⁰² the second is one of the largest plasma pQTL studies conducted in 35,559 Icelanders.¹⁰³"*

Methods (p. 10, paragraph, line 276-283): *"Protective associations between rare deleterious variants and stroke risk traits. For the four genes targeted by inhibitors, VCAM1, F11, KLKB1, and GP1BA, we extracted the associations of rare deleterious variants (MAF<0.01) with stroke and stroke-related traits from the gene-based burden tests in the whole-exome sequencing data of >450,000 UK Biobank participants.(Backman, Nature 2021) As stroke and stroke-related traits, we extracted 30 traits belonging to nine vascular risk factor and disease categories (Supplementary*

Table 35). We applied Bonferroni correction and the corrected p-value threshold was $0.05/4/30=4.17\times 10^{-4}$ (4 and 30 represent the number of tested genes and traits, respectively)

Supplementary Table 33 has been updated to incorporate results with the additional pQTL datasets.

Supplementary Table 35 has been added to show the results of gene-based burden tests of rare deleterious variants (MAF<1%) in genes targeted by inhibitors for stroke and stroke-related traits in UK Biobank participants with whole-exome sequencing.

Referee #1 (Remarks to the Author):

Stroke genetics informs drug discovery and risk prediction across ancestries
Mishra et al.

In this manuscript, Mishra and colleagues conducted the largest GWAS of stroke and stroke subtypes to date in five ancestries, which discovered over 60 novel stroke risk loci. The authors performed a range of follow-up analyses to identify putative causal genes and possible drug targets, and improved polygenic prediction of stroke in European and East Asian populations. Overall this work represents a comprehensive set of analysis that advances our understanding of the genetics of stroke. I have a number of concerns/comments that maybe useful.

- Replication: A major limitation of this study in my opinion is the lack of replication of discovered risk loci. This raises concerns about the reliability of the risk loci especially those that were identified by borrowing power from other GWAS via MTAG. While it may be difficult to find additional independent and sufficiently large cohorts for a replication analysis, the authors could more comprehensively investigate the replication rate (at different levels of stringency) across populations, or in left-out cohorts. I think this direction is under-explored.

We would specifically like to point out that referees have raised concerns about the confidence in the novel reported loci and we ask that you provide further supporting evidence (and replication as far as that is possible).

→ We thank the editor and the referees for raising this point. In order to adress this point we gathered an independent dataset of 89,084 stroke patients (of which 85,546 ischemic strokes; 70.0% EUR, 15.6% AFR, 10.1% EAS, 4.1% HIS, and 0.1% SAS) and 1,013,843 controls, mostly from large biobanks, for external replication (Table A).

Following the referees' advice, we have now followed up genome-side significant stroke risk loci both internally and externally. First, we sought to replicate the 42 stroke risk loci reaching genome-wide significance in individual ancestries in at least one other ancestry group among the discovery

samples. Second, we aimed for replication in the additional independent dataset (Table A). The biobank setting did not allow suitable ischemic stroke subtype analyses.

Table A: GIGASTROKE external follow-up samples

Total ALL discovery		AS	AIS	Controls
EUR	MVP	39 165	39 165	425 512
	FinnGen	9027	8140	115 190
	FinnHosp	1600	1600	800
	Interstroke	1312	1223	1577
	MGB	3745	3745	28 281
	BioVU	4686	3582	60 936
	EstBB	1571	1372	4062
	PMBB	1223	1018	28795
Total EUR		62 329	59 845	665 153
AFR	MVP	11 126	11 126	111 914
	Interstroke	356	245	380
	PMBB	730	647	10309
	SI REN	1691	1691	1743
Total AFR		13 903	13 709	124 346
HIS	MVP	3 234	3 234	48 901
	Interstroke	400	288	422
Total HIS		3 634	3 522	49 323
EAS	Interstroke	162	97	184
	CKBB	5692	5692	10805
	TBB	2064	1399	86283
	CRCS-K and KBA GWAS	1120	1116	77583
Total EAS		9 038	8 304	174 855
SAS	Interstroke	114	100	143
Total SAS		114	100	143
Total ALL follow-up		89 018	85 480	1 013 820

Based on these follow-up results we characterized the level of confidence of identified loci as follows: high confidence in case of significant internal ‘cross-ancestry’ and/or external replication after accounting for the number of loci tested, or nominally significant replication in both internal and external replication, or evidence of involvement in monogenic stroke; intermediate confidence in case of nominal significance in either internal ‘inter-ancestry’ or external replication but not both; and low confidence in the absence of formal replication.

Overall, out of the 60 loci reaching genome-wide significance in the main IVW GWAS meta-analysis, 52 (87%) replicated at $p < 0.05$ with consistent direction, of which 37 (61.7%) with high confidence, and 15 with intermediate confidence (25%). The 8 loci that did not replicate were labeled as “low confidence”. Four of these were ethnic specific and 3 were low frequency variants that were monomorphic in some ancestries, limiting our ability for replication.

Within the secondary analyses (MR-MEGA and MTAG), none of the 3 MR-MEGA loci replicated, although one was borderline significant (Supplementary Table 16). Of the 26 MTAG loci, 18 (69%) replicated with AS or AIS at $p < 0.05$, of which 9 (35%) with high confidence. Of the 8 MTAG loci that did not replicate, 7 showed a consistent directionality (borderline significant for one), and 4 were subtype-specific, limiting our ability for replication with AS or AIS.

While we have clearly labeled “low confidence” variants, we have not removed them from bioinformatics functional follow-up analyses. Indeed, we feel that despite the important worldwide effort that enabled to gather nearly 90,000 additional stroke cases, several issues still affect our ability to replicate some of the identified stroke risk loci:

- limits of statistical power, considering a smaller sample size than in the discovery and the winner’s curse phenomenon;
- we cannot rule out some degree of misclassification in the follow-up samples that were, with two smaller exceptions, nearly exclusively derived from large biobanks with stroke ascertainment based on ICD codes only (Turnbull, *Lancet Reg Health West Pac.* 2022; Rannikmae, *Neurology* 2020), while a large proportion of stroke cases in the discovery were recruited and deeply phenotyped in a hospital-based setting;
- a substantial proportion of genetic risk for stroke is subtype specific, which is not fully captured in the replication because of the limited availability of stroke subtype data

In this revised manuscript we have:

- included the replication results and grading in Supplementary Tables 14-16

- revised Figure 1 to provide information on confidence levels for each locus:

- amended the text as follows:

Results (p. 16, line 436-437): “Next, we conducted a secondary cross-ancestry meta-analysis with MR-MEGA,¹² which accounts for the allelic heterogeneity between ancestries.”

Results (p. 16, line 440-442): “To further enhance statistical power for AIS subtypes, we conducted secondary multi-trait analyses of GWAS (MTAG)¹³ in Europeans and East-Asians, including traits correlated with specific stroke subtypes, (...)”

Results (p. 16, line 450-454): “This brings the number of identified stroke risk loci from primary (IVW) and secondary (MR-MEGA and MTAG) analyses to 89 in total (61 novel), of which 69 were associated

with AS, 45 with AIS, 15 with LAS, 13 with CES, and 10 with SVS (of these 44, 33, 11, 8, and 3 were novel respectively, Fig. 1, Supplementary Table 4, 8, and 9-14)."

Results (p. 16 last paragraph line 455-459 and p. 17 first paragraph, line 460-479): *"Independent follow-up of GWAS signals*

We followed up genome-side significant stroke risk loci both internally and externally. First, we sought to replicate the 42 stroke risk loci reaching genome-wide significance in individual ancestries in at least one other ancestry group among the discovery samples. We successfully replicated, in a consistent direction, 10 of these loci at $p < 1.19 \times 10^{-3}$ (accounting for the number of loci tested), of which 7 were genome-wide significant in EUR, 1 in EAS, and 2 in both EUR and EAS. Additional 15 loci showed nominal association ($p < 0.05$) in at least one other ancestry (Supplementary Table 15).

Second, we gathered an independent dataset of 89,084 AS (of which 85,546 AIS; 70.0% EUR, 15.6% AFR, 10.1% EAS, 4.1% HIS, and 0.1% SAS) and 1,013,843 controls, mostly from large biobanks, for external replication (the biobank setting did not allow suitable ischemic stroke subtype analyses). Out of the 60 loci reaching genome-wide significance in IVW meta-analyses, 48 loci (80%) replicated at $p < 0.05$ with consistent directionality, of which 31 (52%), at $p < 8.2 \times 10^{-4}$ (accounting for the number of loci tested) (Supplementary Table 16). When considering both the internal and external follow-up, 52 (87%) of the 60 IVW loci replicated, of which 37 with "high confidence", and 15 with "intermediate confidence" (Methods, Supplementary Table 14). The 8 loci that did not replicate were labeled as "low confidence" (Methods, Supplementary Table 14). Four of these were ethnic specific and 3 were low frequency variants that were monomorphic in some ancestries, restricting our ability to obtain replication.

Within the secondary analyses, none of the 3 MR-MEGA loci replicated, although one was borderline significant (Supplementary Table 16). Of the 26 MTAG loci, 18 (69%) replicated with AS or AIS at $p < 0.05$, of which 9 (35%) with high confidence ($p < 1.7 \times 10^{-3}$, accounting for 29 secondary loci tested, Supplementary Table 16). Of the 8 MTAG loci that did not replicate, 7 showed a consistent directionality (borderline significant for one), and 4 were subtype-specific and hence with less power to detect associations with AS or AIS."

Discussion (p. 26, line 723-731): *"Our GWAS meta-analyses gathering 110,182 stroke patients and 1,503,898 controls from five different ancestries (33% non European stroke patients) identified 89 risk loci for stroke and stroke subtypes (60 through primary IVW and 29 through secondary MR-MEGA and MTAG analyses), of which 61 were novel. We observed substantial shared susceptibility to stroke across ancestries, with strong correlation of effect sizes. Based on internal cross-ancestry validation and independent external follow-up in 89,084 stroke cases (30% non European) and 1,013,843 controls, mostly from large biobanks with information on AS and AIS only, the level of confidence of these loci was intermediate or high for 87% of primary stroke risk loci and 60% of secondary loci."*

Discussion (p. 28, last paragraph, line 803-809): *"We provided independent validation of the vast majority of identified genome-wide significant associations and graded our loci by level of confidence based on these findings. Despite the notable size of the follow-up study sample, with nearly 90,000*

additional stroke patients, we remain limited in our ability to validate low frequency variants, ancestry- and subtype-specific associations in particular. In regards to the latter, most of the follow-up studies were derived from large biobanks with event ascertainment based on electronic health records and without suitable stroke subtype information.”

Methods (p. 1, line 3-5): *“Information on participating studies (discovery and follow-up), study design, and definition of stroke and stroke subtype is provided in the Supplementary Appendix. Population characteristics of individual studies are provided in Supplementary Table 1.”*

Methods (p. 1, line 9-10): *“Genotyping methods, pre-imputation quality control (QC) of genotypes and imputation methods of individual cohorts (discovery and follow-up) are presented in Supplementary Table 2.”*

Methods (p. 3 second paragraph, line 65-80): *“First, we sought to replicate “internally” the 42 stroke risk loci reaching genome-wide significance in IVW meta-analyses within individual ancestries, in at least one other ancestry group among the discovery samples, considering both nominal replication levels ($p < 0.05$) and multiple testing corrected significance at $p < 1.19 \times 10^{-3}$ ($0.05/42$). Second, we gathered independent datasets totaling 89,084 AS (of which 85,546 AIS; 70.0% EUR, 15.6% AFR, 10.1% EAS, 4.1% HIS, and 0.1% SAS) and 1,013,843 controls for “external” replication of associations with AS and AIS (Supplementary Table 1-2). These comprised 8 biobanks (82,263/930,988 cases/controls) and 4 hospital-based cohorts (6,821/82,855 cases/controls). We considered both nominal replication levels ($p < 0.05$) and multiple testing corrected significance at $p < 8.2 \times 10^{-4}$ ($0.05/60$) and $p < 1.3 \times 10^{-3}$ ($0.05/29$) for follow-up of genome-wide significant loci from the IVW and the MR-MEGA/MTAG meta-analyses respectively. We considered stroke risk loci as high confidence in case of significant internal ‘cross-ancestry’ and/or external replication after accounting for the number of loci tested, or nominally significant replication in both internal and external replication, or evidence of involvement in monogenic stroke; intermediate confidence in case of nominal significance in either internal ‘inter-ancestry’ or external replication but not both; and low confidence in the absence of formal replication.”*

- Contribution of population diversity: Although the authors put together a diverse dataset from 5 ancestries, the majority of the analyses were conducted in European and East Asian samples only. I think there is a missed opportunity to investigate the contributions of genomic diversity to the loci discovery and risk prediction. For example, it's unclear whether increased discovery power can be simply attributed to sample size increase, and it's unclear how the polygenic score performs in non-European non-Asian populations and whether increasing the diversity in the training dataset improves prediction across populations (including in European and Asian samples).

→ We thank Referee 1 for these comments and questions.

- “it's unclear whether increased discovery power can be simply attributed to sample size increase”

We have addressed this point through the following additions:

1- Evaluation of our best integrative polygenic scores (iPGS) in two African ancestry samples that were added with the revision: one large dataset of 107,343 African-American participants from the Million Veteran Program (MVP), of whom 2,227 developed an incident ischemic stroke, and one unique African sample from Nigeria and Ghana (SIREN) with 1,691 ischemic stroke cases and 1,743 controls. Both these cohorts have independent GWAS manuscripts on their unique datasets that are currently being finalized. In order not to interfere with these efforts, considering the challenge to collect samples from these highly underrepresented ancestry groups, we have included their data only in the polygenic score validation (MVP and SIREN) and for replication of our loci (MVP), but have neither meta-analyzed them with the rest of our discovery dataset nor generated African ancestry-specific PGS as we feel this reaches beyond the scope of the present manuscript and could undermine other important ongoing efforts under the aforementioned circumstances.

These important additions of PGS validation in African ancestry populations have been included in the following parts of the manuscript:

Results (p. 24, line 678-691): *“Next, we evaluated the predictive ability of the European-derived GIGASTROKE-based iPGS model in African-American and indigenous African (Nigerian and Ghanaian) datasets. In 107,343 African-American MVP participants, of whom 2,227 developed an AIS, the GIGASTROKE-based iPGS model showed a significant association with AIS incidence (HR per 1 SD, 1.11 [95%CI: 1.06-1.17]; $P = 1.8 \times 10^{-5}$, ΔC -index = 0.003, Supplementary Table 49), although weaker than in European MVP participants (Supplementary Table 43). Participants in the top 1% of the iPGS showed a 1.53-fold higher odds of AIS (95% CI, 1.04–2.25; $P=0.03$ compared to participants in the middle 10% (Figure 4c, Supplementary Table 50). In 1,691 cases and 1,743 controls from the indigenous African (Nigerian and Ghanaian) SIREN case-control study, the GIGASTROKE-based iPGS also showed a significant association with the odds of AIS (OR per 1 SD, 1.09 [95%CI, 1.02-1.17], $P = 0.010$, $\Delta AUC = 0.007$, Supplementary Table 51). The GIGASTROKE-based iPGS model showed a stronger association with AIS and a larger improvement in predictive ability compared to the MEGASTROKE-based iPGS model in both MVP and SIREN (Supplementary Tables 49 and 51).”*

Discussion (p. 27 last paragraph line 778-781, p. 28 first paragraph line 782-791): *“While African ancestry samples were too small to generate an African-specific stroke PGS, the European iPGS showed a significant association with ischemic stroke in both African-American and indigenous African (Nigerian and Ghanaian) participants, although expectedly weaker than in European participants. Individuals in the top 1% of the PGS distribution had a more than 2 to 2.5-fold risk of ischemic stroke in EAS and EUR participants compared to those in the middle 10%, while this risk was 1.5-fold in AFR participants. Although caution is warranted when interpreting risk estimates due to wide confidence intervals, these results suggest that GIGASTROKE-based iPGS models may be useful to stratify individuals exposed to genetically high risk of ischemic stroke, especially in Europeans and East-Asians. Our results highlight the importance of ancestry-specific and cross-ancestry genomic studies for the transferability of genomic risk prediction across populations, and the urgent need to vastly increase the diversity of participants in genomic studies, especially from the most underrepresented regions such as the African continent, to avoid exacerbation of health disparities in the era of precision medicine and precision public health.”*^{59-61”}

Methods (p. 12, line 335-337): *“The iPGS model for Europeans was further evaluated in two external cohorts of European ancestry (MVP and pooled data of clinical trials) as well as in two studies of African-ancestry participants (MVP and SIREN).”*

2- Incremental addition of non-European samples to the meta-analysis:

We have now conducted a secondary analysis whereby we have incrementally added to the previous largest European stroke GWAS (from MEGASTROKE): (1) additional European samples; (2) additional East Asian samples; (3) additional samples of non-European and non-East-Asian ancestry. This analysis shows that the increase in identified risk loci for stroke, proportionately to the increase in sample size, is notably higher when adding East-Asian samples and even higher when adding African, Hispanic, and South-Asian ancestry cohorts.

This has been included in the following sections of the manuscript:

Results (p. 17, line 481-484): *“For the 60 stroke risk loci derived from the IVW meta-analyses we first demonstrated the added value in terms of locus discovery of including non-European samples, showing a clear gain in power beyond sample size increase, compared to the incremental addition of European ancestry samples (Extended Data Figure 2).”*

Extended Data Fig. 2: Increase in power with increase in population diversity

The scatter plot shows the number of loci identified with incremental increase in sample size and population diversity. The diagonal line reflects the increase in number of genome-wide significant loci with increasing sample size of European ancestry only. The increase in number of loci departs from this line when adding non-European ancestry samples; EUR: European; EAS: East-Asian; AFR: African; HIS: Hispanic; SAS: South Asian; BBJ=Biobank Japan; CKB=China Kadoorie biobank. Strat0_EUR: N=22634 stroke cases (European population-based longitudinal cohorts); Strat1_Strat0&EUR: N=26253 stroke cases, adding Spanish and Estonian samples; Strat2_Strat1&EUR: N=32980 stroke cases, adding German and Dutch samples; Strat3_Strat2&EUR: N=42783, adding large European biobanks; Strat4_Strat3&EUR: N=73652 stroke cases, adding MEGASTROKE European; Strat5_Strat4&EAS: N=91303 stroke cases, adding Japanese BBJ; Strat6_Strat5&EAS: N=99661 stroke cases, adding Chinese CKB; Strat7_Strat6&EAS: N=101065 stroke cases, adding other East-Asian samples; Strat8_Strat7&AFR: N=105026 stroke cases, adding African ancestry samples; Strat9_Strat8&HIS: N=106542 stroke cases, adding South-American ancestry samples;

Strat10_Strat9&SAS, N=110182 stroke cases, adding South-Asian ancestry samples

- “it’s unclear whether increasing the diversity in the training dataset improves prediction across populations (including in European and Asian samples)”:

To address this comment we evaluated the predictive ability of two iPGS models. The first was derived from ancestry-specific GWAS summary data, and the second was constructed using cross-ancestry stroke GWAS summary data. Five stroke GWAS (AS, AIS, LAS, SVS, and CES) were incorporated in both iPGS models.

For Europeans, the two iPGS models were constructed using the EstBB training case-control dataset, and they were evaluated using the EstBB validation cohort. Compared to the iPGS model derived from European-specific stroke GWAS, the iPGS model derived from cross-ancestry stroke GWAS showed a larger improvement in C-index (Δ C-index = 0.006 for cross-ancestry model vs. Δ C-index = 0.002 for European-specific model) and, although confidence intervals overlapped, a stronger association with the incidence of AIS events (HR = 1.14 for cross-ancestry model vs. HR = 1.07 for European-specific model).

Table: Comparison of predictive ability of European iPGS models derived from ancestry-specific or cross-ancestry stroke GWAS

Derivation GWAS	#. GWAS	C-index	Δ C-index	HR (95% CI)	P
-----------------	---------	---------	------------------	-------------	---

Ancestry-specific stroke GWAS	5	0.606	0.002	1.07 (1.01–1.13)	0.03
Cross-ancestry stroke GWAS	5	0.610	0.006	1.14 (1.07–1.21)	1.2E-05

CI indicates confidence interval; GWAS, genome-wide association study; HR, hazard ratio. ΔC -index means the improvement in C-index over a base model that includes age, sex, and top 5 genetic principal components.

For East Asians, the two iPGS models were derived using the BBJ training case-control dataset and they were assessed using the BBJ validation case-control dataset. Compared to the iPGS model derived from East Asian-specific stroke GWAS, the iPGS model derived from cross-ancestry stroke GWAS showed a larger improvement in AUC ($\Delta AUC = 0.016$ for cross-ancestry model vs. $\Delta AUC = 0.011$ for East Asian-specific model) and a stronger association with the odds of AIS (OR = 1.29 for cross-ancestry model vs. OR = 1.24 for East Asian-specific model).

Table: Comparison of predictive ability of East Asian iPGS models derived from ancestry-specific or cross-ancestry stroke GWAS

Derivation GWAS	#. GWAS	AUC	ΔAUC	OR (95% CI)	P
Ancestry-specific stroke GWAS	5	0.645	0.011	1.24 (1.17–1.30)	2.5E-15
Cross-ancestry stroke GWAS	5	0.650	0.016	1.29 (1.22–1.36)	2.2E-21

AUC indicates area under the curve; CI, confidence interval; GWAS, genome-wide association study; OR, odds ratio.

ΔAUC means the improvement in AUC over a base model that includes age, sex, and top 5 genetic principal components.

These results show that iPGS models derived from cross-ancestry stroke GWAS consistently had a higher predictive ability than iPGS models derived from ancestry-specific stroke GWAS both in Europeans and East Asians. The greater ability of the cross-ancestry GWAS-based iPGS model may be simply due to a larger sample size of cross-ancestry GWAS compared to that of ancestry-specific GWAS. However, given that >66% of AIS cases included in cross-ancestry GWAS was European ancestry, a substantial difference in the predictive ability of the two iPGS models would be, at least partially, attributable to the increased diversity of derivation GWAS.

These updates have been reflected in the revised manuscript:

- **Supplementary Table 48:** “Predictive ability achieved by PGS models derived from cross-ancestry vs. ancestry-specific stroke GWAS”
- **Results (p. 24, line 675-677):** “Of note, iPGS models derived from cross-ancestry stroke GWAS had a higher predictive ability than iPGS models derived from ancestry-specific stroke GWAS both in Europeans and East Asians (Supplementary Table 48).”

- “it's unclear how the polygenic score performs in non-European non-Asian populations”

We evaluated the predictive ability of the GIGASTROKE-based iPGS model for Europeans in two external non-European non-Asian datasets.

First, we used the Million Veteran Program (MVP) longitudinal cohort to estimate the performance for subjects of African-American ancestry. Survival models were estimated from data spanning the beginning of 2011 (ie start of MVP) to end of 2018 (date of the latest National Death Index [NDI] data). Participants were censored at event (AIS), administrative censoring (Jan 1st, 2019), or death. Participants who had an AIS prior to enrollment in MVP were excluded. Out of 107,343 African-American ancestry MVP participants at baseline, 2,227 developed an incident AIS during follow-up. In parallel we also evaluated the predictive ability of the GIGASTROKE-based iPGS model in European MVP participants. Out of 403,489 European ancestry MVP participants at baseline, 8,392 developed an incident AIS during follow-up. For Europeans, our iPGS showed an evident association with AIS incidence (HR per 1 SD increase, 1.19; $P = 6.9 \times 10^{-52}$) and a certain improvement in C-index (ΔC -index = 0.010). For African-Americans, the GIGASTROKE-based iPGS model showed a significant association with AIS incidence, but the strength of the association was weaker than for Europeans (HR per 1 SD increase, 1.11; $P = 1.8 \times 10^{-5}$). Also, the improvement in C-index was smaller in African-Americans (ΔC -index = 0.003) than in Europeans (ΔC -index = 0.010).

Table: Predictive ability of GIGASTROKE-based iPGS model for Europeans in the MVP cohort, in both European and African-American participants

Population	C-index	ΔC -index	HR (95% CI)	P
European MVP participants	0.645	0.010	1.19 (1.16–1.21)	6.9E-52
African-American MVP participants	0.653	0.003	1.11 (1.06–1.17)	1.8E-05

CI indicates confidence interval; HR, hazard ratio.

ΔC -index means the improvement in C-index over a base model that includes age, sex, and top 5 genetic principal components.

Second, we evaluated the predictive ability of the iPGS model in the indigenous African (Nigerian and Ghanaian) SIREN study, comprising 1,691 ischemic stroke cases and 1,743 controls. Our iPGS showed a significant association with the odds of AIS (OR per 1 SD increase, 1.09; $P = 0.010$) and an improvement in AUC ($\Delta\text{AUC} = 0.007$).

Table: Predictive ability of the GIGASTROKE-based iPGS model for Europeans in the *indigenous African (Nigerian and Ghanaian) SIREN case-control study*

Population	AUC	ΔAUC	OR (95% CI)	P
African SIREN participants	0.548	0.007	1.09 (1.02–1.17)	0.010

AUC indicates area under the curve; CI, confidence interval; OR, odds ratio.

ΔAUC means the improvement in AUC over a base model that includes age, sex, and top 5 genetic principal components.

These results revealed that our iPGS model, which was derived using the cross-ancestry GIGASTROKE summary data and data of EstBB participants of European ancestry, was significantly associated with AIS in populations of African-American and indigenous African (Nigerian and Ghanaian) ancestry. However, our iPGS showed stronger association and larger improvement in predictive ability for Europeans than for Africans. Future directions for filling the gap in the prediction power between ancestries are the inclusion of larger numbers of non-European subjects in the cross-ancestry stroke GWAS and construction of ancestry-specific PGS models for non-Europeans. Therefore, continuous efforts of international collaboration will be of great importance.

These updates have been reflected in the revised manuscript:

- **Supplementary Table 49:** “Improvement of predictive ability achieved by PGS models for African-Americans in MVP”
- **Supplementary Table 50:** “Association of final iPGS model for European with the risk of ischemic stroke in African-Americans from MVP”
- **Supplementary Table 51:** “Improvement of predictive ability achieved by PGS models for African ancestry (Nigerian) SIREN participants”
- **Results (p. 24, line 678-691):** “Next, we evaluated the predictive ability of the European-derived GIGASTROKE-based iPGS model in African-American and indigenous African (Nigerian and Ghanaian) datasets. In 107,343 African-American MVP participants, of whom 2,227 developed an AIS, the GIGASTROKE-based iPGS model showed a significant association with AIS incidence (HR per 1 SD, 1.11 [95%CI: 1.06-1.17]; $P = 1.8 \times 10^{-5}$, $\Delta\text{C-index} = 0.003$, Supplementary Table 49), although weaker than in European MVP participants (Supplementary Table 43). Participants in the top 1% of the iPGS showed a 1.53-fold higher odds of AIS (95% CI, 1.04–2.25; $P=0.03$ compared to participants in the middle 10% (Figure 4c, Supplementary Table 50). In 1,691 cases and 1,743 controls from the indigenous African (Nigerian and Ghanaian) SIREN case-control study, the

GIGASTROKE-based iPGS also showed a significant association with the odds of AIS (OR per 1 SD, 1.09 [95%CI, 1.02-1.17], P = 0.010, Δ AUC = 0.007, Supplementary Table 51). The GIGASTROKE-based iPGS model showed a stronger association with AIS and a larger improvement in predictive ability compared to the MEGASTROKE-based iPGS model in both MVP and SIREN (Supplementary Tables 49 and 51).”

- **Discussion (p. 27 last paragraph line 778-781, p. 25 first paragraph line 782-791):** *“While African ancestry samples were too small to generate an African-specific stroke PGS, the European iPGS showed a significant association with ischemic stroke in both African-American and indigenous African (Nigerian and Ghanaian) participants, although expectedly weaker than in European participants. Individuals in the top 1% of the PGS distribution had a more than 2 to 2.5-fold risk of ischemic stroke in EAS and EUR participants compared to those in the middle 10%, while this risk was 1.5-fold in AFR participants. Although caution is warranted when interpreting risk estimates due to wide confidence intervals, these results suggest that GIGASTROKE-based iPGS models may be useful to stratify individuals exposed to genetically high risk of ischemic stroke, especially in Europeans and East-Asians. Our results highlight the importance of ancestry-specific and cross-ancestry genomic studies for the transferability of genomic risk prediction across populations, and the urgent need to vastly increase the diversity of participants in genomic studies, especially from the most underrepresented regions such as the African continent, to avoid exacerbation of health disparities in the era of precision medicine and precision public health.⁵⁹⁻⁶¹”*
- **Methods (p. 12, line 335-337):** *“The iPGS model for Europeans was further evaluated in two external cohorts of European ancestry (MVP and pooled data of clinical trials) as well as in two studies of African-ancestry participants (MVP and SIREN).”*

- Presentation of significant loci: As there are population-specific GWAS and cross-ancestry meta-analysis for multiple stroke subtypes, it's sometimes difficult to figure out the number of overlapping loci between subtypes and how the total number of (novel) loci was calculated. For example, the authors reported 56 genome-wide significant loci in cross-ancestry meta-analysis (IVW and MR-MEGA). In the next paragraph, per-allele effect sizes were compared across 60 IVW meta-analysis loci. It's unclear where this number (60) came from. When exploring the overlap between stroke risk loci and vascular risk factors, the authors reported 57 of 88 loci had pleiotropic associations, but why not 89 as reported in the abstract? I'd suggest (i) double check all the reported numbers are consistent and explain how they were calculated; and (ii) find a better way to present the discovered loci and their relationships between populations and subtypes to improve the flow of the manuscript (additional figures/tables may be needed in supplementary).

→ We thank the referee for these suggestions and apologies for the confusion. To navigate the loci better we have now complemented Supplementary Tables 4 (now Supplementary Tables 4A) with Supplementary Tables 4B describing GWAS loci discovered in population-specific and/or cross-ancestry meta-analyses, and Supplementary Tables 4C describing loci associated with different stroke phenotypes: AS, AIS, CES, LAS and SVS. We also updated the footnote of supplementary table 4A to define the term novel:

“§ OLD_MEGASTROKE refers to Malik et al Nat Genet 2018 (PMID: 29531354);
OLD_MEGASTROKE+UKBB refers to Malik et al Ann Neurol 2018 (PMID: 30383316) and

OLD_LAC_STROKE refers to Traylor et al Lancet Neurol 2021 (PMID: 33773637); and NOVEL refers to the loci that were not previously reported to be associated with stroke and any of its subtypes at genome-wide significance.”

To improve the flow of manuscript and tone down the MR-MEGA and MTAG results, we have separated the sections describing the 60 loci identified using the primary IVW meta-analysis from the section describing loci identified using secondary analyses (MR-MEGA and MTAG). Moreover, we verified all numbers and clarified the text wherever required.

We have rephrased the following sections of the manuscript for more clarity:

- **Results (p. 15, line 413-430):** *“Using IVW GWAS meta-analysis, we identified variants associated with stroke at genome-wide significance ($p < 5 \times 10^{-8}$) at 60 loci, of which 33 were novel (Fig. 1, Supplementary Table 4). Lead variants at all novel loci were common ($MAF \geq 0.05$), except for low-frequency intronic variants in THAP5 ($MAF = 0.02$, in complete association [$r^2 = 1$] with variants in the 5’UTR of NRCAM) associated with cross-ancestry incident AS/AIS, and in COBL ($MAF = 0.04$) associated with AS/AIS in South-Asians. Out of these 60 loci, the largest number of associations were identified for AS (48 loci [23 novel]) and AIS (45 loci, [18 novel]), of which one with incident AIS only (Supplementary Table 4c). While AIS subtypes were not available in some population-based cohorts (Supplementary Table 1), genome-wide significance was reached for 4 loci for LAS, 8 for CES, and 7 for SVS (of which 1, 3, and 3 were novel respectively, Supplementary Table 4). To our knowledge, our results include the most comprehensive and largest description of stroke genetic risk variants to date in each of the five represented ancestries. In cross-ancestry meta-analyses 53 loci (51 loci after controlling for ancestry specific LD score intercepts) reached genome-wide significance (Supplementary Table 4), while 42 loci were genome-wide significant in individual ancestries (35 in Europeans, 6 in East-Asians, 1 in South-Asians, and 2 in African-Americans, Supplementary Table 4). Using conditional and joint analysis (GCTA-COJO),⁸ we confirmed three independent signals at PITX2 and two at SH3PXD2A (CES in EUR, Supplementary Table 5).¹”*
- **Results (p. 16, line 450-454):** *“This brings the number of identified stroke risk loci from primary (IVW) and secondary (MR-MEGA and MTAG) analyses to 89 in total (61 novel), of which 69 were associated with AS, 45 with AIS, 15 with LAS, 13 with CES, and 10 with SVS (of these 44, 33, 11, 8, and 3 were novel respectively, Fig. 1, Supplementary Table 4, 8, and 9-14).”*

We also apologize for the typo in the following sentence where 88 should read 89:

- **Results (p. 19, line 523-526):** *“In Europeans, the lead variants for stroke at 57 of the 89 primary and secondary risk loci (64.0%) were associated ($P < 5 \times 10^{-8}$) with at least one vascular trait, most frequently blood pressure (33 loci, 37.1%, Extended Data Fig. 3, Supplementary Table 22).”*

- Fine-mapping: it is a little surprising that there was very limited overlap of credible sets between European and East Asian populations, given that effect sizes were highly correlated across populations. Does this indicate that non-European GWAS are still underpowered for reliable fine-mapping analysis? It's also unclear how to functionally interpret the findings from the MENTR analysis which seemed to pinpoint a range of tissues and cell types with no clear convergence of biology.

→ We thank the referee for these questions

- “Does this indicate that non-European GWAS are still underpowered for reliable fine-mapping analysis”:

An extensive analysis of trans-ethnic fine-mapping has been performed in a recent landmark study by Kanai et al. across 148 complex traits in three large-scale biobanks (BioBank Japan [BBJ], FinnGen, and UK Biobank [UKB]) (medrxiv 2021). Using BBJ data alone (corresponding to a subset of the GIGASTROKE East-Asian sample), for both ischemic stroke itself and phenotypes with similar sample sizes like myocardial infarction, the authors identified i) a limited set of high PIP variants and ii) a low number of 95% credible sets overall.

In addition, the limited overlap of EUR and EAS credible sets was seen over multiple phenotypes, leading to the conclusion that high PIP variants are not overlapping across populations. In fact, across the three biobanks, of the 4,518 identified variant-trait pairs with high posterior probability (> 0.9) of causality, only 285 replicated across multiple populations, thus displaying, as in our study, a surprising lack of overlap among fine-mapping results from different ethnicities. Many but not all of the variants with high PIP in one population but not in the other two populations were explained by the fact they are rare or monomorphic in the other two populations.

Beyond situations of population-specific associations, as in the study by Kanai et al, we believe that the main reason for the limited overlap of credible sets between EUR and EAS populations is limited power due to low sample size, especially in stroke subtypes and in EAS populations, enabling use to fine-map only variants with large or at most moderate effect sizes. Nevertheless, we believe that our fine-mapping approach has enabled the identification of some high-confidence causal variants shared across populations, at an important stroke risk locus, some population-enriched fine-mapped variants in EUR, and allelic series of high-impact variants across populations at several loci. We have now added the following sentence in the second to last paragraph of the discussion: “Despite major efforts to enhance non-European contributions to GIGASTROKE we still had limited power for identifying shared causal variants through cross-ancestry fine-mapping.”

- “It's also unclear how to functionally interpret the findings from the MENTR analysis which seemed to pinpoint a range of tissues and cell types with no clear convergence of biology”:

In the submitted manuscript, we showed that the alternative G allele of rs12476527 (5'UTR of *KCNK3*, coding for K⁺ channel protein; increasing stroke risk) was predicted to increase *KCNK3* expression in the kidney cortex tubule cells, despite no eQTL of this variant being reported. This is a lead SNP associated with pulse pressure ($p=2.553 \times 10^{-21}$) and systolic blood pressure ($p=5.892 \times 10^{-50}$) in the previous cross-ancestry study in >750,000 individuals (ref1), suggesting that the kidney involvement is plausible.

ref1. A Giri, et al. *Nat Genet.* 2019.

We also showed that three variants (rs12705390 at *PIK3CG*, rs2483262 at *PRDM16*, and rs2282978 at *CDK6*) were predicted to regulate the expression of a long non-coding RNA and enhancer RNAs. Unfortunately, currently no functions are available for these transcripts; however, many predictions were for many types of endothelial cells, as shown in Supplementary Table 20. We consider that these involvements of endothelial cells in stroke risk are plausible.

Taken together, predictions using MENTR could highlight biologically plausible transcripts and cell types for understanding stroke etiology. The multiple cell types involved reflect the complex and diverse mechanisms underlying stroke etiology that may involve different tissues.

We have now edited the text as follows Results (p. 18, line 502-515):

*"To detect putative causal regulatory variants, we conducted in silico mutagenesis analysis using MENTR, a machine-learning method to pin-point prediction of causal variants on transcriptional changes.³ From CS, we obtained 78 robust predictions of variant-transcript-model sets comprising 13 variants and 19 transcripts (Supplementary Table 20), involving multiple cell types in line with the diversity of mechanisms underlying stroke etiology. For instance, the G allele of rs12476527 (5'UTR of *KCNK3*) was a risk allele for stroke and predicted to increase *KCNK3* expression in kidney cortex tubule cells, despite no eQTL of this variant being reported in GTEx (v8) or eQTLgen (2019-12-23). The same G allele has been associated with higher systolic blood pressure (Giri, Nat Genet 2019). Furthermore, three variants (rs12705390 at *PIK3CG*, rs2282978 at *CDK6*, rs2483262 at *PRDM16*) were predicted to affect expression of a long non-coding RNA and enhancer RNAs, predominantly in endothelial cells, as well as other vascular cells and visceral preadipocytes, while a promoter variant of *SH3PXD2A* was predicted to modulate its expression in macrophages."*

- Data release: it seems that only summary statistics for meta-analysis will be publicly available. I would encourage the authors to release summary statistics for all population-specific GWAS and GWAS for all stroke subtypes. Population-specific data will be hugely useful as the field continues to develop methods (e.g. cross-ancestry fine-mapping and polygenic prediction) that can integrate data from multiple populations for improved genomic discovery and prediction.

→ We thank the referee for this suggestion. We will make summary statistics available for cross-ancestry and ancestry-specific meta-analyses and for stroke subtypes, as suggested by the referee.

The data availability statement has been clarified accordingly: *“Summary statistics for the GWAS meta-analyses of stroke (cross-ancestry and ancestry-specific, for any stroke and stroke subtypes) will be deposited in a public repository and made available by the time of publication. All other data supporting the findings of this study are available either within the article, the supplementary information and supplementary data files, or from the corresponding authors upon reasonable request.”*

- Two-sample Mendelian randomization: Is there sample overlap between the GIGASTROKE study and some vascular risk factor GWAS, thus violating the assumption of two-sample MR? For example, both GIGASTROKE and the blood pressure GWAS included the UK Biobank sample?

→ We thank the referee for this question and we apologize for not having addressed this issue in the first version of the manuscript. Indeed, sample overlap in Mendelian randomization analyses can lead to winner’s curse bias thus inflating the effect sizes of the selected instruments and introducing weak instrument bias. The magnitude of relevant bias in the obtained estimates has been found to be dependent on the magnitude of sample overlap and the strength of the instruments, but for binary traits, the obtained estimates are robust against weak instrument bias even in a one-sample setting, as long as the overlap is restricted to the control sample (Burgess et al, Genet Epidemiol. 2016 Nov; 40(7): 597–608). In our analyses we used the largest available datasets for the exposures of interest, which indeed led to overlap with the GIGASTROKE European sample. However, the overlap is expected to lie predominantly at the level of controls, as the vast majority of studies contributing stroke patients in GIGASTROKE were not included in risk factor GWAS, except for population-based cohorts, in particular UK Biobank that contributes only a small subset of GIGASTROKE stroke patients (<3%). Due to the complexity of the meta-analyzed datasets, it was not possible to estimate the exact magnitude of overlap on the case sample, in order to provide a reliable estimate of the magnitude of the expected relative bias based on the simulations performed by Burgess et al. Notably, there is no overlap in the datasets used for selecting and weighing the instruments in the East Asian cohorts and the GIGASTROKE East Asian sample.

To fully address the Referee’s comment, we have run GWAS analyses for all the exposures of interest within the UK Biobank cohort, after excluding the individuals that were included as stroke cases in GIGASTROKE. As previously recommended, we applied a three-sample Mendelian randomization approach, which is more robust to sample overlap, winner’s curse bias, and weak instrument bias (<https://www.medrxiv.org/content/10.1101/2021.06.28.21259622v1.full.pdf>). Specifically, we (1) selected the instruments from the original GWAS meta-analyses for the exposure traits, (2) weighed them on the basis of the UKB GWASs excluding stroke cases from GIGASTROKE, and (3) tested them on the GIGASTROKE cohort.

Importantly, the Mendelian randomization estimates derived from this approach were remarkably consistent with the estimates derived from the original analyses, thus confirming the robustness of our results to any potential sample overlap between GIGASTROKE and the cohorts used in the GWASs for the exposure traits. This is demonstrated in the correlation plot presented below comparing the results from inverse-variance weighted Mendelian randomization analyses derived from the two approaches.

Figure. Correlation plot comparing the effect estimates (betas) derived from inverse-variance weighted Mendelian randomization analyses using weights from the potentially overlapping GWASs for the exposure traits and using weights from a non-overlapping UK Biobank sample (Pearson's $r=0.96$).

The results from this sensitivity analysis are presented in Supplementary Table 25. Furthermore, we now report on these findings in the Results of our manuscript, as follows:

Results (p. 19, line 536-540): *“Due to limited overlap between the European GIGASTROKE sample and cohorts included in GWAS for the exposure traits, we ran sensitivity analyses weighing our genetic instruments based on a sub-sample of UK Biobank excluding cases included in GIGASTROKE (Burgess et al, Genet Epidemiol. 2016). The remarkable consistency of these with the main analyses confirm their robustness against weak instrument bias (Supplementary Table 25)”*

- SuSiE: Please elaborate on how LD patterns were checked to identify false positives or use a more objective way to filter out false positive signals.

→ We thank the reviewer for this question and apologize for not being clearer in describing our methodology. We illustrate our approach to identify false positives on one specific example: In EAS there was one locus for AS (SH2B3, chr12:11910219) that produced 8 credible sets, which could potentially indicate a false positive finding. To identify potential LD mismatches, we used the diagnostic vignette accompanied with SuSiE (https://cran.r-project.org/web/packages/susieR/vignettes/susierss_diagnostic.html) and compared the expected z-scores vs. observed z-scores.

Figure 1A below shows the SuSiE result before curation, variants are outlined by color according to the credible set.

Figure 1B shows an outlier 12:112645401 which was in high LD with the lead variant ($r^2=0.782$, $z=7.29$) but had observed $z=0.88$, while the expected z -score was 6.69, flagging an inconsistency. In the EAS summary statistics, this variant was missing from 67.25% of the samples ($N_{EAS:112645401}=86,331$, $N_{EAS_total}=263,592$), thus the power for this variant was greatly reduced, leading to statistical fluctuation. Hence, we removed this specific variant and reran the fine-mapping. The results and diagnostic plots are shown below in Figure 2. Only one CS was identified with consistent observed z scores vs expected z scores. This strategy was repeated for all loci. We now include a sentence in the Methods section:

Methods (p. 6, line 149-152): *“Fine-mapping results were checked for potential false-positive findings using a diagnostic procedure implemented in SuSiE. In short, we compared observed and expected z -score for each variant at a given locus and removed the variant if the difference between observed and expected z -score was too high after manual inspection.”*

Figure 1. before curating

Figure 2. after curating

- PheWAS: Consider using PheCodes to reduce noise in ICD codes and combine information from diseases that are individually rare to increase statistical power.

→ We thank the referee for this suggestion and we have now rerun the PheWAS analysis using phecodes that were generated with the PheWAS R-package (see Methods section). We see that all of the previously significant associations remained and are now even stronger (Supplementary Table 36). For the rs2289252 SNP we still detect strong association with pulmonary embolism and see significant association with phlebitis and thrombophlebitis. We also detect borderline significant association with acute pulmonary heart disease and pulmonary embolism and infarction (Extended Data Fig. 8).

We have edited the manuscript accordingly

- **Results (p. 21 last paragraph line 607-611, p. 22 first paragraph line 612-621):** *“To further validate the candidate drugs and estimate their potential side effects, we investigated whether the drug-target genes were associated with stroke-related phenotypes using a phenome-wide association study (PheWAS) approach.³⁴ We conducted PheWAS in Estonian Biobank (EstBB) for the pQTL variants for PROC, VCAM1, F11, KLKB1, and GP1BA genes (Supplementary Table 36). A cis-pQTL for F11, rs2289252, was associated with higher risk of venous thromboembolic disorders ($p < 3.95 \times 10^{-6}$), as previously described,³⁵ and showed suggestive association ($p = 3.44 \times 10^{-3}$) with cerebral artery occlusion with cerebral infarction (Phecode 433.21, Extended Data Fig. 8). Conversely, we observed no significant association with non-stroke-related phenotypes, suggesting the safety of targeting F11. Similar profiles were observed in UK Biobank and FinnGen (<https://r5.finnngen.fi/variant/4-186286227-C-T>), with no significant associations with other disorders and no overlap of subthreshold signals with side-effects reported in clinical trials.³⁶ We further confirmed the association of rs2289252 with venous thromboembolic disorders and having no association with other non-stroke-related phenotypes using the Phenoscanner database (Supplementary Table 37)”.*
- **Methods (p. 11, first paragraph line 285-293):** *“PheWAS analysis was carried out using the R software (4.0.3). We used the PheWAS R package¹⁰⁹ (<https://github.com/PheWAS/PheWAS>) function “createPhenotypes” to translate ICD10 diagnosis codes into phecodes for the PheWAS analysis. We tested the associations between phecodes and genetic variants using logistic regression and adjusting for sex, birth year and 10 genotype PCs. We applied Bonferroni*

correction to select statistically significant associations (number of tested Phecodes: 1809, number of tested SNPs: 7; corrected p-value threshold: $0.05/1809*7 = 3.95 \times 10^{-6}$). Results were visualized using the PheWAS library. To further characterize the associations of the genetic variants with other phenotypes we searched for all 7 SNPs in the PhenoScanner database.^{110,111}”

- **Legend of Extended Data Fig. 8:**

“PheWAS in Estonian biobank for pQTL of drug targets identified as being putative drug targets for stroke in the Mendelian randomization analysis, for which associations reached genome-wide significance ($p=3.95 \times 10^{-6}$): top, PheWAS for rs2289252, a cis-pQTL for F11. Each triangle in the plot represents one phecode and the direction of the triangle represents direction of effect.”

- iPGS analysis: The overall predictive performance of PGS looked fairly weak, requiring an extreme cutoff of 0.1% to reach an OR of 3. Is it possible to assess the prediction performance of iPGS using completely independent training and evaluation datasets? Currently training and evaluation sets were created from the same biobank, which increases the risk of overfitting to specific sample characteristics of the biobank and does not fully assess the generalizability of the iPGS. Why the European iPGS was constructed from 14 secondary traits but the East Asian iPGS was built from many more (37) traits? Would that be better to make a fair comparison between the two iPGS (i.e., using the same set of GWAS for training and evaluation)? It would be helpful to separate the contribution from stroke GWAS and secondary GWAS to the predictions.

→ We thank the referee for these questions:

- “Is it possible to assess the prediction performance of iPGS using completely independent training and evaluation datasets?”:

In the initial submission our evaluation datasets were entirely distinct from the training datasets (and from the GIGASTROKE GWAS dataset), but the referee is referring to the fact that training and evaluation datasets stem from the same study (Estonian Biobank for Europeans and Biobank Japan for East-Asians). Following the referee’s recommendation, we evaluated the predictive ability of the GIGASTROKE-based iPGS model for Europeans and East-Asians in completely independent evaluation datasets, stemming from entirely distinct studies. For comparison, a MEGASTROKE-based iPGS model constructed in a previous study (Abraham et al., 2019) was also assessed in the same independent evaluation datasets.

First, we evaluated the European iPGS models using European-ancestry participants of the Million Veteran Program (MVP) longitudinal cohort. Survival models were estimated from data spanning the beginning of 2011 (ie start of MVP) to end of 2018 (date of the latest National Death Index [NDI] data). Participants were censored at event (AIS), administrative censoring (Jan 1st, 2019), or death. Participants who had an AIS prior to enrollment in MVP were excluded. Out of 403,489 European ancestry participants at baseline, 8,392 had an incident AIS during follow-up. The GIGASTROKE-based iPGS model showed a significant association with the incidence of AIS (HR per 1 SD increase, 1.19; $P = 6.9 \times 10^{-52}$) and a certain improvement in C-index (Δ C-index = 0.010). Compared to the MEGASTROKE-based iPGS model, the association was stronger and the improvement in C-index was larger.

Table: Predictive ability of the European iPGS model assessed using MVP European ancestry participants

PGS model	C-index	Δ C-index	HR (95% CI)	P
MEGASTROKE-based European iPGS model	0.640	0.006	1.13 (1.10–1.15)	5.2E-28
GIGASTROKE-based European iPGS model	0.645	0.010	1.19 (1.16–1.21)	6.9E-52

CI indicates confidence interval; GWAS, genome-wide association study; HR, hazard ratio; PGS, polygenic score.

Δ C-index means the improvement in C-index over a base model that includes age, sex, and top 5 genetic principal components.

Second, we evaluated the European iPGS models using clinical trial data (across the spectrum of cardiometabolic disease, described in our initial submission) in European ancestry participants (51,288 European participants of whom 960 developed an incident AIS over 3 years follow-up). The GIGASTROKE-based iPGS model was significantly associated with the incidence of AIS (HR per 1 SD increase, 1.19 [1.11-1.27]; $P = 3.2 \times 10^{-7}$) and a certain improvement in C-index (Δ C-index = 0.008). Compared to the MEGASTROKE-based iPGS model, the GIGASTROKE-based iPGS model showed a stronger association and a larger improvement in C-index.

Table: Predictive ability of iPGS model assessed using the clinical trials dataset in Europeans

PGS model	C-index	Δ C-index	HR (95% CI)	P
MEGASTROKE-based European iPGS model	0.641	0.006	1.14 (1.06–1.21)	1.8E-04
GIGASTROKE-based European iPGS model	0.644	0.008	1.19 (1.11–1.27)	3.2E-07

CI indicates confidence interval; GWAS, genome-wide association study; HR, hazard ratio; PGS, polygenic score.

Δ C-index means the improvement in C-index over a base model that includes age, sex, and top 5 genetic principal components.

Thus very similar confirmatory results were obtained in the two completely independent evaluation datasets. These suggest an improved predictive performance of the GIGASTROKE-based iPGS model compared to the MEGASTROKE-based iPGS model.

Third, we evaluated the East-Asian iPGS models using East-Asian ancestry participants of the Taiwan BioBank (TBB) study (87,940 participants, of whom 1,391 developed an incident stroke). The GIGASTROKE-based iPGS model was significantly associated with the risk of AIS (OR per 1 SD increase, 1.18; $P = 1.8 \times 10^{-9}$) and a certain improvement in AUC ($\Delta AUC = 0.003$). Again, compared to the MEGASTROKE-based iPGS model, the GIGASTROKE-based iPGS model showed a stronger association and a larger improvement in C-index.

Table: Predictive ability of the European iPGS model assessed using Taiwan Biobank East-Asian ancestry participants

PGS model	AUC	ΔAUC	OR (95%CI)	P
MEGASTROKE-based European iPGS model	0.762	0.001	1.10 (1.04-1.16)	4.88×10^{-4}
GIGASTROKE-based European iPGS model	0.762	0.003	1.18 (1.12-1.25)	1.08×10^{-9}

AUC indicates area under the curve; CI indicates confidence interval; OR, odds ratio; PGS, polygenic score ; ΔAUC means the improvement in AUC compared to a base model that includes age, sex, and top 5 genetic principal components

These updates have been reflected in the revised manuscript:

- **Supplementary Table 43:** “Improvement of predictive ability achieved by PGS models for Europeans in MVP Europeans”
- **Supplementary Table 47:** “Improvement of predictive ability achieved by PGS models for East-Asians in Taiwan Biobank”
- **Supplementary Table 53:** “Improvement of predictive ability achieved by PGS models for Europeans in trial participants”
- **Results (p. 22, line 636-637):** “Participants in the training and evaluation datasets did not overlap and were not included in the input GWAS summary data.”
- **Results (p. 23, line 652-656):** “We further confirmed the GIGASTROKE-based European iPGS model trained in EstBB in 403,489 European-ancestry participants of the Million Veteran Program (MVP) study, of whom 8,392 developed an AIS: HR per SD, 1.19 (95%CI, 1.16–1.21; $P=6.94 \times 10^{-52}$), with a ΔC -index of 0.010 (Supplementary Table 43).”
- **Results (end of p. 23 line 671-672, beginning of p. 24 line 673-674):** “We further confirmed the GIGASTROKE-based East-Asian iPGS model trained in BBJ in 1,399 prevalent AIS cases

and 86,283 controls from the Taiwan Biobank (TBB): OR per SD, 1.18 (95%CI, 1.12–1.25; $P=1.1 \times 10^{-9}$), with a Δ AUC of 0.003 (Supplementary Table 47).”

- **Results (p. 25, end of last paragraph line 715-720):** “Finally, in European trial participants (East-Asians were too few for this analysis), the GIGASTROKE-based iPGS was also significantly associated with increased AIS incidence (HR per 1 SD increase, 1.19 [1.11-1.27]; $P = 3.2 \times 10^{-7}$, Δ C-index = 0.008), performing better than the MEGASTROKE-based iPGS (Supplementary Table 53). Compared to the middle 10% of the subjects, those in the top 1% had a 2.8-fold higher hazard of AIS (HR=2.78 [95% CI, 1.67–4.61]; $P=7.9 \times 10^{-5}$) (Figure 4e and Supplementary Table 54).”
- **Discussion (p. 27, last paragraph line 770-774):** “We improved polygenic risk prediction of stroke and importantly pioneered the exploration of stroke PGS across ancestries. Polygenic scores integrating cross-ancestry and ancestry-specific stroke GWAS with vascular risk factor GWAS (iPGS) showed strong prediction of ischemic stroke risk in European and, importantly, for the first time, in East-Asian where stroke incidence is highest.⁷ These results were confirmed in several independent datasets.”
- **Discussion (p. 28, end of second paragraph line 797):** “We further confirmed the GIGASTROKE iPGS in this clinical trial setting.”
- **Methods (p. 12, line 335-336):** “The iPGS model for Europeans was further evaluated in two external cohorts of European ancestry (MVP and pooled data of clinical trials)”.
- **Methods (p.13, end of first paragraph line 347-348):** “The iPGS model for East Asians was further evaluated in an external study of East Asian-ancestry (TBB).”
 - **“Why the European iPGS was constructed from 14 secondary traits but the East Asian iPGS was built from many more (37) traits”:**

In the initial submission, we had constructed the European iPGS model using the same set of risk factor GWAS as in a previous study using an integrative PGS approach in Europeans (Abraham et al., 2019), in order to more specifically assess the added value of the new (GIGASTROKE) over the older (MEGASTROKE) GWAS dataset. At the same time, we had constructed the East Asian iPGS model by selecting stroke-related phenotypes from East Asian or cross-ancestry GWAS summary data collected in jenger (<http://jenger.riken.jp/en/result>) or pheweb.jp (<https://pheweb.jp>). As a result, the number of risk factor GWAS incorporated in iPGS models were different between Europeans and East Asians.

Following the referee’s comment, in order to minimize the difference in the derivation methods of iPGS models between ancestries, we have now re-constructed European and East Asian iPGS models using the same set of 12 risk factor GWASs (for atrial fibrillation [AF], coronary artery disease [CAD], type 2 diabetes [T2D], systolic and diastolic blood pressure [SBP, DBP], total cholesterol [TC], LDL and HDL cholesterol [LDL-C, HDL-C], triglycerides [TG], body mass index [BMI], height and smoking).

For Europeans, compared to the set of risk factor GWAS used in the previous study (Abraham et al., 2019), we could obtain GWAS summary data with larger sample sizes for several traits (*i.e.*, AF, T2D, SBP, DBP, BMI, and height). We re-constructed the iPGS model using the EstBB training dataset, and subsequently, evaluated the predictive ability in the EstBB validation cohort (and, subsequently, as described above, in MVP and the clinical trial data). Compared to the previous

version of iPGS model, the revised version of iPGS model showed slightly stronger association with the incidence of AIS in EstBB (HR per 1 SD increase, 1.26; $P = 2.0 \times 10^{-15}$), with a slightly larger improvement in C-index (ΔC -index = 0.027). This slight improvement was possibly due to the larger sample size of risk factor GWAS.

Table: Comparison of predictive ability of European iPGS models (previous and revised versions)

PGS model	#. GWAS	C-index	ΔC -index	HR (95% CI)	P
Previous iPGS model	24	0.626	0.022	1.25 (1.18–1.32)	8.2E-14
Revised iPGS model	22	0.631	0.027	1.26 (1.19–1.34)	2.0E-15

CI indicates confidence interval; GWAS, genome-wide association study; HR, hazard ratio; PGS, polygenic score.

ΔC -index means the improvement in C-index over a base model that includes age, sex, and top 5 genetic principal components.

For East Asians, we used jenger (<http://jenger.riken.jp/en/result>) as a resource of GWAS summary data for the selected 12 risk factor traits. The 12 GWAS were included in the previous set of 37 risk factor GWAS. We re-constructed East Asian iPGS model using the BBJ training case-control dataset, and subsequently, evaluated the predictive performance of the iPGS model using the BBJ validation case-control dataset (and, subsequently, as described above, in the TBB data). Compared to the previous version of iPGS model, the revised iPGS model showed similar strength of association with the odds of AIS (OR per 1 SD increase, 1.33; $P = 9.9 \times 10^{-26}$) and similar improvement in AUC ($\Delta AUC = 0.019$).

Table: Comparison of predictive ability of East Asian iPGS models (previous and revised version)

PGS model	#. GWAS	AUC	ΔAUC	OR (95% CI)	P
Previous iPGS model	47	0.654	0.020	1.33 (1.26–1.40)	2.3E-26
Revised iPGS model	22	0.653	0.019	1.33 (1.26–1.40)	9.9E-26

AUC indicates area under the curve; CI, confidence interval; GWAS, genome-wide association study; OR, odds ratio.

Δ AUC means the improvement in AUC over a base model that includes age, sex, and top 5 genetic principal components.

These updates have been reflected in the revised manuscript:

- **Supplementary Tables 40-42, 44-46 have been updated.**
- **Results (beginning of p. 23 line 642-647):** “The iPGS model for Europeans incorporated 10 GIGASTROKE GWAS (all stroke types, using the European and cross-ancestry analysis) and 12 vascular risk trait GWAS (Extended Data Fig. 9, Supplementary Table 40). The iPGS model achieved a Δ C-index of 0.027 (Supplementary Table 41), 93% higher than that for a previously constructed iPGS model for Europeans, derived from 5 MEGASTROKE GWAS and similar vascular risk trait GWAS (Δ C-index=0.014).”
- **Results (p. 23, line 660-664):** “The iPGS model was constructed by integrating 10 GIGASTROKE GWAS and 12 vascular risk trait GWAS (Extended Data Fig. 9, Supplementary Table 44). The iPGS model for East-Asians showed an improvement in AUC (Δ AUC) of 0.019 (Figure 4a and Supplementary Table 45). The age-, sex-, and top 5 PC-adjusted odds ratio (OR) per SD of PGS was 1.33 (95%CI, 1.26–1.40; $P=9.9\times 10^{-26}$) for the iPGS model.”
- **Methods (p. 12, line 327-329):** “To derive the European iPGS model, we incorporated 5 ancestry-specific and 5 cross-ancestry stroke GWAS (AS, AIS, LAS, SVS, and CES) from the GIGASTROKE project, and 12 GWAS of vascular risk traits from other groups (Extended Data Fig. 9).”
- **Method (p.13, first paragraph line 344-347):** “To derive the East-Asian iPGS model, we incorporated 5 ancestry-specific and 5 cross-ancestry stroke GWAS (AS, AIS, LAS, SVS, and CES) from the GIGASTROKE project, and 12 GWAS of vascular risk traits (Extended Data Fig. 9).”
 - “It would be helpful to separate the contribution from stroke GWAS and secondary GWAS to the prediction”:

To separate the contribution of stroke GWAS and risk factor (secondary) GWAS, we compared the performance of iPGS models with and without risk factors GWAS.

The Table below shows the predictive ability of iPGS models for Europeans, which was evaluated in the EstBB validation cohort. The iPGS model derived from stroke GWAS only showed a significant association with the incidence of AIS (HR per 1 SD increase, 1.14; $P = 1.6\times 10^{-5}$) and a modest improvement in C-index (Δ C-index = 0.006). By incorporating risk factor GWAS in addition to stroke GWAS, the iPGS model achieved stronger association with AIS incidence (HR per 1 SD increase, 1.26; $P = 2.0\times 10^{-15}$) and higher improvement in C-index (Δ C-index = 0.027).

Table: Comparison of predictive ability of European iPGS models with and without risk factor GWAS

Derivation GWAS	#. GWASs	C-index	Δ C-index	HR (95% CI)	P
Stroke GWAS	10	0.609	0.006	1.14 (1.07–1.21)	1.6E-05
Stroke GWAS + risk factor GWAS	22	0.631	0.027	1.26 (1.19–1.34)	2.0E-15

CI indicates confidence interval; GWAS, genome-wide association study; HR, hazard ratio. Δ C-index means the improvement in C-index over a base model that includes age, sex, and top 5 genetic principal components.

The following Table below shows the predictive ability of iPGS models for East Asians, which was evaluated in the BBJ validation case-control dataset. The iPGS model derived from stroke GWAS was associated with the odds of AIS (OR per 1 SD increase, 1.30; $P = 3.1 \times 10^{-22}$) and achieved a substantial improvement in predictive ability (Δ AUC = 0.016). By incorporating risk factor GWAS in addition to stroke GWAS, the iPGS model showed slightly stronger association with AIS (OR per 1 SD increase, 1.33; $P = 9.9 \times 10^{-26}$) and slightly higher improvement in AUC (Δ AUC = 0.019).

Table: Comparison of predictive ability of East Asian iPGS models with and without risk factor GWAS

Derivation GWAS	#. GWASs	AUC	Δ AUC	OR (95% CI)	P
Stroke GWAS	10	0.650	0.016	1.30 (1.23–1.37)	3.1E-22
Stroke GWAS + risk factor GWAS	22	0.653	0.019	1.33 (1.26–1.40)	9.9E-26

AUC indicates area under the curve; CI, confidence interval; GWAS, genome-wide association study; OR, odds ratio.

Δ AUC means the improvement in AUC over a base model that includes age, sex, and top 5 genetic principal components.

These results suggested that the contribution of risk factor GWAS was important to improve the predictive ability of iPGS models both in Europeans and East Asians, but the degree of importance seemed to be different between the two ancestries.

These results can be found in Supplementary Tables 41 and 45.

- PGS in clinical setting: Is there any reason not to use the genome-wide PGS or iPGS constructed by P+T, LDpred or PRSs in this analysis?

In the clinical trial setting we had used a GRS approach based on independent genome-wide significant risk loci for stroke, as in a previous study using a similar approach based on an earlier stroke GWAS (Abraham et al., 2019), in order to more specifically assess the added value of the new (GIGASTROKE) over the older (MEGASTROKE) GWAS dataset. Moreover, we found it interesting and complementary to present both an elaborate iPGS and a pragmatic GRS-based approach. However, in this revised version we have now also used the clinical trial data for iPGS evaluation.

In this trial setting, the GIGASTROKE-based iPGS was significantly associated with the incidence of AIS (HR per 1 SD increase, 1.19; $P = 3.17 \times 10^{-7}$) and a certain improvement in C-index (Δ C-index = 0.008). Compared to the MEGASTROKE-based iPGS model, the GIGASTROKE-based iPGS model showed a stronger association and a larger improvement in C-index.

Table: Predictive ability of iPGS model assessed using the datasets of clinical settings

PGS model	C-index	Δ C-index	HR (95% CI)	P
MEGASTROKE-based European iPGS model	0.641	0.006	1.14 (1.06–1.21)	1.76E-04
GIGASTROKE-based European iPGS model	0.644	0.008	1.19 (1.11–1.27)	3.17E-07

Thus, very similar confirmatory results were obtained for the iPGS in the clinical trial setting and in the setting of the population-based MVP cohort. Moreover, as for the GRS-based analysis, these results suggest an improved predictive performance of the GIGASTROKE-based iPGS model compared to the MEGASTROKE-based iPGS model.

These updates have been reflected in the revised manuscript:

- **Supplementary Table 53:** “Improvement of predictive ability achieved by PGS models for Europeans in trial participants”
- **Supplementary Table 54:** “Association of final iPGS model for Europeans with the risk of ischemic stroke in European trial participants”
- **Results (p. 25, end of last paragraph line 715-720):** “Finally, in European trial participants (East-Asians were too few for this analysis), the GIGASTROKE-based iPGS was also significantly associated with increased AIS incidence (HR per 1 SD increase, 1.19 [1.11-1.27]; $P = 3.2 \times 10^{-7}$, ΔC -index = 0.008), performing better than the MEGASTROKE-based iPGS (Supplementary Table 53). Compared to the middle 10% of the subjects, those in the top 1% had a 2.8-fold higher hazard of AIS (HR=2.78 [95% CI, 1.67–4.61]; $P=7.9 \times 10^{-5}$) (Figure 4d and Supplementary Table 54).”
- **Discussion (p. 28, line 797):** “We further confirmed the GIGASTROKE iPGS in this clinical trial setting.”
- **Methods (p. 12, line 335-337):** “The iPGS model for Europeans was further evaluated in two external cohorts of European ancestry (MVP and pooled data of clinical trials) as well as in two studies of African-ancestry participants (MVP and SIREN).”

- Methods Page 15 Line 335: "1,470 IS cases and 40,459 controls in the model training dataset" -> "... model evaluation dataset"

→ We apologize this typo that we have corrected this:

Methods, p. 12, line 338-340: “For the East-Asian iPGS model we used BBJ data for the model training and evaluation. The model training dataset was composed of 577 AIS cases and 9,232 controls, whereas there were 1,470 AIS cases and 40,459 controls in the model evaluation dataset.”

Referee #2 (Remarks to the Author):

Mishra et al. present a genome-wide association study (GWAS) of 110,000 cases of prevalent or incident stroke with extensive computational follow-up analyses. This sample is 40% larger than the previous largest study (MEGASTROKE), and importantly, includes more individuals from diverse ancestries. While the current study reveals incrementally more genomic loci, credible sets, genes, and pathways—which I would argue is mainly of interest to an expert statistical genetics audience—I consider the following findings of particular importance and expect them to appeal to readers across disciplines:

- Dozens of potential drug candidates for stroke are highlighted in the study by combining stroke genetics with public transcriptomic and proteomic data. The large stroke GWAS sample size and innovative use of multiple drug discovery methods lead to compelling evidence for F11 and KLKB1 as drug targets, with the former likely having minimal off-target effects.

- Genetic risk scores from the cross-ancestry GWAS provide meaningful improvement in stroke incidence prediction in addition to standard clinical risk factors, for individuals with EUR ancestry and individuals with EAS ancestry. (Granted I am not qualified to assess whether the selected clinical

risk factors used were appropriate)

I believe the study is methodologically sound and the conclusions are largely supported by the evidence presented. There are a small number of statistical checks I expect the authors to perform to verify their results, as outlined in the “minor comments” section below.

My only issue with the current work is its presentation of newly discovered genomic loci. The authors acknowledge their study lacks a formal replication dataset—which is understandable for a GWAS of this size—but despite this limitation, they are not conservative with their definition of stroke risk loci and genes.

We would specifically like to point out that referees have raised concerns about the confidence in the novel reported loci and we ask that you provide further supporting evidence (and replication as far as that is possible).

→ We thank the editor and the referees for raising this point. In order to address this point we gathered an independent dataset of 89,084 stroke patients (of which 85,546 ischemic strokes; 70.0% EUR, 15.6% AFR, 10.1% EAS, 4.1% HIS, and 0.1% SAS) and 1,013,843 controls, mostly from large biobanks, for external replication (Table A).

Following the referees’ advice, we have now followed up genome-side significant stroke risk loci both internally and externally. First, we sought to replicate the 42 stroke risk loci reaching genome-wide significance in individual ancestries in at least one other ancestry group among the discovery samples. Second, we aimed for replication in the additional independent dataset (Table A). The biobank setting did not allow suitable ischemic stroke subtype analyses.

Table A: GIGASTROKE external follow-up samples

Total ALL discovery		AS	AIS	Controls
EUR	MVP	39 165	39 165	425 512
	FinnGen	9027	8140	115 190
	FinnHosp	1600	1600	800
	Interstroke	1312	1223	1577
	MGB	3745	3745	28 281
	BioVU	4686	3582	60 936
	EstBB	1571	1372	4062
	PMBB	1223	1018	28795
Total EUR		62 329	59 845	665 153
AFR	MVP	11 126	11 126	111 914
	Interstroke	356	245	380
	PMBB	730	647	10 309
	SIREN	1691	1691	1743
Total AFR		13 903	13 709	124 346
HIS	MVP	3 234	3 234	48 901
	Interstroke	400	288	422
Total HIS		3 634	3 522	49 323
EAS	Interstroke	162	97	184
	CKBB	5692	5692	10 805
	TBB	2064	1399	86 283
	CRCS-K and KBA GWAS	1120	1116	77 583
Total EAS		9 038	8 304	174 855
SAS	Interstroke	114	100	143
Total SAS		114	100	143
Total ALL follow-up		89 018	85 480	1 013 820

Based on these follow-up results we characterized the level of confidence of identified loci as follows: high confidence in case of significant internal ‘cross-ancestry’ and/or external replication after accounting for the number of loci tested, or nominally significant replication in both internal and external replication, or evidence of involvement in monogenic stroke; intermediate confidence in case of nominal significance in either internal ‘inter-ancestry’ or external replication but not both; and low confidence in the absence of formal replication.

Overall, out of the 60 loci reaching genome-wide significance in the main IVW GWAS meta-analysis, 52 (87%) replicated at $p < 0.05$ with consistent direction, of which 37 (61.7%) with high confidence, and 15 with intermediate confidence (25%). The 8 loci that did not replicate were labeled as “low confidence”. Four of these were ethnic specific and 3 were low frequency variants that were monomorphic in some ancestries, limiting our ability for replication.

Within the secondary analyses (MR-MEGA and MTAG), none of the 3 MR-MEGA loci replicated, although one was borderline significant (Supplementary Table 16). Of the 26 MTAG loci, 18 (69%) replicated with AS or AIS at $p < 0.05$, of which 9 (35%) with high confidence. Of the 8 MTAG loci that did not replicate, 7 showed a consistent directionality (borderline significant for one), and 4 were subtype-specific, limiting our ability for replication with AS or AIS.

While we have clearly labeled “low confidence” variants, we have not removed them from bioinformatics functional follow-up analyses. Indeed, we feel that despite the important worldwide

effort that enabled to gather nearly 90,000 additional stroke cases, several issues still affect our ability to replicate some of the identified stroke risk loci:

- limits of statistical power, considering a smaller sample size than in the discovery and the winner's curse phenomenon;
- we cannot rule out some degree of misclassification in the follow-up samples that were, with two smaller exceptions, nearly exclusively derived from large biobanks with stroke ascertainment based on ICD codes only (Turnbull, Lancet Reg Health West Pac. 2022; Rannikmae, Neurology 2020), while a large proportion of stroke cases in the discovery were recruited and deeply phenotyped in a hospital-based setting;
- a substantial proportion of genetic risk for stroke is subtype specific, which is not fully captured in the replication because of the limited availability of stroke subtype data

In this revised manuscript we have:

- included the replication results and grading in **Supplementary Tables 14-16**

- revised **Figure 1** to provide information on confidence levels for each locus:

- amended the text as follows:

Results (p. 16, line 436-437): *"Next, we conducted a secondary cross-ancestry meta-analysis with MR-MEGA,¹² which accounts for the allelic heterogeneity between ancestries."*

Results (p. 16, line 440-442): *"To further enhance statistical power for AIS subtypes, we conducted secondary multi-trait analyses of GWAS (MTAG)¹³ in Europeans and East-Asians, including traits correlated with specific stroke subtypes, (...)"*

Results (p. 16, line 450-454): *"This brings the number of identified stroke risk loci from primary (IVW) and secondary (MR-MEGA and MTAG) analyses to 89 in total (61 novel), of which 69 were associated with AS, 45 with AIS, 15 with LAS, 13 with CES, and 10 with SVS (of these 44, 33, 11, 8, and 3 were novel respectively, Fig. 1, Supplementary Table 4, 8, and 9-14."*

Results (p. 16 last paragraph line 455-459 and p. 17 first paragraph, line 460-479): *"Independent follow-up of GWAS signals*

We followed up genome-side significant stroke risk loci both internally and externally. First, we sought to replicate the 42 stroke risk loci reaching genome-wide significance in individual ancestries in at least one other ancestry group among the discovery samples. We successfully replicated, in a consistent direction, 10 of these loci at $p < 1.19 \times 10^{-3}$ (accounting for the number of loci tested), of which 7 were genome-wide significant in EUR, 1 in EAS, and 2 in both EUR and EAS. Additional 15 loci showed nominal association ($p < 0.05$) in at least one other ancestry (Supplementary Table 15).

Second, we gathered an independent dataset of 89,084 AS (of which 85,546 AIS; 70.0% EUR, 15.6% AFR, 10.1% EAS, 4.1% HIS, and 0.1% SAS) and 1,013,843 controls, mostly from large biobanks, for external replication (the biobank setting did not allow suitable ischemic stroke subtype analyses). Out

of the 60 loci reaching genome-wide significance in IVW meta-analyses, 48 loci (80%) replicated at $p < 0.05$ with consistent directionality, of which 31 (52%), at $p < 8.2 \times 10^{-4}$ (accounting for the number of loci tested) (Supplementary Table 16). When considering both the internal and external follow-up, 52 (87%) of the 60 IVW loci replicated, of which 37 with “high confidence”, and 15 with “intermediate confidence” (Methods, Supplementary Table 14). The 8 loci that did not replicate were labeled as “low confidence” (Methods, Supplementary Table 14). Four of these were ethnic specific and 3 were low frequency variants that were monomorphic in some ancestries, restricting our ability to obtain replication.

Within the secondary analyses, none of the 3 MR-MEGA loci replicated, although one was borderline significant (Supplementary Table 16). Of the 26 MTAG loci, 18 (69%) replicated with AS or AIS at $p < 0.05$, of which 9 (35%) with high confidence ($p < 1.7 \times 10^{-3}$, accounting for 29 secondary loci tested, Supplementary Table 16). Of the 8 MTAG loci that did not replicate, 7 showed a consistent directionality (borderline significant for one), and 4 were subtype-specific and hence with less power to detect associations with AS or AIS.”

Discussion (p. 26, line 723-731): “Our GWAS meta-analyses gathering 110,182 stroke patients and 1,503,898 controls from five different ancestries (33% non European stroke patients) identified 89 risk loci for stroke and stroke subtypes (60 through primary IVW and 29 through secondary MR-MEGA and MTAG analyses), of which 61 were novel. We observed substantial shared susceptibility to stroke across ancestries, with strong correlation of effect sizes. Based on internal cross-ancestry validation and independent external follow-up in 89,084 stroke cases (30% non European) and 1,013,843 controls, mostly from large biobanks with information on AS and AIS only, the level of confidence of these loci was intermediate or high for 87% of primary stroke risk loci and 60% of secondary loci.”

Discussion (p. 28, last paragraph, line 803-809): “We provided independent validation of the vast majority of identified genome-wide significant associations and graded our loci by level of confidence based on these findings. Despite the notable size of the follow-up study sample, with nearly 90,000 additional stroke patients, we remain limited in our ability to validate low frequency variants, ancestry- and subtype-specific associations in particular. In regards to the latter, most of the follow-up studies were derived from large biobanks with event ascertainment based on electronic health records and without suitable stroke subtype information.”

Methods (p. 1, line 3-5): “Information on participating studies (discovery and follow-up), study design, and definition of stroke and stroke subtype is provided in the Supplementary Appendix. Population characteristics of individual studies are provided in Supplementary Table 1.”

Methods (p. 1, line 9-10): “Genotyping methods, pre-imputation quality control (QC) of genotypes and imputation methods of individual cohorts (discovery and follow-up) are presented in Supplementary Table 2.”

Methods (p. 3 second paragraph, line 65-80): “First, we sought to replicate “internally” the 42 stroke risk loci reaching genome-wide significance in IVW meta-analyses within individual ancestries, in at least one other ancestry group among the discovery samples, considering both nominal replication

levels ($p < 0.05$) and multiple testing corrected significance at $p < 1.19 \times 10^{-3}$ (0.05/42). Second, we gathered independent datasets totaling 89,084 AS (of which 85,546 AIS; 70.0% EUR, 15.6% AFR, 10.1% EAS, 4.1% HIS, and 0.1% SAS) and 1,013,843 controls for “external” replication of associations with AS and AIS (Supplementary Table 1-2). These comprised 8 biobanks (82,263/930,988 cases/controls) and 4 hospital-based cohorts (6,821/82,855 cases/controls). We considered both nominal replication levels ($p < 0.05$) and multiple testing corrected significance at $p < 8.2 \times 10^{-4}$ (0.05/60) and $p < 1.3 \times 10^{-3}$ (0.05/29) for follow-up of genome-wide significant loci from the IVW and the MR-MEGA/MTAG meta-analyses respectively. We considered stroke risk loci as high confidence in case of significant internal ‘cross-ancestry’ and/or external replication after accounting for the number of loci tested, or nominally significant replication in both internal and external replication, or evidence of involvement in monogenic stroke; intermediate confidence in case of nominal significance in either internal ‘inter-ancestry’ or external replication but not both; and low confidence in the absence of formal replication.”

Main comments:

The authors claim to identify 89 risk loci for stroke and stroke subtypes (61 novel) based on a genome-wide association threshold of 5×10^{-8} .

However, the authors did not adjust any of the input GWAS for their LD-score regression intercept. While they are correct there is no systematic inflation across ancestries, the European GWAS has an inflated intercept of 1.10 and contributes the largest sample to the cross-ancestry meta-analysis. I would expect the standard errors of each ancestry-specific set of GWAS summary statistics to be adjusted for any inflation before meta-analysis to minimize false positive discoveries.

→ We thank referee 2 for their suggestion. We have now additionally performed cross-ancestry GWAS meta-analyses accounting for the LD-score regression intercept observed in each ancestry specific GWAS meta-analysis. With the exception of two loci (THADA, $p = 6.40 \times 10^{-8}$, AS, and LPA, $p = 7.53 \times 10^{-8}$, LAS), all other loci remained genome-wide significant in our analyses. Of note, some loci that did not reach genome-wide significance in cross-ancestry meta-analyses controlling for LD score intercepts for the main phenotype were genome-wide significant in LD score intercept corrected cross-ancestry meta-analyses of other phenotypes and/or were also identified in ancestry-specific analyses (footnote of supplementary table 4). LPA is biologically relevant to stroke pathology, with variants in this gene previously associated with stroke risk. (Langsted, J Am Coll Cardiol 2019) Moreover, in our follow-up of genome-wide significant loci we also observed a strong association of the LPA locus and a nominally significant association of the THADA locus with AS and AIS in independent datasets.

The manuscript has been edited accordingly:

Results of the sensitivity cross-ancestry GWAS meta-analyses accounting for the LD-score regression intercept have been included in Supplementary Table 4.

Results (p 15, line 423-429): *To our knowledge, our results include the most comprehensive and largest description of stroke genetic risk variants to date in each of the five represented ancestries. In cross-ancestry meta-analyses 53 loci (51 loci after controlling for ancestry specific LD score intercepts) reached genome-wide significance (Supplementary Table 4),*

Methods (p 2, line 30-34): *“We applied the covariate adjusted LD score regression (cov-LDSC) method to ancestry-specific GWAS meta-analyses without GC correction to test for genomic inflation and to compute robust SNP-heritability estimates in admixed populations.⁶⁴ We conducted cross-ancestry GWAS meta-analyses without genomic correction and with correction of the LD score intercept for genomic inflation observed in individual ancestry-specific GWAS.”*

Secondly, throughout the manuscript, the authors investigate three stroke subtypes (LAS, CES, SVS) and two stroke summary traits (AS, AIS), but do not adjust for multiple testing of these phenotypes. It is clear these stroke traits are correlated, but the authors do not quantify this correlation, and (from a statistical point of view) treat their five phenotypes as one. The authors should either 1) quantify the (in)dependence of their traits and adjust any significance thresholds for the number independent components (e.g. using genetic correlation and PCA) or 2) qualify statements regarding the number of discovered loci and genes.

→ We thank the Referee for raising this point. To clearly identify the number of independent traits in our analysis, we performed spectral decomposition to calculate V_{eff} (Nyholt, AJHG 2004) and V_{effLi} (Li & Li, Heredity 2005) as implemented in PhenoSpD (Zheng et al, Gigascience 2018), based on the GWAS summary statistics of stroke (sub)types. Both measures have been shown to accurately reflect the observed phenotypic correlation. When using all five traits (AS, AIS, LAS, CES and SVS), the minimum value of V_{eff} and V_{effLi} was 4. When only using ischemic stroke subtypes (LAS, CES and SVS), the minimum value of V_{eff} and V_{effLi} was 2.95. Overall, while AS and AIS are, as expected, highly correlated ($r=0.93$), ischemic stroke subtypes only show a weaker correlation ($r=0.14$ to 0.17). Conceptually, however, we are looking at subtypes of the same phenotypic entity. Moreover, as we have now introduced follow-up analyses (both internally and externally, in nearly 90,000 additional stroke cases), we propose to describe the level of confidence of identified risk loci primarily based on the level of replication.

Thus, as described above, we have now introduced a grading system providing the level of confidence of identified loci (Methods, p. 3, end of second paragraph) : *“We considered stroke risk loci as high confidence in case of significant internal ‘cross-ancestry’ and/or external replication after accounting for the number of loci tested, or nominally significant replication in both internal and external replication, or evidence of involvement in monogenic stroke; intermediate confidence in case of nominal significance in either internal ‘inter-ancestry’ or external replication but not both; and low confidence in the absence of formal replication.”*

Minor comments:

Were summary statistics checked for potential sample overlap before meta-analysis? For example, the METASTROKE analysis appears to contain the WHI cohort, which is also listed as a new dataset in Supplementary Table 1. It would be useful to see the phenotypic correlations between individual cohort GWAS to rule out any sample overlap, using e.g. the cross-trait LD-score regression intercept or alternative methods (see <https://doi.org/10.3389/fgene.2021.665252>),

→ We thank the Referee for raising this point. Indeed, WHI has contributed both to METASTROKE (as part of a meta-analysis performed within the SiGN initiative, combining WHI and numerous additional samples of mostly “prevalent” stroke in a hospital-based setting) and to the incident stroke GWAS meta-analysis. However, for the overall meta-analysis comprising both incident and prevalent stroke cases, only the contribution of WHI to METASTROKE was used. To illustrate this, in Supplementary Table 1 the cells for “incident SiGN (WHI)” (line 20) are empty in the incident+prevalent columns. A similar scenario occurs of samples from the longitudinal HISAYAMA, REGARDS, and JHS studies, which were used for the specific incident stroke GWAS, but had also contributed to a previous meta-analysis of (mostly prevalent) stroke GWAS (Japanese GWAS meta-analysis of MEGASTROKE [Hisayama] and METASTROKE [REGARDS, JHS]). To make this clearer we have added the following sentence in the Methods:

Methods (p. 2, end of first paragraph line 34-38): *“We conducted separate GWAS of incident AS and AIS (N=32,903 and 16,863) in longitudinal population-based cohort studies. For the meta-analysis combining both incident and prevalent stroke studies, a few incident stroke studies were removed, because they were already part of a meta-analysis of stroke GWAS used as an input of the overall meta-analysis (WHI, Hisayama, REGARDS, JHS).”*

Mendelian randomisation can be quite susceptible to horizontal pleiotropy and reverse causality. However, the authors do not test for reverse causality in their MR analysis of vascular risk factors on stroke traits. As they have GWAS summary statistics for all phenotypes, they should perform a bidirectional MR. This step is particularly important when assessing the causal role of BMI and WHR, where alternative pathways could explain the observed association with stroke phenotypes.

→ We thank the Referee for this comment. We share the Referee’s opinion that Mendelian randomization estimates may be influenced by reverse causality, especially when such large datasets are used for performing the analyses. As recommended, to minimize risk for reverse causality, we now performed:

- (i) the Steiger test for directionality in the original analyses
- (ii) bidirectional Mendelian randomization using the stroke phenotypes as exposure.

The Steiger test compares the variance of the exposure (r^2) and the outcome explained by the variants included as instruments to the model. The Steiger test relies on the assumption that if an association derived from Mendelian randomization is due to reverse causality, then the r^2 of the exposure explained from the instruments will be lower than that of the outcome (Hemani et al, PLoS

Genet. 2017 Nov; 13(11): e1007081). In our main analyses, for all of our exposure-outcome traits, the Steiger test confirmed that the directionality of the tested associations was correct and robust to potential reverse causality, both in Europeans and East-Asians (Supplementary Table 24).

As suggested by the reviewer we additionally performed reverse Mendelian randomization analyses (Supplementary Table 26). Using the same criteria for instrument selection as for our exposure traits, we ended up with 21, 23, 1, and 7 genetic instruments for Any stroke, Any ischemic stroke, Large artery stroke, and Cardioembolic stroke, respectively. No variants fulfilled our instrument criteria for Small vessel stroke. Furthermore, because of the substantially lower power in the East Asian GIGASTROKE sample, we performed this analysis only for the European sample. The reverse Mendelian randomization analyses indeed showed some associations between the stroke traits and the risk factors we used as exposures in our original analyses. However, out of the 17 significant associations, all but 1 failed the Steiger test (association between cardioembolic stroke and venous thromboembolism). This analysis showed however high heterogeneity ($p=3.8 \times 10^{-114}$), was not consistent across the different methods, and the Egger intercept was significant ($p=0.009$) implying directional pleiotropy towards a positive association.

All in all, our sensitivity analyses confirm that our Mendelian randomization results are robust to potential reverse causality.

These results from the Steiger test have been added to Supplementary Table 24 and the results from the reverse Mendelian randomization analyses are presented in Supplementary Table 26.

Furthermore, we now report on these findings in the Results of our manuscript, as follows:

Results (p. 19, line 540-542): *“We confirmed directionality with the Steiger test (Supplementary Table 24) and ruled out reverse causation with reverse MR (Supplementary Table 26).”*

Why do the authors use VEGAS to aggregate genome-wide P values into gene-wide association statistics, but then use MAGMA to calculate these gene-wide statistics when selecting genes as GREP input? I also note the MAGMA results are not provided in any Supplementary Tables, and can therefore not be compared with VEGAS2 results.

→ We thank the referee for their comment. We have now run both MAGMA and VEGAS2 and included a table showing gene-based associations derived using both VEGAS2 and MAGMA approaches (**Supplementary Table 6-7**). In line with the literature,(De Leeuw, PLoS Comput Biol 2015) we observed very similar gene-based findings from both VEGAS2 and MAGMA approaches.

→ **The manuscript has been amended as follows:**

Results (p. 16, line 431-435): *“We also performed cross-ancestry gene-based association tests using VEGAS2⁹ and MAGMA¹⁰, which revealed 267 gene-wide significant associations ($p < 2.63 \times 10^{-6}$) in 39 loci, of which 14 were in 8 novel loci not reaching genome-wide significance in the single-variant analyses (AGAP5/SYNPO2L/SEC24C/CHCHD1, CD96, HNRNPA0, MAMSTR, PPM1H, RALGAPA1, USP34, and USP38, Supplementary Table 6-7).”*

Methods (p. 3, line 82-85; p. 4, line 86-93): *“We performed gene-based tests of common variant associations using the using VEGAS29 and MAGMA.10 Both VEGAS2 and MAGMA considered variants in the gene or within 10kb on either side of a gene’s transcription site were used to compute a gene-based p-value. We performed MAGMA tests using default parameters, whereas the VEGAS2 analyses were performed using the ‘-top 10’ parameter that that tests enrichment of the top 10% variants assigned to a gene accounting for LD between variants and total number of variants within a gene. We used 1000 Genomes phase 3 continental reference samples European, East-Asian, African, South-Asian and South-American (for our Hispanic samples), to compute LD between variants for respective ancestry-specific gene-based analyses. We then meta-analyzed ancestry-specific gene-based results, using Stouffer’s method for sample size weighted combination of P-values. Gene-wide significance was defined as $p < 2.72 \times 10^{-6}$, correcting for 18,371 autosomal protein-coding genes tested.”*

We also re-ran the drug target enrichment analysis with GREP using the gene-based association results not only from MAGMA, but also from VEGAS2 (in response to a comment from Referee 1). All previously significant enrichments are maintained, and two additional genes were prioritized: F2 and TFPI, targets of Lepirudin and Dalteparin respectively, both involved in the coagulation process and used for treatment of recurrent thromboembolism.

→ The manuscript has been amended as follows :

Results (p. 20, line 569-577): *“First, using GREP²² we observed significant enrichment of stroke-associated genes (MAGMA¹¹ or VEGAS2¹⁰ false discovery rates [FDR] < 0.05) in drug-target genes for blood and blood-forming organs (Anatomical Therapeutic Chemical Classification System [ATC] B drugs, for AS, AIS, and CES). This encompasses the previously described PDE3A and FGA genes,²⁴ encoding targets for cilostazol (antiplatelet agent) and alteplase (thrombolytic drug acting via plasminogen²⁵), respectively, as well as F11, KLKB1, F2, TFPI, and MUT encoding targets for conestat alfa, ecallantide (both used for hereditary angioedema), lepirudin, dalteparin (both used for recurrent thromboembolism), and vitamin B12, respectively (Supplementary Table 31).”*

Methods (p. 9, line 233-235): “We ran MAGMA²³ and VEGAS2¹⁰ to summarize variant-level p-values into gene-level and used the genes with false discovery rates (FDR) less than 0.05 in either MAGMA or VEGAS2 as the disease-risk genes.”

Supplementary Table 31 has been updated.

Editorial comments:

Line 115 - It is not mentioned what weights were used in Stouffer’s p value meta-analysis of VEGAS2 pathway results.

→ We thank the Referee for this question. In the methods section we now included the text reporting weights used in Stouffer’s p value meta-analysis of VEGAS2 pathway results.

Methods (p. 4, line 97-101): “For each stroke phenotype, we meta-analyzed the ancestry-specific pathway association p-values using Stouffer’s method considering the number of cases in each ancestry specific GWAS; e.g. for AS we considered the number of cases 73,652, 27,413, 3,961, 1,516, and 3,640 in EUR, EAS, AFR, HIS, SAS specific GWAS to combine the respective ancestry-specific pathway association p-values.”

Line 170 - the phrase “checking the LD pattern” does not make it entirely clear which criteria were used to justify removing specific credible sets.

→ We thank the reviewer for this question and apologize for not being clearer in describing our methodology. We illustrate our approach to identify false positives on one specific example: In EAS there was one locus for AS (SH2B3, chr12:111910219) that produced 8 credible sets, which could potentially indicate a false positive finding. To identify potential LD mismatches, we used the diagnostic vignette accompanied with SuSiE (https://cran.r-project.org/web/packages/susieR/vignettes/susierss_diagnostic.html) and compared the expected z-scores vs. observed z-scores.

Figure 1A below shows the SuSiE result before curation, variants are outlined by color according to the credible set.

Figure 1B shows an outlier 12:112645401 which was in high LD with the lead variant ($r^2=0.782$, $z=7.29$) but had observed $z=0.88$, while the expected z-score was 6.69, flagging an inconsistency. In the EAS summary statistics, this variant was missing from 67.25% of the samples ($N_{EAS:112645401}=86,331$, $N_{EAS_total}=263,592$), thus the power for this variant was greatly reduced, leading to statistical fluctuation. Hence, we removed this specific variant and reran the fine-mapping. The results and diagnostic plots are shown below in Figure 2. Only one CS was

identified with consistent observed z scores vs expected z scores. This strategy was repeated for all loci. We now include a sentence in the Methods section:

“Fine-mapping results were checked for potential false-positive findings using a diagnostic procedure implemented in SuSiE. In short, we compared observed and expected z-score for each variant at a given locus and removed the variant if the difference between observed and expected z-score was too high after manual inspection.”

Figure 1. before curating

Figure 2. after curating

Line 269 - Which population/source was used for R2 calculations of proxy instruments in the protein MR & colocalization analysis?

→ We apologize for not having specified this before. We have used 1000G EUR. The methods section has been amended accordingly (p. 10, line 259-261): *“For the lead variants of pQTL that were missing in the stroke GWAS summary statistics, the proxy variants with the largest R2 were used if the R2 was greater than 0.8 (1000G EUR).”*

Figure 2 - error bars are not defined (95% confidence interval or standard error?)

→ We apologize for the omission. Error bars correspond to 95% CI. This has been added to the legend of Figure 2.

The x-axis of Figure 4D is not defined.

→ We apologize for the omission. The x-axis corresponds to time from inclusion in the trial in days. This has been added to the figure and figure legend (now Figure 4e).

Supplementary Table 19 has two sub-tables which are mistakenly labelled 26A and 26B.

→ This has been corrected.

Supplementary Tables 21b, 30, and 38 have coauthor comments which have not been resolved.

→ This has been corrected.

The LDSC label and some of the vascular trait labels in Extended Data Fig 3. are not properly visible in the PDF file.

→ This has been corrected.

Cathie LM Sudlow appears twice in the second tier author list.

→ This has been corrected.

The total participants in the Dutch PSI cohort is reported as 1375 in the Supplementary Appendix, but the number listed in Supplementary Table 1 is 1,377.

→ This has been updated accordingly (N=1,377).

Referee #3 (Remarks to the Author):

In this manuscript, Mishra et al perform a GWAS of stroke and stroke subtypes in over 110,000 cases and 1.5 million controls of diverse ancestries nearly doubling the sample size of the largest prior stroke GWAS (MEGASTROKE). The authors identify many novel stroke loci, and perform a series of downstream analyses including conditional analysis (GCTA COJO), multi trait analysis (MTAG), fine

mapping, enrichment analysis, Mendelian randomization, and a PRS analysis. Most compelling to this referee was the drug discovery analysis, identifying a series of putative stroke drug targets combining multiple lines of evidence. The manuscript represents a landmark paper in the understanding of stroke genetics with substantial novelty in many of its features, and is well written. Most of the conclusions are supported by the data, with appropriate methodology, and the authors cite relevant prior work. However, this referee has several concerns (major/minor) outlined below.

Major concerns:

1) While this manuscript is a multi ancestry meta-analysis, no independent replication for novel genomic loci are provided. While the authors' efforts in aggregating as many cases as possible should be commended, at a minimum some form of internal replication would improve the manuscript. For example, as utilized in a BP GWAS from 2018 (PMID: 30224653), genome-wide significance plus a p value of 0.01 in 2 (roughly) evenly divided strata of participants would reduce concerns for false positive findings. These concerns are additionally important in light of the fact that multiple GWAS in distinct stroke subtypes (AIS, LAS, SVS) are performed without further correction for multiple testing.

We would specifically like to point out that referees have raised concerns about the confidence in the novel reported loci and we ask that you provide further supporting evidence (and replication as far as that is possible).

→ We thank the editor and the referees for raising this point. In order to address this point we gathered an independent dataset of 89,084 stroke patients (of which 85,546 ischemic strokes; 70.0% EUR, 15.6% AFR, 10.1% EAS, 4.1% HIS, and 0.1% SAS) and 1,013,843 controls, mostly from large biobanks, for external replication (Table A).

Following the referees' advice, we have now followed up genome-wide significant stroke risk loci both internally and externally. First, we sought to replicate the 42 stroke risk loci reaching genome-wide significance in individual ancestries in at least one other ancestry group among the discovery samples. Second, we aimed for replication in the additional independent dataset (Table A). The biobank setting did not allow suitable ischemic stroke subtype analyses.

Table A: GIGASTROKE external follow-up samples

Total ALL discovery		AS	AIS	Controls
EUR	MVP	39 165	39 165	425 512
	FinnGen	9027	8140	115 190
	FinnHosp	1600	1600	800
	Interstroke	1312	1223	1577
	MGB	3745	3745	28 281
	BioVU	4686	3582	60 936
	EstBB	1571	1372	4062
	PMBB	1223	1018	28795
Total EUR		62 329	59 845	665 153
AFR	MVP	11 126	11 126	111 914
	Interstroke	356	245	380
	PMBB	730	647	10 309
	SIREN	1691	1691	1743
Total AFR		13 903	13 709	124 346
HIS	MVP	3 234	3 234	48 901
	Interstroke	400	288	422
Total HIS		3 634	3 522	49 323
EAS	Interstroke	162	97	184
	CKBB	5692	5692	10 805
	TBB	2064	1399	86 283
	CRCS-K and KBA GWAS	1120	1116	77 583
Total EAS		9 038	8 304	174 855
SAS	Interstroke	114	100	143
Total SAS		114	100	143
Total ALL follow-up		89 018	85 480	1 013 820

Based on these follow-up results we characterized the level of confidence of identified loci as follows: high confidence in case of significant internal ‘cross-ancestry’ and/or external replication after accounting for the number of loci tested, or nominally significant replication in both internal and external replication, or evidence of involvement in monogenic stroke; intermediate confidence in case of nominal significance in either internal ‘inter-ancestry’ or external replication but not both; and low confidence in the absence of formal replication.

Overall, out of the 60 loci reaching genome-wide significance in the main IVW GWAS meta-analysis, 52 (87%) replicated at $p < 0.05$ with consistent direction, of which 37 (61.7%) with high confidence, and 15 with intermediate confidence (25%). The 8 loci that did not replicate were labeled as “low confidence”. Four of these were ethnic specific and 3 were low frequency variants that were monomorphic in some ancestries, limiting our ability for replication.

Within the secondary analyses (MR-MEGA and MTAG), none of the 3 MR-MEGA loci replicated, although one was borderline significant (Supplementary Table 16). Of the 26 MTAG loci, 18 (69%) replicated with AS or AIS at $p < 0.05$, of which 9 (35%) with high confidence. Of the 8 MTAG loci that did not replicate, 7 showed a consistent directionality (borderline significant for one), and 4 were subtype-specific, limiting our ability for replication with AS or AIS.

While we have clearly labeled “low confidence” variants, we have not removed them from bioinformatics functional follow-up analyses. Indeed, we feel that despite the important worldwide

effort that enabled to gather nearly 90,000 additional stroke cases, several issues still affect our ability to replicate some of the identified stroke risk loci:

- limits of statistical power, considering a smaller sample size than in the discovery and the winner's curse phenomenon;
- we cannot rule out some degree of misclassification in the follow-up samples that were, with two smaller exceptions, nearly exclusively derived from large biobanks with stroke ascertainment based on ICD codes only (Turnbull, Lancet Reg Health West Pac. 2022; Rannikmae, Neurology 2020), while a large proportion of stroke cases in the discovery were recruited and deeply phenotyped in a hospital-based setting;
- a substantial proportion of genetic risk for stroke is subtype specific, which is not fully captured in the replication because of the limited availability of stroke subtype data

In this revised manuscript we have:

- included the replication results and grading in **Supplementary Tables 14-16**

- revised **Figure 1** to provide information on confidence levels for each locus:

- amended the text as follows:

Results (p. 16, line 436-437): *“Next, we conducted a secondary cross-ancestry meta-analysis with MR-MEGA,¹² which accounts for the allelic heterogeneity between ancestries.”*

Results (p. 16, line 440-442): *“To further enhance statistical power for AIS subtypes, we conducted secondary multi-trait analyses of GWAS (MTAG)¹³ in Europeans and East-Asians, including traits correlated with specific stroke subtypes, (...)”*

Results (p. 16, line 450-454): *“This brings the number of identified stroke risk loci from primary (IVW) and secondary (MR-MEGA and MTAG) analyses to 89 in total (61 novel), of which 69 were associated with AS, 45 with AIS, 15 with LAS, 13 with CES, and 10 with SVS (of these 44, 33, 11, 8, and 3 were novel respectively, Fig. 1, Supplementary Table 4, 8, and 9-14).”*

Results (p. 16 last paragraph line 455-459 and p. 17 first paragraph, line 460-479): *“Independent follow-up of GWAS signals”*

We followed up genome-side significant stroke risk loci both internally and externally. First, we sought to replicate the 42 stroke risk loci reaching genome-wide significance in individual ancestries in at least one other ancestry group among the discovery samples. We successfully replicated, in a consistent direction, 10 of these loci at $p < 1.19 \times 10^{-3}$ (accounting for the number of loci tested), of which 7 were genome-wide significant in EUR, 1 in EAS, and 2 in both EUR and EAS. Additional 15 loci showed nominal association ($p < 0.05$) in at least one other ancestry (Supplementary Table 15).

Second, we gathered an independent dataset of 89,084 AS (of which 85,546 AIS; 70.0% EUR, 15.6% AFR, 10.1% EAS, 4.1% HIS, and 0.1% SAS) and 1,013,843 controls, mostly from large biobanks, for external replication (the biobank setting did not allow suitable ischemic stroke subtype analyses). Out

of the 60 loci reaching genome-wide significance in IVW meta-analyses, 48 loci (80%) replicated at $p < 0.05$ with consistent directionality, of which 31 (52%), at $p < 8.2 \times 10^{-4}$ (accounting for the number of loci tested) (Supplementary Table 16). When considering both the internal and external follow-up, 52 (87%) of the 60 IVW loci replicated, of which 37 with “high confidence”, and 15 with “intermediate confidence” (Methods, Supplementary Table 14). The 8 loci that did not replicate were labeled as “low confidence” (Methods, Supplementary Table 14). Four of these were ethnic specific and 3 were low frequency variants that were monomorphic in some ancestries, restricting our ability to obtain replication.

Within the secondary analyses, none of the 3 MR-MEGA loci replicated, although one was borderline significant (Supplementary Table 16). Of the 26 MTAG loci, 18 (69%) replicated with AS or AIS at $p < 0.05$, of which 9 (35%) with high confidence ($p < 1.7 \times 10^{-3}$, accounting for 29 secondary loci tested, Supplementary Table 16). Of the 8 MTAG loci that did not replicate, 7 showed a consistent directionality (borderline significant for one), and 4 were subtype-specific and hence with less power to detect associations with AS or AIS.”

Discussion (p. 26, line 723-731): “Our GWAS meta-analyses gathering 110,182 stroke patients and 1,503,898 controls from five different ancestries (33% non European stroke patients) identified 89 risk loci for stroke and stroke subtypes (60 through primary IVW and 29 through secondary MR-MEGA and MTAG analyses), of which 61 were novel. We observed substantial shared susceptibility to stroke across ancestries, with strong correlation of effect sizes. Based on internal cross-ancestry validation and independent external follow-up in 89,084 stroke cases (30% non European) and 1,013,843 controls, mostly from large biobanks with information on AS and AIS only, the level of confidence of these loci was intermediate or high for 87% of primary stroke risk loci and 60% of secondary loci.”

Discussion (p. 28, last paragraph, line 803-809): “We provided independent validation of the vast majority of identified genome-wide significant associations and graded our loci by level of confidence based on these findings. Despite the notable size of the follow-up study sample, with nearly 90,000 additional stroke patients, we remain limited in our ability to validate low frequency variants, ancestry- and subtype-specific associations in particular. In regards to the latter, most of the follow-up studies were derived from large biobanks with event ascertainment based on electronic health records and without suitable stroke subtype information.”

Methods (p. 1, line 3-5): “Information on participating studies (discovery and follow-up), study design, and definition of stroke and stroke subtype is provided in the Supplementary Appendix. Population characteristics of individual studies are provided in Supplementary Table 1.”

Methods (p. 1, line 9-10): “Genotyping methods, pre-imputation quality control (QC) of genotypes and imputation methods of individual cohorts (discovery and follow-up) are presented in Supplementary Table 2.”

Methods (p. 3 second paragraph, line 65-80): “First, we sought to replicate “internally” the 42 stroke risk loci reaching genome-wide significance in IVW meta-analyses within individual ancestries, in at least one other ancestry group among the discovery samples, considering both nominal replication

levels ($p < 0.05$) and multiple testing corrected significance at $p < 1.19 \times 10^{-3}$ (0.05/42). Second, we gathered independent datasets totaling 89,084 AS (of which 85,546 AIS; 70.0% EUR, 15.6% AFR, 10.1% EAS, 4.1% HIS, and 0.1% SAS) and 1,013,843 controls for “external” replication of associations with AS and AIS (Supplementary Table 1-2). These comprised 8 biobanks (82,263/930,988 cases/controls) and 4 hospital-based cohorts (6,821/82,855 cases/controls). We considered both nominal replication levels ($p < 0.05$) and multiple testing corrected significance at $p < 8.2 \times 10^{-4}$ (0.05/60) and $p < 1.3 \times 10^{-3}$ (0.05/29) for follow-up of genome-wide significant loci from the IVW and the MR-MEGA/MTAG meta-analyses respectively. We considered stroke risk loci as high confidence in case of significant internal ‘cross-ancestry’ and/or external replication after accounting for the number of loci tested, or nominally significant replication in both internal and external replication, or evidence of involvement in monogenic stroke; intermediate confidence in case of nominal significance in either internal ‘inter-ancestry’ or external replication but not both; and low confidence in the absence of formal replication.”

2) From this referee's understanding, the authors identify a series of novel loci in their primary IVW analysis, and then perform MTAG to further identify loci with correlated and/or causal stroke traits. This MTAG analysis identifies a series of additional loci, which the authors then claim as novel stroke loci in the abstract/paper. In the opinion of this referee, this representation of the data is misleading. MTAG results should be interpreted as largely exploratory, and not be claimed as distinctly novel stroke loci. This is particularly true in light of the fact that there is likely substantial overlap among samples in these analyses. While MTAG accounts for sample overlap, these results should not be considered "novel stroke loci" with the same level of supportive evidence.

→ We thank the Referee for this comment. We have edited the text accordingly, toning down the MTAG and MR-MEGA results, which are presented as secondary investigations.

Results (p. 16, line 436-437): “Next, we conducted a secondary cross-ancestry meta-analysis with MR-MEGA,¹² which accounts for the allelic heterogeneity between ancestries.”

Results (p. 16, line 440-442): “To further enhance statistical power for AIS subtypes, we conducted secondary multi-trait analyses of GWAS (MTAG)¹³ in Europeans and East-Asians, including traits correlated with specific stroke subtypes, (...)”

Results (p. 16, line 450-454): “This brings the number of identified primary (IVW) and secondary (MR-MEGA and MTAG) stroke risk loci to 89 in total (61 novel), of which 68 were associated with AS, 50 with AIS, 14 with LAS, 12 with CES, and 10 with SVS (of these 45, 35, 11, 10, and 2 were novel respectively, Fig. 1, Supplementary Table 4, 8, and 9-14).”

We also edited the display of Figure 1 to reflect this, as clarified in the updated legend: “loci are characterized as follows: bold with asterisk for loci with high confidence findings based on replication results (Methods); bold without asterisk for loci with intermediate confidence; not bold for loci with low confidence; italic, with the same aforementioned coding, for loci identified in secondary MR-MEGA and MTAG analyses.”

→ Moreover, we have now followed up all genome-wide significant associations derived from primary and secondary analyses of the discovery set.

Results (p. 17, line 474-479): *“Within the secondary analyses, none of the 3 MR-MEGA loci replicated, although one was borderline significant (Supplementary Table 16). Of the 26 MTAG loci, 17 (66%) replicated with AS or AIS at $p < 0.05$, of which 9 (35%) with high confidence ($p < 1.7 \times 10^{-3}$, accounting for number of loci tested, Supplementary Table 16). Of the 8 MTAG loci that did not replicate, 7 showed a consistent directionality (borderline significant for one), and 4 were subtype-specific and hence underpowered to detect associations with AS or AIS.”*

3) In some cases Mendelian randomization analyses are performed with binary exposures -> binary outcomes. There is literature that suggests that this can lead to violation of the exclusion restriction assumption (PMID: 30039250), and that these results should be interpreted in terms of liability (PMID: 34570226). The nuance here is important and should be appropriately addressed by the authors in their results/figures.

→ We thank the Referee for highlighting this important methodological issue and we apologize for not having appropriately addressed this nuance in the original version of our article. To address the Referee's comment, we now:

Acknowledge the limitation of using binary exposure and outcomes in our Mendelian randomization analyses due to violation of the exclusion restriction assumption in the Results, as follows:

Results (p. 19, line 546-548): *“Of note, Mendelian randomization analyses performed with binary exposures should be interpreted with caution due to the potential violations of the exclusion restriction assumption.^{17”}*

Throughout the manuscript and the figure/tables, we now interpret our findings derived from Mendelian randomization for binary exposure traits in terms of genetic liability to these traits. Accordingly, we have rephrased our results, as well as the table and figure legends to address this issue.

4) A substantial amount of the paper focuses on the multi-ancestry applicability of the PRS generated by the authors. While this is interesting, much of this data is more limited in novelty in light of the recent GLGC paper recently published in Nature (PMID: 34887591).

→ We thank the referee for raising this point.

We would like to respectfully point out that our approach to the derivation of polygenic score (PGS) models was substantially different from recent genetics studies including the GLGC publication (PMID: 34887591) because we found that widely used state-of-the-art algorithms, such as P+T, LDpred and PRS-CS, generated PGS models with a limited ability to predict AIS events. Therefore, we sought to incorporate not only AIS GWAS, but also other stroke GWAS (AS, LAS, SVS, and CES) and GWAS of 12 stroke-related phenotypes (*i.e.*, atrial fibrillation [AF], coronary artery disease [CAD],

type 2 diabetes [T2D], systolic and diastolic blood pressure [SBP, DBP], total cholesterol [TC], LDL and HDL cholesterol [LDL-C, HDL-C], triglycerides [TG], body mass index [BMI], height and smoking).

In brief, we created tens of candidate PGS models using the state-of-the-art algorithms (P+T, LDpred, and PRS-CS) with varying parameters for each input GWAS summary data. Then, for each input GWAS, we selected the best PGS model that showed the highest predictive ability in terms of area under the curve (AUC) or C-index using a training dataset of AIS cases and controls. Subsequently, we combined the best PGS models derived from GWAS of various traits with a machine learning method, elastic net logistic regression. The elastic net regression model was referred to as integrative PGS (iPGS) model in the present study, and was trained using the training dataset. Finally, we evaluated the predictive ability of the iPGS model using independent validation datasets.

For Europeans, we constructed the iPGS model using the EstBB case-control dataset of European ancestry, and assessed the predictive ability of the iPGS model using the EstBB validation cohort of European ancestry. For comparison, we evaluated the predictive ability of a PGS model derived from AIS GWAS alone and an iPGS model derived from GIGASTROKE-based stroke GWAS (10 GWAS of AS, AIS, LAS, SVS, and CES for each of cross-ancestry and ancestry-specific). The full iPGS model was derived from 22 GWAS, including 10 GIGASTROKE GWAS and 12 GWAS of stroke-related phenotypes.

The PGS model derived from AIS GWAS alone showed a significant association with the incidence of AIS (HR per 1 SD increase, 1.08; $P = 0.008$) and a modest improvement in C-index (Δ C-index = 0.003). The iPGS model derived from 10 GIGASTROKE GWAS showed a stronger association (HR per 1 SD increase, 1.14; $P = 1.6 \times 10^{-5}$) and larger improvement in C-index (Δ C-index = 0.006). Moreover, the full iPGS model derived from GIGASTROKE and stroke-related GWAS showed an even stronger association (HR per 1 SD increase, 1.26; $P = 2.0 \times 10^{-15}$) and larger improvement in C-index (Δ C-index = 0.027).

Table: Comparison of predictive ability of European PGS models

Derivation GWAS	#. GWAS	C-index	Δ C-index	HR (95% CI)	P
AIS GWAS	1	0.606	0.003	1.08 (1.02–1.15)	0.008
GIGASTROKE GWAS	10	0.609	0.006	1.14 (1.07–1.21)	1.6E-05
GIGASTROKE + risk factor GWAS	22	0.631	0.027	1.26 (1.19–1.34)	2.0E-15

For East Asians, we constructed the full iPGS model using the BBJ case-control training dataset of East Asian ancestry, and evaluated the predictive ability of the iPGS model using the BBJ case-control validation dataset of East Asian ancestry. We compared the predictive performance of the full iPGS

model with that of a PGS model derived from AIS GWAS alone, and that of an iPGS model derived from GIGASTROKE GWAS.

The full iPGS model showed the strongest association with the odds of AIS (OR per 1 SD increase, 1.33; $P = 9.9 \times 10^{-26}$) and the largest improvement in AUC ($\Delta\text{AUC} = 0.019$), followed by the iPGS model derived from GIGASTROKE GWAS and a PGS model derived from AIS GWAS alone, in a descending order.

Table: Comparison of predictive ability of East Asian PGS models

Derivation GWAS	#. GWAS	AUC	ΔAUC	OR (95% CI)	P
AIS GWAS	1	0.643	0.009	1.22 (1.15–1.28)	3.9E-13
GIGASTROKE GWAS	10	0.650	0.016	1.30 (1.23–1.37)	3.1E-22
GIGASTROKE + risk factor GWAS	22	0.653	0.019	1.33 (1.26–1.40)	9.9E-26

These European and East-Asian analyses consistently demonstrated that the iPGS models had a superior predictive ability compared to the standard PGS models derived from a single GWAS.

To our understanding, the GLGC publication (PMID: 34887591) reported the predictive performance of the PGS models derived from single GWAS, and the approaches to the derivation of PGS models were substantially different from the present study. Our approach enabled us to overcome the limited predictive ability of standard PGS by constructing iPGS models and applying this for the first time to a cross-ancestry setting. As such, we believe that the present study would have a particular novelty for stroke risk prediction.

5) While the authors use SUSIE/finemapping techniques to identify putative causal variants, minimal effort is spent in an attempt to identify candidate causal genes at each locus for all identified loci. More recent GWAS manuscripts often integrate multiple lines of evidence (software like PoPs, colocalization with eQTL data in appropriate tissue(s), nearest gene, TWAS, protein altering variants in high LD with sentinel variants, etc) to interpret and identify what they believe is the likely causal gene acting at a given locus. In many cases this may be unclear, and can be stated.

→ To further elicit the most likely causal gene at each locus, and to expand on the extensive bioinformatics follow-up of stroke risk loci summarized in Supplementary Table 30, we performed PoPs for all 60 genome-wide significant loci in the IVW meta-analysis. We used the lead SNP and

SNPs within \pm 500kb for the analysis and selected the gene with the highest PoPs score based on MAGMA results for EUR and EAS separately. As an additional filtering step, we only considered genes for which the PoPs score was in the top 10% of PoPs scores over all genes in the respective analysis. Consistent results across EUR and EAS are reported.

The manuscript has been amended as follows, to summarize more clearly the numerous bioinformatics follow-up analyses we conducted:

Results (p. 22, line 622-627): *“Overall, combining evidence from genomics-driven drug discovery approaches, characterization of stroke risk loci (missense variants, TWAS, PWAS, COLOC, pathway enrichment, MR with pQTL, MENTR, and PoPS³⁸), and prior knowledge from monogenic disease models and experimental data, we found evidence for potential functional implication of 47 genes to be prioritized for further functional follow-up, with evidence from multiple approaches for 17 genes (Supplementary Table 30).”*

6) This referee believes the interpretation of the FGA locus as a direct link to alteplase (a tissue plasminogen activator) is a bit of a reach. While FGA and a large component of thrombus (fibrin mesh) are certainly critical in this pharmacologic mechanism, the nuance of alteplase and its effect in plasminogen in thrombus degradation is completely overlooked. This should be mentioned in greater detail to the reader.

→ We thank the reviewer for raising this point and agree that the link between the FGA locus and alteplase (a tissue plasminogen activator) is via the conversion of plasminogen into plasmin, which in turn degrades fibrin, rather than a direct action of alteplase on FGA and that this link should be made more clear. Accordingly, we now expanded on the Results as follows

Results (page 20, lines 572-574): *“This encompasses the previously described PDE3A and FGA genes, 24 encoding targets for cilostazol (antiplatelet agent) and alteplase (thrombolytic drug acting via plasminogen²⁵),”*

We would like to avoid going into more mechanistic detail as the discovery of FGA as a drug target for alteplase and other thrombolytic agents has previously been reported (Malik, Nat Genet 2018) and because of limitations in space. However, we have added the respective reference from DRUGBANK, which provides more background on the underlying mechanism (<https://go.drugbank.com/drugs/DB00009>). We would be prepared to expand on this further if the reviewer and editors feel this is required.

Minor concerns:

1) There appears to be a theme of highlighting novel loci with [] following a numerical statement of overall loci. This referee found this presentation distracting.

→ This has been edited for more clarity

Results (p. 15, line 418-430): “Out of these 60 loci, the largest number of associations were identified for AS (48 loci [23 novel]) and AIS (45 loci, [18 novel]), of which one with incident AIS only (Supplementary Table 4c). While AIS subtypes were not available in some population-based cohorts (Supplementary Table 1), genome-wide significance was reached for 4 loci for LAS, 8 for CES, and 7 for SVS (of which 1, 3, and 3 were novel respectively, Supplementary Table 4). To our knowledge, our results include the most comprehensive and largest description of stroke genetic risk variants to date in each of the five represented ancestries. In cross-ancestry meta-analyses 53 loci reached genome-wide significance (Supplementary Table 4), while 42 loci were genome-wide significant in individual ancestries (35 in Europeans, 6 in East-Asians, 1 in South-Asians, and 2 in African-Americans, Supplementary Table 4). Using conditional and joint analysis (GCTA-COJO),⁸ we confirmed three independent signals at PITX2 and two at SH3PXD2A (CES in EUR, Supplementary Table 5).¹”

Results (p. 16, line 450-454): “This brings the number of identified stroke risk loci from primary (IVW) and secondary (MR-MEGA and MTAG) analyses to 89 in total (61 novel), of which 69 were associated with AS, 45 with AIS, 15 with LAS, 13 with CES, and 10 with SVS (of these 44, 33, 11, 8, and 3 were novel respectively, Fig. 1, Supplementary Table 4, 8, and 9-14).”

2) Single cell data from the pFC is briefly mentioned, but it is not clear to this referee what this information/data adds to improve the paper. This is particularly evident given how strong other sections of the manuscript are.

→ We thank the referee for the question and have now added a single-cell enrichment analysis.

Results (p. 20, line 559-562): “Overall, we observed a significant enrichment in brain vascular endothelial cells and astrocytes mostly, possibly reflecting the importance of both vascular pathology and brain response to the vascular insult in modulating stroke susceptibility. (Extended Data Fig. 7, Supplementary Table 28-29).”

Methods (p. 8, line 211-219): “We also conducted a cell-type enrichment analysis using the STEAP pipeline (Single cell Type Enrichment Analysis for Phenotypes <https://github.com/ComPopBio/STEAP>). This is an extension to CELLECT and uses S-LDSC,⁹⁰ MAGMA,¹¹ and H-MAGMA⁹¹ for enrichment analysis. Stroke GWAS summary statistics were first munged. Then, expression specificity profiles were calculated using human and mouse single cell RNA-seq databases (Supplementary Table W). Cell-type enrichment was calculated with three models: MAGMA, H-MAGMA (incorporating chromatin interaction profiles from human brain tissues in MAGMA) and stratified LD score regression. P-values were corrected for the number of independent cell-types in each database (Bonferroni correction).”

Supplementary Table 28: “RNA-seq datasets used in the Single cell Type Enrichment Analysis for Phenotypes (STEAP) pipeline”

Supplementary Table 29: “Single cell enrichment analysis”

Extended Data Fig. 7: “Single-cell gene expression/enrichment analysis and proteome-wide association study (PWAS) of stroke in brain tissue”

3) The PheWAS analysis is somewhat limited and could easily be expanded in light of the plethora of open source/access tools that can be used for this purpose (Phenoscaner, ieu open GWAS project, etc).

→ We thank the referee for this suggestion. We have now searched the Phenoscaner database for all the 7 SNPs included in PheWAS analysis and added a new Supplementary Table 37 with the results.

The Phenoscanner results helped to further confirm that we detect no significant association with non-stroke-related phenotypes for the F11 cis-pQTL rs2289252 and confirmed the seen significant associations with venous thromboembolic disorders.

We have edited the manuscript accordingly:

Results (p. 21, line 607-611 and p. 22, line 612-621): *“To further validate the candidate drugs and estimate their potential side effects, we investigated whether the drug-target genes were associated with stroke-related phenotypes using a phenome-wide association study (PheWAS) approach.³⁴ We conducted PheWAS in Estonian Biobank (EstBB) for the pQTL variants for PROC, VCAM1, F11, KLKB1, and GP1BA genes (Supplementary Table 36). A cis-pQTL for F11, rs2289252, was associated with higher risk of venous thromboembolic disorders ($p < 3.95 \times 10^{-6}$), as previously described,³⁵ and showed suggestive association ($p = 3.44 \times 10^{-3}$) with cerebral artery occlusion with cerebral infarction (Phecode 433.21, Extended Data Fig. 8). Conversely, we observed no significant association with non-stroke-related phenotypes, suggesting the safety of targeting F11. Similar profiles were observed in UK Biobank and FinnGen (<https://r5.finnngen.fi/variant/4-186286227-C-T>), with no significant associations with other disorders and no overlap of subthreshold signals with side-effects reported in clinical trials.³⁶ We further confirmed the association of rs2289252 with venous thromboembolic disorders and having no association with other non-stroke-related phenotypes using the Phenoscanner database (Supplementary Table 37).”*

Methods (p. 11, line 285-293): *“PheWAS analysis was carried out using the R software (4.0.3). We used the PheWAS R package¹⁰⁹ (<https://github.com/PheWAS/PheWAS>) function “createPhenotypes” to translate ICD10 diagnosis codes into phecodes for the PheWAS analysis. We tested the associations between phecodes and genetic variants using logistic regression and adjusting for sex, birth year and 10 genotype PCs. We applied Bonferroni correction to select statistically significant associations (number of tested Phecodes: 1809, number of tested SNPs: 7; corrected p-value threshold: $0.05/1809 \times 7 = 3.95 \times 10^{-6}$). Results were visualized using the PheWAS library. To further characterize the associations of the genetic variants with other phenotypes we searched for all 7 SNPs in the PhenoScanner database.^{110,111}”*

4) Figure 2b was somewhat blurry on the PDF version of the article this referee read. The resolution on this image could be enhanced

→ We apologize for this and have improved the resolution. The full scale figure is also uploaded separately.

5) In reference to point #1, the discussion mentions that independent validation is available through the use of population studies/clinical trials. This referee believes this sentence is misleading and should be amended.

→ This has now been amended with the addition of a large external follow-up study described in detail above.

Reviewer Reports on the First Revision:

Referees' comments:

Referee #1 (Remarks to the Author):

The authors have sufficiently addressed my concerns.

Referee #2 (Remarks to the Author):

All my concerns have been adequately addressed.

I thank the authors for their detailed responses, and commend them on their effort to gather a substantial replication cohort and expand the drug discovery section.

Correcting for multiple testing of the stroke phenotypes is important to limit false positive discoveries. However, as the authors argue, the inclusion of an external cohort to explicitly replicate putative loci removes the need for a conservative discovery threshold. I agree with this point: having a nominal discovery threshold is acceptable given the presence of a replication cohort. In addition, the tiered levels of confidence provided by the authors is a useful and transparent way of presenting the results.

Please see some additional comments below:

Are the P values in the external replication dataset one-sided or two-sided? As the authors are attempting to confirm consistency of SNP effect direction with the discovery dataset, I believe a one-sided test is appropriate here.

The replication analysis would benefit from a side-by-side visual representation of discovery and replication effect sizes, similar to figure 3 from <https://doi.org/10.1038/s41467-017-00934-5>.

The authors apply several thresholds for colocalisation, e.g. TWAS PP.H4 ≥ 0.75 ; PWAS PP.H4 > 0.75 ; pQTL MR PP.H4 > 0.80 . Is there a reason to use different thresholds?

Line 419 - In line with other text in parentheses, I would expect "(48 loci [23 novel])" to be written as "(48 loci, 23 novel)".

Line 580 - The authors state their Trans-Phar correlation analysis is 'significant', but they should explicitly state the significance threshold here is FDR < 0.10 .

Line 1333 - "0.05/1809*7" should be "0.05/(1809*7)"

For ease of navigation, the Supplementary Tables file would benefit from a table of contents, listing the names of all tables within the file.

Supplementary Table 6 - the legend incorrectly states the table is sorted by the genomic location of the gene instead of P value.

Supplementary Table 14 - Is the choice to use VRAI/FAUX instead of TRUE/FALSE deliberate for the internal and external replication columns? I would prefer the authors to use the latter.

Supplementary Table 33 - The CSF-based replication of the KLKB1xAIS MR result contains BORDELIN [sic] instead of TRUE or FALSE for the colocalisation test. The authors should either define suggestive colocalisation formally (e.g. $PP.H4 > 0.75$) and apply this definition consistently across findings, or stick to their binary notation and report the borderline result as FALSE.

Referee #3 (Remarks to the Author):

In the revised manuscript, the authors have laudably responded to this reviewer's comments/concerns. This is particularly evident in the effort to obtain independent replication, as well as providing a clear distinction between primary (IVW) and secondary (MTAG/MRMEGA) GWAS signals. Overall, the changes largely improve the paper. I have three minor comments related to the genomics driven drug discovery/drug target analysis that have come to light with the addition of new data.

Minor comments:

- 1) The new gene burden results from UK Biobank highlighted by the authors suggest that loss of function in F11 is protective for venous thromboembolism, and by extension, for stroke and its subtypes. In the context of stroke, while F11 remains a logical target for cardioembolic stroke (and theoretically, large artery stroke though I assume this was tested and results were not significant), the authors also report a statistically significant effect for "any stroke" - a phenotype inclusive of hemorrhagic stroke. This is highlighted in figure 3, and is depicted in supplementary table 33. While presumably the AS signal is being driven by CES and/or LAS, this needs to be further clarified in the text. After reading the revised version of the manuscript, this reviewer felt that a non-clinical audience could gather that F11 is a target for AS and potentially, hemorrhagic stroke. I suspect this is not what the authors are suggesting.
- 2) By extension, the same could be concluded for PROC in Fig 3 and Supplementary table 33
- 3) The methods section describing the pQTL MR does not clearly define the stroke subtypes being tested (presumably AS, AIS, SVS, CES based on figure 3, but then Supplementary Table 33 also includes LAS in the outcomes). It would be helpful to clearly state this both in the results section, but more importantly in the methods. There is also no specific mention of the statistical significance thresholds used for this analysis (Bonferroni based on the number of stroke subtypes and proteins being tested?)

Author Rebuttals to First Revision:

Referees' comments:

Referee #1 (Remarks to the Author):

The authors have sufficiently addressed my concerns.

Referee #2 (Remarks to the Author):

All my concerns have been adequately addressed.

I thank the authors for their detailed responses, and commend them on their effort to gather a substantial replication cohort and expand the drug discovery section.

Correcting for multiple testing of the stroke phenotypes is important to limit false positive discoveries. However, as the authors argue, the inclusion of an external cohort to explicitly replicate putative loci removes the need for a conservative discovery threshold. I agree with this point: having a nominal discovery threshold is acceptable given the presence of a replication cohort. In addition, the tiered levels of confidence provided by the authors is a useful and transparent way of presenting the results.

Please see some additional comments below:

Are the P values in the external replication dataset one-sided or two-sided? As the authors are attempting to confirm consistency of SNP effect direction with the discovery dataset, I believe a one-sided test is appropriate here.

→ We thank Referee 2 for this comment. The p-values provided in the replication were two-sided. We agree with the Referee that a one-sided test would have been sufficient. However, this would lead to a large number of changes throughout the manuscript, tables and figures and would require more time. If the editor and referees agree we suggest sticking to our current results and adding this information in the **Methods (p. 37, line 80-81):** “Two-sided p-values were used for both discovery and replication analysis.” This approach is conservative and will therefore not have increased our type I error rate. It is also consistent with other GWAS in the literature (e.g. Sinnott-Armstrong et al., Nat Genet, 2021; Wyss et al., Nat Communications, 2018).

The replication analysis would benefit from a side-by-side visual representation of discovery and replication effect sizes, similar to figure 3 from <https://doi.org/10.1038/s41467-017-00934-5>.

→ We thank Referee 2 for this suggestion. We have now added an Extended data figure 2 which consists of two panels for the 48 replicated signals from the IVW meta-analysis:

Extended data figure 2: Graphical representation of replication results:

Shown are effect sizes and 95% confidence intervals for the 48 replicated IVW loci for (A) cross-ancestry discovery and replication association results for any stroke and (B) cross-ancestry discovery and replication association results for any ischemic stroke. OR=odds ratio; 95% CI=95% confidence interval.

The authors apply several thresholds for colocalisation, e.g. TWAS $PP.H4 \geq 0.75$; PWAS $PP.H4 > 0.75$; pQTL MR $PP.H4 > 0.80$. Is there a reason to use different thresholds?

→ We thank Referee 2 for this question. Both 0.75 and 0.80 PP.H4 thresholds are commonly used for TWAS, PWAS and pQTL MR analyses in the literature. For consistency across the manuscript, as suggested by the Referee, we have now adopted 0.75 as the PP.H4 threshold for all analyses (consistent with other recent studies using pQTL-based MR analyses, e.g. Storm, Nat Commun 2021; Kia, JAMA Neurol 2021; Wu, J Transl Med 2022). Following these changes two additional associations (LAMC2 with SVS and KLKB1 with AIS) became significant in our pQTL MR colocalization analyses. We have updated Supplementary Tables 33-38 accordingly and the manuscript as follows:

Abstract (p. 13, line 389-392): “Using a novel three-pronged approach,⁴ we provided genetic evidence for putative drug effects, highlighting F11, KLKB1, PROC, GP1BA, LAMC2, and VCAM1 as possible targets, with drugs already under investigation for stroke for F11 and PROC.”

Results (p. 20, line 620- page 21 line 652): “Third, we used protein quantitative trait loci (pQTL) for 218 drug-target proteins as instruments for MR and found evidence for causal associations of 9 plasma proteins with stroke risk (4 cis-pQTL, 6 trans-pQTL), of which 7 were supported by colocalization analyses, with no evidence for reverse causation using the Steiger test (PROC, VCAM1, F11, KLKB1, MMP12, GP1BA, and LAMC2 **Supplementary Table 33**). All of these replicated (at FDR < 0.05), with consistent directionality using at least one independent plasma pQTL resource and CSF pQTL for PROC and KLKB1, with evidence for colocalization for PROC, F11, KLKB1, and MMP12, except GP1BA (for which both concordant and discordant directionality was observed) and LAMC2 (pQTL available in one replication dataset only, FDR=0.08). Using public drug databases we curated drugs targeting those proteins in a direction compatible with a beneficial therapeutic effect against stroke based on MR estimates and identified such drugs for VCAM1, F11, KLKB1, GP1BA, LAMC2 (inhibitors) and PROC (activators, **Supplementary Table 34**). Drugs targeting F11 (NCT04755283, NCT04304508, NCT03766581) and PROC (NCT02222714) are currently under investigation for stroke, and our results provided genetic support for this. Of note, F11 and KLKB1 are adjacent genes with a long range linkage disequilibrium pattern and complex co-regulation,²⁸ as illustrated here by the presence of a shared trans-pQTL in KNG1 (**Supplementary Table 33**). Additional studies are needed to disentangle causal associations and the most appropriate drug target in this region.^{29,30} Next, for the five genes targeted by inhibitors, VCAM1, F11, KLKB1, GP1BA, and LAMC2 we examined the associations of rare deleterious variants (MAF<0.01) with stroke and stroke-related traits applying gene-based burden tests to whole-exome sequencing data from >450,000 UK Biobank participants to support potential therapeutic targets for inhibitors.³¹ We observed one significant protective association of rare deleterious variants in F11 with venous thromboembolism (OR=0.471, $p=2.46\times 10^{-4}$), in a direction concordant with that of MR estimates (**Supplementary Table 35**). To further validate the candidate drugs and estimate their potential side effects, we investigated whether the drug-target genes were associated with stroke-related phenotypes using a phenome-wide association study (PheWAS) approach. We conducted PheWAS in the Estonian Biobank (EstBB) for the pQTL variants for the PROC, VCAM1, F11, KLKB1, GP1BA, and LAMC2 genes. A cis-pQTL for F11, rs2289252, was associated with higher risk of venous thromboembolic disorders ($p<3.45\times 10^{-6}$), as previously described,³² and showed suggestive association ($p=3.44\times 10^{-3}$) with cerebral artery occlusion with cerebral infarction (Phecode 433.21, **Extended Data Fig. 9, Supplementary Table 36**).”

Discussion (p. 26, line 795-812): “Our results provide genetic evidence for putative drug effects using three independent approaches, with converging results from two methods (gene enrichment analysis and pQTL-based MR) for drugs targeting F11 and KLKB1. F11 and F11a inhibitors (e.g. abelacimab, BAY 2433334, BMS-986177) are currently explored in phase-2 trials for primary or secondary stroke prevention (NCT04755283, NCT04304508, NCT03766581). pQTL-based MR suggested PROC as a potential drug target for stroke. A recombinant variant of human activated protein C (encoded by PROC) was found to be safe for the treatment of acute ischemic stroke following thrombolysis, mechanical thrombectomy or both in phase 1 and 2 trials (3K3A-APC, NCT02222714),^{43,44} and is poised for an upcoming phase 3 trial. 3K3A-APC is proposed as a neuroprotectant, with evidence for protection of white matter tracts and oligodendrocytes from ischemic injury in mice.⁴⁵ Weaker evidence was found for GP1BA, VCAM1 and LAMC2 as potential drug targets for stroke, with evidence for colocalization in only one pQTL dataset. Anfibatide, a GPIIb/IIIa antagonist, reduced blood-brain barrier disruption following ischemic stroke in mice⁴⁶ and is being tested as an antiplatelet drug in myocardial infarction (NCT01585259). While specific VCAM1 inhibitors are not available, probucol, a lipid lowering drug with pleiotropic effects including VCAM1 inhibition was tested for secondary prevention of atherosclerotic events in CAD patients (PROSPECTIVE, UMIN000003307).⁴⁷”

Methods (p. 44, line 285-286): “We considered that colocalization occurred when the maximum posterior probability (i.e., PP.H4) was greater than 0.75 or R^2 was greater than 0.8.”

Line 419 - In line with other text in parentheses, I would expect “(48 loci [23 novel])” to be written as “(48 loci, 23 novel)”.

→ We apologize for this oversight and have corrected this accordingly.

Results (p. 15, line 447-449): “Out of these 60 loci, the largest number of associations were identified for AS (48 loci, 23 novel) and AIS (45 loci, 18 novel), of which one with incident AIS only (Supplementary Table 4c).”

Line 580 - The authors state their Trans-Phar correlation analysis is ‘significant’, but they should explicitly state the significance threshold here is FDR < 0.10.

→ We thank the Referee for this comment and have clarified the text accordingly:

Results (p. 20, line 612-616): “Second, we used Trans-Phar²⁴ to test the negative correlations between genetically determined case-control gene expression associated with stroke (TWAS using all GTEX v7 tissues¹⁷) and compound-regulated gene expression profiles. At FDR<0.10, we observed significant negative correlations for BRD.A22514244 (for SVS; drug target unknown) and GR.32191 (for CES, **Supplementary Table 32**).”

*Line 1333 - "0.05/1809*7" should be "0.05/(1809*7)"*

→ We thank the Referee for this comment and have clarified the text accordingly (please note that following another comment from the referee the number of SNPs tested has increased to 8)

Methods (p. 44, line 305-309): *"We applied Bonferroni correction to select statistically significant associations (number of tested Phecodes: 1809, number of tested SNPs: 8; corrected p-value threshold: $0.05/[1809 \times 8] = 3.45 \times 10^{-6}$). Results were visualized using the PheWAS library. To further characterize the associations of the genetic variants with other phenotypes we searched for all 8 SNPs in the PhenoScanner database.^{98,99"}*

For ease of navigation, the Supplementary Tables file would benefit from a table of contents, listing the names of all tables within the file.

→ We thank the Referee for this suggestion and have added a table of contents in the first sheet of the supplementary tables file.

Supplementary Table 6 - the legend incorrectly states the table is sorted by the genomic location of the gene instead of P value.

→ We thank the Referee for noticing this inconsistency, which we have corrected: The Suppl. Table 6 footnote *"Sorted by genomic location of a gene (chromosome and start position)"* has now been updated to *"Sorted by VEGAS2-P"*

Supplementary Table 14 - Is the choice to use VRAI/FAUX instead of TRUE/FALSE deliberate for the internal and external replication columns? I would prefer the authors to use the latter.

→ We thank the Referee for this question. We have updated this to TRUE/FALSE in Supplementary Table 14, as well as in Supplementary Table 33a and Supplementary Table 33b (this issue was related to French versions of Excel)

Supplementary Table 33 - The CSF-based replication of the KLKB1xAIS MR result contains BORDELIN [sic] instead of TRUE or FALSE for the colocalisation test. The authors should either define suggestive colocalisation formally (e.g. $PP.H4 > 0.75$) and apply this definition consistently across findings, or stick to their binary notation and report the borderline result as FALSE.

→ We thank the Referee for this question. In response to question 3 by the Referee we have now adopted 0.75 as the PP.H4 threshold for our TWAS, PWAS and pQTL MR analyses. Following these

changes, the association of KLKB1 with AIS, became significant in our pQTL MR colocalization analysis. We have updated Supplementary Table 33b accordingly.

Referee #3 (Remarks to the Author):

In the revised manuscript, the authors have laudably responded to this reviewer's comments/concerns. This is particularly evident in the effort to obtain independent replication, as well as providing a clear distinction between primary (IVW) and secondary (MTAG/MRMEGA) GWAS signals. Overall, the changes largely improve the paper. I have three minor comments related to the genomics driven drug discovery/drug target analysis that have come to light with the addition of new data.

Minor comments:

1) The new gene burden results from UK Biobank highlighted by the authors suggest that loss of function in F11 is protective for venous thromboembolism, and by extension, for stroke and its subtypes. In the context of stroke, while F11 remains a logical target for cardioembolic stroke (and theoretically, large artery stroke though I assume this was tested and results were not significant), the authors also report a statistically significant effect for "any stroke" - a phenotype inclusive of hemorrhagic stroke. This is highlighted in figure 3, and is depicted in supplementary table 33. While presumably the AS signal is being driven by CES and/or LAS, this needs to be further clarified in the text. After reading the revised version of the manuscript, this reviewer felt that a non-clinical audience could gather that F11 is a target for AS and potentially, hemorrhagic stroke. I suspect this is not what the authors are suggesting.

→ We thank the Referee for this note of caution.

Indeed, associations between F11 pQTL and large artery stroke were non-significant (only significant results are reported in Supplementary Table 33a).

We have also clarified that association results with AS cannot be extrapolated to ICH.

Results (p.20, line 601–614): “We used a three-pronged approach for genomics-driven discovery of drugs for prevention or treatment of stroke (**Methods, Fig. 3**).⁴ First, using GREP²² we observed significant enrichment of stroke-associated genes (MAGMA⁹ or VEGAS2⁸ false discovery rates [FDR] <0.05) in drug-target genes for blood and blood-forming organs (Anatomical Therapeutic Chemical Classification System [ATC] B drugs, for AS, AIS, and CES). This encompasses the previously described PDE3A and FGA genes,¹ encoding targets for cilostazol (antiplatelet agent) and alteplase (thrombolytic drug acting via plasminogen²³), respectively, as well as F11, KLKB1, F2, TFPI, and MUT encoding targets for conestat alfa, ecallantide (both used for hereditary angioedema), lepirudin, dalteparin (both used to treat recurrent thromboembolism), and vitamin B12, respectively (**Supplementary Table 31**). Of note, results for AS are likely driven by AIS (the vast majority of AS in the current study) and cannot be extrapolated to ICH. Second, we used Trans-Phar²⁴ to test the negative correlations between genetically determined case-control gene expression associated with stroke (TWAS using all GTEX v7 tissues¹⁷) and compound-regulated gene expression profiles.”

2) By extension, the same could be concluded for PROC in Fig 3 and Supplementary table 33

→ We thank the Referee for this note of caution. Please see response to the previous comment.

3) The methods section describing the pQTL MR does not clearly define the stroke subtypes being tested (presumably AS, AIS, SVS, CES based on figure 3, but then Supplementary Table 33 also includes LAS in the outcomes). It would be helpful to clearly state this both in the results section, but more importantly in the methods. There is also no specific mention of the statistical significance thresholds used for this analysis (Bonferroni based on the number of stroke subtypes and proteins being tested?)

→ We thank the Referee for this question. We have now clarified the stroke phenotypes tested in pQTL MR analyses (AS, AIS, SVS, CES, LAS) in the Methods, as well as the statistical significance thresholds used in this context (FDR <0.05).

We made following changes in Methods (p. 43 line 265- p.44 line 291):

“Endophenotype Mendelian randomization

To pin-point the disease-causing proteins that were targeted by existing drugs, we performed MR analysis (specifically, Wald ratio test) by using lead variants in protein quantitative trait loci (pQTL) as instrumental variables and 5 stroke phenotypes as outcomes: AS, AIS, CES, LAS, SVS. We used the tier 1 lead variants defined by Zheng et al.⁸⁶ to avoid confounding by horizontal pleiotropy. The tier 1 variants were summarized from five pQTL studies (N=997 to 6,861)⁸⁷⁻⁹¹ and excluded the variants with heterogeneous effect sizes among the studies or the number of associated proteins larger than five. We restricted the lead variants to the variants associated with drug-target proteins. For the lead variants of pQTL that were missing in the stroke GWAS summary statistics, the proxy variants with the largest R^2 were used if the R^2 was greater than 0.8 (1000G EUR). In total, we used 277 lead variants for 218 drug-target proteins for MR and considered the FDR < 0.05 threshold to identify significant associations. We used the “TwoSampleMR” R package⁹² for MR analysis. As post-MR quality controls, we performed (i) directionality check of causal relationships by Steiger filtering⁹³ and (ii) colocalization analysis for the proteins with FDR < 0.05. To examine colocalization assuming multiple causal variants per locus, coloc⁷⁴ was applied to the decomposed signals by SuSiE¹² for the variants within 500 kb upstream and downstream of the lead variants (coloc + SuSiE).⁹⁴ If SuSiE did not converge after 10,000 iterations, coloc was used instead. Coloc + SuSiE and coloc were run with their respective default parameters. For the two pQTL studies without public summary statistics,^{87,91} we compared the R^2 between the lead variants of the pQTL study and the stroke GWAS. We considered that colocalization occurred when the maximum posterior probability (i.e., PP.H4) was greater than 0.75 or R^2 was greater than 0.8.

To provide further support for our findings we conducted MR analyses with two additional recent independent pQTL datasets, using the same methodology and significance thresholds (FDR<0.05 for MR and PP.H4>0.75 for colocalization as above: one study comprised both plasma (N=529) and cerebrospinal fluid (CSF, N=835) pQTL datasets,⁹⁵ the second is one of the largest plasma pQTL studies conducted in 35,559 Icelanders.⁹⁶